# A quinolone *N*-oxide antibiotic selectively targets *Neisseria gonorrhoeae* via its toxin–antitoxin system

Ann-Kathrin Mix[1], Thi Hong Nhung Nguyen[2], Tamara Schuhmacher[1], Dávid Szamosvári[2], Petra Muenzner[1], Paula Haas[1], Lydia Heeb[1], Haleluya T. Wami[3], Ulrich Dobrindt[3], Yasar Özge Delikkafa[4], Thomas U. Mayer [4,5], Thomas Böttcher [2] ✉ & Christof R. Hauck [1,5] ✉

Gonorrhoea is a major sexually transmitted infection and the emergence of multidrug-resistant *Neisseria gonorrhoeae* poses a global health threat. To identify candidate antibiotics against *N. gonorrhoeae*, we screened *Pseudomonas aeruginosa*-derived secondary metabolites and found that 2-nonyl-4-quinolone *N*-oxide (NQNO) abrogated growth of *N. gonorrhoeae* in vitro. NQNO did not impair growth of commensal *Neisseriae*, vaginal lactobacilli or viability of human cells. Mechanistically, NQNO disrupted the electron transport chain, depleted ATP and NADH levels and increased oxidative stress. This triggered activation of a toxin–antitoxin system, release of the endogenous Zeta1 toxin and bacterial death. In a mouse model of infection, topical application of NQNO prevented colonization by *N. gonorrhoeae*. Chemical modification yielded 3-methyl NQNO, which exhibited nanomolar potency against multidrug-resistant strains, lack of resistance development and significantly reduced pathogen numbers during experimental infection of mice. These findings show the potential for selective killing of bacterial pathogens such as multidrug-resistant *N. gonorrrhoeae* through activation of endogenous toxins.

With >80 million people affected annually, gonorrhoea is one of the most common sexually transmitted infections[1]. Caused by the human-restricted pathogen *Neisseria gonorrhoeae* (also called the gonococcus), gonorrhoea is characterized by an acute inflammatory response in the genital tract, but can also lead to disseminated disease and infertility in both sexes[2]. As gonorrhoea could be controlled by antibiotic treatment, it has been disregarded as a serious infectious disease. However, *N. gonorrhoeae* is naturally transformable and readily acquires resistance genes[3]. An example is the acquisition of tetracycline resistance in the form of the *tetM* gene and its worldwide spread via a conjugative plasmid (pTetM), which is highly prevalent in many countries[4,5].

Due to widespread multidrug resistance in *N. gonorrhoeae*, the only remaining treatment options are based on a combination of azithromycin and ceftriaxone, an extended-spectrum cephalosporin requiring parenteral application[6]. Strains resistant to these antibiotics have been reported and the current rise in extensive drug-resistant gonococcal strains is of global concern[7]. As a result of increased efforts to combat *N. gonorrhoeae*[8], several compounds are in advanced stages of clinical development[9]. However, these compounds are directed

[1]Lehrstuhl Zellbiologie, Department of Biology, University of Konstanz, Konstanz, Germany. [2]Microbial Biochemistry, Faculty of Chemistry, Institute for Biological Chemistry and Centre for Microbiology and Environmental Systems Science, University of Vienna, Vienna, Austria. [3]Institut für Hygiene, Universität Münster, Münster, Germany. [4]Lehrstuhl Molekulare Genetik, Department of Biology, University of Konstanz, Konstanz, Germany. [5]Konstanz Research School Chemical Biology, Universität Konstanz, Konstanz, Germany. ✉e-mail: Thomas.Boettcher@univie.ac.at; christof.hauck@uni-konstanz.de

against known targets such as DNA gyrase or the prokaryotic ribosome, which have shown resistance development in the past.

Most antibiotics originate from bacterial secondary metabolites involved in microbial competition[10,11]. In this regard, we investigated a class of secondary metabolites released by *Pseudomonas aeruginosa*, the 2-alkyl-4-quinolones (AQs) and their derivatives to determine whether they might have antimicrobial activity. While 2-heptyl-3-hydroxy-4-quinolone serves as a quorum-sensing signal in *P. aeruginosa*[12], other AQ derivatives, such as the 2-alkyl-4-quinolone *N*-oxides (AQNOs), do not function as signalling molecules in *Pseudomonas*[13,14]. Some of the naturally occurring AQNOs are implicated in autolysis of *P. aeruginosa*[15] or show antibacterial activity against *Staphylococcus aureus*[16–19]. Therefore, we speculated that these compounds might act against other microorganisms and conducted an extended screen.

Here we report that 2-nonyl-4-quinolone *N*-oxide (NQNO) has strong antibiotic activity against *N. gonorrhoeae*, including multidrug-resistant gonococcal strains. NQNO does not act on most non-pathogenic neisserial species or unrelated lactobacilli, which live as commensals on human mucosal surfaces. Topical application of NQNO or its improved derivative to the genital tract of female mice was well tolerated and significantly reduced pathogen numbers. Mechanistically, NQNO-mediated inhibition of the electron transport chain resulted in increased reactive oxygen levels, which in turn triggered degradation of the Epsilon1 antitoxin. This involvement of the zeta1–epsilon1 toxin–antitoxin (TA) system can explain the selective vulnerability of *N. gonorrhoeae*. While TA systems have been proposed as anti-infective targets[20], our study identifies a highly selective natural antibiotic and its synthetic derivative that act via an endogenous toxin to combat this high-priority pathogen.

## Results

### NQNO has antibiotic activity against *N. gonorrhoeae* isolates

As soluble factors released by *P. aeruginosa* can affect other microbes, we evaluated the general impact of *P. aeruginosa* strain PAO1 on diverse bacteria by cross-streak assays. While *Klebsiella pneumoniae* was not affected by co-culture with *P. aeruginosa*, growth of enteropathogenic *E. coli* and *N. gonorrhoeae* strain MS11 was strongly inhibited (Extended Data Fig. 1a). Extending this basic growth inhibition test to a variety of gonococcal isolates and additional neisserial species revealed that *N. gonorrhoeae* was generally more sensitive towards *P. aeruginosa*-derived factors than other neisserial species (Extended Data Fig. 1b). Indeed, when *N. gonorrhoeae* was incubated with sterile *P. aeruginosa* culture supernatants, gonococcal growth was suppressed and the bacteria appeared deteriorated, while *N. cinerea* was not affected (Extended Data Fig. 1c). Having previously observed bactericidal activity of *P. aeruginosa* AQNOs[19,21], we tested whether *P. aeruginosa* mutants lacking key enzymes in quinolone biosynthesis exhibited reduced growth inhibition of gonococci. Indeed, the inhibitory effect of *P. aeruginosa* PAO1 on gonococcal growth was diminished upon compromising quinolone biosynthesis (Extended Data Fig. 1d). These data suggest that *P. aeruginosa* releases factors that inhibit the growth of *N. gonorrhoeae* and that gonococci might be particularly sensitive towards AQs or AQ derivatives.

Therefore, we synthesized NQNO and *trans*-Δ[1]-NQNO as well as their corresponding AQs, which are among the most abundant quinolones produced by *P. aeruginosa*[19] (Fig. 1a). The synthetic compounds (5–50 μM) were applied to broth cultures of pathogenic *N. gonorrhoeae* or commensal *N. cinerea*. Strikingly, 2-nonyl-4-quinolone (NQ), *trans*-Δ[1]-2-nonenyl-4-quinolone (*trans*-Δ[1]-NQ) and 2-nonyl-4-quinolone *N*-oxide (NQNO) completely inhibited growth of *N. gonorrhoeae* MS11 (Fig. 1b). While *trans*-Δ[1]-NQNO was considerably less active, NQNO inhibited growth of *N. gonorrhoeae* MS11 even at the lowest concentration of 5 μM (~1.5 μg ml⁻¹) (Fig. 1b). In contrast to the effect on gonococci, neither of the compounds had a major impact on the growth of non-pathogenic *N. cinerea* (Fig. 1b). Growth assays with varying concentrations of NQNO

were independently repeated for *N. gonorrhoeae* MS11 and *N. cinerea*, consistently showing gonococcal growth inhibition at NQNO concentrations ≥5 μM (Extended Data Fig. 2a). The effect of NQNO on growth of gonococcal cultures was corroborated by determining colony-forming units (c.f.u.s) instead of optical density (OD$_{550}$) (Extended Data Fig. 2b). Extending from this initial analysis, we performed in vitro growth inhibition assays using a variety of gonococcal isolates from local infections or disseminated disease (Fig. 1c, Extended Data Fig. 3 and Supplementary Table 1). Furthermore, we tested the susceptibility of various commensal *Neisseria* species inhabiting human or non-human primate mucosal surfaces (Fig. 1c and Extended Data Fig. 4a). The results underlined the striking selectivity of AQs, in particular NQNO and *trans*-Δ[1]-NQ, for inhibiting growth of *N. gonorrhoeae*. While all gonococcal strains were strongly impaired in their growth, most commensal *Neisseriae* showed negligible sensitivity to NQNO (Fig. 1c). An exception was seen for *N. lactamica*, a species generally regarded as commensal but closely related to *N. gonorrhoeae*[22] (Fig. 1c). Motivated by the pronounced growth inhibitory effect of the quinolones on *N. gonorrhoeae*, we tested their effect on major commensals of the human female genital tract. Importantly, none of the Gram-positive lactobacilli tested (*L. delbruecki, L. gasseri, L. paragasseri, L. homini, L. jensenii, Limosilactobacillus vaginalis*) was inhibited by NQNO (Extended Data Fig. 4b). Therefore, our combined results reveal potent anti-gonococcal activity of *Pseudomonas*-produced AQ(NO)s in general and a highly selective activity of NQNO in particular, which suggested that this compound could be a promising agent to limit gonococcal infections.

### NQNO affects the gonococcal electron transport chain

The 2-heptyl-quinolone *N*-oxide (HQNO) of *Pseudomonas* has been implicated in interfering with respiratory chain cytochromes in prokaryotic and eukaryotic cells[15,23,24]. Similar to HQNO, NQNO has also been reported to stall electron delivery to cytochrome b by blocking the quinol oxidation site (Qp site) of this membrane protein[25]. In line with an inhibitory effect on the respiratory chain, we observed a severe reduction in ATP levels within 20 min upon treatment of *N. gonorrhoeae* with NQNO, whereas ATP levels were unaffected in *N. cinerea* (Fig. 2a). Interestingly, Antimycin A, a known antagonist of cytochrome b-dependent electron transport in mammals, also affected ATP synthesis in *N. gonorrhoeae*, while it did not reduce ATP levels in *N. cinerea*, indicating that the respiratory chain of these related microorganisms differs in its vulnerability towards small compounds (Fig. 2a). Concomitant with the depletion of ATP, NQNO diminished the NADH levels in *N. gonorrhoeae* in a dose-dependent manner (Fig. 2b). NADH depletion by NQNO again mimicked the phenotype induced by the cytochrome b inhibitor Antimycin A, and both treatments selectively affected gonococcal NADH levels, but not NADH levels in *N. cinerea* (Fig. 2b). As blockage of electron transfer to cytochrome b should lead to elevated production of reactive oxygen species (ROS)[26], we measured oxidation of lucigenin in whole bacteria treated with NQNO. Within 15 min of NQNO treatment, reactive oxygen levels in *N. gonorrhoeae* increased in a dose-dependent manner and, upon application of 50 μM NQNO, were comparable to ROS levels observed upon Antimycin A treatment (Fig. 2c,d). *N. cinerea* responded neither to NQNO nor to Antimycin A (Fig. 2c), further strengthening the idea that the respiratory chain of gonococci is particularly sensitive to this natural product. Importantly, NQNO at concentrations up to 50 μM did not affect the viability of human cervix carcinoma cells (HeLa cells, ME-180 cells) or immortalized human vaginal epithelial cells (hVECs) in vitro (Extended Data Fig. 5a,b). Under cell culture conditions in serum-containing medium at 37 °C, NQNO was stable over several days (Extended Data Fig. 5c). These findings indicate that there could be a sufficiently large therapeutic window to allow the application of NQNO in human tissues.

### NQNO inhibits growth of multidrug-resistant *N. gonorrhoeae*

Candidate therapeutics for treating gonorrhoea should inhibit growth of antibiotic-resistant clinical isolates. Therefore, we tested the activity

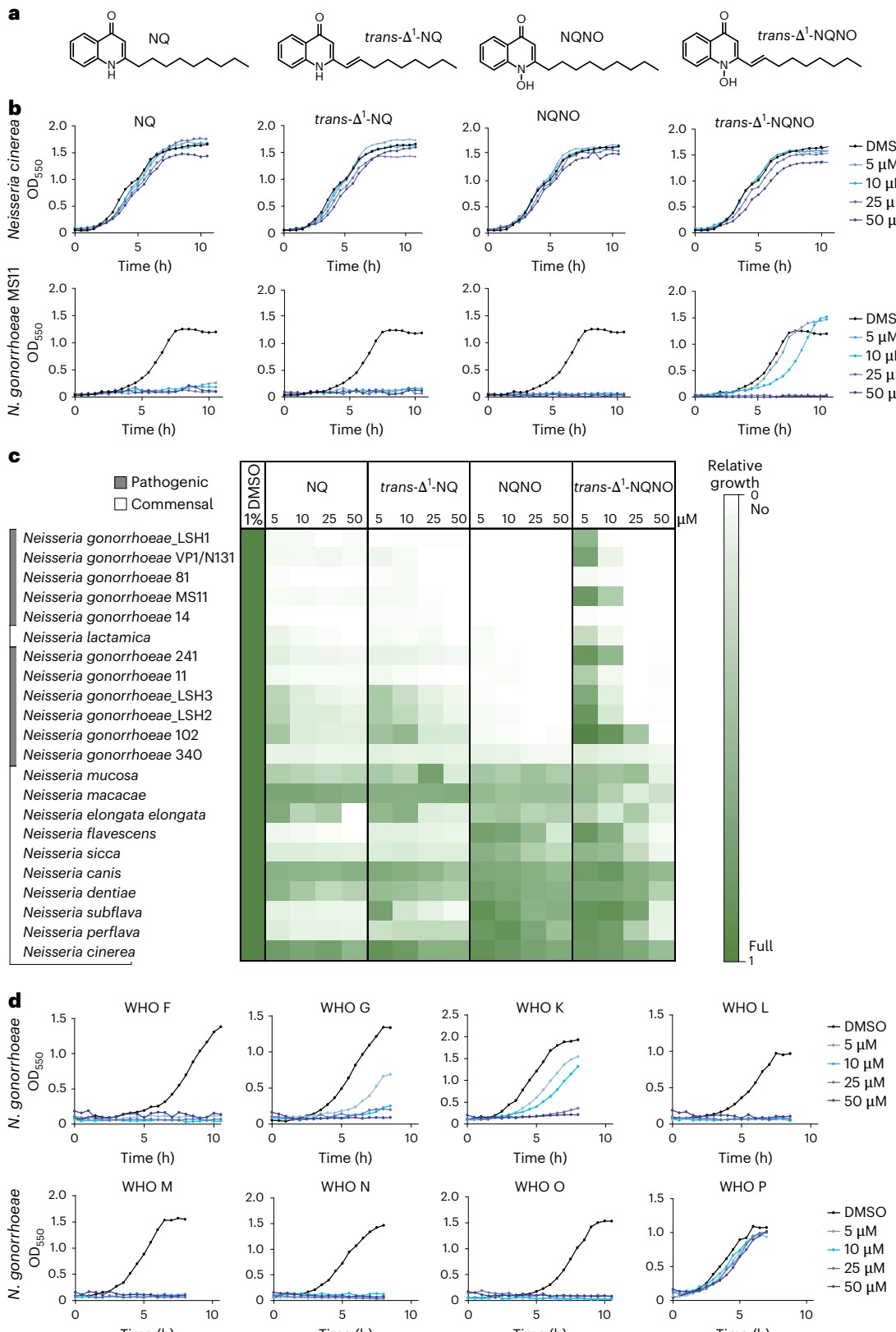

**Fig. 1 | NQNO has selective antibiotic activity against *N. gonorrhoeae*.**
**a**, Structure of the synthesized 2-alkyl-4-quinolones and corresponding *N*-oxides of *P. aeruginosa*. **b**, Growth curves of *N. gonorrhoeae* MS11 and *N. cinerea* incubated with 5, 10, 25 or 50 μM of NQ, *trans*-Δ¹-NQ, NQNO or *trans*-Δ¹-NQNO; control cultures were grown with 1% solvent (DMSO). Growth was monitored using $OD_{550}$ readings every 30 min. Data are representative of 3 independent experiments. **c**, Effect of NQ, *trans*-Δ¹-NQ, NQNO or *trans*-Δ¹-NQNO on the growth

of different gonococcal strains and *Neisseria* species. Growth was measured as in **b** and quantified using area under the growth curve. The colour gradient indicates growth compared to control (1% DMSO), from no growth (90–100% inhibition of growth, white) to full growth (0–10% inhibition of growth, dark green). **d**, Growth of *N. gonorrhoeae* WHO reference strains in the presence of 5, 10, 25 or 50 μM of NQNO. Control cultures were grown with 1% DMSO. Growth was monitored as in **b**.

of NQNO on a range of antibiotic-resistant strains compiled by the WHO[27]. Similar to *N. gonorrhoeae* MS11, most of the drug-resistant gonococcal strains were also affected by NQNO (Fig. 1d). It is important to point out that the efficacy of NQNO was not impaired by existing fluoroquinolone resistance of several WHO strains. This is best exemplified by strain WHO N, which was resistant against ciprofloxacin and rosoxacin (Extended Data Fig. 6a), but was highly sensitive towards NQNO (Fig. 1d). Conversely, strain WHO P was resistant to NQNO (Fig. 1d), yet was fully susceptible to ciprofloxacin (Extended Data Fig. 6a). While fluoroquinolones share the basic 4-quinolone ring system with AQs, they do not interfere with the respiratory chain, but rather act on DNA gyrase and topoisomerase[28]. Among the WHO collection, strain WHO P exhibited NQNO resistance (up to 50 μM) and strain WHO K showed partial resistance to NQNO (up to 10 μM) (Fig. 1d). As WHO P was the only strain in this collection with known resistance to azithromycin, we tested an additional set of multidrug-resistant strains either highly resistant to azithromycin (Azm) or to 3rd generation cephalosporins (Ceph). While the Azm-resistant isolate NCTC 13799 and the Ceph-resistant strain N231 (ref. 29) were fully susceptible to NQNO, the Ceph-resistant strain N316 (ref. 30) showed intermediate resistance to NQNO (Extended Data Fig. 6b). Accordingly, azithromycin resistance does not explain the lack of NQNO-mediated growth inhibition of strain WHO P. Nevertheless, these results indicate that resistance to NQNO can occur among gonococcal strains. Knowing the molecular basis of this resistance could help to elucidate the selective mode of action of NQNO. Therefore, we passaged the NQNO-sensitive strain MS11 over the course of several days in liquid medium containing sublethal concentrations (2.5 μM) of NQNO. Aliquots of these NQNO-conditioned bacteria were tested daily for their growth at higher concentrations (>5 μM) of NQNO (Extended Data Fig. 6c). Only after prolonged conditioning for 8 days, growth of NQNO-resistant gonococci could be observed and two independent clones (MS11-R1 and MS11-R2) were further characterized. The MS11-derived clone R1 showed intermediate levels, while clone R2 showed full resistance up to 50 μM of NQNO (Fig. 2e). Shotgun whole-genome sequences of MS11-R1 and -R2 as well as of the MS11 parental strain were generated and compared. Both NQNO-resistant MS11-derived clones shared several gene deletions as well as missense mutations in hypothetical genes (Supplementary Table 2). In addition, both NQNO-resistant isolates lost the large conjugative pTetM plasmid and concomitantly lost their tetracycline resistance (Supplementary Table 2 and Extended Data Fig. 6d). Surprisingly, both MS11-derived NQNO-resistant strains still showed increased ROS production and reduced NADH levels in response to NQNO (Fig. 2f,g). Therefore, despite the NQNO-mediated impairment of the respiratory chain in *N. gonorrhoeae*, which does not occur in *N. cinerea*, this metabolic interference is not sufficient to explain the susceptibility of gonococci to NQNO.

## The ζ1/ε1 TA system mediates gonococcal NQNO sensitivity

The 25.2 MDa pTetM plasmid, which mediates high-level tetracycline resistance and which was lost in NQNO-resistant MS11 subclones, is derived from an ancestral 24.5 MDa conjugative plasmid found only in *N. gonorrhoeae*[31]. Besides components needed for plasmid maintenance and conjugation, two zeta–epsilon (ζ1/ε1 and ζ2/ε2) toxin–antitoxin (TA) systems are encoded on this extrachromosomal DNA[32]. Ngo ζ1/ε1 is highly conserved among different *N. gonorrhoeae* isolates, while the ζ2/ε2 genes exhibit sequence variation[32]. Gonococcal zeta–epsilon proteins are related to type II TA systems found mainly in Gram-positive bacteria, where these toxins are activated by physiological stress such as nutrient starvation, and induce stasis or programmed cell death of microorganisms[33,34]. As the gain in NQNO resistance in strains MS11-R1 and MS11-R2 was accompanied by loss of the pTetM plasmid, we investigated the prevalence of the ζ1/ε1 genes in these strains and the parent MS11 (Fig. 3a). Not surprisingly, the MS11 parental strain encoded ζ1/ε1 genes, while the NQNO-resistant strains MS11-R1 and MS11-R2 lacked these factors (Fig. 3a). Interestingly, the same correlation was observed for the strains of the WHO collection: the NQNO-sensitive WHO strains encoded the ζ1/ε1 genes, while strains WHO K and WHO P, which showed resistance to NQNO, lacked this operon (Fig. 3a).

Ngo ζ1 toxin encodes a phosphotransferase, which is activated upon proteolytic degradation of the labile ε1 antitoxin and modifies precursors for cell wall synthesis, leading to membrane fragility[35]. Indeed, NQNO treatment of *N. gonorrhoeae* MS11 resulted in increased membrane permeability, while the integrity of the *N. cinerea* membrane was unaltered by NQNO (Fig. 3b,c). The effect of NQNO on cell wall dynamics was also observed by scanning electron microscopy, which revealed irregular surface structures and membrane debris of NQNO-treated *N. gonorrhoeae* MS11, while *N. cinerea* again was unaffected (Extended Data Fig. 7a). To demonstrate the role of the zeta–epsilon TA system in NQNO-mediated inhibition of gonococcal growth, we conjugated the pTetM plasmid back into the MS11-derived, fully NQNO-resistant clone MS11-R2, which lacks pTetM (Fig. 3d). The introduction of the zeta/epsilon-encoding plasmid rendered the pTetM-positive clones sensitive to NQNO (Fig. 3e). Moreover, MS11-R2 pTetM conjugants displayed increased membrane permeability upon NQNO treatment (Fig. 3f and Extended Data Fig. 7b). These results demonstrate that pTetM-encoded determinants, most probably the zeta1–epsilon1 TA system, render gonococci particularly sensitive to NQNO.

To unambiguously pinpoint the contribution of the zeta1–epsilon1 TA system, we introduced either the pTetM plasmid or the isolated *zeta1/epsilon1* operon into WHO P (Extended Data Fig. 7c–e). Both the plasmid-encoded (WHO P pTetM A and WHO P pTetM B) and the chromosomally encoded *zeta1/epsilon1* genes (WHO P ζ1/ε1) were

**Fig. 2 | NQNO affects the gonococcal electron transport chain. a**, *Neisseria gonorrhoeae* and *N. cinerea* were incubated with 0, 25 or 50 μM of NQNO for the indicated time or with Antimycin A for 120 min. ATP levels were determined relative to the DMSO control. Data are shown as mean ± s.e.m. of $n = 3$ (biological replicates) independent experiments for 10, 20, 30, 45 min, 120 min Antimycin A and $n = 2$ (biological replicates) for 120 min, with each experiment conducted in technical triplicates; *$P < 0.05$, **$P < 0.01$, ***$P < 0.001$ significant differences compared with control using one-sample Student's $t$-test; also shown are 95% confidence intervals (CI) and exact two-tailed $P$ values. **b**, *N. gonorrhoeae* and *N. cinerea* were incubated with indicated amounts of NQNO or Antimycin A for 120 min and NADH levels were determined. NADH levels were normalized to the DMSO control. Bars indicate means ± s.e.m. ($n = 3$ biological replicates) with each experiment conducted in technical triplicates. Significance was evaluated using two-way ANOVA with Sidak's multiple comparisons test; NS, not significant; *$P < 0.05$, **$P < 0.01$, ***$P < 0.001$; 95% CIs and exact two-tailed $P$ values also shown. **c**, Production of ROS upon treatment of the indicated bacteria with 5 and 50 μM NQNO was evaluated. Dashed line represents luminescence level of DMSO-treated control cultures. Antimycin A was used as a known respiratory chain inhibitor and ROS inducer. Data indicate means ± s.e.m. ($n = 4$ biological replicates) with each experiment conducted in technical triplicates. Luminescence values measured every other minute over the course of 40 min. **d**, Total ROS production was determined by measuring the AUC from 3 independent experiments as in **c**. Dashed line represents ROS level of DMSO-treated control cultures. Bars indicate means ± s.e.m.; $n = 4$ biological replicates, with each experiment conducted in technical triplicates. Unpaired Students $t$-test; *$P < 0.05$; 95% CI and exact two-tailed $P$ values also shown. **e**, NQNO-resistant clones *N. gonorrhoeae* (Ngo) MS11-R1 and MS11-R2 as well as the MS11 parent strain were grown in the presence of the indicated concentrations of NQNO or 1% solvent (DMSO). Growth was monitored using $OD_{550}$ readings every 30 min ($n = 1$). Data representative of 3 independent experiments. **f**, Production of ROS by *N. gonorrhoeae* MS11-R1 or MS11-R2 treated with NQNO (50 μM) or Antimycin A (36 μM) was measured as in **c**. Data are shown as mean ± s.e.m.; $n = 3$ biological replicates, with each experiment conducted in technical triplicates. **g**, *N. gonorrhoeae* (Ngo) MS11-R1 or MS11-R2 were incubated with DMSO, NQNO or Antimycin A for 120 min as indicated. Determination of NADH levels and statistics as in **b**.

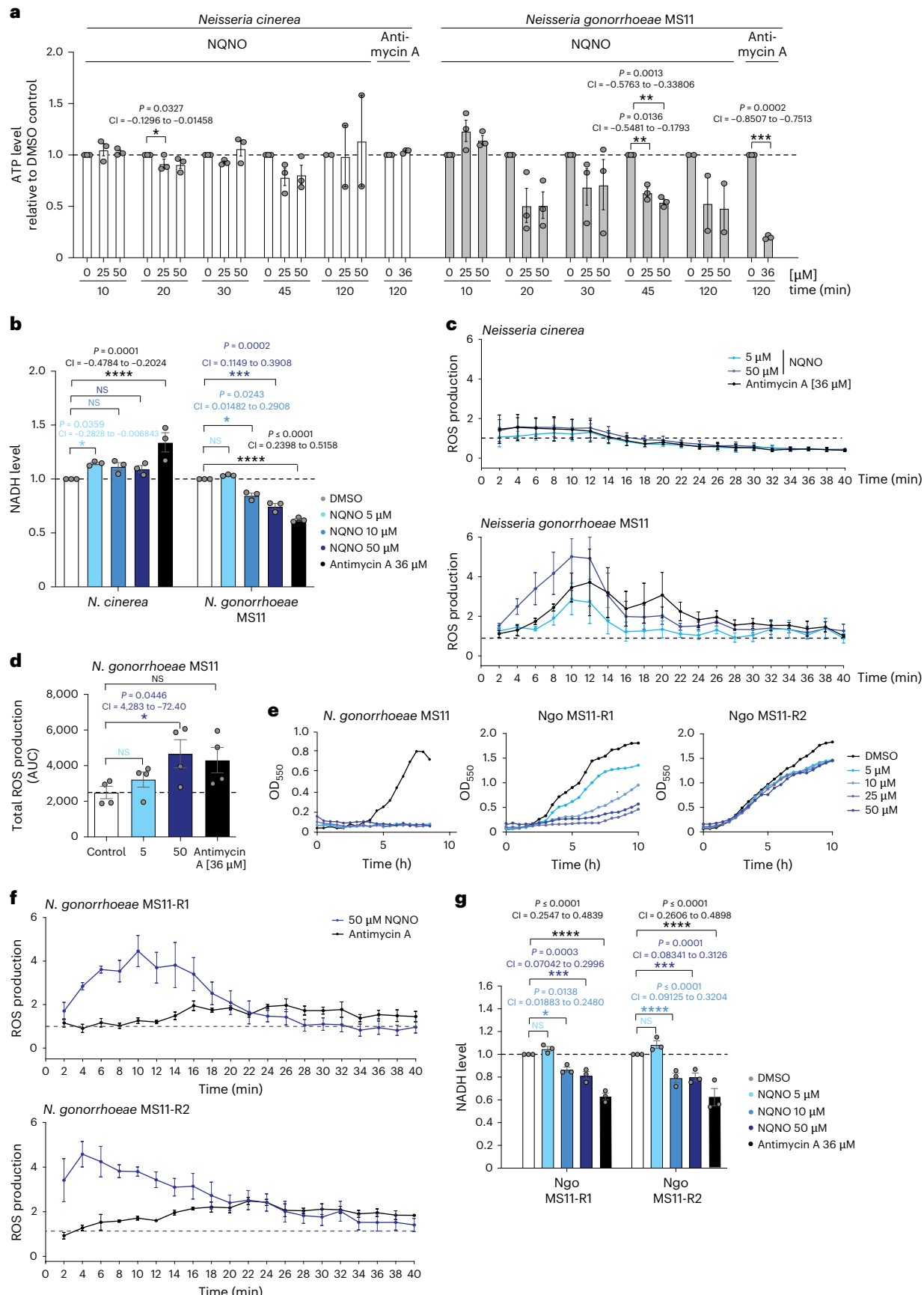

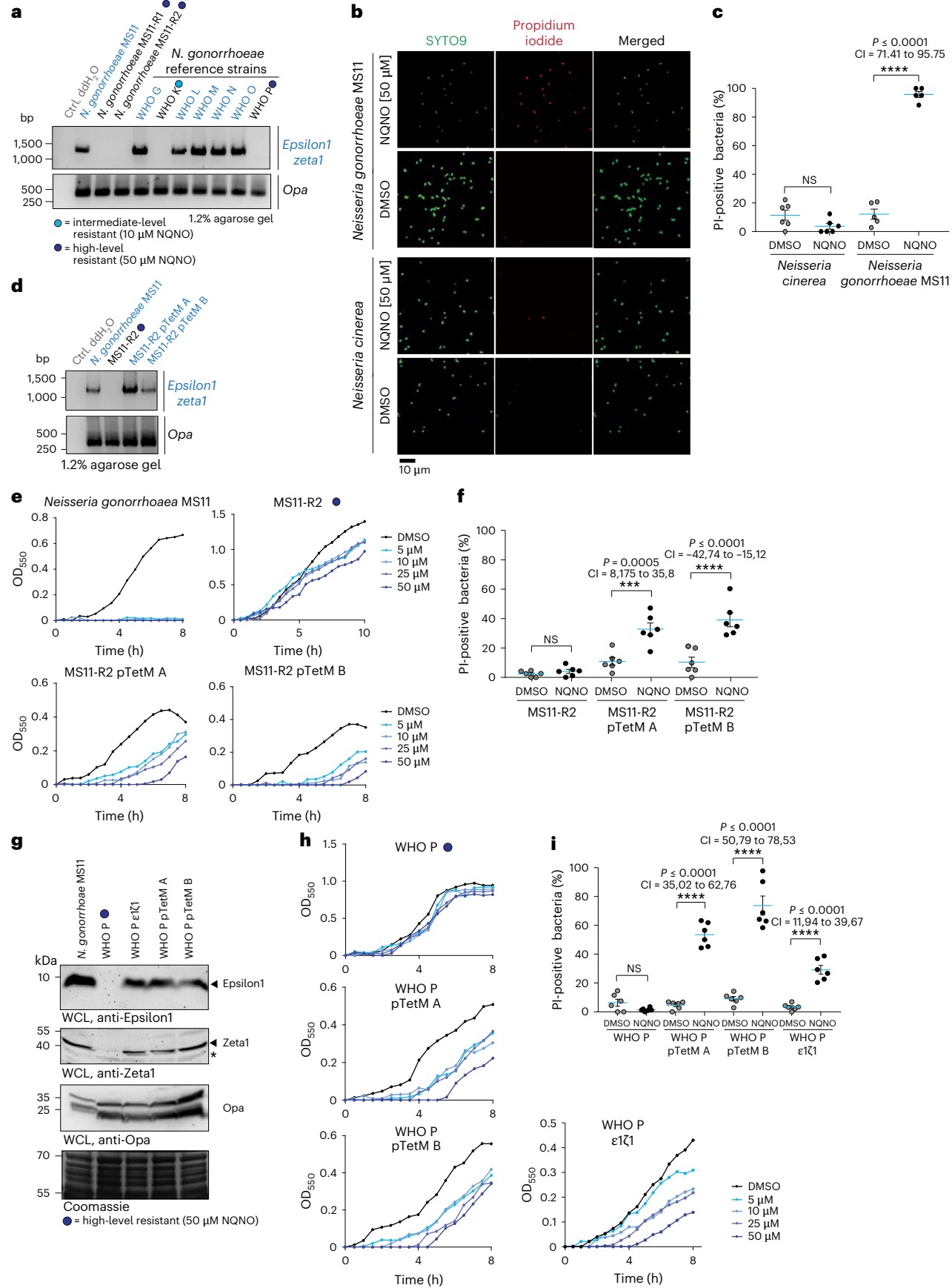

**Fig. 3 | Growth inhibition by NQNO requires the presence of the zeta toxin.**
**a**, Presence of the *epsilon1/zeta1* operon in *N. gonorrhoeae* strains was determined by PCR. As control, PCR for *opa* genes was conducted with the same DNA samples. Blue font marks strains harbouring the *epsilon1/zeta1* locus, blue dots mark high and intermediate NQNO-resistant strains as indicated. **b**, *N. gonorrhoeae* MS11 and *N. cinerea* were treated with 50 μM NQNO or DMSO for 30 min and stained with SYTO9 (all bacteria) and propidium iodide (bacteria with disintegrated cell wall). **c**, Quantification of **b** showing the percentage of propidium iodide-positive bacteria (% PI-positive bacteria) of the total SYTO9-stained bacterial cells. Mean ± s.e.m. (*n* = 6 microscope fields). Significance was analysed using ordinary-pairing one-way ANOVA with Tukey's multiple comparisons test of the mean; $^{NS}P > 0.10$, ***$P < 0.001$ ****$P < 0.0001$; 95% CI and exact two-tailed *P* values also shown. **d**, pTetM plasmid was conjugated in MS11-R2 and two independent conjugants were selected. Successful conjugation of pTetM was verified by PCR of the *epsilon1/zeta1* operon or *opa* genes as in **a**. **e**, Growth of strains MS11, MS11-R2 (NQNO-resistant) and pTetM-positive conjugants MS11-R2 pTetM A and MS11-R2 pTetM B in the presence of 5, 10, 25 or 50 μM NQNO. Control cultures were grown in 1% DMSO. Growth was monitored

using OD$_{550}$ readings every 30 min. Data represent 3 independent replicates. **f**, Strains as in **e** were treated with 50 μM NQNO or solvent control (DMSO) for 30 min and stained, enumerated and statistically analysed as in **c**. **g**, The pTetM plasmid was conjugated into WHO P and two independent clones (WHO P pTetM A and WHO P pTetM B) were selected. WHO P was also transformed with a construct encoding *epsilon1/zeta1* and the ErmC resistance (WHO P ε1ζ1). Expression of ε1 and ζ1 in MS11, WHO P, WHO P ε1ζ1 and WHO P pTetM conjugants was verified by western blotting of whole-cell lysates (WCL) with antibodies against ε1 (top panel) or ζ1 (second panel). The asterisk indicates a non-specific band reacting with the polyclonal anti-ζ1 antiserum. Western blotting of the same lysates with a monoclonal antibody against Opa protein (third panel) and Coomassie staining of the membrane (bottom panel) served as loading controls. Blue dot indicates NQNO-resistant strains. **h**, Growth of strains from **g** in the presence of 5, 10, 25 or 50 μM NQNO. Control cultures were grown in 1% DMSO. OD$_{550}$ was monitored every 30 min. Data represent 3 independent replicates. **i**, Strains as in **g** were treated with 50 μM NQNO or solvent control (DMSO) for 30 min and stained, enumerated and statistically analysed as in **c**.

expressed at equal levels (Fig. 3g). Most importantly, ζ1/ε1-expressing derivatives of WHO P became NQNO sensitive, while the parent strain was fully resistant up to 50 μM (Fig. 3h). Similar to the introduction of pTetM into strain MS11-R2, the presence of the Zeta1–Epsilon1 TA pair resulted in increased membrane permeability in WHO P upon NQNO exposure (Fig. 3i and Extended Data Fig. 8e). Our genetic data link the NQNO susceptibility of gonococci to the presence of the *zeta1/epsilon1* operon. Furthermore, the observed membrane permeability phenotype is in line with the known activity of the Zeta toxin. Therefore, we hypothesized that NQNO promotes Zeta1 activity to compromise cell wall synthesis and to stall gonococcal growth.

### NQNO triggers degradation of the ε1 antitoxin

To monitor the action of NQNO on the ζ/ε toxin–antitoxin pair, we first confirmed the expression of ζ/ε proteins in the different NQNO-sensitive strains (Fig. 4a). Upon treatment of *N. gonorrhoeae* MS11 with NQNO, levels of the Epsilon antitoxin decreased within the first 60 min, while the amount of Zeta toxin remained unchanged (Fig. 4b). In line with the idea that the Epsilon antitoxin is destabilized by reactive oxygen species produced in NQNO-treated gonococci, the Epsilon antitoxin expressed in *E. coli* was rapidly degraded in response to oxidative stress, but not upon NQNO treatment (Fig. 4c,d). The differences in NQNO-induced effects on growth or Epsilon stability between gonococci and *E. coli* (Fig. 4d,e) might be explained by different permeability of their outer membranes. However, NQNO did not affect the stability of purified recombinant Epsilon protein (Fig. 4f), further supporting the idea that NQNO does not act directly on Epsilon stability, but rather acts

indirectly via NQNO-induced oxidative stress in susceptible bacteria such as *N. gonorrhoeae*.

Importantly, NQNO-triggered degradation of the Epsilon antitoxin was observed in several gonococcal strains including the multidrug-resistant WHO G and WHO N strains, as well as the MS11-R2 pTetM A conjugant (Fig. 4g). In contrast to the NQNO-induced disappearance of the Epsilon antitoxin in *N. gonorrhoae* MS11, treatment of the same strain with ciprofloxacin did not affect levels of the Epsilon antitoxin relative to the Zeta toxin (Fig. 4h). This finding again highlights the mechanistic differences between AQs and fluoroquinolones, but also indicates that Epsilon degradation is a downstream consequence of NQNO action on the electron transport chain. Indeed, specific interruption of ubiquinon-mediated electron delivery to cytochrome b1 by Antimycin A also resulted in rapid Epsilon antitoxin depletion, while the Zeta toxin was unaffected (Fig. 4h). Together, these results suggest that NQNO- or Antimycin A-induced oxidative stress impinges on the labile gonococcal Epsilon1 antitoxin. By unleashing the endogenous Zeta1 toxin from inhibition by its cognate antitoxin, the enzymatic activity of Zeta1 leads to disruption of cell wall synthesis and integrity, thereby providing a mechanistic explanation for the NQNO-induced phenotype and the selective toxicity of this compound to gonococci.

### NQNO blocks colonization of the female genital tract

While *N. gonorrhoeae* is a strictly human-specific bacterial pathogen, the infection of the female genital tract can be modelled in oestrogen-treated mice[36–38]. In addition, host-specific aspects of the interaction of gono-cocci with the intact vaginal epithelium can be observed in mice

**Fig. 4 | NQNO triggers degradation of the Epsilon antitoxin. a**, Whole-cell lysates prepared from indicated *N. gonorrhoeae* strains were probed with polyclonal antibodies against ε1 (top) or against ζ1 (bottom). Strains resistant to NQNO are indicated with a blue dot; 2.5 ng of recombinant His-tagged ε1 was loaded as a positive control. The asterisk indicates a non-specific band reacting with the anti-ζ1 antiserum. **b**, Left: *N. gonorrhoeae* MS11 was treated with 50 μM NQNO or DMSO for the indicated time and lysates were probed as in **a**. Blot representative of 3 independent replicates. The asterisk indicates a non-specific band reacting with the polyclonal anti-ζ1 antiserum. Right: graph showing the ratio of ε1 vs ζ1 protein levels. Bars represent mean ± s.e.m. (*n* = 3 biological replicates); *$P < 0.05$, **$P < 0.01$; ordinary-pairing one-way ANOVA with Dunnett's multiple comparisons test; 95% CI and exact two-tailed *P* values also shown. **c**, *E. coli* expressing ε1 was treated for 45 or 60 min with 1 mM or 5 mM H$_2$O$_2$ or left untreated. Lysates were probed with rabbit polyclonal antibodies raised against ε1 (top). *E. coli* lysates taken before (BI) and after (AI) induction of ε1 expression by IPTG were used as controls. The blots were reprobed with a monoclonal antibody against GAPDH (bottom). Blot representative of 3 independent replicates. Graph at the bottom shows the ratio of ε1 protein levels

vs GAPDH. Bars represent mean ± s.e.m. (*n* = 3 biological replicates). Statistics as in **b**. **d**, *E. coli* expressing ε1 was treated for 45 or 60 min with 25 μM or 50 μM NQNO, 1% DMSO or left untreated. Lysates were probed as in **c**. Bars represent means ± s.e.m. (*n* = 4 or 3 biological replicates); statistics as in **b**. **e**, Growth curves of *E. coli* incubated with 25 or 50 μM NQNO; control cultures were grown in the presence of 1% DMSO. Growth was monitored using OD$_{600}$ readings every 30 min. Data representative for 3 independent replicates. **f**, Purified ε1 protein was incubated for the indicated times with solvent (1% DMSO), 50 μM NQNO or 1 mM H$_2$O$_2$ and samples were analysed for the presence of ε1 with an anti-His-tag antibody. Data representative of 3 independent replicates. **g**, *N. gonorrhoeae* strains WHO G, WHO N and MS11-R2 pTetM A were treated for 0 or 60 min with 50 μM NQNO (N) or with DMSO (D) and lysates were probed as in **a**. Graph at the bottom shows the ratio of ε1 vs ζ1 protein levels. Bars represent means ± s.e.m. (*n* = 5 biological replicates); statistics as in **b**. **h**, *N. gonorrhoeae* MS11 was treated with 50 μM NQNO, 36 μM Antimycin A (AA), 30 μM ciprofloxacin (Cipro) or with DMSO for 1 h and lysates were probed as in **a**. Bars represent means ± s.e.m. (*n* = 3 biological replicates); statistics as in **b**.

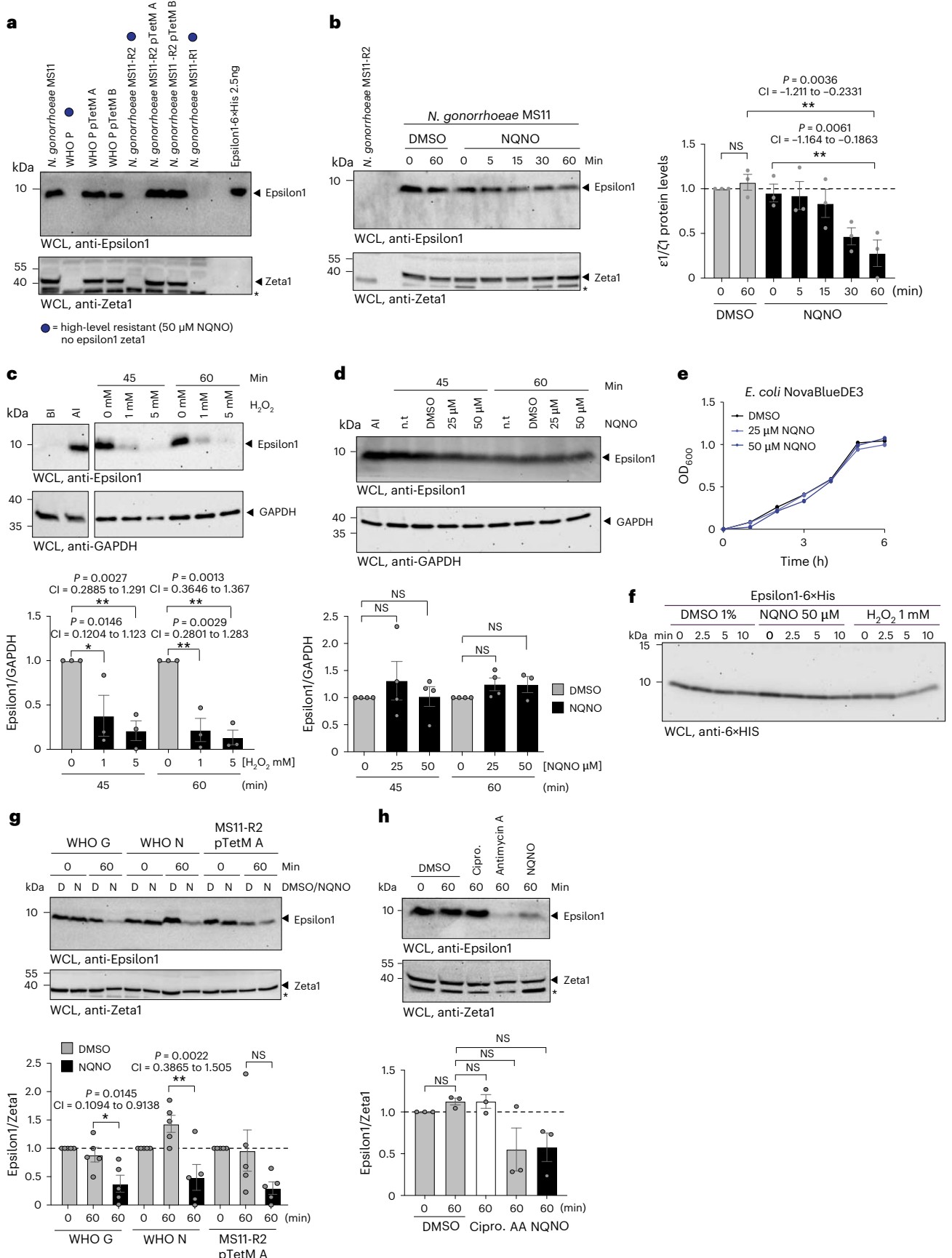

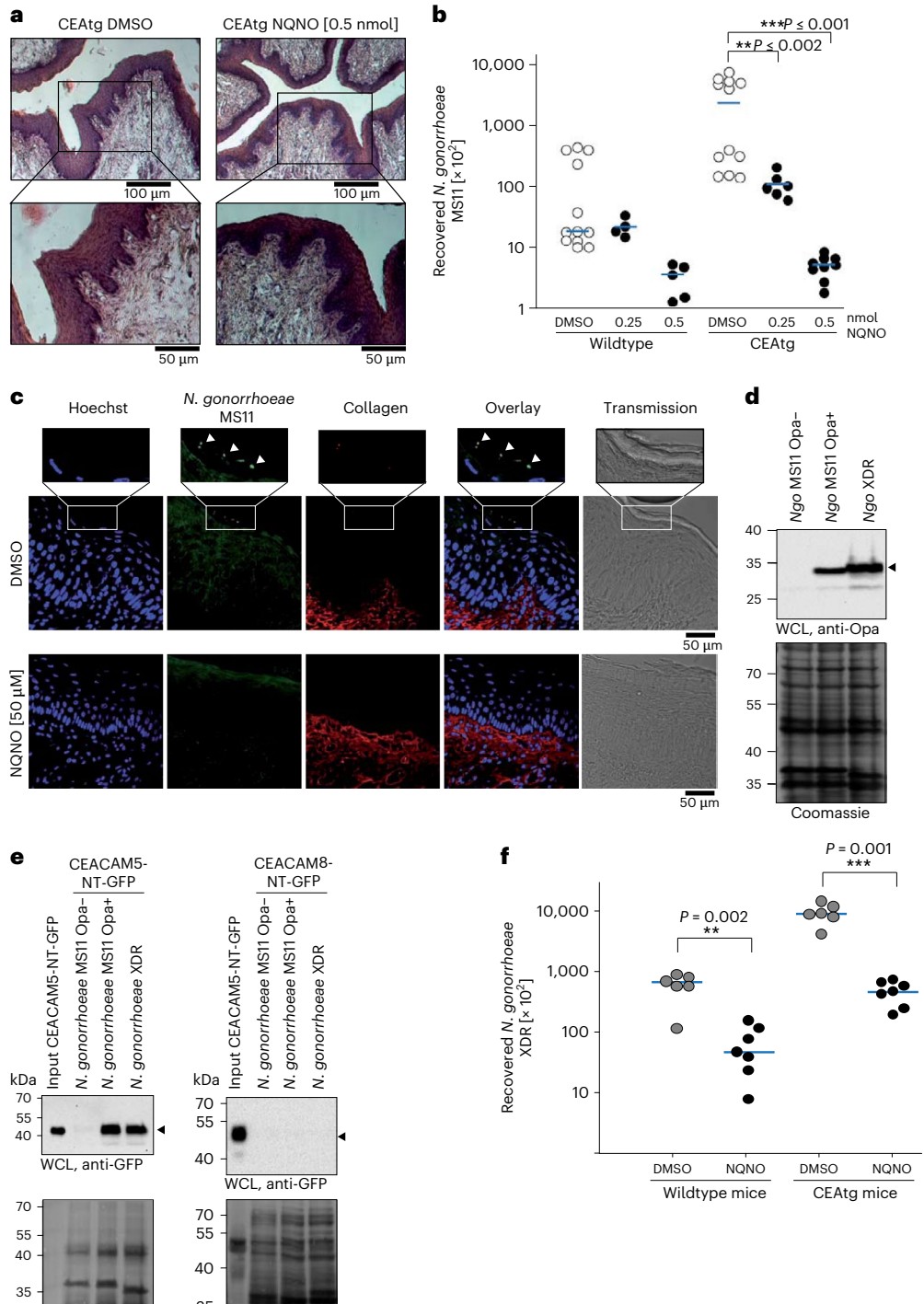

**Fig. 5 | NQNO prevents gonococcal colonization in vivo. a**, Haematoxylin and eosin (H&E) staining of tissue sections derived from murine upper vaginal tract of CEAtg mice 16 h after application of NQNO or solvent. Top panel shows ×10 magnification, bottom panel ×20. Data are representative of 3 independent biological replicates. **b**, Wild-type or CEAtg female mice were infected with Opa$_{CEA}$-expressing *N. gonorrhoeae* MS11 for 1 h before being treated with 0.25 nmol or 0.5 nmol NQNO in 10 μl DMSO (prophylactic treatment). Controls received 10 μl DMSO. After 24 h, bacteria were re-isolated and plated. Data points show bacterial colonies isolated from individual mice and horizontal blue bars indicate median (*n* = 4–12). Differences between groups were determined using unpaired Mann–Whitney rank-sum test; **\*P* < 0.01, \*\*\*P* < 0.001. **c**, CEAtg female mice were infected as in **b** before treatment with DMSO or 0.5 nmol NQNO. One day later, genital tracts were excised, fixed and frozen. Cryosections were co-stained against *N. gonorrhoeae* (green) and against collagen (red). Cell nuclei were visualized by Hoechst staining (blue). Magnified images (top row) show numerous gonococci (white arrowheads) associated with the mucosal surface of the vaginal epithelium of DMSO-treated mice. No gonococci were detected in mice treated with 0.5 nmol NQNO (bottom panels). **d**, Opa expression by *N. gonorrhoeae* XDR was verified by western blotting. Non-opaque (MS11 Opa-) and Opa$_{CEA}$-expressing (MS11 Opa+) gonococci served as controls. **e**, Soluble CEA-NT-GFP or CEACAM8-NT-GFP (input) were incubated with bacteria from **d**. After washing, CEACAM-binding to bacteria was analysed by western blotting using α-GFP antibodies (top panels). Equivalent amounts of bacteria in the different samples were verified by Coomassie staining (bottom panels). **f**, Wild-type or CEAtg female mice were infected with Opa$_{CEA}$-expressing *N. gonorrhoeae* XDR and received prophylactic treatment with NQNO as in **a**. Data points show bacterial colonies isolated from individual mice and horizontal bars indicate median (*n* = 6–7). Statistics as in **b**.

expressing human CEA, a receptor for gonococcal Opa$_{CEA}$ adhesins[39,40]. Therefore, we tested whether the topical application of NQNO to the genital tract of female mice can prohibit infection. In this way, NQNO could serve as a potential post-exposure prophylaxis, a form of treatment advocated for high risk settings[41,42]. Importantly, mice receiving 0.5 nmol NQNO in 10 µl of DMSO or receiving 10 µl DMSO alone did not show any signs of discomfort, and the local application did not lead to erosion of the stratified vaginal epithelium or to the induction of inflammatory responses (Fig. 5a). Next, we infected CEA-transgenic (CEAtg) female mice and littermate wild-type controls with Opa$_{CEA}$-expressing *N. gonorrhoeae* MS11. At 1 h after infection, the mice were either treated with 10 µl of the solvent control (DMSO), or with 0.25 nmol or 0.5 nmol NQNO dissolved in 10 µl DMSO ('prophylactic treatment'). As expected, gonococci could be re-isolated from wild-type mice, and 50–100-fold more bacteria could be re-isolated from CEA-expressing mice when the animals were treated with DMSO (Fig. 5b). In contrast, the number of recovered gonococci was severely reduced when mice were treated with NQNO (Fig. 5b). Immunohistological analysis of tissue sections from the upper vaginal tract confirmed the presence of gonococci attached to the apical surface of epithelial cells in DMSO-treated CEAtg mice (Fig. 5c). In contrast, treatment with 0.5 nmol NQNO diminished gonococci below the level that could be detected by microscopy (Fig. 5c). To investigate whether growth inhibition by NQNO is sufficient to prevent colonization by extended-drug-resistant gonococci, we first selected an Opa protein-expressing and CEA-binding variant of a cefixime- and ciprofloxacin-resistant isolate from Austria (Ngo XDR[30]) (Fig. 5d,e)[43–45]. Infection of female mice with Ngo XDR allowed recovery of gonococci 1 day after infection with elevated numbers of gonococci isolated from CEAtg mice (Fig. 5f). Importantly, when mice were treated topically with 0.5 nmol NQNO, the numbers of recovered Ngo XDR from either wild-type or CEAtg mice were significantly reduced (Fig. 5f). Together, these results demonstrate that NQNO can inhibit gonococcal infection in vivo, including infection by a clinical isolate with extended-drug resistance.

## 3-methyl-NQNO exhibits selective gonococcicidal activity

To get insight into the structure–activity relationship of NQNO as a prerequisite for further improving its efficacy, we synthesized a series of NQNO and NO derivatives with modifications of the quinolone core (Extended Data Fig. 8). When tested for their potential to inhibit gonococcal growth, all modifications at positions 5, 6, 7 and 8 of the aromatic ring system with methyl or halogen groups resulted in severe or complete loss of activity against gonococci (Extended Data Fig. 8). Strikingly, introduction of a methyl group at the 3-position yielded 3-methyl-NQNO (3Me-NQNO) (Fig. 6a), which showed increased potency against *N. gonorrhoeae* (Fig. 6b and Extended Data Fig. 8). Indeed, 3Me-NQNO concentrations as low as 0.5 µM led to complete abrogation of gonococcal growth, while growth of *N. cinerea* was not compromised by 3Me-NQNO over a wide range of concentrations (Fig. 6b and Extended Data Fig. 9a). Moreover, 3Me-NQNO at concentrations up to 50 µM did not show any detrimental effect on commensal lactobacilli from the vaginal tract (Extended Data Fig. 9b). Similar to the situation with NQNO, 3Me-NQNO caused a rapid drop in NADH content of gonococci, but not in *N. cinerea* (Fig. 6c). When added to exponentially growing *N. gonorrhoeae* cultures, a concentration as low as 100 nM 3Me-NQNO stalled gonococcal growth (Fig. 6d). Already at a concentration of 2.5 µM, 3Me-NQNO was bactericidal and induced rapid killing of the pathogens, while *N. cinerea* was not affected (Fig. 6e). Given its potency against gonococci, it is important to note that 3Me-NQNO at concentrations up to 50 µM did not interfere with the metabolic activity of human cervical carcinoma cells (Extended Data Fig. 10a,b). Moreover, hVECs were not compromised in their growth upon application of 10 µM 3Me-NQNO over 2 days (Extended Data Fig. 10c). Motivated by the increased efficacy, we wondered whether 3Me-NQNO is not only able to work in a prophylactic manner to prevent, but is also able to treat an established gonococcal infection. To this end, we first infected estradiol-treated, female CEA-transgenic mice with Opa$_{CEA}$-expressing *N. gonorrhoeae* MS11 for 24 h. After gonococcal infection had been established, the animals were treated by topical application of either 0.5 nmol 3Me-NQNO in 10 µl DMSO ('therapeutic treatment') or with 10 µl of the solvent DMSO. Another 24 h later, that is, 48 h after the initial infection, gonococci were re-isolated from the vaginal tract. Importantly, topical application of 3Me-NQNO strongly diminished the recovery of gonococci compared with DMSO-treated controls, and reduced the pathogen burden in the genital tract (Fig. 6f,g). In three out of seven female mice, the therapeutic treatment with 3Me-NQNO completely eradicated the pathogen from the genital tract within 24 h (Fig. 6f). As resistance development is a major impediment in fighting gonorrhoea with antibiotics, we tested whether continued subMIC (sub-minimum inhibitory concentration) exposure will result in 3Me-NQNO resistance. Strikingly, prolonged exposure of *N. gonorrhoeae* MS11 to sublethal concentrations of 3Me-NQNO (100 nM, 14 iterations over the course of 2 weeks) did not lead to the emergence of resistant clones (Fig. 6h). Together, these findings demonstrate that the NQNO derivative 3Me-NQNO has exceptional potency and selectivity, opening up exciting opportunities for combating *N. gonorrhoeae*.

## Discussion

Secondary metabolites released by *P. aeruginosa* are known to harm bacteria and eukaryotic cells. In particular, AQNOs can inhibit the

**Fig. 6 | 3-methyl-NQNO eradicates gonococci from the female genital tract.**
**a**, Structure of 3-methyl-NQNO (3Me-NQNO). **b**, Growth of *N. gonorrhoeae* MS11 and *N. cinerea* in the presence of the indicated concentrations of 3Me-NQNO (0.1–50 µM). Growth was determined using OD$_{550}$ readings every 30 min over 8 h and quantified by the area under the growth curve. Colour gradient indicates growth of compound-treated cultures compared to 1% DMSO control and ranges from no growth (90–100% inhibition, white) to full growth (0–10% inhibition, dark green). Shown is a representative experiment conducted in technical triplicates. **c**, *N. gonorrhoeae* and *N. cinerea* were incubated with DMSO or 3Me-NQNO for 120 min and NADH levels were determined. NADH levels were normalized to the DMSO control. Bars indicate mean ± s.e.m.; $n = 3$ biological replicates, with each experiment conducted in technical triplicates. Significance was evaluated using two-way ANOVA with Dunnett's correction; 95% CI and exact two-tailed $P$ values are indicated; *$P < 0.05$, **$P < 0.01$, ***$P < 0.001$, ****$P < 0.0001$. **d**, Growth of *N. gonorrhoeae* MS11 and *N. cinerea*, with indicated amounts of 3Me-NQNO added during exponential growth (arrow). **e**, Bacteria were treated as in **d** and the number of viable bacteria was determined by dilution plating and colony count. The time given on the $x$ axis refers to the time points after addition of the indicated amounts of 3Me-NQNO or DMSO to log-phase cultures. Bars represent mean ± s.e.m. ($n = 3$ biological replicates). **f**, CEAtg mice were infected with Opa$_{CEA}$-expressing *N. gonorrhoeae* MS11 for 24 h before receiving 0.5 nmol 3Me-NQNO in 10 µl DMSO (therapeutic treatment). Controls received 10 µl solvent only (DMSO). At 2 days after infection, bacteria colonizing the genital tract were re-isolated using a cotton swab and plated. Data points show bacterial colonies isolated from individual mice with mean ± s.e.m. ($n = 7$). Differences between groups were determined using unpaired Mann–Whitney rank-sum test with 95% confidence level; ***$P < 0.001$. **g**, Cryosections of genital tracts of CEAtg female mice infected and treated as in **f** were stained against *N. gonorrhoeae* (red) and cell nuclei were visualized by Hoechst staining (blue). In DMSO-treated mice, numerous gonococci (arrowheads) are associated with the mucosal surface of the vaginal epithelium, while no gonococci were detected in mice treated with 3Me-NQNO. Scale bars, 10 µm. **h**, Selection for 3Me-NQNO resistance over the course of 14 days. Gonococci conditioned in 3Me-NQNO-containing (0.1 µM) medium for the indicated number of days were used to inoculate medium containing 0.2, 0.5, 1, 2.5 or 5 µM 3Me-NQNO and growth was monitored using OD$_{550}$ readings. Open bars reflect the growth of conditioned bacteria grown in 1% DMSO-containing medium. Shown is a representative experiment performed twice with similar results.

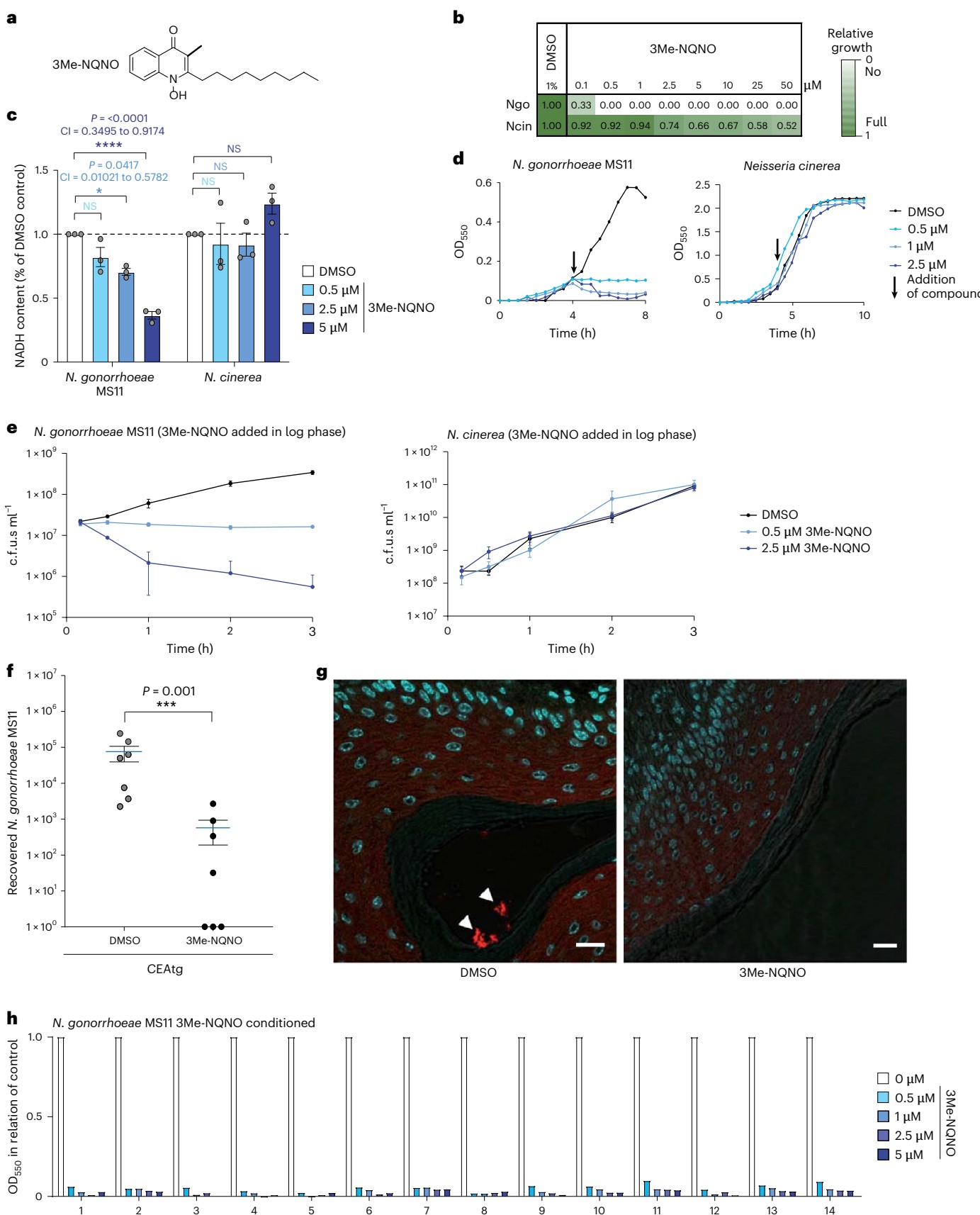

respiratory chain complex III by interfering with ubiquinone reduction at the essential cytochrome b subunit (*cyt b*)[25]. Despite their relatively conserved nature, respiratory chain complexes are blocked by commercial fungicides such as azoystrobin or stigmatellin[46], indicating that they are tangible targets for interfering with microbial growth. Here we describe an unexpected vulnerability of the *N. gonorrhoeae* respiratory chain to the *P. aeruginosa*-derived natural product NQNO. Low micromolar concentrations of NQNO and nanomolar concentrations of 3-methyl-modified NQNO completely blocked growth of *N. gonorrhoeae*. In contrast, a diverse set of related commensal *Neisseriae* was not affected. The closest relative of gonococci, *Neisseria meningitidis*, was not evaluated in this study. Given the high degree of homology between meningococci and gonococci, the initial perturbation of the primary metabolism by NQNO might occur in both species.

Clearly, NQNO treatment of gonococci resulted in a rapid drop in ATP and NADH levels, and induced reactive oxygen species. These responses are in line with the idea that the respiratory chain is the primary target of NQNO. Indeed, similar alterations can be observed upon treatment of gonococci with Antimycin A, a known inhibitor of complex III. However, the perturbation of gonococcal energy metabolism by NQNO is not sufficient to inhibit gonococcal growth as examplified by the NQNO-resistant isolates derived from strain MS11. The peculiar sensitivity of gonococci to NQNO-mediated metabolic interference is due to the presence and activation of the zeta toxin, the active component of a toxin–antitoxin (TA) system encoded on the large conjugative pTetM plasmid[32]. Indeed, transformation of the NQNO-resistant strain WHO P with pTetM or with the *zeta1/epsilon1* operon sensitizes WHO P to NQNO. While TA systems comprising two protein components (type II TA systems) are widespread in various microbes[47–49], the ζ/ε TA system is not present in other Gram-negative bacteria, and the closest homologues of Ngo ζ/ε are found in Gram-positive bacteria[34,50]. Intriguingly, the phenotype of NQNO-treated gonococci is in agreement with the known action of the Ngo ζ toxin on bacterial cell wall dynamics[35]. Moreover, metabolic perturbations such as amino acid starvation or ROS generation are known triggers of toxin activation in numerous TA systems[51,52]. A common mechanism in this regard is the accelerated proteolytic turnover of the antitoxin by stress-induced proteases such as Lon or ClpP, which are involved in degradation of the *Streptococcus pyogenes* Epsilon antitoxin[53,54]. In line with these findings, we also observe that the metabolic perturbations caused by NQNO in *N. gonorrhoeae* impact the stability of the Ngo ε antitoxin and lead to ζ toxin release. Accordingly, NQNO sets off a chain of events culminating in the activation of an endogenous toxin, which affects the bacterial cell wall, an established target of numerous microbicidal compounds.

The involvement of an endogenous TA system explains the selectivity of NQNO for gonococci and provides NQNO with a mode of action that differs from currently used antibiotics. This important feature of NQNO is underscored by the fact that the chloramphenicol–erythromycin–tetracyclin-resistant MS11 laboratory strain, the cefixime–ceftriaxone–spectinomycin–ciprofloxacin-resistant clinical isolates from Slovenia[29] and Austria[30], as well as the high-azithromycin-resistant isolate NCTC 13799, were equally susceptible to NQNO. These desirable properties of NQNO are slightly tarnished by the occurrence of naturally NQNO-resistant strains such as WHO P, which lack the pTetM plasmid. While the occurrence of WHO P-like azithromycin-resistant strains has increased in Europe in recent years[55], pTetM-positive and therefore NQNO-sensitive strains are highly prevalent in other parts of the world[56–58]. Moreover, given the recent emergence of untreatable gonococcal infections[59,60], it appears timely to evaluate the therapeutic applicability of NQNO. Indeed, derivatization of NQNO at the 3-position of the quinolone ring, as found in 2-alkyl-4-quinolone *N*-oxides of *Burkholderia* species[14], can further boost the efficacy against gonococci without losing selectivity. Therefore, this class of compounds might serve as a starting point to derive pathogen-selective compounds, sparing beneficial commensal bacteria such as lactobacilli from antibiotic action.

Topical application of antibiotic-containing gels is the current standard therapy for bacterial vaginosis[61,62]. Whether the local administration of NQNO-containing gel preparations, as employed in the murine infection model, can control gonococcal infections in humans, needs to be explored in the future. However, with a prophylactic vaccine against *N. gonorrhoeae* currently not in sight[63], and with increased prevalence of multidrug-resistant *N. gonorrhoeae* potentially leading to fewer and fewer treatment options[59], these natural products hold promise as potent, selective anti-gonococcal agents.

## Methods

### Growth of bacterial strains

*Neisseria* strains were cultured in PPM medium (15 g l$^{-1}$ proteose peptone, 1 g l$^{-1}$ soluble starch, 5 g l$^{-1}$ NaCl, 4 g l$^{-1}$ KH$_2$PO$_4$, 1 g l$^{-1}$ K$_2$HPO$_4$, pH 7.5) for liquid culture and on GC plates (BD Difco, GC Medium Base), both supplemented with 1% IsoVitale vitamin mix (100 g l$^{-1}$ glucose, 10 g l$^{-1}$ glutamine, 26 g l$^{-1}$ L-cysteine, 100 mg l$^{-1}$ carboxylase, 250 mg l$^{-1}$ NAD, 500 µl l$^{-1}$ Fe(NO$_3$)$_3$, 150 mg l$^{-1}$ arginine, 3 mg l$^{-1}$ thiamine-HCl, 10 mg l$^{-1}$ vitamin B12, 13 mg l$^{-1}$ *p*-amino benzoic acid, 1.1 g l$^{-1}$ L-cystine, 1 g l$^{-1}$ adenine, 500 mg l$^{-1}$ uracil, 30 mg l$^{-1}$ guanine). Strains were either cultivated at 37 °C and 220 rpm (liquid medium) or at 37 °C and 5% CO$_2$ (solid medium). Opa protein expression by gonococcal strains and Opa-mediated binding to human CEACAMs was determined essentially as previously decribed[43].

*Escherichia coli* strains, *P. aeruginosa* and *K. pneumoniae* were cultured in LB medium (10 g l$^{-1}$ tryptone, 5 g l$^{-1}$ yeast extract, 5 g l$^{-1}$ NaCl, 15 g l$^{-1}$ agar for plates; pH 7.0). *P. aeruginosa* PAO1 mutants were obtained from Colin Manoil (University of Washington, Genome Sciences, Seattle, WA).

*Lactobacillus* strains used in this study were grown on Columbia blood agar (42.5 g l$^{-1}$ Columbia agar base, 5% defibrinated horse blood) for 24–48 h at 37 °C with 5% CO$_2$ or in prereduced MRS liquid medium (10 g l$^{-1}$ casein peptone, 10 g l$^{-1}$ meat extract, 5 g l$^{-1}$ yeast extract, 20 g l$^{-1}$ glucose, 1 g l$^{-1}$ Tween 80, 1 mg l$^{-1}$ resazurin, 2 g l$^{-1}$ K$_2$HPO$_4$, 5 g l$^{-1}$ Na-acetate, 2 g l$^{-1}$ (NH$_4$)$_3$ citrate, 0.2 g l$^{-1}$ MgSO$_4$ × 7H$_2$O, 0.05 g l$^{-1}$ MnSO$_4$ × H$_2$O, pH 6.2–6.5; 0.5 g l$^{-1}$ cysteine-HCl was added after the medium cooled under a CO$_2$ atmosphere) at 37 °C without shaking. *Limosilactobacillus vaginalis* was cultured in the same liquid medium with shaking at 150 rpm.

The origins of all strains used in this study are listed in Supplementary Tables 3 and 4.

### Cross-streak assay

Bacteria were grown on agar plates and incubated overnight at 37 °C and 5% CO$_2$. Bacteria were collected and diluted in PBS to an OD$_{550}$ of 0.5 (*Neisseria*) or OD$_{600}$ of 0.5 (*Pseudomonas*, *Klebsiella*, *Escherichia*). Filter strips soaked with the indicated bacterial suspensions were stamped on GC or LB plates in a cross pattern. Plates were incubated overnight at 37 °C and 5% CO$_2$. Images of plates were taken using the ChemiDoc Touch Imaging System from BioRad, and inhibition zone length was determined with Inkscape 0.91.

### Co-culturing of *N. gonorrhoeae* with *P. aeruginosa* culture supernatants

Of an overnight culture of *P. aeruginosa* PAO1, 120 µl was used to inoculate 8 ml LB medium. The culture was incubated for 24 h at 37 °C at 150 rpm to allow maximum production of AQNOs[14]. Then, bacteria were pelleted and the supernatant containing the *Pseudomonas*-conditioned medium was sterile filtered and saved. *N. gonorrhoeae* MS11 was grown on GC plates overnight at 37 °C and 5% CO$_2$. Bacteria (4 × 10$^6$) from a 4-h preculture (in PPM, 37 °C, 220 rpm) were centrifuged on poly-L-lysine-coated coverslips (3,300*g* × 15 min) and incubated with a 1:1 mixture of PPM and *Pseudomonas*-conditioned medium for 3 h. Samples were fixed and processed for scanning electron microcopy.

## Scanning electron microscopy

Bacteria were grown on GC plates and incubated overnight at 37 °C and 5% $CO_2$. Bacteria ($4 \times 10^6$) from a 4-h preculture (in PPM, 37 °C, 220 rpm) were centrifuged on poly-L-lysine-coated coverslips ($3,300g \times 15$ min). NQNO was added at the respective concentration, and bacteria were incubated for 3 h at 37 °C and 5% $CO_2$. Bacteria were fixed with $2 \times 300$ µl fixans (3% formaldehyde, 2% glutardialdehyde, 0.09 M sucrose, 0.01 M $CaCl_2$, 0.01 M $MgCl_2$ in 0.1 M HEPES, pH 7.2) for 5 and 25 min, respectively. Samples were washed twice with 1 ml 0.1 M HEPES. Dehydration was performed in a graded series of ethanol (30%, 50%, 70%, 80%, 90%, 96%, $3 \times 100\%$) for at least 10 min each. Dehydrated samples were critical-point dried with liquid $CO_2$ using a Baltec CPD 030 (Baltec). Finally, samples were mounted on stubs with silver-coating polish (drying overnight at room temperature), sputtered with 6 nm platinum using Quorum Q 150R ES (Quorum Technologies) in a low-pressure argon atmosphere and imaged using a Zeiss Auriga 40 Crossbeam FIB-FESEM.

## Synthesis of AQNOs

The saturated AQs were synthesized via Conrad–Limpach cyclization and the unsaturated *trans*-$\Delta^1$-NQ by Camps cyclization as described previously[19]. The *N*-oxides were obtained by locking the 4-hydroxyquinolins as ethyl carbonates followed by *N*-oxidation with mCPBA (m-chloroperoxybenzoic acid) and deprotection. A list of names, abbreviations and structures of all synthetic compounds is given in Supplementary Table 5. The details of compound synthesis together with the characterization and confirmation by NMR spectroscopy and high-resolution mass spectrometry are described in Supplementary Data File 1.

## Bacterial growth assays in the presence of AQs, AQNOs or antibiotics

Bacteria were grown on agar plates as described above. Before exposure, bacteria were precultured in 5 ml liquid medium for at least 2 h, then collected and resuspended in PBS. Optical density at 550 nm ($OD_{550}$; *N. gonorrhoeae* and commensal *Neisseriae*) or 600 nm ($OD_{600}$; all other microorganisms) was determined and $4 \times 10^7$ bacteria were inoculated in 5 ml of the respective growth medium. AQ or AQNOs dissolved at 5 mM in DMSO were added to the indicated final concentrations. DMSO was adjusted to 1% (v/v) final concentration in all samples including controls. Samples were incubated at 37 °C for 8–10 h until control cultures reached stationary phase or until they reached an OD of 2.5. Optical density was determined every 0.5 h. Growth inhibition was quantified by the calculation of the area under the curve (AUC) using Prism5 (GraphPad) with the start OD at $T_0$ subtracted. The calculated AUC of the DMSO control was set at 1. To visualize multiple samples, a colour gradient was used to indicate strong inhibition/no growth (white) to weak inhibition/full growth (dark green).

## Stability of NQNO in solvent and in cell culture medium

Two samples of NQNO were dissolved in deuterated methanol ($CD_3OD$) to reach a final concentration of 25 mM. Samples were kept at 25 °C or 37 °C for 28 days, proton NMR spectra were measured with the same acquisition parameters every week and the spectra compared with the ones from day 0.

Stability of 100 µM NQNO in Dulbecco's modified Eagle's medium (DMEM) supplemented with 10% fetal calf serum at 24 °C and 37 °C was evaluated over a period of 4 days. Samples were taken at the beginning (day 0) and every second day. Samples were analysed by LC–MS and the integrals of the detected NQNO (by using extracted ion chromatograms) compared to the integral of the control (100 µM NQNO in $H_2O$), which was set to 100%.

## ATP measurement

*Neisseria* species were grown in the presence of NQNO or DMSO for 10, 20, 30, 45 or 120 min. Bacteria were collected by centrifugation for 10 min at 4,000 rpm and 4 °C, resuspended in 1 ml PBS and snap frozen. Samples were lysed by sonication for 1 min at maximum intensity on ice. Supernatant was collected by centrifugation for 10 min at 13,500 rpm at 4 °C. ATP assay was performed using 10 µl of supernatant, 100 µl ATP assay buffer (25 mM Tris pH 7.8, 4 mM EGTA, 20 mM $MgSO_4$, 1 mM dithiothreitol, pH 7.8), 20 µM luciferin and 1 µg firefly luciferase (Sigma, L9420; $10 \times 10^{10}$ units $mg^{-1}$ protein) per well. Luminescence was measured using a Thermo Fisher Varioskan Flash spectrophotometer.

## Determination of NADH levels

*Neisseria* species were grown in the presence of the indicated concentrations of NQNO, 3Me-NQNO, Antimycin A or DMSO for 1 h. DMSO concentration was adjusted to 1% in all samples. Resazurin assay was performed using $1 \times 10^6$ bacteria and 0.02 g $l^{-1}$ resazurin sodium (Sigma-Aldrich) in PBS per well. Fluorescence was measured after 2 h of incubation in flat-bottom transparent 96-well plate using excitation at 459 nm and emission at 590 nm in a Varioskan Flash spectrophotometer (Thermo Fisher).

## Detection of ROS

Log-phase bacteria were collected by centrifugation for 10 min at 4,000 rpm at room temperature and washed in oxidative burst buffer (8 g $l^{-1}$ NaCl, 0.2 g $l^{-1}$ KCl, 0.62 g $l^{-1}$ $KH_2PO_4$, 1.14 g $l^{-1}$ $Na_2HPO_4$, 1 g $l^{-1}$ glucose, 50 mg BSA in double-distilled $H_2O$; pH 7.2). Bacteria at $1 \times 10^7$ per well were seeded in a white 96-well plate in 50 µl oxidative burst buffer. NQNO, Antimycin A or DMSO at 2× final concentration were premixed in a separate plate in 50 µl oxidative burst buffer containing 100 µM lucigenin (Santa Cruz, CAS 2315-97-1). The lucigenin solution was added to the bacteria and luminescence was recorded using a Thermo Fisher Varioskan Flash spectrophotometer.

## Selection for NQNO and 3Me-NQNO resistance

To select for NQNO or 3Me-NQNO resistance, *N. gonorrhoeae* MS11 was grown continuously in the presence of sublethal concentrations of NQNO or 3Me-NQNO[64]. To this end, 5 ml PPM medium containing 2.5 µM NQNO or 0.1 µM 3Me-NQNO was inoculated with *N. gonorrhoeae* strain MS11 at an $OD_{550}$ of 0.2, and incubated for 8 h at 37 °C and 220 rpm (conditioned bacteria). After this growth phase, conditioned bacteria were either streaked on GC agar plates without antibiotic or used to inoculate 5 ml PPM containing 0, 5, 10, 25 or 50 µM NQNO or 0.2, 0.5, 1, 2.5 or 5 µM 3Me-NQNO. Agar plates and liquid cultures were incubated at 37 °C and 5% $CO_2$. Growth of the conditioned bacteria in PPM was monitored by reading the $OD_{550}$ after 14 h. The conditioned gonococci grown overnight on GC agar were used for the next round of conditioning with 2.5 µM NQNO or 0.1 µM 3Me-NQNO for 8 h, and then again evaluated for sensitivity to higher concentrations of NQNO or 3Me-NQNO. This procedure was repeated for a total of 10 days (NQNO) or 14 days (3Me-NQNO). Single clones MS11-R1 and MS11-R2 were derived after 8 days of conditioning in 2.5 µM NQNO.

## Genome determination and comparison

Paired-end libraries (250 bp) were prepared from isolated genomic DNA of parent strain MS11-N309 and its NQNO-resistant mutants MS11-R1 and MS11-R2 using the Nextera XT DNA Library Preparation kit (Illumina). The libraries were sequenced on a MiSeq sequencing system (Illumina) using v.2 sequencing chemistry. FastQC (v.0.11.5)[65] was used to analyse the quality of the raw sequencing data. The raw reads were trimmed using Sickle v.1.33 (https://github.com/najoshi/sickle). SPAdes (v.3.10.1)[66] was used for the assembly of the raw reads. The raw FASTq files and draft genome sequences of the three strains MS11_N309 (accession no. SAMN19108268), MS11-R1 (N568) (accession no. SAMN19108269) and MS11-R2 (N569) (accession no. SAMN19108270) were combined in Bioproject number PRJNA728975 and are publicly available from the NCBI GenBank. The accessible links are: *N. gonorrhoeae* MS11-N309: https://www.ncbi.nlm.nih.gov/nuccore/

JAHBBP000000000; *N. gonorrhoeae* MS11-R1: https://www.ncbi.nlm.nih.gov/nuccore/JAHBBO000000000; *N. gonorrhoeae* MS11-R2: https://www.ncbi.nlm.nih.gov/nuccore/JAHBBN000000000. For gene prediction and automatic annotation, we employed Prokka (v.1.12)[67] and the NCBI Prokaryotic Genome Annotation Pipeline (PGAP release 5.2)[68]. For subsequent comparative genomic analysis, we used the PGAP annotation.

## Genome comparison

The quality trimmed reads of the parent strain MS11-N309 and the two NQNO-resistant mutants MS11-R1 and MS11-R12 were aligned to the reference genome of *N. gonorrhoeae* MS11 (accession number NC_022240; https://www.ncbi.nlm.nih.gov/nuccore/NC_022240.1/) with the BWA-MEM algorithm from the Burrows–Wheeler Aligner (BWA) software package (v.0.7.17)[69]. The produced alignment was sorted with SortSam, and PCR duplicates were marked with MarkDuplicates using Picard (v.2.17.3) (http://broadinstitute.github.io/picard/) for downstream analysis.

The Genome Analysis Toolkit's (GATK) (v.3.8.0) RealignerTargetCreator and IndelRealigner tools were used to perform realignment around the Indels[70]. The GATK BaseRecalibrator was used to perform base quality score recalibration. Variant calling was then performed using the GATK HaplotypeCaller with default parameters, except for the ploidy that was set to 1 according to GATK best practices. The sorted single-nucleotide polymorphisms and InDels were then filtered using GATK VariantFiltration with the filtering criteria recommended by GATK. To further remove sequencing bias, only variants with a minimum read depth of 10 were considered. The effects of the variants were then annotated using SnpEff (v.4.3s)[71].

## Cultivation of eukaryotic cells

Immortalized human vaginal epithelial cells (hVECs, line MS74) were obtained from A. J. Schaeffer (Feinberg School of Medicine, Northwestern University, Chicago, IL) and were derived from vaginal tissue of a post-menopausal woman. The cell line was created through immortalization of the cells with human papilloma virus 16, E6 and E7 genes according to ref. 72. hVECs were cultured on gelatin-coated cell culture dishes in DMEM supplemented with 10% fetal calf serum, 1% non-essential amino acids and 1% pyruvate at 37 °C in 5% $CO_2$, and subcultured when 60% confluency was reached. HeLa S3 (DSMZ ACC 161) and ME-180 (ATCC HTB-33) cervical carcinoma cells were cultured in DMEM supplemented with 10% fetal calf serum and passaged every 2–3 days.

## Measurement of metabolic activity of human cells

Plates (96-well) were coated overnight at 4 °C with 0.1% gelatin in PBS before use. hVECs, HeLa cells or ME-180 cells ($2 \times 10^4$) were seeded in 100 μl of their respective growth medium supplemented with the indicated concentrations of NQNO, 3Me-NQNO or DMSO. Cultures were grown for 1 or 2 days at 37 °C and 5% $CO_2$. Next, 10 μl of MTT solution (12 mM in PBS) were added to each well and cells were incubated for an additional 2 h at 37 °C. After removing the MTT-containing growth medium, 100 μl isopropanol was added to each well. The formazan produced by cellular metabolism was allowed to dissolve overnight and $OD_{550}$ was measured using a Varioskan Flash spectrophotometer (Thermo Fisher).

## Detection of *epsilon1 zeta1* genes

Genomic DNA was extracted from *Neisseria* strains. Presence of the toxin–antitoxin system *epsilon1 zeta1* was tested via PCR by using primers *epsilon1* forward (5′-TATCATATGAATAAAGTTGAGCCC-3′) and *zeta1* reverse (5′-TATAAGCTTTTATCTGCTGATGGATTTTTTGGC-3′) at 58 °C annealing temperature and 90 s elongation time. Opa loci were amplified with primers Opa_for (5′-GCAGATTATGCCCGTTACAG-3′) and Opa_rev (5′-GTTTTGAAGCGGGTGTTTTCC-3′) at 55 °C annealing

temperature and 45 s elongation time. PCR fragments were separated in 1.2% agarose gel using 1 kb DNA ladder for size comparison

## Membrane integrity assay

Bacteria were grown overnight at 37 °C and 5% $CO_2$ and used to inoculate PPM medium. After 2 h culture, bacteria at an $OD_{550}$ of 0.04 were collected and washed in 1 ml PBS. Bacteria were incubated with 50 μM NQNO or 1% DMSO as solvent control at 37 °C and 330 rpm for 30 min. Staining was done using 1.67 μM SYTO9 green fluorescent nucleic acid stain (Invitrogen) and 1.4 μM propidium iodide (Sigma-Aldrich). Bacteria were fixed on a coverslip by centrifugation (2,000 *g*, 2 min, room temperature). Samples were analysed using a Leica SP5 confocal microscope by excitation at 600–630 nm for propidium iodide and at 454–510 nm for SYTO9 and acquired in *xyz* mode with a 1,024 × 1,024 pixel format and 100 Hz scanning speed. All images were analysed in ImageJ/Fiji software. The percentage of PI-stained bacterial cells in relation to SYTO9-stained cells was enumerated from independent microscope pictures (*n* = 5 or 6) for each condition.

## Conjugation of pTetM in NQNO-resistant *Neisseria* strains

To transfer pTetM by conjugation, the pTetM-deficient, NQNO-resistant strain *N. gonorrhoeae* MS11-R2 was first made resistant to rifampicin by successive growth on increasing concentrations of the antibiotic. Next, the rifampicin-sensitive donor strain (*N. gonorrhoeae* MS11-F3 pilus, pTetM, N554) and the acceptor strain (*N. gonorrhoeae* MS11-R2 Rif^R) were precultured for 3 h in PPM medium containing 100 μg ml⁻¹ DNase (AppliChem) and 0.9% $NaHCO_3$. Conjugation between the donor strain ($4 \times 10^7$ bacteria) and the acceptor strain ($2 \times 10^7$ bacteria) was conducted using the filter mating technique based on a 0.2-μm-pore-size nitrocellulose membrane. Gonococci placed on the membrane were co-incubated for 24 h on a GC plate before bacteria were resuspended from the membrane in 500 μl PPM medium and plated on a GC plate containing tetracycline (12.5 mg l⁻¹) and rifampicin (0.5 mg l⁻¹) to select for the pTetM conjugated acceptor strain. Two independent clones (MS11-R2 pTetM A and MS11-R2 pTetM B) were propagated.

## Generation of pNEISS epsilon1/zeta1 and transformation in *N. gonorrhoeae*

A gene strand containing multiple copies of the neisserial DNA uptake sequence (DUS), an *opa* promoter and a LIC cloning site flanked on each side by 200 bp sequences of the gonococcal lactoferrin-binding protein A (LbpA) gene (which is truncated in ~50% of gonococci and is not essential for viability or virulence[73,74]) was synthesized by Eurofins Genomics and inserted into the HindIII/SacI sites of pBluescript SK. The erythromycin resistance gene ErmC encoded in plasmid Hermes8 (ref. 75) was amplified with primers ErmC-BamH1-for 5′-AATGGATCCAGGAGGAAAAAATAAAGAGGGTTATAATGAACGAG-3′ and ErmC-XbaI-rev 5′-AATTCTAGACTTACTTATTAAATAATTTATAGCTATTG-3′ and inserted into the corresponding sites of the gene strand to yield pNEISS. The *epsilon1/zeta1* coding sequences were amplified by PCR from pTetM isolated from *N. gonorrhoeae* MS11 using primers Epsilon1-LIC-for 5′-ACTCCTCCCCCGTTAGGATTTTATTATGAATAAAGTTGAGCC-3′ and Zeta1-LIC-rev 5′-CCCCACTAACCCGTTATCTGCTGATGGATTTTTTGG-3′ and inserted via ligation-independent cloning downstream of the *opa* promoter into vector pNEISS.

For transformation, a fragment containing DUS, *opa* promotor, *epsilon1/zeta1*, ErmC and the LbpA flanking sequences was amplified by PCR using primers pNEISS-TROPIC-for 5′-CTCGAGGTCGACGGTATCG-3′ and pNEISS-TROPIC_rev 5′-GGGAACAAAAGCTGGAGC-3′ at 58 °C annealing temperature and 3 min elongation time. The PCR fragment was used to transform *N. gonorrhoeae* by electroporation. To this end, $9 \times 10^8$ log-phase gonococci were washed twice with 1 ml of ice-cold electroporation buffer (EP; 15% glycerin, 272 mM sucrose, 1 mM HEPES, pH 7.2) before being taken up in 40 μl ice-cold EP. Upon

addition of 100 ng of purified PCR fragment, electroporation with a single pulse of 2,300 V, 400 ohm and 25 µF at an electrode gap of 2 mm was run on an ECM630 Exponential Decay Wave electroporation system (BTX). *Neisseria* were allowed to recover in 1 ml prewarmed PPM medium for 150 min at 37 °C and 220 rpm. Bacteria were selected for 48 h on GC agar with 28 mg l⁻¹ erythromycin. Erm-resistant clones were checked for the presence of *epsilon1*/*zeta1* by PCR. Chromosomal integration of *epsilon1*/*zeta1* into the *lbpA* locus was verified by PCR of genomic DNA using LbpA-for 5′-CGAGGCTGAAGTTCCTGCCCG-3′ and Zeta1-rev 5′-CAATGGCAGTTACGAAATTACACC-3′ primers (Extended Data Fig. 6b). As a control, *opa* loci were amplified using primers Opa-for 5′-GCAGATTATGCCCGTTACAG-3′ and Opa-rev 5′-GTTTTGAAGCGGGTGTTTCC-3′. Expression of Epsilon1 and Zeta1 was also verified by western blotting.

## Antibodies used in this study
The following primary and secondary antibodies were employed at the indicated dilutions for western blotting (WB): monoclonal antibody (mAB) against 6×His (H8, Thermo Fisher, 1:1,000 WB); mAb against GAPDH (clone GA1R, Invitrogen, 1:2,000 WB); mAb against GFP (clone JL-8, Clontech, 1:5,000); mAb α-Opa (clone 4B12/C11, Developmental Studies Hybridoma Bank, University of Iowa, USA; a generous gift from M. Achtman, WB: 1:2,000). Rabbit polyclonal antibody (pRb) against the synthetic Epsilon1 peptide C-EKNRRMMTDEAFRKEVEKRLYAG was produced by PSL (Germany), affinity purified against the cognate peptide and used at 1 µg ml⁻¹ for WB; pRb against the synthetic Zeta1 peptides AKKEYSKQRVVTNSK-C and KIVGINQDRNSEFIDK-C was produced by PSL and affinity purified using both peptides (1 µg ml⁻¹ pRb was used for WB). HRP-conjugated goat anti-mouse IgG (115-035-146, 1:10,000 WB) and HRP-conjugated goat anti-rabbit IgG (111-035-003, 1:5,000 WB) were from Jackson Immunoresearch.

## Cloning and expression of Epsilon1 in *E. coli*
The cDNA of gonococcal *epsilon1* was amplified from pTetM plasmid isolated from *N. gonorrhoeae* MS11 with primers 4069_NcoI_Epsilon1_fw 5′-ATACCATGGATGAATAAAGTTGAGCCCCAAG-3′ and 4113_XhoI_Epsilon1_rev 5′-TATCTCGAGTTGCTCCTTATTTGC-3′. The resulting PCR fragment was cloned into vector pET28a (Novagen, MerckBiosciences) via NcoI and XhoI restriction sites allowing expression of Epsilon1 with a C-terminal 6×His-tag. For the expression of neisserial Epsilon1 without a tag, the *epsilon1* cDNA was amplified as above with the primers 4520_e-antitoxin_NcoI_for 5′-ATACCATGGGCAATAAAGTTGAGCCCCAAG-3′ and 4521_e-antitoxin_XhoI_rev 5′-ATACTCGAGTTATTGCTCCTTATTTGCCGC-3′. The amplicon harboured an extra glycine after the start methionine and a STOP codon at the 3′ end. The resulting PCR fragment was cloned into vector pCDFduett (Novagen, MerckBiosciences) via NcoI and XhoI restriction sites.

## Expression of recombinant Epsilon1-6×His
Epsilon1-6×His was expressed in *E. coli* BL21 Rosetta (DE3) upon induction with 0.5 mM IPTG at an OD₆₀₀ of 0.75 for 4 h at 37 °C and 200 rpm in LB medium with 50 g l⁻¹ kanamycin. After centrifugation at 4,700 rpm for 20 min at 4 °C, the bacterial pellet was resuspended on ice in 20 ml buffer A (50 mM Na phosphate, 1 M NaCl, pH 8, supplemented with 10 µg ml⁻¹ Pefabloc, 10 µg ml⁻¹ aprotinin, 1 µM PMSF and 5 µg ml⁻¹ leupeptin) and sonicated at 4 °C three times for 2 min. Lysate was cleared by centrifugation for 1 h at 16,000 rpm at 4 °C. Supernatant was filtered using a 0.2-µm-pore-size filter. Epsilon1-6×His was purified using His-FF column (GE Healthcare) and dialysed against 50 mM Tris, pH 7.5, 150 mM NaCl and 10% glycerol. The amount and the purity of proteins were analysed via SDS-gel electrophoresis. To investigate stability of the purified protein, 50 ng purified Epsilon1-6×His was incubated with 1% DMSO, 50 µM NQNO, or 1 mM H₂O₂ in PBS supplemented with freshly added protease inhibitors (10 µg ml⁻¹ leupeptin,

20 µg ml⁻¹ aprotinin, 20 µg ml⁻¹ Pefabloc and 20 µM benzamidine) for the indicated time at room temperature. Samples were separated by SDS–PAGE and analysed by western blotting using mouse monoclonal anti-6×His antibodies.

## Epsilon degradation assay in *E. coli*
pCDFduett epsilon1 was transformed into *E. coli* Nova Blue (DE3). Bacteria were grown on LB plates with streptomycin (50 g l⁻¹) overnight at 37 °C and used to inoculate a liquid culture (OD₆₀₀ of 0.3) in LB medium containing 50 g l⁻¹ streptomycin. Bacteria were grown until an OD₆₀₀ of 0.6 and a sample was taken before induction (BI). The remaining culture was induced with 0.01 mM IPTG for 30 min and a second sample was then taken (after induction, AI). The IPTG induction was stopped by pelleting the bacteria and suspending them in LB medium containing 50 g l⁻¹ streptomycin. The culture was split into several samples, which received 0 mM, 1 mM or 5 mM H₂O₂, 25 µM or 50 µM NQNO or 1% DMSO. Bacteria were lysed after 45 and 60 min, and degradation of epsilon1 was analysed via western blotting and densitometric analysis using ImageJ/Fiji software. The amount of epsilon1 was compared to the expression levels of GAPDH.

## Epsilon degradation in *N. gonorrhoeae*
*Neisseria* spp. were grown overnight at 37 °C and 5% CO₂ on GC plates. Upon preculture in PPM for ~2 h, 4 × 10⁷ bacteria were used to inoculate 5 ml PPM. Cultures were grown for 3 h until they reached the log phase. NQNO, Antimycin A or ciprofloxacin were added at the indicated concentrations and in all samples, DMSO was adjusted to 1% final concentration. After the indicated times, bacteria were collected by centrifugation (3,000g, 8 min, 4 °C) and lysed in modified RIPA buffer (1% Triton X-100, 0.1% SDS, 1% deoxycholic acid, 50 mM HEPES, pH 7.4, 150 mM NaCl, 10% glycerol, 1.5 mM MgCl₂, 1 mM EGTA) supplemented with freshly added protease inhibitors (10 µg ml⁻¹ leupeptin, 20 µg ml⁻¹ aprotinin, 20 µg ml⁻¹ Pefabloc and 20 µM benzamidine). Lysates were cleared (13,000 rpm, 4 °C, 20 min) and the total protein content was determined via the bicinchoninic acid protein assay kit (Pierce, Thermo Fisher). Equal amounts of each lysate were separated by SDS–PAGE and analysed by western blotting with rabbit polyclonal anti-epsilon1 antiserum or anti-zeta1 antiserum, and densitometric analysis using ImageJ/Fiji software. The amount of epsilon1 was quantified relative to protein levels of the zeta toxin.

## Screening of AQNO and AQ derivatives in 96-well format
*Neisseria* were precultured in 10 ml PPM medium for 2 h at 37 °C and 220 rpm to reach a final OD₅₅₀ of 0.3. Precultures were diluted to an OD₅₅₀ of 0.2. The 96-well compound plate was prepared by adding 1 µl of a 0.5 mM compound stock solution in DMSO to 100 µl bacterial suspension at OD₅₅₀ 0.1 in PPM medium. The final compound concentration was 5 µM in 1% DMSO. Accordingly, the solvent control was 1% DMSO. Each 96-well plate had 6 wells holding the solvent control and 8 wells holding the positive control (5 µM NQNO). The plates were then incubated for 6 h at 37 °C at 550 rpm. Growth was monitored every 30 min using OD₅₅₀ readings (Tecan Sunrise). The experiment was performed in three independent replicates. Data evaluation was done by subtracting the starting OD₅₅₀ (time point 0 h) from OD₅₅₀ at 6 h (negative values were set to zero). The calculated OD₅₅₀ was normalized to the average OD₅₅₀ of the positive control (5 µM NQNO). Values >1 indicate reduced growth inhibition compared with 5 µM NQNO, while vaues <1 indicate a growth inhibition stronger than the one observed for 5 µM NQNO.

## In vivo infection of CEAtg mice
C57BL/6J mice carrying the complete human CEA gene (CEAtg mice)[76] and wild-type C57BL/6J mice (obtained from Elevage Janvier) were kept under specific pathogen-free conditions under a 12-h light cycle in the animal facility of the University of Konstanz. Experiments involving

animals were performed in accordance with the German Law for the Protection of Animal Welfare. The animal care and use protocol, including the protocol for experimental vaginal infection of female mice, was approved by the appropriate state ethics committee and state authorities regulating animal experiments (Regierungspräsidium Freiburg, Germany) under permit file number G-19/147. Experimental vaginal infection of female mice with *N. gonorrhoeae* was performed as previously described[39] for prohibiting bacterial infections with NQNO (prophylactic treatment) or using an adapted schedule as described below to treat established infections with 3Me-NQNO (therapeutic treatment). For prophylactic treatment, mice were subcutaneously injected with 17-β-estradiol (50 µg per mouse) in corn oil 4 days and 2 days before infection. The drinking water was supplemented with 40 mg per 100 ml drinking water trimethoprim sulphate (Infectotrimed, Infectopharm). Mice were inoculated intravaginally with $10^8$ c.f.u. of $Opa_{CEA}$-expressing gonococci (strain *N. gonorrhoeae* MS11 or Ngo XDR) suspended in 10 µl of PPM medium supplemented with 5% IsoVitale vitamin mix. The CEACAM-binding $Opa_{CEA}$-expressing Ngo XDR was visually selected and functionally tested for CEACAM binding as described previously[43]. Prophylactic treatment was started at 1 h after infection by applying 10 µl of 25 µM or 50 µM NQNO (corresponding to 0.25 nmol or 0.5 nmol NQNO) in PPM medium supplemented with 5% IsoVitale vitamin mix and 1% DMSO intravaginally. Control mice received 10 µl PPM medium supplemented with 5% IsoVitale vitamin mix and 1% DMSO. Mice were randomly assigned to the control or treatment groups, but experimenters were not blinded to the conditions of the experiment.

In the case of therapeutic treatment with 3Me-NQNO, the animals were subcutaneously injected with 17-β-estradiol (50 µg per mouse) in corn oil 4 days and 2 days before infection. To suppress commensal overgrowth during the extended infection, mice also received chloramphenicol (500 µg per mouse) and streptomycin (2.4 mg per mouse) 4 days and 2 days before infection. On day 0, CEAtg mice were inoculated intravaginally with $10^8$ c.f.u. of $Opa_{CEA}$-expressing gonococci (strain *N. gonorrhoeae* MS11) suspended in 10 µl of PPM medium supplemented with 5% IsoVitale vitamin mix. At 24 h after infection, treatment was started by applying 10 µl of 50 µM 3Me-NQNO (corresponding to 0.5 nmol 3Me-NQNO) in PPM medium supplemented with 5% IsoVitale vitamin mix and 1% DMSO intravaginally. Control mice received 10 µl PPM medium supplemented with 5% IsoVitale vitamin mix and 1% DMSO.

At 24 h after infection (prophylactic treatment) or 48 h after infection (therapeutic treatment), the mucosa-associated bacteria were re-isolated by cotton swabs and vaginal lavage using 20 µl PPM medium supplemented with 5% IsoVitale vitamin mix. Serial dilutions of re-isolated bacteria were plated on GC plates without antibiotics (to detect growth of commensal bacteria), on GC agar containing chloramphenicol (10 µg ml$^{-1}$) and erythromycin (7 µg ml$^{-1}$) for MS11, or on GC plates containing ciprofloxacin (10 µg ml$^{-1}$) for Ngo XDR, and the colonies were counted after 20 h of incubation at 37 °C and 5% $CO_2$.

**Immunohistochemistry of tissue samples**
The genital tract of infected mice was excised and immediately fixed with 4% paraformaldehyde for at least 24 h. The fixed tissue was sequentially transferred to 10% sucrose, 0.05% cacodylic acid for 1 h at 4 °C, to 20% sucrose for 1 h at 4 °C, and then to 30% sucrose at 4 °C overnight. Organs were mounted in the embedding medium (Cryo-M-Bed, Bright Instrument) and frozen at −20 °C. Sections (10-µm thick) were cut at −20 °C using a cryostat (Vacutom HM500, Microm). Sections were stained with a mouse monoclonal antibody against collagen type IV (clone M3F7, dilution 1:200) together with a polyclonal rabbit antibody against *N. gonorrhoeae* (dilution 1:100) (IG-511, Immunoglobe). Detection of the primary antibodies was done with a combination of Cy5-conjugated goat anti-rabbit antibody (1:250) and Cy3-conjugated goat anti-mouse antibody (1:250) (both

from Jackson Immunoresearch). Cell nuclei were visualized by the addition of Hoechst 33342 (1:30,000, Life Technologies) in the final staining step.

**Test of bactericidal effect of 3Me-NQNO**
*Neisseria* spp. were inoculated at an $OD_{550}$ of 0.2 in 5 ml PPM. After 4 h of culturing, 0.5 µM or 2.5 µM 3Me-NQNO was added to the log-phase cultures. A volume of 1 ml of culture was taken after 15 min, 30 min, 1 h, 2 h and 3 h, bacteria were washed in PBS and dilutions ($10^0$ to $10^{-6}$) were plated on GC plates. Colonies were enumerated after overnight incubation at 37 °C and 5% $CO_2$.

**Statistics**
All data are presented as mean ± s.e.m. Statistical significance was determined as indicated in the figure legends using a two-tailed Student's *t*-test with Prism5 (GraphPad) or one-way and two-way analysis of variance (ANOVA) by multiple comparison to the DMSO sample using Dunnett's multiple comparisons or Sidak's multiple comparisons test using Prism7 (GraphPad) (*$P < 0.05$, **$P < 0.01$, ***$P < 0.001$). For the statistical evaluation of in vivo colonization, differences between groups were analysed using Mann–Whitney rank-sum test in SigmaStat4 (SysStat). For all applied statistical tests if not differently indicated, *P* values were set as *$P < 0.05$, **$P < 0.01$, ***$P < 0.001$, ****$P < 0.0001$ with a 95% confidence interval and two-tailed *P* values.

**Reporting summary**
Further information on research design is available in the Nature Portfolio Reporting Summary linked to this article.

## Data availability
All data supporting the findings of this study are available within the paper and its Supplementary Information. Raw numerical data of bacterial growth curves, metabolic measurements and image quantifications have been deposited on Zenodo at https://doi.org/10.5281/zenodo.14887998 (ref. 77). The raw FASTq files and draft genome sequences of the three strains MS11_N309 (accession no. SAMN19108268), MS11-R1 (N568) (accession no. SAMN19108269) and MS11-R2 (N569) (accession no. SAMN19108270) were combined in Bioproject number PRJNA728975 and are publicly available in the NCBI GenBank. The accessible links are: *N. gonorrhoeae* MS11-N309: https://www.ncbi.nlm.nih.gov/nuccore/JAHBBP000000000; *N. gonorrhoeae* MS11-R1: https://www.ncbi.nlm.nih.gov/nuccore/JAHBBO000000000; *N. gonorrhoeae* MS11-R2: https://www.ncbi.nlm.nih.gov/nuccore/JAHBBN000000000. As a reference genome, the publicly available genome of *N. gonorrhoeae* MS11 was used (accession number NC_022240; https://www.ncbi.nlm.nih.gov/nuccore/NC_022240.1/). Source data are provided with this paper.

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

## Acknowledgements

We thank M. Laumann and P. Bergmann (Electron Microscopy Center of the University of Konstanz), S. Müller (Screening Center, University of Konstanz) and K. Tegelkamp (Münster) for technical support; G. Keefer (Konstanz) for advice on mouse experiments; M. Unemo (Örebro University, Örebro, Sweden), C. Manoil (University of Washington, Seattle, WA), T. F. Meyer (Max-Planck-Institute for Infection Biology, Berlin, Germany) and T. Oelschläger (University of Würzburg, Würzburg, Germany) for providing bacterial strains, and A. J. Schaeffer (Feinberg School of Medicine, Northwestern University, Chicago, IL) for hVEC cells. This study was supported by Deutsche Forschungsgemeinschaft funds to C.R.H. (HA 2856/10-1 and HA 2856/11-1) and to U.D. (SFB1009/B05), by the BW Stiftung 'Wirkstoffforschung' to C.R.H., T.B. and T.U.M., and by the Emmy Noether programme (DFG), an EU FP7 Marie Curie Zukunftskolleg Incoming Fellowship and Fonds der Chemischen Industrie (FCI) to T.B. D.S. was supported by a Konstanz Research School Chemical Biology (KoRS-CB) fellowship. The funders were not involved in the design, collection, analysis and interpretation of data; in the writing of the manuscript; or in the decision to submit the manuscript for publication.

## Author contributions

A.-K.M., T.H.N.N., T.S., D.S., T.U.M., T.B. and C.R.H. conceived the study and designed the experiments. A.-K.M., T.H.N.N., T.S., D.S., P.H., L.H. and Y.O.D. performed the experiments. P.M. and A.-K.M. performed

the in vivo infections in the murine model. H.T.W. and U.D. performed genome analysis. A.-K.M., T.H.N.N., T.S., D.S., T.U.M., T.B. and C.R.H. evaluated the data. C.R.H. wrote the paper. All authors read and approved the final manuscript.

## Funding

## Competing interests

T.S., D.S., P.M., T.B. and C.R.H. declare that the University of Konstanz has filed a patent application (WO/2019/121392, filing date 14 December 2018, publication date 27 June 2019, United States; no. 16768828) on the use of NQNO to treat bacterial infections, with these authors listed as inventors. The remaining authors declare no competing interests.

## Additional information

**Extended data** is available for this paper at https://doi.org/10.1038/s41564-025-01968-y.

**Correspondence and requests for materials** should be addressed to Thomas Böttcher or Christof R. Hauck.

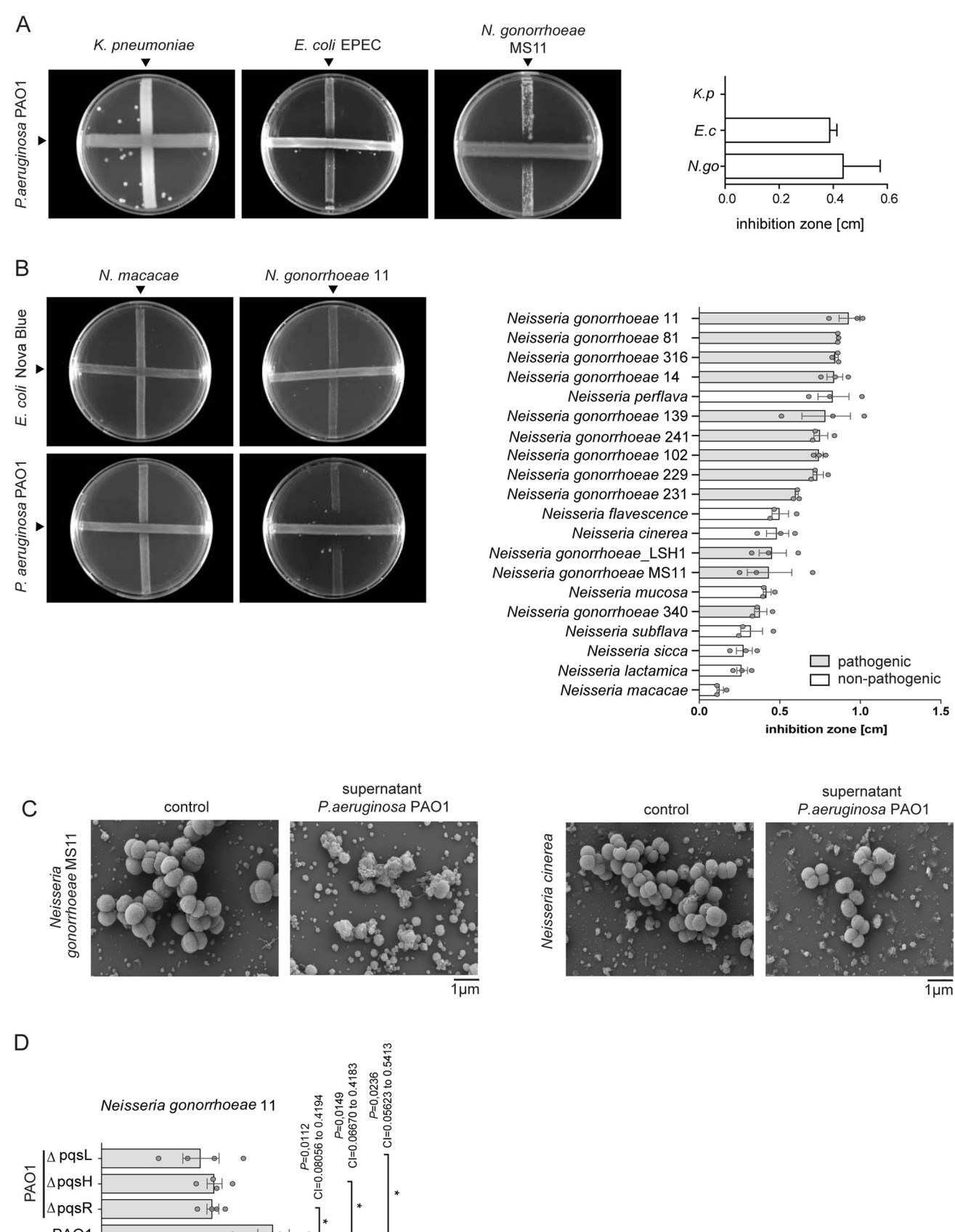

**Extended Data Fig. 1 | See next page for caption.**

**Extended Data Fig. 1 | *Pseudomonas aeruginosa* metabolites inhibit growth of *Neisseria gonorrhoeae*. (a)** *Klebsiella pneumoniae*, enteropathogenic *Escherichia coli* (EPEC) or *Neisseria gonorrhoeae* MS11 were streaked on appropriate agar growth plates together with a horizontal cross-streak of *Pseudomonas aeruginosa* strain PAO1 and incubated for 24 h. Representative images of agar growth plates are shown. Bars in the right panel display mean length of the growth inhibition zone for the indicated bacteria ± SEM (n = 3, biological replicates). **(b)** *N. gonorrhoeae* 11 or *Neisseria macacae* (vertical) were cross-streaked with *E. coli* Nova Blue or with *P. aeruginosa* PAO1 (horizontal) and incubated for 24 h. Shown are representative images. Bars in the right panel display mean length of the growth inhibition zone for the indicated Neisseriae ± SEM (n = 3, biological replicates) grown with *P. aeruginosa* PAO1. Pathogenic *N. gonorrhoeae* strains are highlighted by grey bars, commensal *Neisseriae* are displayed with open bars. **(c)** *N. gonorrhoeae* MS11 and *N. cinerea* were incubated without (control) or with culture supernatants of *Pseudomonas aeruginosa* PAO1 and analyzed by scanning electron microscopy. Scale bars represent 1 μm. **(d)** *N. gonorrhoeae* 11 was cross-streaked with either wild-type *P. aeruginosa* PAO1 or with isogenic mutants in AQ synthesis genes (Δ*pqsL*, Δ*pqsH* or Δ*pqsR*). Bars represent mean length of the growth inhibition zone ± SEM (n = 4, biological replicates). Significance was evaluated with unpaired, parametric Student's t-test, *p < 0.05, confidence interval 95%, P value two-tailed.

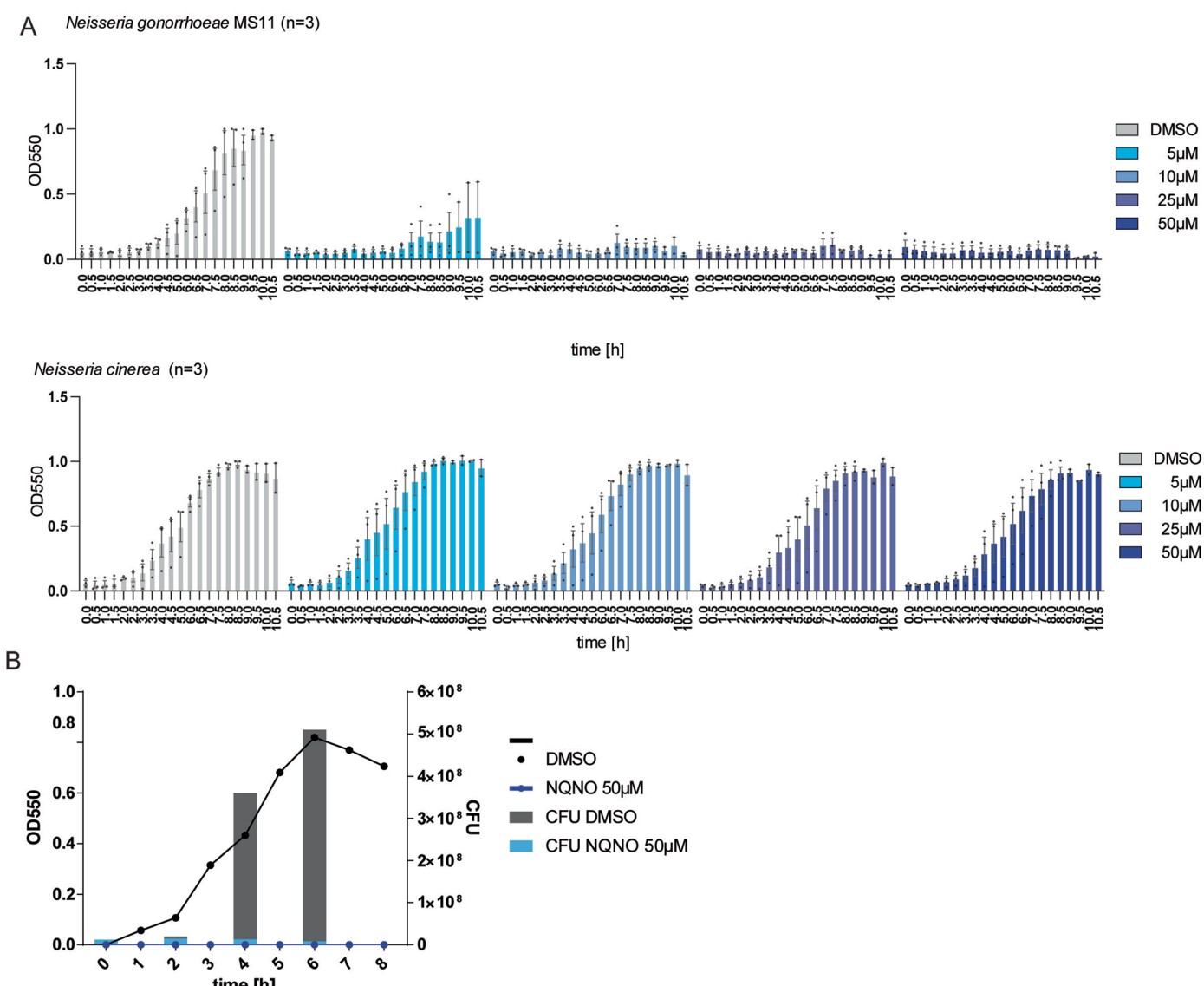

**Extended Data Fig. 2 | *Neisseria gonorrhoeae* is selectively inhibited by AQ derivatives. (a)** Growth curves of *Neisseria gonorrhoeae* MS11 and *N. cinerea* treated with the indicated concentrations of NQNO or the solvent alone (DMSO). $OD_{550}$ readings were taken every 30 min over the course of 10.5 h. Bars represent mean $OD_{550}$ values ± SEM (n = 3, biological replicates). **(b)** Growth curves of *Neisseria gonorrhoeae* MS11 incubated with 50 μM NQNO or DMSO were determined as in **(a)**. In parallel, aliquots of the same samples were taken at the indicated time points and plated to determine colony forming units (cfu). Black bars indicate cfu of DMSO-treated, red bars are derived from NQNO-treated cultures.

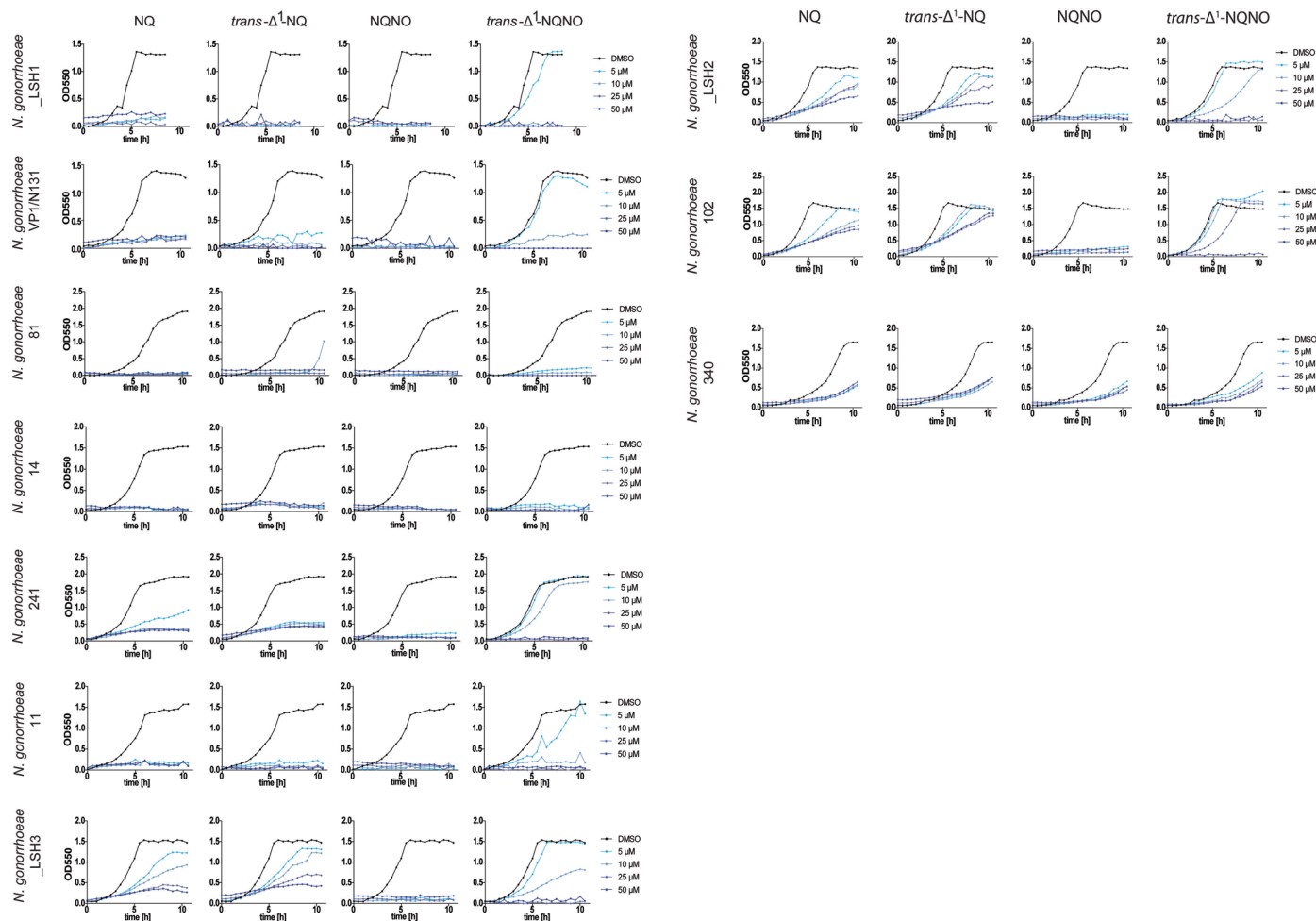

**Extended Data Fig. 3 | AQ derivatives inhibit growth of various *Neisseria gonorrhoeae* strains.** The indicated *Neisseria gonorrhoeae* strains were grown in the presence of 5, 10, 25, or 50 μM of NQ, *trans*-Δ¹-NQ, NQNO, or *trans*-Δ¹-NQNO in PPM medium. Growth was monitored by $OD_{550}$ readings every 30 min over the course of 10 h. Control cultures received solvent (1% DMSO) only.

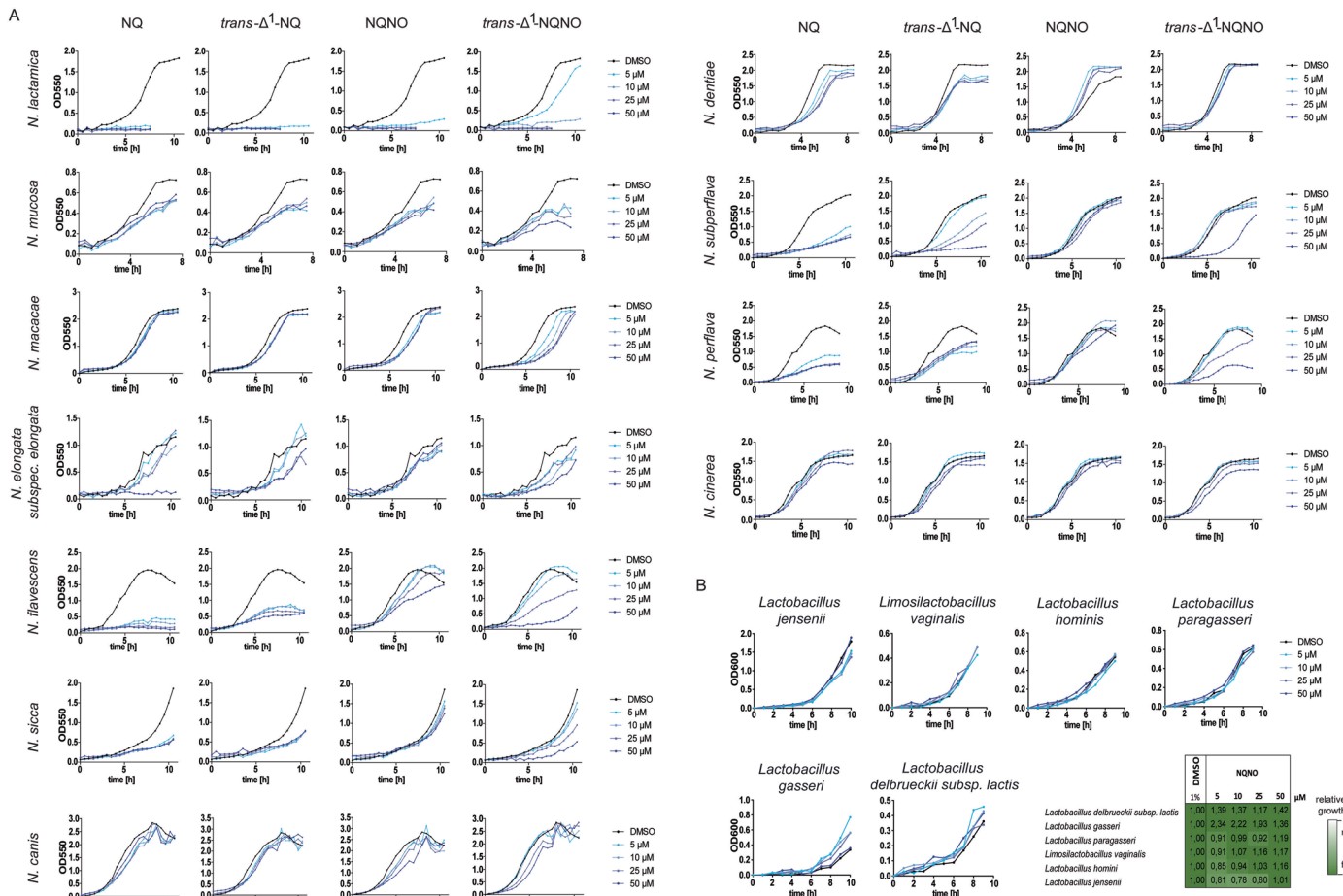

**Extended Data Fig. 4 | Commensal *Neisseria* strains and commensals from the vaginal tract are insensitive to AQ derivatives. (a)** The indicated commensal *Neisseria* species were grown in the presence of 5, 10, 25, or 50 µM of NQ, *trans*-Δ¹-NQ, NQNO or *trans*-Δ¹-NQNO in PPM medium. Growth was monitored by OD₅₅₀ readings every 30 min over the course of 8–10 h. Control cultures received solvent (1% DMSO) only. **(b)** *Lactobacillus* strains were exposed to the indicated concentrations of NQNO. Control cultures received solvent (1% DMSO) only.

Growth was monitored by OD₆₀₀ readings every 30 min over the course of 10 h as in **(a)**. Growth was quantified by the area under the growth curve and normalized to growth of the DMSO-treated sample. Values are displayed in the table. The color gradient indicates growth compared to control (1% DMSO) from no growth (90–100 % inhibition of growth - white) to full growth (0–10 % inhibition of growth – dark green).

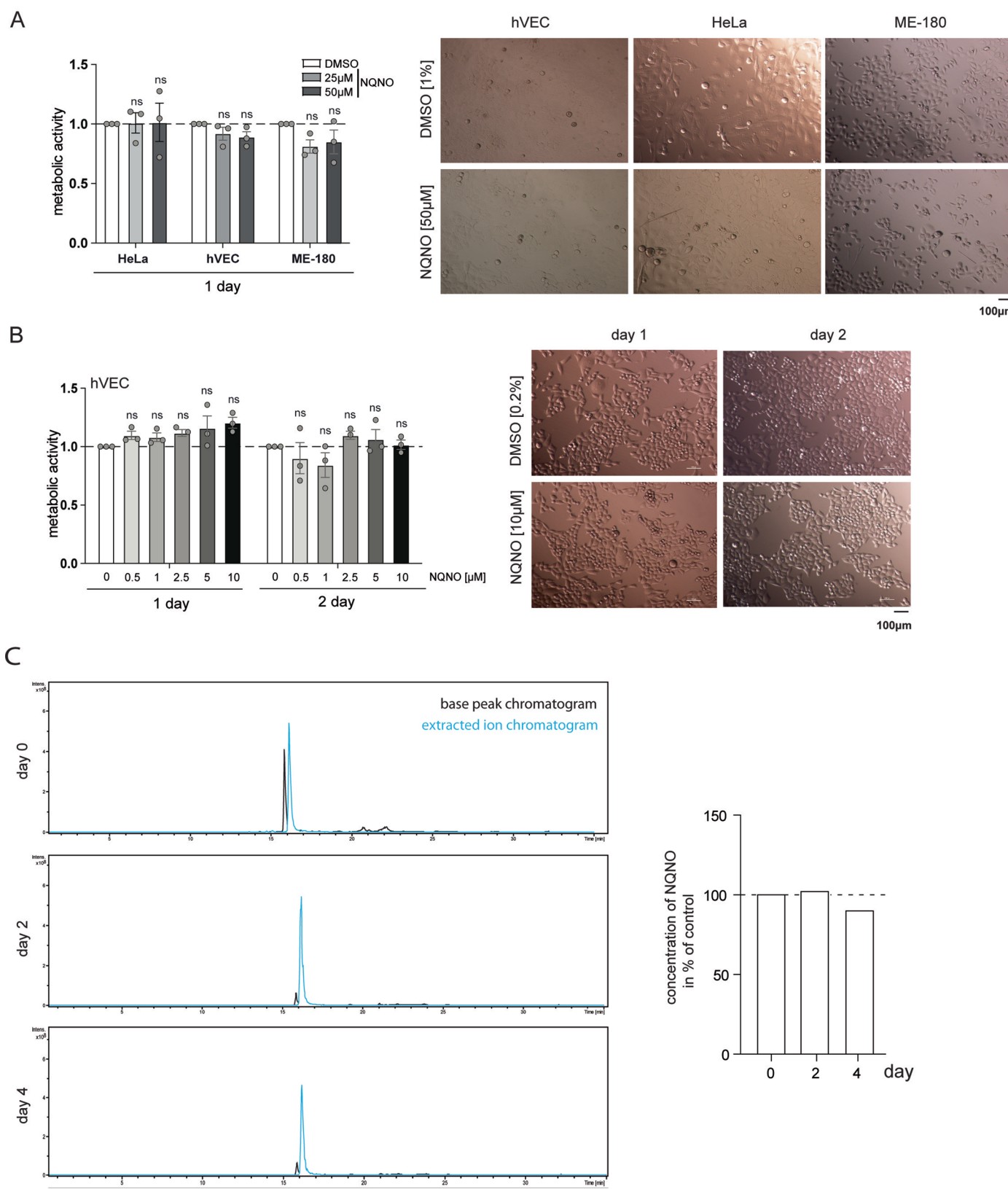

**Extended Data Fig. 5 | See next page for caption.**

**Extended Data Fig. 5 | NQNO does not harm isolated human cells and is stable in serum containing medium at 37 °C. (a)** Measurement of metabolic activity of HeLa, hVEC, or ME180 cells treated with 25 µM or 50 µM NQNO for 24 h. Samples were normalized to solvent (DMSO) treated cells. Bars represent mean ± SEM (n = 3, biological replicates) with each experiment conducted in technical triplicates. Statistical analysis by 2-way ANOVA with Tukey's multiple comparisons test with 95% confidence interval, * $p < 0.05$, ** $p < 0.01$. Representative microscopic images of the different cell lines grown for one day in the presence of 50 µM NQNO or 1% DMSO are depicted on the right. **(b)** Measurement of metabolic activity of hVEC treated with 0.5, 1, 2.5, 5 or 10 µM NQNO for 1 or 2 days respectively. Samples were normalized to solvent (DMSO) treated cells. Bars represent mean ± SEM (n = 3, biological replicates) with each experiment conducted in technical triplicates. Statistical analysis as in **(a)**. Representative microscopic images of hVEC cultures are depicted on the right. **(c)** 100 µM NQNO in Dulbecco's modified Eagle's medium (DMEM) supplemented with 10 % fetal calf serum was incubated at 37 °C. Samples were taken at day 0 and every second day and analyzed by LC-MS. The integrals of the NQNO signal (by using ion chromatograms) were normalized to the integral of the control (100 µM NQNO in $H_2O$) and are depicted in the bar graph on the right.

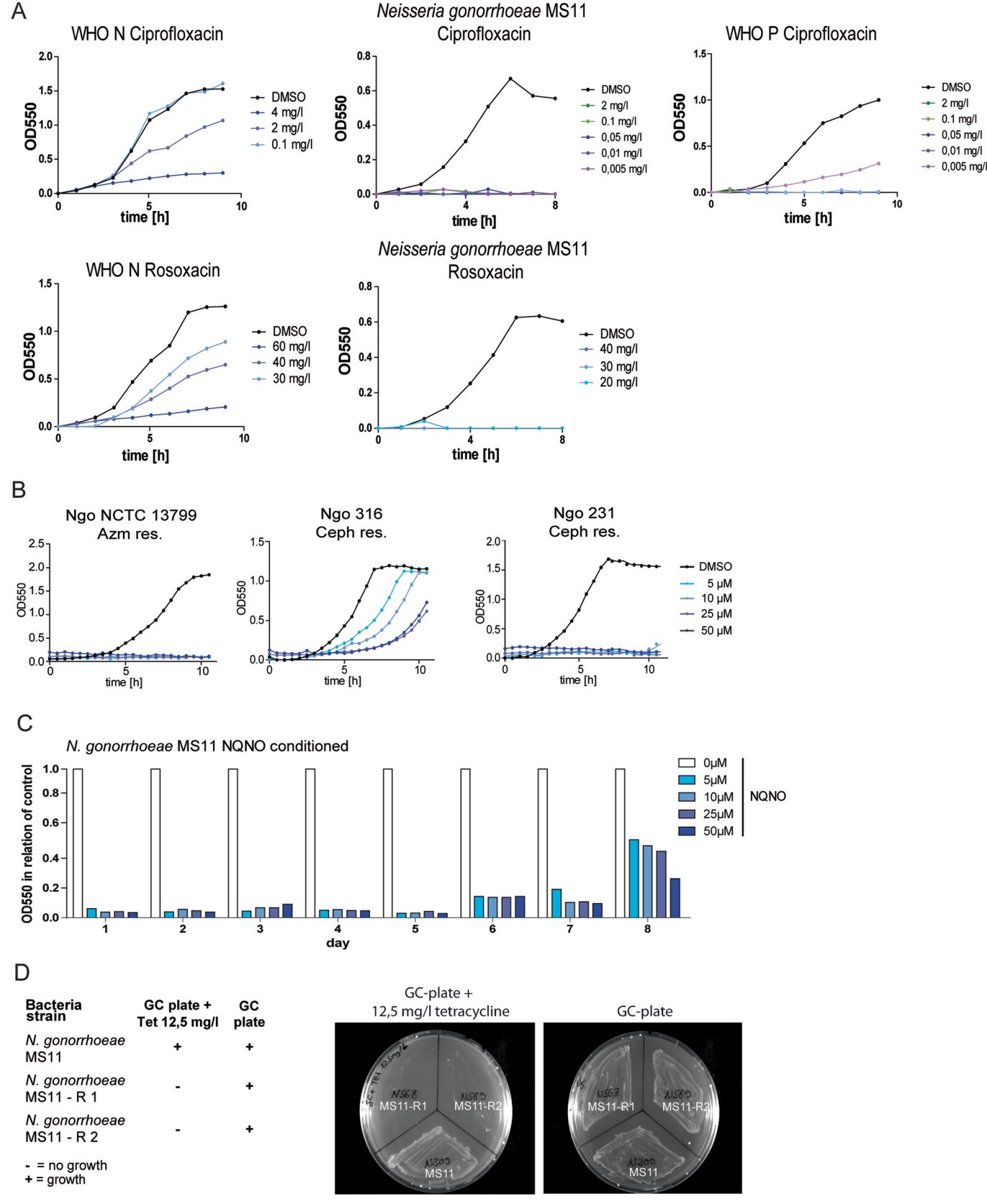

**Extended Data Fig. 6 | See next page for caption.**

**Extended Data Fig. 6 | NQNO resistance does not correlate with existing fluoroquinolone or azithromycin resistance, but is linked to the loss of pTetM. (a)** *N. gonorrhoeae* MS11, WHO N, or WHO P were grown in the presence of indicated concentrations of ciprofloxacin or rosoxacin. Control cultures were grown in presence of 1% DMSO. Growth was monitored by OD$_{550}$ readings every 60 min over the course of 8–10 h. Shown are representative growth curves. **(b)** Growth of azithromycin (Azm) and 3$^{rd}$ generation cephalosporin (Ceph) resistant *N. gonorrhoeae* (Ngo) strains in the presence of different NQNO concentrations as in **(a)**. **(c)** Selection for NQNO resistance. *N. gonorrhoeae* MS11 was conditioned in 1 μM NQNO-containing medium for the indicated number of days, before inoculating medium containing 0, 5, 10, 25 or 50 μM NQNO. Growth was monitored by OD$_{550}$ readings every 60 min over the course of 8 h. Bars indicate area under the growth curve in relation to the DMSO control. Shown is a representative experiment performed twice with similar results. **(d)** Growth of *N. gonorrhoeae* MS11, MS11-R1, and MS11-R2 on GC-plates supplemented or not with 12.5 mg/l tetracycline. Data are representative of three independent replicates.

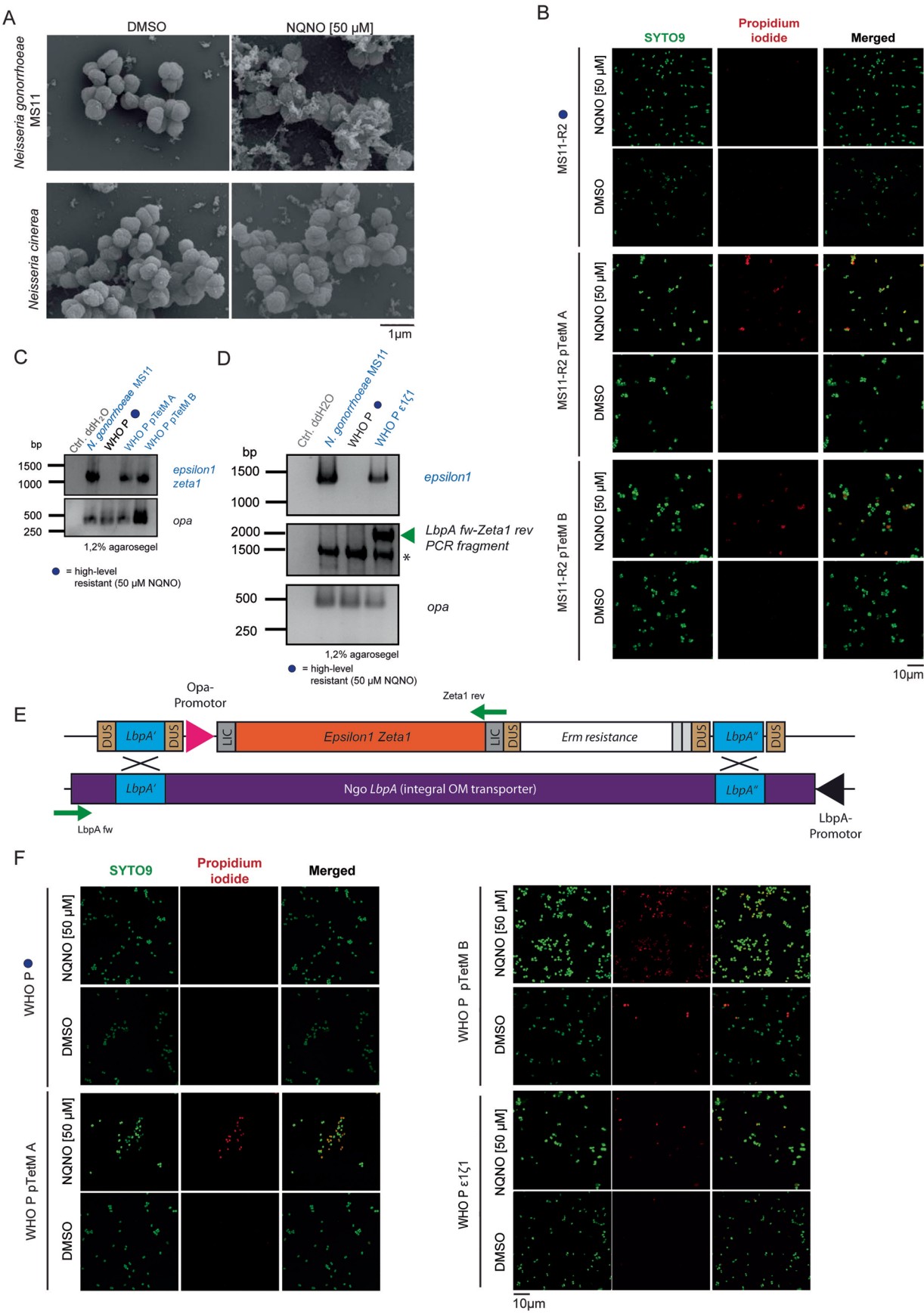

**Extended Data Fig. 7 | See next page for caption.**

**Extended Data Fig. 7 | The presence of the ζ1 toxin correlates with NQNO-induced membrane permeability in gonococci. (a)** *N. gonorrhoeae* MS11 and *N. cinerea* treated for 4 h with 50 μM NQNO or solvent only (DMSO) were analyzed by scanning electron microscopy. Scale bar represents 1 μm. **(b)** *N. gonorrhoeae* MS11-R2 (NQNO-resistant) and its pTetM-positive conjugants MS11-R2 pTetM A and MS11-R2 pTetM B were grown for 30 min in the presence of 50 μM NQNO or 1% DMSO. Bacteria were stained with SYTO9 (all bacteria) and propidium iodide (bacteria with disintegrated cell wall) and membrane integrity was analyzed by confocal microscopy. Scale bar, 10 μm. **(c)** Presence of the *epsilon1/zeta1* operon in WHO P conjugated with the pTetM plasmid or transformed with the ε1/ζ1 construct was verified by PCR of the *epsilon1/zeta1* locus using total DNA. As a control, PCR for *opa* genes was conducted with the same DNA sample. Red dot marks NQNO resistant strains. **(d)** Successful chromosomal integration of the *epsilon1/zeta1* operon in WHO P ε1/ζ1 was verified by PCR of genomic DNA isolated from *N. gonorrhoeae* MS11, the parent strain WHO P, or a WHO P ε1/ζ1 transformant. PCR was performed with primers LbpA_fw and Zeta1_rev indicated in **(e)** yielding a 2 kb amplicon. Control PCRs for the *epsilon1/zeta1* operon and *opa* genes was conducted as in **(c)**. **(e)** Schematic overview of the insertion of the opa promotor driven *epsilon1zeta1* operon and the erythromycin resistance gene (ErmC) into the *Neisseria gonorrhoeae* chromosome at the *LbpA* locus. Arrows indicate the location of primers used for PCR verification. DUS, neisserial DNA uptake sequence. **(f)** *N. gonorrhoeae* WHO P, two independent pTetM conjugants (WHO P pTetM A and WHO P pTetM B), and WHO P ε1/ζ1 were grown for 30 min in the presence of 50 μM NQNO or 1% DMSO. Membrane integrity was analyzed as in **(b)**. Scale bar, 10 μm.

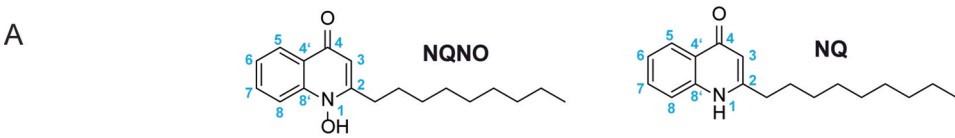

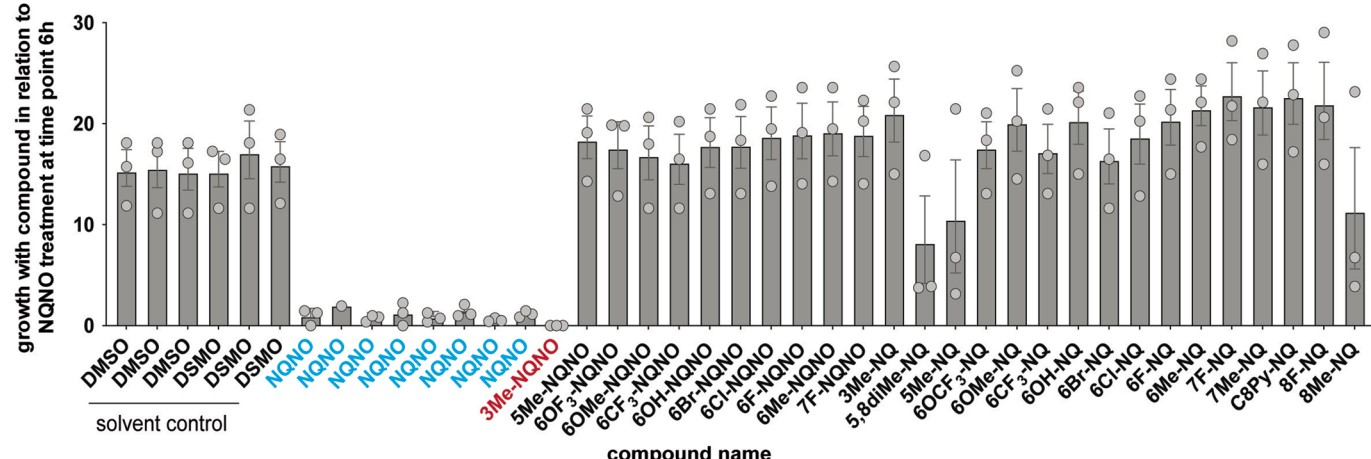

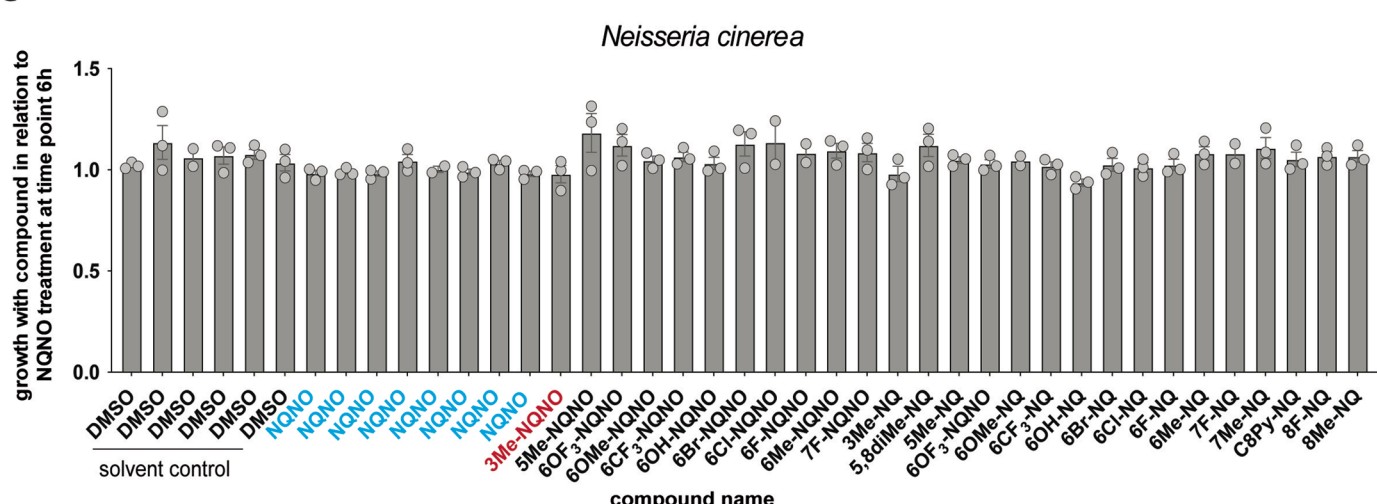

**Extended Data Fig. 8 | Growth inhibitory potential of NQNO derivatives towards *N. gonorrhoeae* and *N. cinerea*. (a)** General structure and nomenclature of NQNO and NO derivatives synthesized in this study. A detailed list of all compound names, abbreviatons, and chemical structures can be found in Supplementary Table 3. **(b)** *N. gonorrhoeae* MS11 or **(c)** *N. cinerea* were grown in the presence of 5 μM of the indicated NQNO derivatives for 6 h in 96-well plates. Six wells received 0.1% DMSO only (solvent control), while eight wells received 5 μM NQNO and served as positive control (blue font). The OD$_{550}$ at time point 0 h of each well was subtracted from the OD$_{550}$ measured after 6 h of growth, the area under the curve (AUC) was determined, and the value was normalized to the growth observed in the presence of 5 μM NQNO. Therefore, values > 1 indicate increased growth compared to 5 μM NQNO, while values < 1 indicate a growth inhibition stronger than the one observed with 5 μM NQNO. Data show mean ± SEM (n = 3, biological replicates). The only compound with an increased inhibitory effect towards *N. gonorrhoeae*, 3Me-NQNO, is indicated in red.

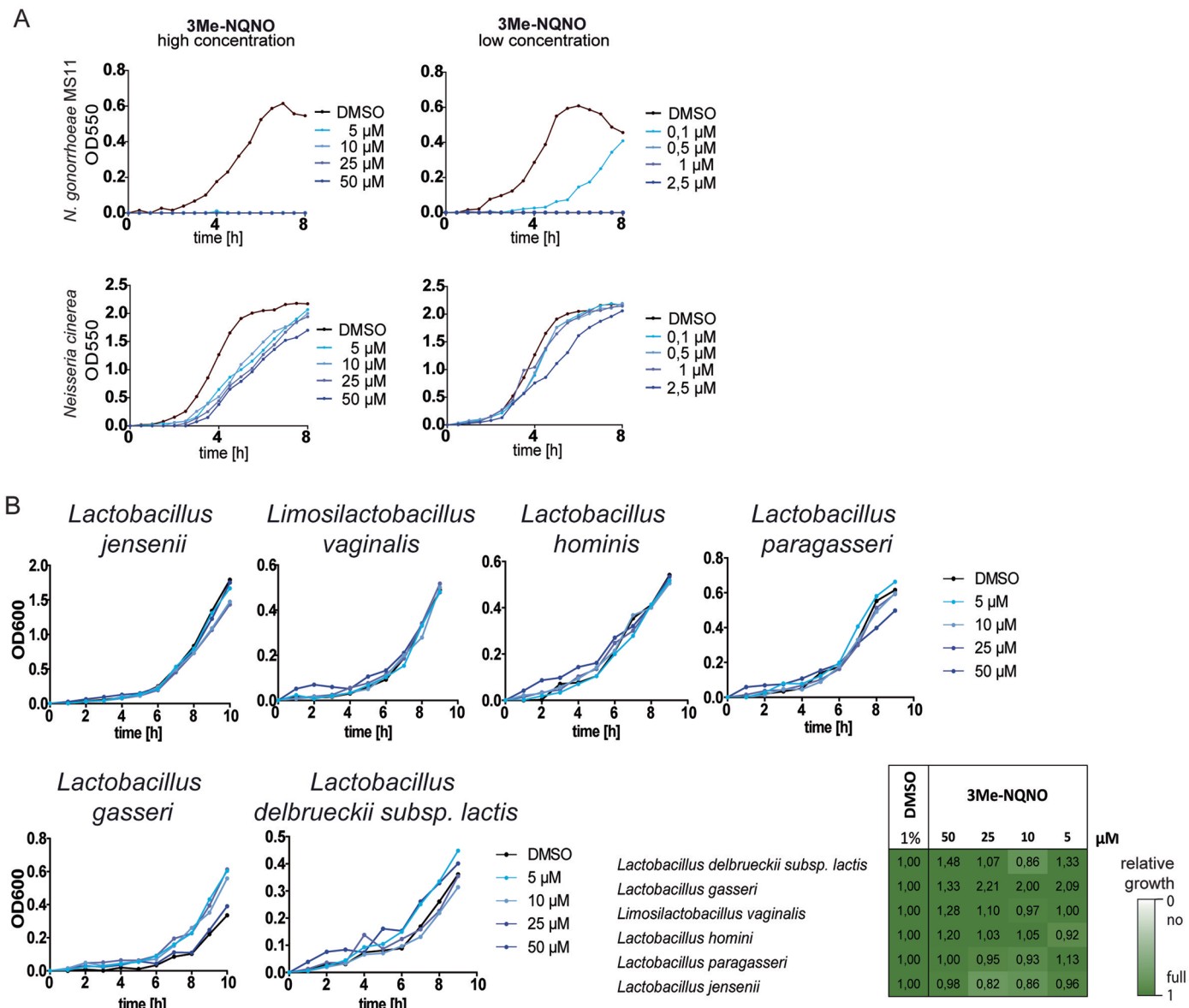

**Extended Data Fig. 9 | 3-methyl-NQNO has potent and selective antibiotic activity against *N. gonorrhoeae*. (a)** Growth of *N. gonorrhoeae* MS11 and *N. cinerea* in the presence of the indicated concentrations of 3Me-NQNO (left: high concentration range 50–5 µM; right: low concentration range: 2.5 – 0.1 µM). Growth was determined by OD₅₅₀ readings every 30 min over 8 h. Shown is a representative experiment repeated twice with similar results. **(b)** The indicated

*Lactobacillus* species were grown in the presence of 5, 10, 25, or 50 µM 3Me-NQNO. Control cultures received solvent (1% DMSO) only. Growth was monitored by OD₆₀₀ readings every 30 min. Growth was also quantified by the area under the growth curve. The color gradient used indicates growth of compound-treated cultures compared to 1% DMSO control and ranges from no growth (90–100 % inhibition - white) to full growth (0–10 % inhibition – dark green).

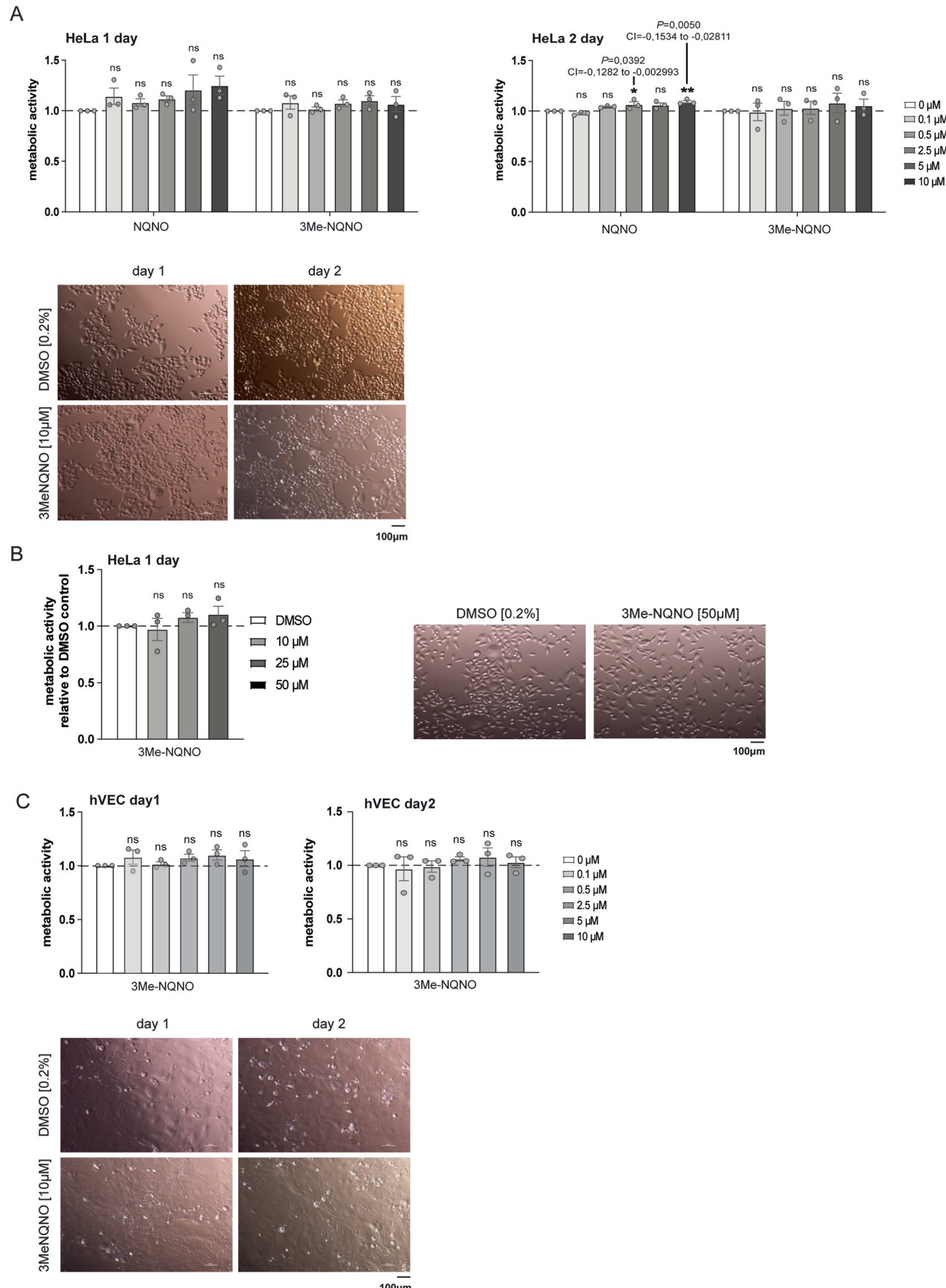

**Extended Data Fig. 10 | See next page for caption.**

**Extended Data Fig. 10 | 3-methyl-NQNO does not harm human epithelial cells. (a)** Human cervical carcinoma cells (HeLa) were grown for one or two days in the presence of the indicated concentrations of NQNO or 3Me-NQNO. Controls received 0.2% DMSO. Metabolic activity of the cells was measured by MTT assays. Bars depict the mean ± SEM (n = 3, biological replicates) with each experiment conducted in technical triplicates. Statistical analysis by 2-way ANOVA Dunnett's multiple comparisons test with confidence interval of 95%. P value two tailed, ns – not significant, * p < 0.05, ** p < 0.01. Panel below shows representative images of DMSO or 3Me-NQNO (10 μM) treated cultures after 1 or 2 days. Scale bar, 100 μm. **(b)** HeLa cells were grown for one day in the presence of 10 μM, 25 μM, or 50 μM 3Me-NQNO. Controls received 0.2 % DMSO. Metabolic activity of the cells was measured by MTT assays. Bars depict the mean ± SEM (n = 3, biological replicates) with each experiment conducted in technical triplicates. Statistical analysis as in **(a)**. Right panel shows representative images of control cultures and cultures treated with 50 μM 3Me-NQNO. **(c)** Immortilized human vaginal epithelial cells (hVEC) were grown for 1 or 2 days in the presence of the indicated concentrations of NQNO or 3Me-NQNO. Controls received 0.2% DMSO. Metabolic activity of the cells was measured by MTT assays. Bars depict the mean ± SEM (n = 3, biological replicates) with each experiment conducted in technical triplicates. Statistical analysis as in **(a)**. Panel below shows representative images of DMSO or 3Me-NQNO (10 μM) treated cultures after 1 or 2 days. Scale bar, 100 μm.

# Reporting Summary

## Statistics

For all statistical analyses, confirm that the following items are present in the figure legend, table legend, main text, or Methods section.

| n/a | Confirmed | |
|---|---|---|
| ☐ | ☒ | The exact sample size (*n*) for each experimental group/condition, given as a discrete number and unit of measurement |
| ☐ | ☒ | A statement on whether measurements were taken from distinct samples or whether the same sample was measured repeatedly |
| ☐ | ☒ | The statistical test(s) used AND whether they are one- or two-sided *Only common tests should be described solely by name; describe more complex techniques in the Methods section.* |
| ☒ | ☐ | A description of all covariates tested |
| ☐ | ☒ | A description of any assumptions or corrections, such as tests of normality and adjustment for multiple comparisons |
| ☐ | ☒ | A full description of the statistical parameters including central tendency (e.g. means) or other basic estimates (e.g. regression coefficient) AND variation (e.g. standard deviation) or associated estimates of uncertainty (e.g. confidence intervals) |
| ☐ | ☒ | For null hypothesis testing, the test statistic (e.g. *F*, *t*, *r*) with confidence intervals, effect sizes, degrees of freedom and *P* value noted *Give P values as exact values whenever suitable.* |
| ☒ | ☐ | For Bayesian analysis, information on the choice of priors and Markov chain Monte Carlo settings |
| ☒ | ☐ | For hierarchical and complex designs, identification of the appropriate level for tests and full reporting of outcomes |
| ☐ | ☒ | Estimates of effect sizes (e.g. Cohen's *d*, Pearson's *r*), indicating how they were calculated |

*Our web collection on statistics for biologists contains articles on many of the points above.*

## Software and code

Policy information about availability of computer code

**Data collection**   For gonococcal genome sequencing data obtained by the MiSeq sequencing system (Illumina), FastQC v0.11.5 (Andrews, S. (2010). FastQC: A Quality Control Tool for High Throughput Sequence Data. http://www.bioinformatics.babraham.ac.uk/projects/fastqc/) was used to analyze the quality of the raw sequencing data. The raw reads were trimmed by Sickle v1.33 (https://github.com/najoshi/sickle). SPAdes v3.10.1 (Bankevich, A., Nurk, S., Antipov, D., Gurevich, A.A., Dvorkin, M., Kulikov, A.S., Lesin, V.M., Nikolenko, S.I., Pham, S., Prjibelski, A.D., et al. (2012). SPAdes: a new genome assembly algorithm and its applications to single-cell sequencing. J Comput Biol 19, 455-477.) was used for the assembly of the raw reads.

**Data analysis**   For gene prediction and automatic annotation we employed Prokka (v1.12) (Seemann, T. (2014). Prokka: rapid prokaryotic genome annotation. Bioinformatics 30, 2068-2069.) and the NCBI Prokaryotic Genome Annotation Pipeline (PGAP release 5.2) (Tatusova, T., DiCuccio, M., Badretdin, A., Chetvernin, V., Nawrocki, E.P., Zaslavsky, L., Lomsadze, A., Pruitt, K.D., Borodovsky, M., and Ostell, J. (2016). NCBI prokaryotic genome annotation pipeline. Nucl Acids Res 44, 6614-6624.). For subsequent comparative genomic analysis, we used the PGAP annotation. The quality trimmed reads of the parent strain MS11 N309 and the two NQNO-resistant mutants MS11-R1 and MS11-R2 were aligned to the reference genome of N. gonorrhoeae MS11 (accession number NC_022240; https://www.ncbi.nlm.nih.gov/nuccore/ NC_022240.1/) with the BWA-MEM algorithm from the Burrows-Wheeler Aligner (BWA) software package (v0.7.17) (Li, H., and Durbin, R. (2010). Fast and accurate long-read alignment with Burrows-Wheeler transform. Bioinformatics 26, 589-595). The produced alignment was sorted with SortSam, and PCR duplicates were marked with MarkDuplicates using Picard (v2.17.3) (http://broadinstitute.github.io/picard/), for downstream analysis.
The Genome Analysis Toolkit's (GATK) (v3.8.0) RealignerTargetCreator and IndelRealigner tools were used to perform realignment around the Indels (McKenna, A., Hanna, M., Banks, E., Sivachenko, A., Cibulskis, K., Kernytsky, A., Garimella, K., Altshuler, D., Gabriel, S., Daly, M., et al. (2010). The Genome Analysis Toolkit: a MapReduce framework for analyzing next-generation DNA sequencing data. Genome Res 20, 1297-1303.). The GATK BaseRecalibrator was used to perform base quality score recalibration. Variant calling was then performed using the

GATK HaplotypeCaller with the default parameters except for the ploidy that was set to 1, according to GATK Best Practices. The sorted SNPs and InDels were then filtered using the GATK VariantFilteration with the filtering criteria recommended by GATK. To further remove sequencing bias only variants with a minimum read depth of 10 were considered. The effects of the variants were then annotated using SnpEff (v4.3s) (Cingolani, P., Platts, A., Wang le, L., Coon, M., Nguyen, T., Wang, L., Land, S.J., Lu, X., and Ruden, D.M. (2012). A program for annotating and predicting the effects of single nucleotide polymorphisms, SnpEff: SNPs in the genome of Drosophila melanogaster strain w1118; iso-2; iso-3. Fly (Austin) 6, 80-92).

For manuscripts utilizing custom algorithms or software that are central to the research but not yet described in published literature, software must be made available to editors and reviewers. We strongly encourage code deposition in a community repository (e.g. GitHub). See the Nature Portfolio guidelines for submitting code & software for further information.

## Data

Policy information about availability of data

All manuscripts must include a data availability statement. This statement should provide the following information, where applicable:
- Accession codes, unique identifiers, or web links for publicly available datasets
- A description of any restrictions on data availability
- For clinical datasets or third party data, please ensure that the statement adheres to our policy

The draft genome sequences of the three strains MS11_N309 (accession no. SAMN19108268), MS11-R1 (N568) (accession no. SAMN19108269) and MS11-R2 (N569) (accession no. SAMN19108270) were combined in Bioproject number PRJNA728975 and are publicly available from the NCBI GenBank. The accessible links are:
N. gonorrhoeae MS11-N309: https://www.ncbi.nlm.nih.gov/nuccore/JAHBBP000000000
N. gonorrhoeae MS11-R1: https://www.ncbi.nlm.nih.gov/nuccore/JAHBBO000000000
N. gonorrhoeae MS11-R2: https://www.ncbi.nlm.nih.gov/nuccore/JAHBBN000000000

As a reference genome, the publicly available genome of N. gonorrhoeae MS11 was used (accession number NC_022240; https://www.ncbi.nlm.nih.gov/nuccore/NC_022240.1/)
All other primary data are contained within the manuscript, including the extended supplementary information about chemical synthesis of all compounds and their quality control.

## Human research participants

Policy information about studies involving human research participants and Sex and Gender in Research.

| | |
|---|---|
| Reporting on sex and gender | *Use the terms sex (biological attribute) and gender (shaped by social and cultural circumstances) carefully in order to avoid confusing both terms. Indicate if findings apply to only one sex or gender; describe whether sex and gender were considered in study design whether sex and/or gender was determined based on self-reporting or assigned and methods used. Provide in the source data disaggregated sex and gender data where this information has been collected, and consent has been obtained for sharing of individual-level data; provide overall numbers in this Reporting Summary. Please state if this information has not been collected. Report sex- and gender-based analyses where performed, justify reasons for lack of sex- and gender-based analysis.* |
| Population characteristics | *Describe the covariate-relevant population characteristics of the human research participants (e.g. age, genotypic information, past and current diagnosis and treatment categories). If you filled out the behavioural & social sciences study design questions and have nothing to add here, write "See above."* |
| Recruitment | *Describe how participants were recruited. Outline any potential self-selection bias or other biases that may be present and how these are likely to impact results.* |
| Ethics oversight | *Identify the organization(s) that approved the study protocol.* |

Note that full information on the approval of the study protocol must also be provided in the manuscript.

# Field-specific reporting

Please select the one below that is the best fit for your research. If you are not sure, read the appropriate sections before making your selection.

☒ Life sciences   ☐ Behavioural & social sciences   ☐ Ecological, evolutionary & environmental sciences

For a reference copy of the document with all sections, see nature.com/documents/nr-reporting-summary-flat.pdf

# Life sciences study design

All studies must disclose on these points even when the disclosure is negative.

| | |
|---|---|
| Sample size | Based on initial experiments with the first generation of NQNO compounds, we calculated the effect size (Cohen's d) on recovered viable bacteria from NQNO-treated versus control (DMSO)-treated animals to be between 2 and 3. Based on an effect size of 3, we have performed |

an a priori power analysis with a minimum probability for the type I-error of p<0.05 (the probability that we reject H0 despite it is correct) to predict, how large a sample size we need to achieve this significance level. The a priori analysis indicated that the sample size should consist of at least 4 animals per group.

Data exclusions | No data were excluded

Replication | All experiments contained technical and biological replicates as detailed in Material&Methods and Figure legends.

Randomization | Animals were assigned randomly to the treatment or control group by drawing labeled cards from a blinded container

Blinding | Data collection and analysis were performed by experimentators, who were not blind to the conditions of the experiments.

# Reporting for specific materials, systems and methods

We require information from authors about some types of materials, experimental systems and methods used in many studies. Here, indicate whether each material, system or method listed is relevant to your study. If you are not sure if a list item applies to your research, read the appropriate section before selecting a response.

## Materials & experimental systems

| n/a | Involved in the study |
|---|---|
| ☐ | ☒ Antibodies |
| ☐ | ☒ Eukaryotic cell lines |
| ☒ | ☐ Palaeontology and archaeology |
| ☐ | ☒ Animals and other organisms |
| ☒ | ☐ Clinical data |
| ☒ | ☐ Dual use research of concern |

## Methods

| n/a | Involved in the study |
|---|---|
| ☒ | ☐ ChIP-seq |
| ☒ | ☐ Flow cytometry |
| ☒ | ☐ MRI-based neuroimaging |

## Antibodies

Antibodies used | The following primary and secondary antibodies were employed at the indicated dilutions for Western blotting (WB). Monoclonal antibody (mAB) against 6x His (clone H8; Thermo Fisher Scientific; 1:1.000 WB); mAb against GAPDH (clone GA1R, Invitrogen, 1:2.000 WB); mAb against GFP (clone JL-8, Clontech, Palo Alto, CA, 1:5000); mAb α-Opa (clone 4B12/C11, Developmental Studies Hybridoma Bank, University of Iowa, USA, a generous gift of M. Achtman, WB: 1:2000). Rabbit polyclonal antibody (pRb) against the synthetic epsilon1 peptide C-EKNRRMMTDEAFRKEVEKRLYAG was produced by PSL GmbH (Heidelberg, Germany), affinity purified against the cognate peptide and used at 1µg/ml for WB; pRb against the synthetic zeta1 peptides AKKEYSKQRVVTNSK-C and KIVGINQDRNSEFIDK-C was produced by PSL GmbH (Heidelberg, Germany) and affinity purified using both peptides (1µg/ml pRb was used for WB). HRP-conjugated goat anti-mouse IgG (cat-no: 115-035-146; 1:10.000 WB) and HRP-conjugated goat anti-rabbit IgG (cat-no: 111-035-003; 1:5.000 WB) were from Jackson Immunoresearch (West Grove, PA).

Validation | The polyclonal antisera against epsilon1 and zeta1 were affinity purified against the synthetic peptides used to raise the antibodies. For validation, the polyclonal antibodies against epsilon1/zeta1 were used in Western Blotting assays with bacterial lysates obtained from gonococcal strains lacking the toxin/antitoxin system (negative control) or expressing the toxin/antitoxin. The validation data are part of the manuscript.
For validation, the monoclonal antibodies against Opa proteins were used in Western Blotting assays with bacterial lysates obtained from gonococcal strains lacking Opa expression (negative control) or expressing a panel of different Opa proteins. The validation data have been published as part of a previous study (Roth A, Mattheis C, Muenzner P, Unemo M, Hauck CR. Innate recognition by neutrophil granulocytes differs between Neisseria gonorrhoeae strains causing local or disseminating infections. Infect Immun. 2013 Jul;81(7):2358-70. doi: 10.1128/IAI.00128-13).
The commercially available monoclonal antibodies have been validated by the suppliers.

## Eukaryotic cell lines

Policy information about cell lines and Sex and Gender in Research

Cell line source(s) | Primary human vaginal epithelial cells (hVEC line MS74) were obtained from A.J. Schaeffer (Feinberg School of Medicine, Northwestern University, Chicago, IL) and are derived from vaginal tissue of a post-menopausal woman. The cell line was created through immortalization of the cells with human papilloma virus 16, E6 and E7 genes according to (Rajan, N., Pruden, D.L., Kaznari, H., Cao, Q., Anderson, B.E., Duncan, J.L., and Schaeffer, A.J. (2000). Characterization of an immortalized human vaginal epithelial cell line. J Urol 163, 616-622).
HeLa S3 cells were obtained from DSMZ, Braunschweig, Germany (ACC 161) and ME-180 cells were obtained from ATCC, Rockville, MD (ATCC HTB-33)

Authentication | The authentication was performed at the source by immunostaining for cytokeratins (hVEC) or STR analysis according to the global standard ANSI/ATCC ASN-0002.1-2021 (HeLa S3 and ME-180)

| Mycoplasma contamination | All eukaryotic cell lines were screened in-house by a PCR-test for the presence of mycoplasma (Uphoff CC, Drexler HG (2005) Detection of mycoplasma contaminations. Methods Mol Biol 290: 13-23) and were tested to be Mycoplasma free. |
|---|---|
| Commonly misidentified lines (See ICLAC register) | no commonly misidentified cell lines were used. |

# Animals and other research organisms

Policy information about studies involving animals; ARRIVE guidelines recommended for reporting animal research, and Sex and Gender in Research

| Laboratory animals | C57BL/6J mice carrying the complete human CEA gene (CEAtg mice) (Eades-Perner, A.M., van der Putten, H., Hirth, A., Thompson, J., Neumaier, M., von Kleist, S., and Zimmermann, W. (1994). Mice transgenic for the human carcinoembryonic antigen gene maintain its spatiotemporal expression pattern. Cancer Res. 54, 4169–4176.) and wildtype C57BL/6J mice (obtained from Elevage Janvier, Le Genest Saint Isle, France) were kept under specified pathogen-free conditions under a 12-h light cycle in the animal facility of University of Konstanz in accordance with national and institutional guidelines. Only female mice between 6 – 8 weeks of age were used. |
|---|---|
| Wild animals | The study did not involve wild animals |
| Reporting on sex | Only female mice were used, as in vivo infections were studied in a vaginal infection model. |
| Field-collected samples | The study did not involve samples collected from the field |
| Ethics oversight | Experiments involving animals were performed in accordance with the German Law for the Protection of Animal Welfare. The animal care and use protocol, including the protocol of experimental vaginal infection of female mice, was approved by the appropriate state ethics committee and state authorities regulating animal experiments (Regierungspraesidium Freiburg, Germany) under the permit file number G-19/147. |

Note that full information on the approval of the study protocol must also be provided in the manuscript.

