## [Peer Review File · Nature Microbiology]

A quinolone N-oxide antibiotic selectively targets *Neisseria gonorrhoeae* via its toxin-antitoxin system

Corresponding Author: Professor Christof Hauck

Version 0:

Reviewer comments:

Reviewer #1

(Remarks to the Author)

With this manuscript, Boettcher, Hauck, and colleagues provide surprising and convincing evidence on a highly specific antimicrobial activity of 2-alkyl-4-quinolones N-oxides (AQNOs) against 25.2 MDa pTetM plasmid carrying *Neisseria gonorrhoea* variants. The inhibition of respiratory chain cytochromes by AQNOs, as the authors describe, has been described in several reports in the past. However, it is new and exciting, that downstream effects of this interference can exert strain specific lethality in *N. gonorrhoea* but not against other neisserial species. In a convincing approach they started with *Pseudomonas aeruginosa* / *N. gonorrhoea* battle experiments enabling identification of AQ(NO)s as anti-gonococcal compounds secreted by *Pseudomonas aeruginosa*. Synthetic compounds were produced, their bacteriostatic or bactericidal activity was monitored and NQNO was shown to be the most potent candidate. As expected, NQNO treatment caused massive perturbation of cytosolic ATP and NADH levels accompanied with ROS production in gonococcal strains but not in *N. cinerea* which they showed to be resistant against AQ(NO)s. By generating NQNO resistant strains which in fact have lost the entire pTetM plasmid, they concluded that the ROS dependent activation of Type II epsilon zeta TA systems is required for gonococcal sensitivity. In fact, clinical variants of *N. gonorrhoea* strains that they identified to be resistant to NQNO lacked the encoding locus. Although ROS dependent activation of Type II TA systems has been reported in the past, this is, to my knowledge, the first time that unleashing a toxin from its antitoxin eventually would tip the scale in favor of exerting a bactericidal phenotype. Most importantly, they established synthesis of a highly potent NQNO lead compound, demonstrate anti-gonococcal efficacy of NQNOs in a mouse model and provided first insights for a potential therapeutic application of their compounds.

In general, the experimental work is well designed, the manuscript is conclusive and convincing. However, as described below, some aspects should be reconsidered in a more appropriate and/or careful way. If resolved, this would strengthen the impact of the manuscript.

Comment 1) The genetic load region of the 25.2 MDa pTetM plasmid contains two paralogues of the epsilon/zeta system and the authors only demonstrate activation of the epsilon_1/zeta_1 system. Is zeta_2 unleashed similarly upon ROS dependent degradation of epsilon_2? If the authors only provide experimental evidence on the mechanism of activation of zeta_1, care should be taken when generalizing their observation. At least, it would be better if they refer to epsilon_1/zeta_1 throughout the manuscript.

Comment 2) Is NQNO sensitivity of *N. gonorrhoea* indeed exclusively the result of zeta_1 toxin activation? In fact, loss of the entire 25.2 MDa pTetM plasmid was observed when raising strains resistant to NQNO and clinical variants that proved not to be sensitive to their compounds most likely do also lack the plasmid. Can the author exclude that sensitivity to their quinolone compounds is established from the plasmid by any other means than the zeta_1 activity? In fact, TA systems were originally described as plasmid addiction modules that ensure stable plasmid maintenance. It would be interesting to see, whether a catalytically inactive zeta_1 toxin encoded from a stably maintained plasmid also gives rise to resistance. In fact, sensitivity could be due to synergistic effects and not just by zeta_1 toxin unleashing as the authors try to suggest.

Comment 3) Mechanism of action of zeta toxins: Zeta toxins have been shown to phosphorylate a number of cytosolic, UDP-activated sugar compounds and were proposed to exert toxicity by interfering with bacterial peptidoglycan synthesis in *Neisseria gonorrhoea*. However, a zeta toxin from *Streptococcus pyogenes* was proposed to be a kind of UDP-N-acetylglucosamine stimulated ATP hydrolase (for instance: <https://doi.org/10.3389/fmicb.2017.01130>). On the other hand, a zeta toxin has been identified in eukaryotic *Leishmania donovani* recently (<https://doi.org/10.1002/1873-3468.13429>), an organism that does not produce peptidoglycan at all. Although the authors do not go much into detail when describing the observed gonococcal phenotype, it would be appropriated and fair if the authors would occasionally discuss their observation with a more wholistic view on existing literature.

Comment 4) Are there any unwanted off-target effects of NQNO in eukaryotes? The authors decided for topical application of NQNO in mice which apparently did not exert any detrimental effect to the mucosa and the entire genital track. Furthermore,

NQMO was tested for potential off-target effects on vaginal epithelial cells as well as cervix carcinoma cells in vitro. Can the authors report on pharmacological parameters that would exclude for instance oral administration? For instance, what is the lethal dose for mice or how strong is the hepatic clearance in model systems? Although not essential for the general conclusions drawn in the manuscript, it would give a better view on the potential of NQNOs in anti-gonococcal therapy.

Reviewer #2

(Remarks to the Author)

I'm glad for the opportunity to review "Quinolone-N-oxides kill multidrug resistant *N. gonorrhoeae* by unleashing the endogenous zeta toxin" by Mix et al. The authors have shown that 2-nonyl-4-quinolone N-oxide might have strong antimicrobial activity and specially against *Neisseria gonorrhoeae*. The manuscript is very comprehensive with some intriguing findings.

Minor comments:

Please replace the WHO, 2016 reference with "Global progress report on HIV, viral hepatitis and sexually transmitted infections, 2021. May 20, 2021".

The same for da Costa-Lourenco et al., 2017 – replace with Unemo, M., Seifert, H. S., Hook, E. W., 3rd, Hawkes, S., Ndowa, F., & Dillon, J. R. (2019). *Gonorrhoea*. Nature reviews. Disease primers, 5(1), 79.

Row 81-85: Please revise though the conjugative plasmid carrying TetM is not present in a large fraction in all *Neisseria gonorrhoeae* but can differ a lot between regions. The authors should revise the sentence to that it is present in large fractions in some countries. This is also stated by Cehovin et al.

Authors should include a reference for Row 102-103 stating "Another group of AQ derivatives comprises the 2-alkyl-4-quinolone N-oxides (AQNOs), which do not function as signalling molecules in *Pseudomonas*."

Row 104: Print out the abbreviation for HQNO and NQNO when mentioned in text for the first time.

Row 512-513: Is there a reason the authors used OD550 for *Neisseria* spp?

Row 533: Please include the source of the NQNO. If this was synthesized, please include a section for this.

Row 564: Is this OD600?

Row 630: I suggest that the authors refer to the 500 bp paired-end to have instead 250 bp paired-end reads.

Genome determination and comparison section: The authors is strongly advised to upload the raw fastq files to the Bioproject as well.

Row 135: Authors state that they tested other "pathogenic *Neisseriae*". But I can't see that the authors have tested *P. aeruginosa* factors against *N. meningitidis*. Why is this?

Figure 1A: The authors refer to this electron microscopy scan to conclude that *Neisseria cinerea* is not affected by PAO1. How can the reader discern this? To me, both strains are affected in a similar way including intact cells. Furthermore, I imagine that the authors have chosen a very well representative, even the best, scan photo.

Figure S1. Why are different strains used in the different experiments? I.e in figure S1A the authors have used MS11, while in S1B and S1C they have used Ngo 11. I would rather that the authors include all tested *N. gonorrhoeae* isolates in the figure.

The authors need to include figure texts for supplementary figures.

Row 163-165: Pefloxacin is not a relevant antibiotic for the treatment of *Neisseria gonorrhoeae*. Please include ciprofloxacin MICs instead.

Major comments

The manuscript includes an unreasonable number of different figures. As reader it is very confusing when the authors use A to even H of unrelated figures. E.g how is figure 1A related to 1B? The authors could revise these as e.g figure 1 and 2 instead. This should be applied to all figures.

The authors are very clear that *N. cinerea* is resistant to NQNO. However, the authors use *N. cinerea* data to draw conclusions for all commensal *Neisseria*. This should be revised and clear that NQNO does affect commensals in various degrees, eg see row 198-200.

The authors could include limitations of the study in Discussion, eg the authors suggest that NQNO could be used via local vaginal administration. However, authors have not tested many of the bacteria in the vaginal microbiota. The authors could still include this hypothesis but include the limitations of the study. Furthermore, using antibacterial compounds freely, especially prior to exposure could lead to rapid resistance in the gonococcal population. The authors can include and discuss around this possibility and the importance of antibiotic stewardship.

Finally, although the manuscript is well written, it draws very large conclusions that would need further studies with a larger, up-to-date collection of gonococcal isolates. The tested isolates are mainly lab strains or isolates no longer circulating in the gonococcal population. The gonococcal population have shifted towards azithromycin resistance (due to change in treatment recommendations) and it is worrying that the WHO P (azithromycin resistant strain that have clonally expanded) is resistant to NQNO. The authors could mention how this can be addressed in future studies.

Reviewer #3

(Remarks to the Author)

The *P. aeruginosa* alkyl quinolone N-oxides (AQNOs) were originally discovered as potential antibiotics in the 1950s. Their low in vivo antibacterial efficacy with activity primarily against Gram positives combined with their cytotoxicity for eukaryotic cells (electron transport inhibition) generally prevented the development of AQNOs as therapeutic agents. Mix et al make some interesting observations on the selective susceptibility of *Neisseria gonorrhoeae* for NQNO compared with the resistance of commensal *Neisseria* strains. The selectivity observed was due to the release of a zeta toxin from its cognate antitoxin, the

genes for which are carried on a large pTetM conjugative plasmid commonly found in gonococci but not in commensal *Neisseria*. In a female mouse genital tract colonization and infection model, topical NQNO blocked colonization. In vitro screening of a selection of modified AQNOs yielded a 3Me-substituted analogue with nanomolar efficacy. Given that *N. gonorrhoeae* is a major global public health concern and a World Health Organization (WHO) priority antibiotic-resistant pathogen, this is an important topic given the urgent need for new therapeutics for the treatment and prevention of gonorrhoea.

Specific points

1. Line 151 and 191. Why did the authors select the C9 compound NQNO rather than the more commonly investigated C7 compound HQNO? Was there any difference in potency?
2. Fig. S1. (C), the inhibition data for the *P. aeruginosa* pqsH mutant is surprising given that it still makes AQNOs. Although the pqsL mutant doesn't make AQNOs it will still produce NHQ which the authors show has very good activity against *N. gonorrhoeae*.
3. Lines 217-219. Why did the authors only treat the hVEC and HeLa cells with a maximum of 10 µM which is very close to the gonococcal MIC of 5 µM and well below the therapeutic dose that would require administration. The claim (line 117) that effective doses do not harm epithelial cells in vitro should be qualified given the low concentrations tested.
4. Lines 274 to 276. There are two similar zeta/epsilon toxin-antitoxin systems on the plasmid, why did the authors only focus on the zeta 1/epsilon 1 system since both would be lost when the plasmid was lost during NQNO conditioning and conversely restored when the plasmid was re-introduced. Deletion of each system individually would be informative. It is also possible that other conjugative plasmid genes are involved. Why not introduce the zeta 1/epsilon 1 system alone on a plasmid into a resistant gonococcus strain?
5. Line 294. The Syto9 and PI fluorescence in Figs 4B and 4F should be quantified.
6. Lines 311-313 also 338-339. The authors cannot rule out from the data presented that NQNO may act on both the electron chain and directly on one or both of the TA systems.
7. Line 361-2 and Fig. 6C. The authors state that NQNO eradicated the gonococci below the level that could be detected by microscopy. However, there is a very clear green signal towards the top of the panel
8. Line 370. Fig 6E should be 6F.
9. Lines 373-374. The authors only tested one strain in vivo so cannot claim NQNO also suppressed viability of extended drug resistant isolates in vivo.
10. Line 376. There is little in the way of a systematic SAR. Given the increased potency of the 3Me AQNO derivative, the synthesis of further 3 position analogues would be informative as would a shorter 2 position C9 alkyl chain to make the compound more drug-like.
11. Line 396-397. The authors only tested 14 repeat subcultures for increased resistance which here could relate either to loss of the conjugative plasmid or a mutation in the target of NQNO – probably the quinone reduction site cytochrome bc1 sub-unit. Many additional subcultures are likely to be required for resistance to emerge.

Decision Letter:

29th March 2023

Dear Professor Hauck,

Thank you for your patience while your manuscript "Quinolone-N-oxides kill multidrug resistant *N. gonorrhoeae* by unleashing the endogenous zeta toxin" was under peer-review at Nature Microbiology. It has now been seen by 3 referees, whose expertise and comments you will find at the of this email. You will see from their comments below that while they find your work of interest, some important points are raised. We are very interested in the possibility of publishing your study in Nature Microbiology, but would like to consider your response to these concerns in the form of a revised manuscript before we make a final decision on publication.

In particular, you will see that several referees raise ask for further validation of which of the two zeta-epsilon toxin-antitoxin systems encoded on pTetM were involved in the bactericidal effects of NQNO, or whether any additional plasmid-encoded factors might also contribute. Based on comments from Referee #2, we would encourage you to test whether NQNO is effective against clinically relevant azithromycin-resistant *N. gonorrhoeae* strains if possible. In subsequent discussion, one referee also found the mechanism driving NQNO-induced antitoxin degradation unclear and questioned whether this was due to ROS or a direct effect of NQNO upon the antitoxin. We would also encourage you to posit more of a mechanism of action, if possible. The rest of the issues outlined in the referees' reports are clear and should be straightforward to address.

Please include a data availability statement as a separate section after Methods but before references, under the heading "Data Availability". This section should inform readers about the availability of the data used to support the conclusions of your study. This information includes accession codes to public repositories (data banks for protein, DNA or RNA sequences, microarray,

proteomics data etc...), references to source data published alongside the paper, unique identifiers such as URLs to data repository entries, or data set DOIs, and any other statement about data availability. At a minimum, you should include the following statement: "The data that support the findings of this study are available from the corresponding author upon request", mentioning any restrictions on availability. If DOIs are provided, we also strongly encourage including these in the Reference list (authors, title, publisher (repository name), identifier, year). For more guidance on how to write this section please see: <http://www.nature.com/authors/policies/data/data-availability-statements-data-citations.pdf>

* If you have not done so already we suggest that you begin to revise your manuscript so that it conforms to our Article format instructions at <http://www.nature.com/nmicrobiol/info/final-submission>. Refer also to any guidelines provided in this letter.

When submitting the revised version of your manuscript, please pay close attention to our [href="https://www.nature.com/nature-portfolio/editorial-policies/image-integrity">Digital Image Integrity Guidelines](https://www.nature.com/nature-portfolio/editorial-policies/image-integrity) and to the following points below:

Link Redacted

Note: This url links to your confidential homepage and associated information about manuscripts you may have submitted or be reviewing for us. If you wish to forward this e-mail to co-authors, please delete this link to your homepage first.

Nature Microbiology is committed to improving transparency in authorship. As part of our efforts in this direction, we are now requesting that all authors identified as 'corresponding author' on published papers create and link their Open Researcher and Contributor Identifier (ORCID) with their account on the Manuscript Tracking System (MTS), prior to acceptance. This applies to primary research papers only. ORCID helps the scientific community achieve unambiguous attribution of all scholarly contributions. You can create and link your ORCID from the home page of the MTS by clicking on 'Modify my Springer Nature account'. For more information please visit [please visit www.springernature.com/orcid](http://www.springernature.com/orcid).

If you wish to submit a suitably revised manuscript we would hope to receive it within 6 months. If you cannot send it within this time, please let us know. We will be happy to consider your revision, even if a similar study has been accepted for publication at Nature Microbiology or published elsewhere (up to a maximum of 6 months).

Yours sincerely,

Reviewer Expertise:

Referee #1:
Referee #2:
Referee #3:

Reviewer Comments:

Reviewer #1 (Remarks to the Author):

With this manuscript, Boettcher, Hauck, and colleagues provide surprising and convincing evidence on a highly specific antimicrobial activity of 2-alkyl-4-quinolones N-oxides (AQNOs) against 25.2 MDa pTetM plasmid carrying *Neisseria gonorrhoea*

variants. The inhibition of respiratory chain cytochromes by AQNOs, as the authors describe, has been described in several reports in the past. However, it is new and exciting, that downstream effects of this interference can exert strain specific lethality in *N. gonorrhoea* but not against other neisserial species. In a convincing approach they started with *Pseudomonas aeruginosa* / *N. gonorrhoea* battle experiments enabling identification of AQ(NO)s as anti-gonococcal compounds secreted by *Pseudomonas aeruginosa*. Synthetic compounds were produced, their bacteriostatic or bactericidal activity was monitored and NQNO was shown to be the most potent candidate. As expected, NQNO treatment caused massive perturbation of cytosolic ATP and NADH levels accompanied with ROS production in gonococcal strains but not in *N. cinerea* which they showed to be resistant against AQ(NO)s. By generating NQNO resistant strains which in fact have lost the entire pTetM plasmid, they concluded that the ROS dependent activation of Type II epsilon zeta TA systems is required for gonococcal sensitivity. In fact, clinical variants of *N. gonorrhoea* strains that they identified to be resistant to NQNO lacked the encoding locus. Although ROS dependent activation of Type II TA systems has been reported in the past, this is, to my knowledge, the first time that unleashing a toxin from its antitoxin eventually would tip the scale in favor of exerting a bactericidal phenotype. Most importantly, they established synthesis of a highly potent NQNO lead compound, demonstrate anti-gonococcal efficacy of NQNOs in a mouse model and provided first insights for a potential therapeutic application of their compounds.

In general, the experimental work is well designed, the manuscript is conclusive and convincing. However, as described below, some aspects should be reconsidered in a more appropriate and/or careful way. If resolved, this would strengthen the impact of the manuscript.

Comment 1) The genetic load region of the 25.2 MDa pTetM plasmid contains two paralogues of the epsilon/zeta system and the authors only demonstrate activation of the epsilon_1/zeta_1 system. Is zeta_2 unleashed similarly upon ROS dependent degradation of epsilon_2? If the authors only provide experimental evidence on the mechanism of activation of zeta_1, care should be taken when generalizing their observation. At least, it would be better if they refer to epsilon_1/zeta_1 throughout the manuscript.

Comment 2) Is NQNO sensitivity of *N. gonorrhoea* indeed exclusively the result of zeta_1 toxin activation? In fact, loss of the entire 25.2 MDa pTetM plasmid was observed when raising strains resistant to NQNO and clinical variants that proved not to be sensitive to their compounds most likely do also lack the plasmid. Can the author exclude that sensitivity to their quinolone compounds is established from the plasmid by any other means than the zeta_1 activity? In fact, TA systems were originally described as plasmid addiction modules that ensure stable plasmid maintenance. It would be interesting to see, whether an catalytically inactive zeta_1 toxin encoded from a stably maintained plasmid also gives rise to resistance. In fact, sensitivity could be due to synergistic effects and not just by zeta_1 toxin unleashing as the authors try to suggest.

Comment 3) Mechanism of action of zeta toxins: Zeta toxins have been shown to phosphorylate a number of cytosolic, UDP-activated sugar compounds and were proposed to exert toxicity by interfering with bacterial peptidoglycan synthesis in *Neisseria gonorrhoea*. However, a zeta toxin from *Streptococcus pyogenes* was proposed to be a kind of UDP-N-acetylglucosamine stimulated ATP hydrolase (for instance: <https://doi.org/10.3389/fmicb.2017.01130>). On the other hand, a zeta toxin has been identified in eukaryotic *Leishmania donovani* recently (<https://doi.org/10.1002/1873-3468.13429>), an organism that does not produce peptidoglycan at all. Although the authors do not go much into detail when describing the observed gonococcal phenotype, it would be appropriated and fair if the authors would occasionally discuss their observation with a more wholistic view on existing literature.

Comment 4) Are there any unwanted off-target effects of NQNO in eukaryotes? The authors decided for topical application of NQNO in mice which apparently did not exert any detrimental effect to the mucosa and the entire genital track. Furthermore, NQNO was tested for potential off-target effects on vaginal epithelial cells as well as cervix carcinoma cells in vitro. Can the authors report on pharmacological parameters that would exclude for instance oral administration? For instance, what is the lethal dose for mice or how strong is the hepatic clearance in model systems? Although not essential for the general conclusions drawn in the manuscript, it would give a better view on the potential of NQNOs in anti-gonococcal therapy.

Reviewer #2 (Remarks to the Author):

I'm glad for the opportunity to review "Quinolone-N-oxides kill multidrug resistant *N. gonorrhoeae* by unleashing the endogenous zeta toxin" by Mix et al. The authors have shown that 2-nonyl-4-quinolone N-oxide might have strong antimicrobial activity and specially against *Neisseria gonorrhoeae*. The manuscript is very comprehensive with some intriguing findings.

Minor comments:

Please replace the WHO, 2016 reference with "Global progress report on HIV, viral hepatitis and sexually transmitted infections, 2021. May 20, 2021".

The same for da Costa-Lourenco et al., 2017 – replace with Unemo, M., Seifert, H. S., Hook, E. W., 3rd, Hawkes, S., Ndowa, F., & Dillon, J. R. (2019). Gonorrhoea. Nature reviews. Disease primers, 5(1), 79.

Row 81-85: Please revise though the conjugative plasmid carrying TetM is not present in a large fraction in all *Neisseria gonorrhoeae* but can differ a lot between regions. The authors should revise the sentence to that it is present in large fractions in some countries. This is also stated by Cehovin et al.

Authors should include a reference for Row 102-103 stating "Another group of AQ derivatives comprises the 2-alkyl-4-quinolone N-oxides (AQNOs), which do not function as signalling molecules in *Pseudomonas*."

Row 104: Print out the abbreviation for HQNO and NQNO when mentioned in text for the first time.

Row 512-513: Is there a reason the authors used OD550 for *Neisseria* spp?

Row 533: Please include the source of the NQNO. If this was synthesized, please include a section for this.

Row 564: Is this OD600?

Row 630: I suggest that the authors refer to the 500 bp paired-end to have instead 250 bp paired-end reads.
Genome determination and comparison section: The authors is strongly advised to upload the raw fastq files to the Bioproject as well.
Row 135: Authors state that they tested other "pathogenic Neisseriae". But I can't see that the authors have tested *P. aeruginosa* factors against *N. meningitidis*. Why is this?
Figure 1A: The authors refer to this electron microscopy scan to conclude that *Neisseria cinerea* is not affected by PAO1. How can the reader discern this? To me, both strains are affected in a similar way including intact cells. Furthermore, I imagine that the authors have chosen a very well representative, even the best, scan photo.
Figure S1. Why are different strains used in the different experiments? I.e in figure S1A the authors have used MS11, while in S1B and S1C they have used Ngo 11. I would rather that the authors include all tested *N. gonorrhoeae* isolates in the figure. The authors need to include figure texts for supplementary figures.
Row 163-165: Pefloxacin is not a relevant antibiotic for the treatment of *Neisseria gonorrhoeae*. Please include ciprofloxacin MICs instead.

Major comments

The manuscript includes an unreasonable number of different figures. As reader it is very confusing when the authors use A to even H of unrelated figures. E.g how is figure 1A related to 1B? The authors could revise these as e.g figure 1 and 2 instead. This should be applied to all figures.
The authors are very clear that *N. cinerea* is resistant to NQNO. However, the authors use *N. cinerea* data to draw conclusions for all commensal *Neisseria*. This should be revised and clear that NQNO does affect commensals in various degrees, eg see row 198-200.
The authors could include limitations of the study in Discussion, eg the authors suggest that NQNO could be used via local vaginal administration. However, authors have not tested many of the bacteria in the vaginal microbiota. The authors could still include this hypothesis but include the limitations of the study. Furthermore, using antibacterial compounds freely, especially prior to exposure could lead to rapid resistance in the gonococcal population. The authors can include and discuss around this possibility and the importance of antibiotic stewardship.

Finally, although the manuscript is well written, it draws very large conclusions that would need further studies with a larger, up-to-date collection of gonococcal isolates. The tested isolates are mainly lab strains or isolates no longer circulating in the gonococcal population. The gonococcal population have shifted towards azithromycin resistance (due to change in treatment recommendations) and it is worrying that the WHO P (azithromycin resistant strain that have clonally expanded) is resistant to NQNO. The authors could mention how this can be addressed in future studies.

Reviewer #3 (Remarks to the Author):

The *P. aeruginosa* alkyl quinolone N-oxides (AQNOs) were originally discovered as potential antibiotics in the 1950s. Their low in vivo antibacterial efficacy with activity primarily against Gram positives combined with their cytotoxicity for eukaryotic cells (electron transport inhibition) generally prevented the development of AQNOs as therapeutic agents. Mix et al make some interesting observations on the selective susceptibility of *Neisseria gonorrhoea* for NQNO compared with the resistance of commensal *Neisseria* strains. The selectivity observed was due to the release of a zeta toxin from its cognate antitoxin, the genes for which are carried on a large pTetM conjugative plasmid commonly found in gonococci but not in commensal *Neisseria*. In a female mouse genital tract colonization and infection model, topical NQNO blocked colonization. In vitro screening of a selection of modified AQNOs yielded a 3Me-substituted analogue with nanomolar efficacy. Given that *N. gonorrhoeae* is a major global public health concern and a World Health Organization (WHO) priority antibiotic-resistant pathogen, this is an important topic given the urgent need for new therapeutics for the treatment and prevention of gonorrhoea.

Specific points

1. Line 151 and 191. Why did the authors select the C9 compound NQNO rather than the more commonly investigated C7 compound HQNO? Was there any difference in potency?
2. Fig. S1. (C), the inhibition data for the *P. aeruginosa* pqsH mutant is surprising given that it still makes AQNOs. Although the pqsL mutant doesn't make AQNOs it will still produce NHQ which the authors show has very good activity against *N. gonorrhoeae*.
3. Lines 217-219. Why did the authors only treat the hVEC and Hela cells with a maximum of 10 microM which is very close to the gonococcal MIC of 5 microM and well below the therapeutic dose that would require administration. The claim (line 117) that effective doses do not harm epithelial cells in vitro should be qualified given the low concentrations tested.
4. Lines 274 to 276. There are two similar zeta/epsilon toxin-antitoxin systems on the plasmid, why did the authors only focus on the zeta 1/epsilon 1 system since both would be lost when the plasmid was lost during NQNO conditioning and conversely restored when the plasmid was re-introduced. Deletion of each system individually would be informative. It is also possible that other conjugative plasmid genes are involved. Why not introduce the zeta 1/epsilon 1 system alone on a plasmid into a resistant gonococcus strain?
5. Line 294. The Syto9 and PI fluorescence in Figs 4B and 4F should be quantified.
6. Lines 311-313 also 338-339. The authors cannot rule out from the data presented that NQNO may act on both the electron chain and directly on one or both of the TA systems.
7. Line 361-2 and Fig. 6C. The authors state that NQNO eradicated the gonococci to below the level that could be detected by microscopy. However, there is a very clear green signal towards the top of the panel
8. Line 370. Fig 6E should be 6F.
9. Lines 373-374. The authors only tested one strain in vivo so cannot claim NQNO also suppressed viability of extended drug

resistant isolates in vivo.

10. Line 376. There is little in the way of a systematic SAR. Given the increased potency of the 3Me AQNO derivative, the synthesis of further 3 position analogues would be informative as would a shorter 2 position C9 alkyl chain to make the compound more drug-like.

11. Line 396-397. The authors only tested 14 repeat subcultures for increased resistance which here could relate either to loss of the conjugative plasmid or a mutation in the target of NQNO – probably the quinone reduction site cytochrome bc1 sub-unit. Many additional subcultures are likely to be required for resistance to emerge.

Version 1:

Reviewer comments:

Reviewer #1

(Remarks to the Author)

The authors have added substantial and significant additional data to this manuscript addressing most of the reviewer's concern and provide convincing arguments in their rebuttal letter if not included.

I would like to congratulate the authors for this outstanding achievement.

Reviewer #2

(Remarks to the Author)

I would like to express my gratitude to the authors for their thoughtful and comprehensive responses to my comments. I appreciate the effort and diligence applied to address the points raised. However, I have a few additional comments.

Row 135 comment: Please include a sentence or two in regards of limitations in the discussion section regarding *N. meningitidis*.

Figure S1B – Weren't all the isolates in Table S1 subjected to this methodology? If that's the case, it would be prudent to extend the supplementary material by an additional two pages, despite it already surpassing 50 pages. Providing the inhibition zones for all strains would significantly enhance the comprehensiveness of our data. A dedicated table detailing these inhibition zones would be invaluable. Lastly, for clarification, the label in Figure B should read '*N. gonorrhoeae* MS11,' not '*N. gonorrhoeae* 11,' correct?

Row 163-165: This reviewer notes that the authors do not advocate for pefloxacin as a recommended treatment. The authors state:

"Strikingly, 2-nonyl-4-quinolone (NQ), trans- Δ 1 162 -2-nonyl-4-quinolone (trans- Δ 1 163 -NQ) and 2-nonyl-4-quinolone N-oxide (NQNO) completely inhibited growth of *N. gonorrhoeae* (Fig. 1C). While trans- Δ 1-NQNO was considerably less active, NQNO inhibited growth of *N. gonorrhoeae* even at the lowest concentration of 5 μ M (~ 1.5 μ g/ml) (Fig. 1C). This effective concentration is in the range of minimum inhibitory concentrations (MICs) observed for clinically relevant antibiotics such as pefloxacin or spectinomycin (Bengtsson-Palme and Larsson, 2016)."

When the authors use the term "clinically relevant" alongside a genuinely clinically relevant antibiotic for *N. gonorrhoeae*, such as spectinomycin, it may lead readers to infer that pefloxacin is/has been clinically relevant for gonorrhoea treatment. Please clarify this statement.

Final comment: The isolates that have undergone clonal expansion in recent years are the WHO P-like strains. Meanwhile, high-level resistance to azithromycin appears more sporadically. This doesn't necessarily suggest that the resistance mechanisms to azithromycin are causing the decreased susceptibility to NQNO. It may instead indicate that the WHO P-like genomic backbone, with its myriad traits, is also proliferating. This backbone might encompass a consistent TA system. That said, this reviewer is not advocating for additional experiments. Instead, it would be beneficial if the authors could address this point in the discussion section.

Reviewer #3

(Remarks to the Author)

The authors have addressed all my key points and added new data to support their conclusions with respect to the second toxin/antitoxin system, the stability of the antitoxin, quantification of membrane damage and extended the concentrations and cell lines used to evaluate toxicity of the quinolone N-oxides. Although the precise molecular basis by which the quinolone N-oxides release the anti-toxin from the toxin remains to be established, the authors have made a very novel and interesting finding of broad interest with significant translational potential in the AMR context.

Minor points:

Line 104. Evidence that the AQNOs are not signal molecules in *Pseudomonas* was directly addressed in Rampioni et al (2016) PLOS Pathogens doi: 10.1371/journal.ppat.1006029 rather in the two Szamosvaró et al papers cited.

Lines 348 and 351 and Fig. 6F. 'disappearance' in these sentences is incorrect. The epsilon antitoxin is reduced not absent from these blots.

Fig. 6 F and G. There is a band on the zeta 1 blots marked with a star. This is presumably a protein that cross-reacts with the antibody used? This should be explained in the legend.

Reviewer #4

(Remarks to the Author)

The manuscript by Mix et al investigates the activity of 2-nonyl-4-quinolone N-oxide (NQNO) against *N. gonorrhoeae*. The study reveals that NQNO demonstrates anti-gonococcal activity without impairing growth of commensal *Neisseria* or viability of human cells. In vivo, application of NQNO resulted in reducing the bacterial burden in the vaginal tract. Mechanistically, NQNO was found to disrupt the gonococcal electron transport chain with the release of zeta toxin from its epsilon antitoxin. Chemical modification of NQNO yielded a more potent derivative. While most of the experimental work is well designed, and the manuscript is well written, I have outlined some comments and critiques below that should be considered to enhance the manuscript's impact

Lines 96-97: "Most antibiotics in current use are derived from bacterial secondary metabolites, which have evolved in the frame of microbial competition." This sentence requires supporting references to substantiate the claim.

Line 127: please correct the typo to "of this global of concern pathogen".

Fig. S1C: difference is small

Lines 172-174: "a highly consistent growth inhibition of gonococci by NQNO concentrations > 2.5 μM was observed (Suppl. Fig. S2A)". According to Suppl. Fig. S2A, HQNO was not tested at the concentration of 2.5 μM (the lowest concentration tested was 5 μM). So, this sentence should be rephrased to "a highly consistent growth inhibition of gonococci by NQNO concentrations \geq 5 μM

Fig. 3D: there are no standard deviation values shown in the figure.

Lines 280-281: "two zeta-epsilon (ζ_1/ϵ_1 and ζ_2/ϵ_2) toxin/anti-toxin (TA) systems are encoded on this extrachromosomal DNA".

Why was ζ_1/ϵ_1 system the only system investigated and ζ_2/ϵ_2 was not investigated?

Lines 309-311: "The introduction of the zeta/epsilon-encoding plasmid rendered the resulting pTetM-positive clones sensitive to NQNO (Fig. 4F)". As shown in Fig. 4F, it seems that the growth of MS11-R2 pTetM A and MS11-R2 pTetM B is normally reduced (OD550 of the negative control (DMSO) did not exceed 0.4 as compared to about 1.4 for MS11-R2). In presence of HQNO, the OD550 of MS11-R2 pTetM A and MS11-R2 pTetM B is increasing by time and it reaches values comparable to that of MS11-R2. A similar trend is shown for MS11-R2 (growth increases by time with HQNO treatment). Consequently, it is not reasonable to conclude that "The introduction of the zeta/epsilon-encoding plasmid (pTetM A and pTetM B) rendered the resulting pTetM-positive clones sensitive to NQNO". The incubation time should be extended beyond 8 hours to observe potential differences between the conjugants and the parent strains. Furthermore, given the low OD550 values observed in the case of the conjugant strains, relying solely on OD values for drawing conclusions may not be sufficient. It is essential to include CFU (colony-forming unit) counts, as even minor discrepancies in OD values may not necessarily reflect significant differences in CFU counts.

Lines 325-326: Similar to the previous comment, Fig. 5B shows that the growth of WHO P pTetM A, WHO P pTetM B, and WHO P ζ_1/ϵ_1 is normally reduced (OD550 of the negative control reaches up to 0.4-0.5 only as compared to about 1.0 for WHO P). In presence of HQNO, the OD550 of WHO P pTetM A, WHO P pTetM B, and WHO P ζ_1/ϵ_1 is increasing by time and it reaches values comparable to that of DMSO (~0.25-4). A similar trend is shown for WHO P (growth increases by time with HQNO treatment). Consequently, it is not reasonable to conclude that "Most importantly, Zeta1/Epsilon1-expressing derivatives of WHO P became NQNO-sensitive, while the parent strain was fully resistant up to 50 μM (Fig. 5B)". The incubation time should be extended beyond 8 hours to see if there will be significant difference between the conjugants and the parent strains. Also, since the OD550 values are low in case of the conjugant strains, we cannot depend on the OD values alone for drawing conclusions. CFU counts are important to be included since this small difference in OD values may not reflect a significant difference in the CFU counts.

Lines 341-347: The reason for switching to the use of *E. coli* instead of *N. gonorrhoeae* to investigate the degradative effect of reactive oxygen species on Epsilon antitoxin is not clear and should be clarified in the manuscript.

Fig. 6D illustrates that NQNO treatment did not impact the growth of *E. coli*. It's important to note that the outer membrane of *E. coli* differs from that of *N. gonorrhoeae*, which can lead to variations in the susceptibility of these bacteria to different agents. Some agents may exhibit inhibitory activity against *N. gonorrhoeae* without affecting *E. coli* or with reduced activity against *E. coli* due to permeability. This phenomenon is evident in the case of HQNO, which did not inhibit *E. coli* while displaying inhibitory effects on *N. gonorrhoeae*. Additionally, the results presented in Fig. S1 indicate that *N. gonorrhoeae* exhibits a larger inhibition zone compared to the minimal inhibition observed for *E. coli*. Therefore, the use of *E. coli* in this experiment may not support the conclusion that "NQNO does not act directly on Epsilon stability but rather acts indirectly via NQNO-induced oxidative stress." These experiments should ideally be conducted using *N. gonorrhoeae*, as the effects of NQNO on *E. coli* may be limited due to its reduced activity/permeability against this bacterium.

Fig. 6D: there are no standard deviation values shown in the figure.

Lines 360-362: The proposed mechanism of NQNO action is mediated by unleashing the endogenous Zeta1 toxin from inhibition by its cognate epsilon1 antitoxin, and the enzymatic activity of Zeta1 will lead to disruption of cell wall synthesis and integrity. To further substantiate this conclusion, additional assays demonstrating the disruption of cell wall synthesis and integrity in response to NQNO treatment should be included. Relying solely on electron microscopy may not be sufficient to confirm this mechanism conclusively.

I have several concerns regarding the animal experiment. Firstly, the treatment was administered after only 1 hour, which is a very short duration for *N. gonorrhoeae* to attach to the vaginal tract and establish infection. Additionally, this timeframe may not be clinically applicable. Topical application of *Neisseria* is not supported and is unlikely to clear *Neisseria* infections. Moreover, any topical treatment, including mild antiseptic solutions or mild H₂O₂, applied in this model topically is likely to result in a significant reduction in bacterial load.

Fig. 7B: Why is the number of mice inconsistent between groups? For instance, in the experiment of CEAtg mice, DMSO: 12 mice, NQNO (25 μ M): 6 mice, and NQNO (50 μ M): 9 mice. For wild-type mice, DMSO: 12 mice, 25 μ M NQNO: 4 mice, and 50 μ M NQNO: 5 mice. The same comment applies to the second mouse experiment (Fig. 7F).

No control antibiotics (e.g. ceftriaxone and azithromycin) were used in both mouse experiments.

Lines 417-419: "At concentrations of 0.5 μ M and higher, 3Me-NQNO was bactericidal and induced rapid killing of the pathogens (Fig. 8E)". Fig. 8F shows that at the concentrations of 0.5 μ M, 3Me-NQNO was bacteriostatic, and did not reduce the bacterial count. The concentration of 2.5 μ M resulted in reduction of the bacterial count by about 3 log₁₀ after 3 hours. So, at this concentration only, 3Me-NQNO can be considered bactericidal. Please, modify this sentence accordingly.

Fig. 8E and 8F: The figures do not show standard deviation values. How many replicates were used for these experiments?

Lines 422-424: Toxicity of 3Me-NQNO was measured up to 10 μ M while it showed activity at the concentrations of 2.5 – 5 μ M. To establish a high therapeutic index for the agent, cytotoxicity should be evaluated at higher concentrations, such as those several folds higher than the active concentration (e.g., 100 μ M or higher). This will help provide a more comprehensive assessment of the agent's safety profile and therapeutic potential.

Lines 494-495: "the experimental infection of female mice does not recapitulate gonorrhea and symptomatic disease has not been observed in this model". This information is inaccurate. *N. gonorrhoeae* female mouse model established by Jerse et al 1999 showed symptoms of gonorrhea such as: 1) Gonorrheal inflammation occurred in over 80% of infected mice, 2) *N. gonorrhoeae* was recovered from vaginal swabs for an average of 12 to 13 days following infection, 3) Vaginal smears showed presence of neutrophils recruited to the site of infection.

Lines 500-510: discuss the use of topically applied antimicrobials for sexually transmitted diseases, including *N. gonorrhoeae*. However, it's important to note that the case of *N. gonorrhoeae* is different due to specific criteria for anti-gonococcal therapeutics. For *N. gonorrhoeae*, patient compliance is crucial, and there are certain criteria that should be considered. These criteria include the route of administration, which should be either oral or intramuscular (IM), and the treatment duration. For urogenital *N. gonorrhoeae*, the treatment duration should be 1 day for the oral route and up to 3 days for anorectal and oropharyngeal *N. gonorrhoeae* (for the oral route), or 1 day for all types of *N. gonorrhoeae* infections (for the IM route). These criteria reflect the specific challenges and requirements for treating *N. gonorrhoeae* effectively.

I would recommend adding a paragraph discussing the limitations of the study such as the pre-existing resistance (presence of *N. gonorrhoeae* strains with resistance to NQNO e.g. WHO P and strategies to overcome this resistance).

A general comment on the materials and methods section: most of the experimental protocols presented do not have citations. References for each experiment need to be added.

Lines 629-635: "Determination of NADH levels". The experiment also lacks a positive control (an agent that is known to reduce the NADH levels in the bacteria) which provides the level of confidence in the results obtained and ensure that the results are reproducible and robust. The same comment applies to the experiments of "Detection of reactive oxygen species (ROS)"

Lines 647-662: "Selection for NQNO and 3Me-NQNO resistance". The proposed method requires citation, and the experiment also lacks a positive control (an agent that is known to be susceptible to resistance development such as rifampicin, tetracycline or ciprofloxacin) which confirms that the assay is right, and the results obtained are reproducible and robust. Additionally, in this method, bacteria are exposed daily to the same conditions (conditioning with 2.5 μ M NQNO or 0.1 μ M 3Me-NQNO for 8 h) and then evaluated for sensitivity towards different concentrations of NQNO or 3Me-NQNO. This technique may not enhance the development of bacterial resistant mutants since bacteria are exposed to exactly the same conditions every day. The most commonly used method is through serial passages where the bacterial inoculum is prepared from cultures with a subinhibitory concentration of the test agent (these concentrations are not the same every day and may be different according to the MICs of each passage).

Lines 898-899: A single dose of 17- β -estradiol was administered. To establish the gonococcal infection, three injections of 17- β -estradiol 3-benzoate are commonly administered (one injection every other day) or a slow-release 17- β -estradiol 3-benzoate pellet is implanted to establish a continuous higher estradiol level which is crucial for *N. gonorrhoeae* infection. Also, mice were infected 4 days after estradiol dose. The estradiol level decreases gradually by time. So, after 4 days, the level will be very low given that a single dose was administered. So, these conditions are not ideal to establish a successful gonococcal colonization. Mice should be inoculated with *N. gonorrhoeae* 2 days after estradiol treatment and receive a second dose of estradiol in the same day of infection to ensure that levels of estradiol remain high.

Streptomycin is not included in the animal protocol described. Streptomycin is important in the gonococcal infection model to prevent the growth of commensal Gram-negative bacteria which, if present, prevent the gonococcal colonization. Vancomycin is also injected to mice suppress Gram-positive flora permitting successful colonization of *N. gonorrhoeae*.

Lines 908-910: "samples collected were plated onto GC agar containing chloramphenicol (10 μ g/ml) and erythromycin (7 μ g/ml)". The antibiotics added to the plates does not contain antifungal agent like nystatin to inhibit *Candida* which can be present in the vaginal fluid. Also, colistin is frequently added as one of the antibiotics to the GC plates to inhibit Gram-negative bacteria, including *Pseudomonas* species.

N. cinerea is not affected by NQNO or 3Me-NQNO unlike *N. gonorrhoeae*. Based upon the proposed mechanism, it is very important to show the genetic differences between *N. cinerea* and *N. gonorrhoeae* including the electron transport systems and Zeta/epsilon toxin/antitoxin systems and show the differences which make *N. cinerea* resistant to these agents.

Decision Letter:

13th December 2023

Dear Professor Hauck,

Thank you for your patience while your manuscript "Quinolone-N-oxides kill multidrug resistant *N. gonorrhoeae* by unleashing

the endogenous zeta toxin" was under peer review at Nature Microbiology. It has now been seen by our referees, whose expertise and comments you will find at the end of this email. In the light of their advice, we have decided that we cannot offer to publish your manuscript in Nature Microbiology.

From the reports, you will see that while several of the referees were largely satisfied with the revised manuscript, Referee #4, who provided additional expertise in *Neisseria* infection models to assess the in vivo aspects of your study, raised several technical concerns about this aspect of the study. They were concerned about some of the methods used, and the relevance of the approach. Unfortunately, these concerns were significant enough to preclude publication of the work in Nature Microbiology. To address these concerns would require considerable revision of the manuscript at this stage in the review process, as it would likely involve redoing much of the vivo analyses, without guarantee that this additional experimental work would enable the same conclusions to be drawn.

I am sorry that we cannot be more positive on this occasion, but hope that you find the referees' comments helpful when preparing your paper for resubmission elsewhere. Although you have opted out of transfer consultations, we could also consult with our colleagues at Nature Communications and Communications Medicine to see if they would be interested in seeing a revised manuscript, and what their priorities for revision might be. This might ultimately lead to a more expedited revision and publication process.

Reviewer Expertise:

Referee #1: TA systems

Referee #2: *Neisseria*, AMR

Referee #3: *Pseudomonas* signalling molecules, antimicrobial development

Referee #4: *Neisseria* infection, mouse models

Reviewers Comments:

Reviewer #1 (Remarks to the Author):

The authors have added substantial and significant additional data to this manuscript addressing most of the reviewer's concern and provide convincing arguments in their rebuttal letter if not included.

I would like to congratulate the authors for this outstanding achievement.

Reviewer #2 (Remarks to the Author):

I would like to express my gratitude to the authors for their thoughtful and comprehensive responses to my comments. I appreciate the effort and diligence applied to address the points raised. However, I have a few additional comments.

Row135 comment: Please include a sentence or two in regards of limitations in the discussion section regarding *N. meningitidis*.

Figure S1B – Weren't all the isolates in Table S1 subjected to this methodology? If that's the case, it would be prudent to extend the supplementary material by an additional two pages, despite it already surpassing 50 pages. Providing the inhibition zones for all strains would significantly enhance the comprehensiveness of our data. A dedicated table detailing these inhibition zones would be invaluable. Lastly, for clarification, the label in Figure B should read '*N. gonorrhoeae* MS11,' not '*N. gonorrhoeae* 11,' correct?

Row 163-165: This reviewer notes that the authors do not advocate for pefloxacin as a recommended treatment. The authors state:

"Strikingly, 2-nonyl-4-quinolone (NQ), trans- Δ 1 162 -2-nonenyl-4-quinolone (trans- Δ 1 163 -NQ) and 2-nonyl-4-quinolone N-oxide (NQNO) completely inhibited growth of *N. gonorrhoeae* (Fig. 1C). While trans- Δ 1-NQNO was considerably less active, NQNO inhibited growth of *N. gonorrhoeae* even at the lowest concentration of 5 μ M (~ 1.5 μ g/ml) (Fig. 1C). This effective concentration is in the range of minimum inhibitory concentrations (MICs) observed for clinically relevant antibiotics such as pefloxacin or spectinomycin (Bengtsson-Palme and Larsson, 2016)."

When the authors use the term "clinically relevant" alongside a genuinely clinically relevant antibiotic for *N. gonorrhoeae*, such as spectinomycin, it may lead readers to infer that pefloxacin is/has been clinically relevant for gonorrhoea treatment. Please clarify this statement.

Final comment: The isolates that have undergone clonal expansion in recent years are the WHO P-like strains. Meanwhile, high-level resistance to azithromycin appears more sporadically. This doesn't necessarily suggest that the resistance mechanisms to azithromycin are causing the decreased susceptibility to NQNO. It may instead indicate that the WHO P-like genomic backbone, with its myriad traits, is also proliferating. This backbone might encompass a consistent TA system. That said, this reviewer is not advocating for additional experiments. Instead, it would be beneficial if the authors could address this point in the discussion section.

Reviewer #3 (Remarks to the Author):

The authors have addressed all my key points and added new data to support their conclusions with respect to the second toxin/antitoxin system, the stability of the antitoxin, quantification of membrane damage and extended the concentrations and cell lines used to evaluate toxicity of the quinolone N-oxides. Although the precise molecular basis by which the quinolone N-oxides release the anti-toxin from the toxin remains to be established, the authors have made a very novel and interesting finding of broad interest with significant translational potential in the AMR context.

Minor points:

Line 104. Evidence that the AQNOs are not signal molecules in *Pseudomonas* was directly addressed in Rampioni et al (2016) PLOS Pathogens doi: 10.1371/journal.ppat.1006029 rather in the two Szamosvaro et al papers cited.

Lines 348 and 351 and Fig. 6F. 'disappearance' in these sentences is incorrect. The epsilon antitoxin is reduced not absent from these blots.

Fig. 6 F and G. There is a band on the zeta 1 blots marked with a star. This is presumably a protein that cross-reacts with the antibody used? This should be explained in the legend.

Reviewer #4 (Remarks to the Author):

The manuscript by Mix et al investigates the activity of 2-nonyl-4-quinolone N-oxide (NQNO) against *N. gonorrhoeae*. The study reveals that NQNO demonstrates anti-gonococcal activity without impairing growth of commensal *Neisseria* or viability of human cells. In vivo, application of NQNO resulted in reducing the bacterial burden in the vaginal tract. Mechanistically, NQNO was found to disrupt the gonococcal electron transport chain with the release of zeta toxin from its epsilon antitoxin. Chemical modification of NQNO yielded a more potent derivative. While most of the experimental work is well designed, and the manuscript is well written, I have outlined some comments and critiques below that should be considered to enhance the manuscript's impact

Lines 96-97: "Most antibiotics in current use are derived from bacterial secondary metabolites, which have evolved in the frame of microbial competition." This sentence requires supporting references to substantiate the claim.

Line 127: please correct the typo to "of this global of concern pathogen".

Fig. S1C: difference is small

Lines 172-174: "a highly consistent growth inhibition of gonococci by NQNO concentrations > 2.5 μM was observed (Suppl. Fig. S2A)". According to Suppl. Fig. S2A, HQNO was not tested at the concentration of 2.5 μM (the lowest concentration tested was 5 μM). So, this sentence should be rephrased to "a highly consistent growth inhibition of gonococci by NQNO concentrations \geq 5 μM

Fig. 3D: there are no standard deviation values shown in the figure.

Lines 280-281: "two zeta-epsilon (ζ_1/ϵ_1 and ζ_2/ϵ_2) toxin/anti-toxin (TA) systems are encoded on this extrachromosomal DNA". Why was ζ_1/ϵ_1 system the only system investigated and ζ_2/ϵ_2 was not investigated?

Lines 309-311: "The introduction of the zeta/epsilon-encoding plasmid rendered the resulting pTetM-positive clones sensitive to NQNO (Fig. 4F)". As shown in Fig. 4F, it seems that the growth of MS11-R2 pTetM A and MS11-R2 pTetM B is normally reduced (OD550 of the negative control (DMSO) did not exceed 0.4 as compared to about 1.4 for MS11-R2). In presence of HQNO, the OD550 of MS11-R2 pTetM A and MS11-R2 pTetM B is increasing by time and it reaches values comparable to that of MS11-R2. A similar trend is shown for MS11-R2 (growth increases by time with HQNO treatment). Consequently, it is not reasonable to conclude that "The introduction of the zeta/epsilon-encoding plasmid (pTetM A and pTetM B) rendered the resulting pTetM-positive clones sensitive to NQNO". The incubation time should be extended beyond 8 hours to observe potential differences between the conjugants and the parent strains. Furthermore, given the low OD550 values observed in the case of the conjugant strains, relying solely on OD values for drawing conclusions may not be sufficient. It is essential to include CFU (colony-forming unit) counts, as even minor discrepancies in OD values may not necessarily reflect significant differences in CFU counts.

Lines 325-326: Similar to the previous comment, Fig. 5B shows that the growth of WHO P pTetM A, WHO P pTetM B, and WHO P ζ_1/ϵ_1 is normally reduced (OD550 of the negative control reaches up to 0.4-0.5 only as compared to about 1.0 for WHO P). In presence of HQNO, the OD550 of WHO P pTetM A, WHO P pTetM B, and WHO P ζ_1/ϵ_1 is increasing by time and it reaches values comparable to that of DMSO (~0.25-4). A similar trend is shown for WHO P (growth increases by time with HQNO treatment). Consequently, it is not reasonable to conclude that "Most importantly, Zeta1/Epsilon1-expressing derivatives of WHO P became NQNO-sensitive, while the parent strain was fully resistant up to 50 μM (Fig. 5B)". The incubation time should be extended beyond 8 hours to see if there will be significant difference between the conjugants and the parent strains. Also, since the OD550 values are low in case of the conjugant strains, we cannot depend on the OD values alone for drawing conclusions. CFU counts are important to be included since this small difference in OD values may not reflect a significant difference in the CFU counts.

Lines 341-347: The reason for switching to the use of *E. coli* instead of *N. gonorrhoeae* to investigate the degradative effect of reactive oxygen species on Epsilon antitoxin is not clear and should be clarified in the manuscript.

Fig. 6D illustrates that NQNO treatment did not impact the growth of *E. coli*. It's important to note that the outer membrane of *E. coli* differs from that of *N. gonorrhoeae*, which can lead to variations in the susceptibility of these bacteria to different agents. Some agents may exhibit inhibitory activity against *N. gonorrhoeae* without affecting *E. coli* or with reduced activity against *E.*

coli due to permeability. This phenomenon is evident in the case of HQNO, which did not inhibit *E. coli* while displaying inhibitory effects on *N. gonorrhoeae*. Additionally, the results presented in Fig. S1 indicate that *N. gonorrhoeae* exhibits a larger inhibition zone compared to the minimal inhibition observed for *E. coli*. Therefore, the use of *E. coli* in this experiment may not support the conclusion that "NQNO does not act directly on Epsilon stability but rather acts indirectly via NQNO-induced oxidative stress." These experiments should ideally be conducted using *N. gonorrhoeae*, as the effects of NQNO on *E. coli* may be limited due to its reduced activity/permeability against this bacterium.

Fig. 6D: there are no standard deviation values shown in the figure.

Lines 360-362: The proposed mechanism of NQNO action is mediated by unleashing the endogenous Zeta1 toxin from inhibition by its cognate epsilon1 antitoxin, and the enzymatic activity of Zeta1 will lead to disruption of cell wall synthesis and integrity. To further substantiate this conclusion, additional assays demonstrating the disruption of cell wall synthesis and integrity in response to NQNO treatment should be included. Relying solely on electron microscopy may not be sufficient to confirm this mechanism conclusively.

I have several concerns regarding the animal experiment. Firstly, the treatment was administered after only 1 hour, which is a very short duration for *N. gonorrhoeae* to attach to the vaginal tract and establish infection. Additionally, this timeframe may not be clinically applicable. Topical application of *Neisseria* is not supported and is unlikely to clear *Neisseria* infections. Moreover, any topical treatment, including mild antiseptic solutions or mild H₂O₂, applied in this model topically is likely to result in a significant reduction in bacterial load.

Fig. 7B: Why is the number of mice inconsistent between groups? For instance, in the experiment of CEAtg mice, DMSO: 12 mice, NQNO (25 μ M): 6 mice, and NQNO (50 μ M): 9 mice. For wild-type mice, DMSO: 12 mice, 25 μ M NQNO: 4 mice, and 50 μ M NQNO: 5 mice. The same comment applies to the second mouse experiment (Fig. 7F).

No control antibiotics (e.g. ceftriaxone and azithromycin) were used in both mouse experiments.

Lines 417-419: "At concentrations of 0.5 μ M and higher, 3Me-NQNO was bactericidal and induced rapid killing of the pathogens (Fig. 8E)". Fig. 8F shows that at the concentrations of 0.5 μ M, 3Me-NQNO was bacteriostatic, and did not reduce the bacterial count. The concentration of 2.5 μ M resulted in reduction of the bacterial count by about 3 log₁₀ after 3 hours. So, at this concentration only, 3Me-NQNO can be considered bactericidal. Please, modify this sentence accordingly.

Fig. 8E and 8F: The figures do not show standard deviation values. How many replicates were used for these experiments?

Lines 422-424: Toxicity of 3Me-NQNO was measured up to 10 μ M while it showed activity at the concentrations of 2.5 – 5 μ M. To establish a high therapeutic index for the agent, cytotoxicity should be evaluated at higher concentrations, such as those several folds higher than the active concentration (e.g., 100 μ M or higher). This will help provide a more comprehensive assessment of the agent's safety profile and therapeutic potential.

Lines 494-495: "the experimental infection of female mice does not recapitulate gonorrhea and symptomatic disease has not been observed in this model". This information is inaccurate. *N. gonorrhoeae* female mouse model established by Jerse et al 1999 showed symptoms of gonorrhea such as: 1) Gonorrheal inflammation occurred in over 80% of infected mice, 2) *N. gonorrhoeae* was recovered from vaginal swabs for an average of 12 to 13 days following infection, 3) Vaginal smears showed presence of neutrophils recruited to the site of infection.

Lines 500-510: discuss the use of topically applied antimicrobials for sexually transmitted diseases, including *N. gonorrhoeae*. However, it's important to note that the case of *N. gonorrhoeae* is different due to specific criteria for anti-gonococcal therapeutics. For *N. gonorrhoeae*, patient compliance is crucial, and there are certain criteria that should be considered. These criteria include the route of administration, which should be either oral or intramuscular (IM), and the treatment duration. For urogenital *N. gonorrhoeae*, the treatment duration should be 1 day for the oral route and up to 3 days for anorectal and oropharyngeal *N. gonorrhoeae* (for the oral route), or 1 day for all types of *N. gonorrhoeae* infections (for the IM route). These criteria reflect the specific challenges and requirements for treating *N. gonorrhoeae* effectively.

I would recommend adding a paragraph discussing the limitations of the study such as the pre-existing resistance (presence of *N. gonorrhoeae* strains with resistance to NQNO e.g. WHO P and strategies to overcome this resistance).

A general comment on the materials and methods section: most of the experimental protocols presented do not have citations. References for each experiment need to be added.

Lines 629-635: "Determination of NADH levels". The experiment also lacks a positive control (an agent that is known to reduce the NADH levels in the bacteria) which provides the level of confidence in the results obtained and ensure that the results are reproducible and robust. The same comment applies to the experiments of "Detection of reactive oxygen species (ROS)"

Lines 647-662: "Selection for NQNO and 3Me-NQNO resistance". The proposed method requires citation, and the experiment also lacks a positive control (an agent that is known to be susceptible to resistance development such as rifampicin, tetracycline or ciprofloxacin) which confirms that the assay is right, and the results obtained are reproducible and robust. Additionally, in this method, bacteria are exposed daily to the same conditions (conditioning with 2.5 μ M NQNO or 0.1 μ M 3Me-NQNO for 8 h) and then evaluated for sensitivity towards different concentrations of NQNO or 3Me-NQNO. This technique may not enhance the development of bacterial resistant mutants since bacteria are exposed to exactly the same conditions every day. The most commonly used method is through serial passages where the bacterial inoculum is prepared from cultures with a subinhibitory concentration of the test agent (these concentrations are not the same every day and may be different according to the MICs of each passage).

Lines 898-899: A single dose of 17- β -estradiol was administered. To establish the gonococcal infection, three injections of 17- β -estradiol 3-benzoate are commonly administered (one injection every other day) or a slow-release 17- β -estradiol 3-benzoate pellet is implanted to establish a continuous higher estradiol level which is crucial for *N. gonorrhoeae* infection. Also, mice were infected 4 days after estradiol dose. The estradiol level decreases gradually by time. So, after 4 days, the level will be very low given that a single dose was administered. So, these conditions are not ideal to establish a successful gonococcal colonization. Mice should be inoculated with *N. gonorrhoeae* 2 days after estradiol treatment and receive a second dose of estradiol in the same day of infection to ensure that levels of estradiol remain high.

Streptomycin is not included in the animal protocol described. Streptomycin is important in the gonococcal infection model to prevent the growth of commensal Gram-negative bacteria which, if present, prevent the gonococcal colonization. Vancomycin is also injected to mice suppress Gram-positive flora permitting successful colonization of *N. gonorrhoeae*.

Lines 908-910: "samples collected were plated onto GC agar containing chloramphenicol (10 μ g/ml) and erythromycin (7 μ g/ml)". The antibiotics added to the plates does not contain antifungal agent like nystatin to inhibit *Candida* which can be present in the vaginal fluid. Also, colistin is frequently added as one of the antibiotics to the GC plates to inhibit Gram-negative

bacteria, including *Pseudomonas* species.

N. cinera is not affected by NQNO or 3Me-NQNO unlike *N. gonorrhoeae*. Based upon the proposed mechanism, it is very important to show the genetic differences between *N. cinera* and *N. gonorrhoeae* including the electron transport systems and Zeta/epsilon toxin/antitoxin systems and show the differences which make *N. cinera* resistant to these agents.

***Nature Communications* is the Nature Portfolio flagship Open Access journal. If you would like this work to be considered for publication there, you can easily transfer the manuscript by following the instructions below. It is not necessary to reformat your paper. Once all files are received, the editors at *Nature Communications* will assess your manuscript's suitability for potential publication; they aim to provide feedback quickly, with a median decision time of 8 days for first editorial decisions on suitability. Since your paper has been peer reviewed at this journal, the referee reports will also be transferred and assessed by the editorial team. In some cases, papers are accepted without further peer review, providing a rapid path to publication. The journal is also proud to offer double blind and transparent peer review options. For 2021, the 2-year impact factor for *Nature Communications* is 17.694 and the 2-year median is 10 (for further information on journal impact factors, please visit our http://www.nature.com/npg_company_info/journal_metrics.html>Nature journals metrics page). Our [open access pages](http://www.nature.com/ncomms/open_access/index.html) contain information about article processing charges, open access funding, and advice and support from Springer Nature.

** Although we cannot offer to publish your manuscript, we believe the editors at our sister journal, *Communications Biology*, will find it interesting and recommend you transfer it there. *Communications Biology* is a selective Nature Portfolio title publishing Open Access research that brings new insight in all areas of the biological sciences. [Additional journal metrics and information can be found here](https://www.nature.com/commsbio/journal-information/journal-impact)>. Their editors prioritise good author service, fast peer review (in 2021, the median time to decision after first review was 40 days), and are happy to answer any questions you may have <mailto:commsbio@nature.com>>(commsbio@nature.com). The journal has an Impact Factor of 6.548, a CiteScore of 6.0 and a Scimago quartile ranking of Q1.

Please note that *Communications Biology* is a fully open-access journal and an article processing charge will apply to any papers accepted for publication. Our [open access pages](https://www.nature.com/commsbio/about/open-access)> contain information about article processing charges, open access funding, and advice and support from Springer Nature.

If you wish to transfer your manuscript to *Communications Biology*, please use our manuscript transfer portal using the link below to initiate the transfer to this journal (or to another journal of your choice in the Nature Research portfolio). If you transfer to Nature-branded journals or to the Communications journals, you will not have to re-supply manuscript metadata and files. This link can only be used once and remains active until used. For more information, please see our [manuscript transfer FAQ](https://www.nature.com/nature-portfolio/for-authors/transfer)> page.

***Scientific Reports* publishes primary research from all areas of the natural and clinical sciences that is judged to be scientifically valid and technically sound, whatever the considered significance. If you would like this work to be considered for publication in *Scientific Reports*, you can easily transfer the manuscript by following the instructions below. Your manuscript will be handled by an academic scientist who is an Editorial Board Member and will manage the peer review process and decide whether a paper should be accepted for publication. Most submissions are peer reviewed by one or more referees as well as the editorial board member and you can expect to receive an editorial decision within 56 days. Over 55% of the papers are published following peer review.

To discover more about this journal and, should you wish, have your paper considered the Editorial Board of *Scientific Reports*, please use the link to the manuscript transfer service provided in the footnote below. Please see our [open access pages](http://www.nature.com/ncomms/open_access/index.html)> for information about article processing charges, open access funding, and advice and support from Springer Nature.

Although we cannot publish your paper, it may be appropriate for another journal in the Nature Portfolio. If you wish to explore the journals and transfer your manuscript please use our manuscript transfer portal>. You will not have to re-supply manuscript metadata and files, unless you wish to make modifications. For more information, please see our [manuscript transfer FAQ](http://www.nature.com/authors/author_resources/transfer_manuscripts.html?WT.mc_id=EMI_NPG_1511_AUTHORTRANSF&WT.ec_id=AUTHOR)> page.

Version 2:

Decision Letter:

9th February 2024

Dear Professor Hauck,

Thank you for your letter asking us to reconsider our decision on your Article entitled "Quinolone-N-oxides kill multidrug resistant *N. gonorrhoeae* by unleashing the endogenous zeta toxin". My colleagues and I have looked through the rebuttal and proposed

plans to revise the paper carefully and although at this point we are unable to reverse our decision, we would be willing to reconsider once you have the data and have revised the manuscript.

We appreciate the proposed plans to revise several key points through additional experimental work and edits to the text, and we appreciate that some concerns have been met by existing data within the study. However there are several points that we would like you to further consider in the revised manuscript. Regarding Referee #4's point on 1h being quite a short treatment time, our interpretation of this was more that in real life it would be very unlikely to catch an infection so soon after initial exposure. Testing efficacy at a later time point would be more reflective of a "real" treatment, thus we would strongly encourage you to investigate a later time point experimentally. Furthermore, to address the concerns over the number of mice used per group we would ask you to provide a power analysis. Lastly, we would like to try to reassure you of the referee's independence and ask you to check over the rebuttal and response to this referee before resubmitting.

We really appreciate all the efforts so far to revise what is a very interesting manuscript. I would like to reiterate that we are interested in seeing another appeal once you have all the final data and text revisions in place. We cannot make any guarantees about our decision before seeing these data, however we would be willing to consider the study again.

Version 3:

Reviewer comments:

Reviewer #4

(Remarks to the Author)

I would like to thank the authors for addressing some of my concerns. However, two issues still need to be addressed.

First, the difference between the outer membranes of *E. coli* and *Neisseria*, and the impact this has on the conclusions of the experiment. I was surprised that the authors did not find relevant literature to support this distinction, so I am including some references (1-3) that may help in selecting the correct strains and designing an appropriate experiment. The outer membranes of *E. coli* and *Neisseria* differ significantly in composition, function, and ability to allow the passage of drugs or compounds. *Neisseria* has a lipooligosaccharide (LOS), while *E. coli* has lipopolysaccharide (LPS). LOS is shorter and less complex. The porins in *Neisseria* differ from those in *E. coli*. The *E. coli* strain used in the study does not have the highest permeability and possesses intrinsic resistance mechanisms, including the outer membrane and efflux systems, which prevent most inhibitors from crossing (4) (Randall et al., 2013, *Antimicrob Agents Chemother* 57(1): 637–639). When these intrinsic resistance mechanisms are artificially compromised, such as by permeabilizing the outer membrane or deleting the efflux systems, agents can cross and demonstrate antibacterial activity. To confirm the conclusion that NQNO treatment does not affect *E. coli*, experiments should be conducted using *E. coli* strains with compromised outer membranes and efflux systems (4) (Randall et al., 2013, *Antimicrob Agents Chemother* 57(1): 637–639).

The second major issue is the animal experiment, which remains seriously flawed. While this reviewer acknowledges the potential benefits of prophylaxis, it is important to note that neither prophylactic nor treatment measures for urogenital *Neisseria* infections involve topical application, as it is ineffective. Another significant issue is the lack of positive controls. The authors' explanation for not including a positive control antibiotic in the treatment group raises concerns about the experiment's rigor and validity. Positive controls in drug discovery are essential for validating the experimental setup's ability to detect an effect. Without positive control, it is unclear whether the treatment's lack of effectiveness/clearance is due to the compound itself or flaws in the experimental design. Given that the experimental setting used is non-standard, including a positive control is crucial to establish a valid baseline. The claim that animal research ethics laws prohibit the sacrifice of mice for known facts suggests a misunderstanding of ethical guidelines.

1. Wolf-Watz, H.; Elmros, T.; Normark, S.; Bloom, G. D., Cell envelope of *Neisseria gonorrhoeae*. A comparative study with *Escherichia coli*. *Br J Vener Dis* 1976, 52 (2), 142-5.
2. Morse, S. A.; Cacciapuoti, A. F.; Lysko, P. G., Physiology of *Neisseria gonorrhoeae*. *Adv Microb Physiol* 1979, 20, 251-320.
3. Morse, S. A., *Neisseria gonorrhoeae*: physiology and metabolism. *Sex Transm Dis* 1979, 6 (1), 28-37.
4. Randall, C. P.; Mariner, K. R.; Chopra, I.; O'Neill, A. J., The target of daptomycin is absent from *Escherichia coli* and other gram-negative pathogens. *Antimicrob Agents Chemother* 2013, 57 (1), 637-9.

Decision Letter:

1st October 2024

Dear Christof

Thank you for your patience while your revised manuscript "A selective antibiotic exploits the endogenous zeta toxin to protect from multidrug resistant *N. gonorrhoeae*" was under peer review at Nature Microbiology. It has now been seen by the referee, whose expertise and comments you will find at the end of this email. In the light of their advice, we have decided that we cannot offer to publish your manuscript in Nature Microbiology.

From the reports, you will see that unfortunately the reviewer still had major concerns about the in vivo work, including the lack of certain controls and the overall relevance of the approach for treatment of *N. gonorrhoea* infection. Unfortunately, these criticisms remain sufficiently important as to preclude publication of your work in *Nature Microbiology*.

I am sorry that we cannot be more positive on this occasion, and I appreciate that this must be a very disappointing decision but we felt that these concerns were too strong for us to overrule. We hope that you find the referees' comments helpful when preparing your paper for resubmission elsewhere.

Although we cannot offer to publish your manuscript, I have discussed your manuscript and the reviewers' comments with our colleagues at *Nature Communications*. They would be happy, in principle, to publish your work, under the condition of the removal of the in vivo murine experimental work, given the remaining concerns from Reviewer #4. Should you wish to have your revised paper considered by *Nature Communications*, please use the link to the Springer Nature manuscript transfer service provided below once the revision is ready, and include a point-by-point response to the reviewers' concerns.

Your handling editor at *Nature Communications* would be Dr. Hayleah Pickford (hayleah.pickford@nature.com). If there is anything you would like to discuss before transferring the paper and its reviews, please don't hesitate to contact her by e-mail.

Please note that *Nature Communications* is a fully open access journal. For information about article processing charges, open access funding, and advice and support from Springer Nature, please consult the *Nature Communications* Open Access page (<https://www.nature.com/ncomms/open-access>).

To transfer your manuscript please use our manuscript transfer portal. You will not have to re-supply manuscript metadata and files, unless you wish to make modifications. For more information, please see our [manuscript transfer FAQ](http://www.nature.com/authors/author_resources/transfer_manuscripts.html?WT.mc_id=EMI_NPG_1511_AUTHORTRANSF&WT.ec_id=AUTHOR) page.

Reviewers Comments:

Reviewer #4 (Remarks to the Author):

I would like to thank the authors for addressing some of my concerns. However, two issues still need to be addressed.

First, the difference between the outer membranes of *E. coli* and *Neisseria*, and the impact this has on the conclusions of the experiment. I was surprised that the authors did not find relevant literature to support this distinction, so I am including some references (1-3) that may help in selecting the correct strains and designing an appropriate experiment. The outer membranes of *E. coli* and *Neisseria* differ significantly in composition, function, and ability to allow the passage of drugs or compounds. *Neisseria* has a lipooligosaccharide (LOS), while *E. coli* has lipopolysaccharide (LPS). LOS is shorter and less complex. The porins in *Neisseria* differ from those in *E. coli*. The *E. coli* strain used in the study does not have the highest permeability and possesses intrinsic resistance mechanisms, including the outer membrane and efflux systems, which prevent most inhibitors from crossing (4) (Randall et al., 2013, *Antimicrob Agents Chemother* 57(1): 637–639). When these intrinsic resistance mechanisms are artificially compromised, such as by permeabilizing the outer membrane or deleting the efflux systems, agents can cross and demonstrate antibacterial activity. To confirm the conclusion that NQNO treatment does not affect *E. coli*, experiments should be conducted using *E. coli* strains with compromised outer membranes and efflux systems (4) (Randall et al., 2013, *Antimicrob Agents Chemother* 57(1): 637–639).

The second major issue is the animal experiment, which remains seriously flawed. While this reviewer acknowledges the potential benefits of prophylaxis, it is important to note that neither prophylactic nor treatment measures for urogenital *Neisseria* infections involve topical application, as it is ineffective. Another significant issue is the lack of positive controls. The authors' explanation for not including a positive control antibiotic in the treatment group raises concerns about the experiment's rigor and validity. Positive controls in drug discovery are essential for validating the experimental setup's ability to detect an effect. Without positive control, it is unclear whether the treatment's lack of effectiveness/clearance is due to the compound itself or flaws in the experimental design. Given that the experimental setting used is non-standard, including a positive control is crucial to establish a valid baseline. The claim that animal research ethics laws prohibit the sacrifice of mice for known facts suggests a misunderstanding of ethical guidelines.

1. Wolf-Watz, H.; Elmros, T.; Normark, S.; Bloom, G. D., Cell envelope of *Neisseria gonorrhoeae*. A comparative study with *Escherichia coli*. *Br J Vener Dis* 1976, 52 (2), 142-5.
2. Morse, S. A.; Cacciapuoti, A. F.; Lysko, P. G., Physiology of *Neisseria gonorrhoeae*. *Adv Microb Physiol* 1979, 20, 251-320.
3. Morse, S. A., *Neisseria gonorrhoeae*: physiology and metabolism. *Sex Transm Dis* 1979, 6 (1), 28-37.
4. Randall, C. P.; Mariner, K. R.; Chopra, I.; O'Neill, A. J., The target of daptomycin is absent from *Escherichia coli* and other gram-negative pathogens. *Antimicrob Agents Chemother* 2013, 57 (1), 637-9.

Version 4:

Reviewer comments:

Reviewer #6

(Remarks to the Author)

Decision Letter:

Our ref: NMICROBIOL-23010150D-Z

11th December 2024

Dear Dr. Hauck,

Thank you for submitting your revised manuscript "A selective antibiotic exploits the endogenous zeta toxin to protect from multidrug resistant *N. gonorrhoeae*" (NMICROBIOL-23010150D-Z). It has now been seen by a new referee who was satisfied with the rebuttal and responses provided. The reviewer finds that the paper has improved in revision, and therefore we'll be happy in principle to publish it in Nature Microbiology, pending minor revisions to comply with our editorial and formatting guidelines.

We are now performing detailed checks on your paper and will send you a checklist detailing our editorial and formatting requirements in about a week. We will do our best to get this to you before the holiday break. Please do not upload the final materials and make any revisions until you receive this additional information from us.

Thank you again for your interest in Nature Microbiology. Please do not hesitate to contact me if you have any questions.

Sincerely,

Version 5:

Decision Letter:

19th February 2025

Dear Professor Hauck,

I am pleased to accept your Article "A quinolone N-oxide antibiotic selectively targets *Neisseria gonorrhoeae* via its toxin-antitoxin system" for publication in Nature Microbiology. Thank you for having chosen to submit your work to us and many congratulations.

After the grant of rights is completed, you will receive a link to your electronic proof via email with a request to make any corrections within 48 hours. If, when you receive your proof, you cannot meet this deadline, please inform us at rjsproduction@springernature.com immediately. You will not receive your proofs until the publishing agreement has been received through our system.

Authors may need to take specific actions to achieve <https://www.springernature.com/gp/open-research/funding/policy-compliance-faqs> compliance with funder and institutional open access mandates. If your

research is supported by a funder that requires immediate open access (e.g. according to [Plan S principles](https://www.springernature.com/gp/open-research/plan-s-compliance)) then you should select the gold OA route, and we will direct you to the compliant route where possible. For authors selecting the subscription publication route, the journal's standard licensing terms will need to be accepted, including [self-archiving policies](https://www.nature.com/nature-portfolio/editorial-policies/self-archiving-and-license-to-publish). Those licensing terms will supersede any other terms that the author or any third party may assert apply to any version of the manuscript.

With kind regards,

P.S. Click on the following link if you would like to recommend Nature Microbiology to your librarian <http://www.nature.com/subscriptions/recommend.html#forms>

** Visit the Springer Nature Editorial and Publishing website at http://editorial-jobs.springernature.com?utm_source=ejP_NMicro_email&utm_medium=ejP_NMicro_email&utm_campaign=ejp_NMicro for more information about our career opportunities. If you have any questions please click [here](mailto:editorial.publishing.jobs@springernature.com).**

Open Access This Peer Review File is licensed under a Creative Commons Attribution 4.0 International License, which permits use, sharing, adaptation, distribution and reproduction in any medium or format, as long as you give appropriate credit to the original author(s) and the source, provide a link to the Creative Commons license, and indicate if changes were made. In cases where reviewers are anonymous, credit should be given to 'Anonymous Referee' and the source. The images or other third party material in this Peer Review File are included in the article's Creative Commons license, unless indicated otherwise in a credit line to the material. If material is not included in the article's Creative Commons license and your

intended use is not permitted by statutory regulation or exceeds the permitted use, you will need to obtain permission directly from the copyright holder.

**Reviewer Comments:****Reviewer #1 (Remarks to the Author):**

With this manuscript, Boettcher, Hauck, and colleagues provide surprising and convincing evidence on a highly specific antimicrobial activity of 2-alkyl-4-quinolones N-oxides (AQNOs) against 25.2 MDa pTetM plasmid carrying *Neisseria gonorrhoea* variants. The inhibition of respiratory chain cytochromes by AQNOs, as the authors describe, has been described in several reports in the past. However, it is new and exciting, that downstream effects of this interference can exert strain specific lethality in *N. gonorrhoea* but not against other neisserial species. In a convincing approach they started with *Pseudomonas aeruginosa* / *N. gonorrhoea* battle experiments enabling identification of AQ(NO)s as anti-gonococcal compounds secreted by *Pseudomonas aeruginosa*. Synthetic compounds were produced, their bacteriostatic or bactericidal activity was monitored and NQNO was shown to be the most potent candidate. As expected, NQNO treatment caused massive perturbation of cytosolic ATP and NADH levels accompanied with ROS production in gonococcal strains but not in *N. cinerea* which they showed to be resistant against AQ(NO)s. By generating NQNO resistant strains which in fact have lost the entire pTetM plasmid, they concluded that the ROS dependent activation of Type II epsilon zeta TA systems is required for gonococcal sensitivity. In fact, clinical variants of *N. gonorrhoea* strains that they identified to be resistant to NQNO lacked the encoding locus. Although ROS dependent activation of Type II TA systems has been reported in the past, this is, to my knowledge, the first time that unleashing a toxin from its antitoxin eventually would tip the scale in favor of exerting a bactericidal phenotype. Most importantly, they established synthesis of a highly potent NQNO lead compound, demonstrate anti-gonococcal efficacy of NQNOs in a mouse model and provided first insights for a potential therapeutic application of their compounds.

In general, the experimental work is well designed, the manuscript is conclusive and convincing. However, as described below, some aspects should be reconsidered in a more appropriate and/or careful way. If resolved, this would strengthen the impact of the manuscript.

Comment 1) The genetic load region of the 25.2 MDa pTetM plasmid contains two paralogues of the epsilon/zeta system and the authors only demonstrate activation of the epsilon_1/zeta_1 system. Is zeta_2 unleashed similarly upon ROS dependent degradation of epsilon_2? If the authors only provide experimental evidence on the mechanism of activation of zeta_1, care should be taken when generalizing their observation. At least, it would be better if they refer to epsilon_1/zeta_1 throughout the manuscript.

The reviewer is completely right in pointing out that the pTetM plasmid encodes two closely related toxin/antitoxin systems, while we only focus our attention on the zeta1/ epsilon1 TA pair. In response to this comment and also to the reviewer's comment #2 below, we have now transformed the NQNO-resistant and multidrug-resistant strain WHO P not only with the full-size pTetM plasmid (see former Fig. 4D-E), but also with a construct encoding the

zeta1/epsilon1 operon only. This allowed us to i) test the contribution of the zeta1/epsilon1 to NQNO sensitivity in the absence of zeta2/epsilon2 and ii) to test the contribution of the zeta1/epsilon1 to NQNO sensitivity in the absence of other open reading frames contained on the pTetM-plasmid. As shown in novel Fig. 5 and Suppl. Fig. S6, the construct is chromosomally integrated in strain WHO P and the zeta1 toxin / epsilon1 antitoxin are expressed at levels comparable to strains harboring the pTetM plasmid. Most importantly, selective expression of Zeta1/Epsilon1 renders the NQNO-resistant strain WHO P now sensitive to NQNO and this sensitivity is similar to the sensitivity observed after transformation of this strain with the complete pTetM plasmid. Accordingly, we can conclude that Zeta1 toxin / Epsilon1 antitoxin expression is required for the NQNO sensitivity of *N. gonorrhoeae*.

Comment 2) Is NQNO sensitivity of *N. gonorrhoea* indeed exclusively the result of zeta_1 toxin activation? In fact, loss of the entire 25.2 MDa pTetM plasmid was observed when raising strains resistant to NQNO and clinical variants that proved not to be sensitive to their compounds most likely do also lack the plasmid. Can the author exclude that sensitivity to their quinolone compounds is established from the plasmid by any other means than the zeta_1 activity? In fact, TA systems were originally described as plasmid addiction modules that ensure stable plasmid maintenance. It would be interesting to see, whether an catalytically inactive zeta_1 toxin encoded from a stably maintained plasmid also gives rise to resistance. In fact, sensitivity could be due to synergistic effects and not just by zeta_1 toxin unleashing as the authors try to suggest.

Again, the reviewer points to an important caveat of our initial analysis, where we only could demonstrate that determinants carried on the pTetM plasmid are responsible for the NQNO-sensitivity of the investigated gonococcal strains. As described in our response to comment #1 of the reviewer, we have now complemented the NQNO-resistant and multidrug-resistant strain WHO P with a construct that exclusively provides the zeta1/epsilon1 operon. As now shown in novel Fig. 5, the zeta1/epsilon1-transformed WHO P is becoming sensitive to NQNO and the level of NQNO sensitivity of WHO P with a chromosomally encoded zeta1/epsilon1 operon is comparable to the NQNO-sensitivity previously established by transformation of WHO P with the complete pTetM-plasmid. These novel results clearly demonstrate that expression of the zeta1/epsilon1 TA pair is sufficient to confer NQNO sensitivity to a NQNO-resistant strain of *N. gonorrhoeae*. These results do not rule out that additional gonococcal determinants contribute to the bacteriostatic effect of NQNO, but clearly help to demarcate an essential role of zeta1/epsilon1 in this process.

To incorporate these important novel data, we have included an additional Main Figure (new Fig. 5) and novel Supplementary data (Suppl. Fig. 6) together with an additional paragraph in the Results section on page 14, line 315ff. which now reads:

“The Zeta1/Epsilon1 TA system mediates the increased NQNO sensitivity of gonococci

To unambiguously pinpoint the contribution of the zeta-epsilon TA system, we focused on the multidrug-resistant WHO P strain, which lacks the complete pTetM plasmid including both

zeta/epsilon operons and which exhibits NQNO resistance (Fig. 3B and Fig. 4A). To this end, we introduced either the pTetM plasmid or the isolated zeta1/epsilon1 operon into WHO P (Suppl. Fig. S6). Both, the plasmid-encoded (WHO P pTetM A and WHO P pTetM B) and the chromosomally-encoded zeta1/epsilon1 genes (WHO P $\zeta1/\epsilon1$) were expressed at equal levels (Fig. 5A). Most importantly, Zeta1/Epsilon1-expressing derivatives of WHO P became NQNO-sensitive, while the parent strain was fully resistant up to 50 μ M (Fig. 5B). Similar to the introduction of pTetM into strain MS11-R2 (Fig. 4F), the presence of the Zeta1/Epsilon1 TA pair resulted in dramatically increased membrane permeability in WHO P upon NQNO exposure (Fig. 5C and D). Our genetic data tightly link the NQNO-susceptibility of gonococci to the presence of the zeta1/epsilon1 operon. Furthermore, the observed membrane permeability phenotype is in line with the known activity of the Zeta toxin. Therefore, we hypothesized that NQNO might promote Zeta1 activity to compromise cell wall synthesis and to stall gonococcal growth.”

Comment 3) Mechanism of action of zeta toxins: Zeta toxins have been shown to phosphorylate a number of cytosolic, UDP-activated sugar compounds and were proposed to exert toxicity by interfering with bacterial peptidoglycan synthesis in *Neisseria gonorrhoea*. However, a zeta toxin from *Streptococcus pyogenes* was proposed to be a kind of UDP-N-acetylglucosamine stimulated ATP hydrolase (for instance: <https://doi.org/10.3389/fmicb.2017.01130>). On the other hand, a zeta toxin has been identified in eukaryotic *Leishmania donovani* recently (<https://doi.org/10.1002/1873-3468.13429>), an organism that does not produce peptidoglycan at all. Although the authors do not go much into detail when describing the observed gonococcal phenotype, it would be appropriated and fair if the authors would occasionally discuss their observation with a more wholistic view on existing literature.

The reviewer suggests that we should allow a broader discussion of zeta toxin action in various organisms. Clearly, we have already in the initial manuscript pointed to the established mechanism of action of the gonococcal zeta toxin and cited the relevant literature for the gonococcal zeta/epsilon TA system including mechanistic and structural studies (Mutschler et al. (2011). A novel mechanism of programmed cell death in bacteria by toxin-antitoxin systems corrupts peptidoglycan synthesis. *PLoS Biol* 9, e1001033; Rocker et al. (2018). The ng_zeta1 toxin of the gonococcal epsilon/zeta toxin/antitoxin system drains precursors for cell wall synthesis. *Nature Commun* 9, 1686). We also had eluded to the prominent role of zeta/epsilon orthologues in Gram positive bacteria and explicitly cited work on the related staphylococcal zeta/epsilon (PezA/T) TA system (Brzowska, I., and Zielenkiewicz, U. (2013). Regulation of toxin-antitoxin systems by proteolysis. *Plasmid* 70, 33-41. Brzowska, I., and Zielenkiewicz, U. (2014). The ClpXP protease is responsible for the degradation of the Epsilon antidote to the Zeta toxin of the streptococcal pSM19035 plasmid. *J Biol Chem* 289, 7514-7523). Also in the initial submission, we had cited several authoritative reviews on TA systems, which cover the full breadth of potential toxin mechanisms and downstream consequences (Gerdes, K., Christensen, S.K., and Lobner-Olesen, A. (2005). Prokaryotic toxin-antitoxin stress response

loci. *Nature Rev Microbiol* 3, 371-382; Harms, A., Brodersen, D.E., Mitarai, N., and Gerdes, K. (2018). Toxins, Targets, and Triggers: An Overview of Toxin-Antitoxin Biology. *Mol Cell* 70, 768-784; Hayes, F. (2003). Toxins-antitoxins: plasmid maintenance, programmed cell death, and cell cycle arrest. *Science* 301, 1496-1499) as well as additional references, which refer to examples of Type II TA systems (Kolodkin-Gal, I., and Engelberg-Kulka, H. (2006). Induction of *Escherichia coli* chromosomal mazEF by stressful conditions causes an irreversible loss of viability. *J Bacteriol* 188, 3420-3423; Leplae, R., Geeraerts, D., Hallez, R., Guglielmini, J., Dreze, P., and Van Melderen, L. (2011). Diversity of bacterial type II toxin-antitoxin systems: a comprehensive search and functional analysis of novel families. *Nucl Acids Res* 39, 5513-5525; Mutschler, H., and Meinhart, A. (2011). epsilon/zeta systems: their role in resistance, virulence, and their potential for antibiotic development. *J Mol Med (Berl)* 89, 1183-1194). We do not see that further expansion to related proteins, in particular zeta-related proteins of eukaryotes without clear functional assignment (<https://doi.org/10.1002/1873-3468.13429>), would strengthen the discussion section. Therefore and also due to space limitations, we have refrained from further expanding and elaborating on the interesting theme of zeta toxin action in various microorganisms, which might be best addressed in a timely review article on this topic.

Comment 4) Are there any unwanted off-target effects of NQNO in eukaryotes? The authors decided for topical application of NQNO in mice which apparently did not exert any detrimental effect to the mucosa and the entire genital track. Furthermore, NQNO was tested for potential off-target effects on vaginal epithelial cells as well as cervix carcinoma cells in vitro. Can the authors report on pharmacological parameters that would exclude for instance oral administration? For instance, what is the lethal dose for mice or how strong is the hepatic clearance in model systems? Although not essential for the general conclusions drawn in the manuscript, it would give a better view on the potential of NQNOs in anti-gonococcal therapy.

As you might expect, these questions are also of enormous interest to us and, as mentioned by the reviewer, we have tried to clarify the initial (and for us accessible) parameters with our *in vitro* cytotoxicity assays, the *in vivo* histopathology as well as the investigation into compound stability in the presence and absence of serum, which were already included in the initial manuscript. We are not equipped for detailed pharmacokinetic and pharmacodynamic analysis and have only begun to assess critical parameters of NQNO (LogP, mutagenesis (Ames test), interference with cytochrome P450 enzymes, hERG inhibition) with the help of contract research organisations. So far, none of these initial tests indicates any major limitation in the further development of NQNO-derived compounds for *in vivo* use. As these data do not add to the novel insight reported in this manuscript, we will report those data together with the further chemical diversification of NQNO-related compounds in a future manuscript.

Reviewer #2 (Remarks to the Author):

I'm glad for the opportunity to review "Quinolone-N-oxides kill multidrug resistant *N. gonorrhoeae* by unleashing the endogenous zeta toxin" by Mix et al. The authors have shown that 2-nonyl-4-quinolone N-oxide might have strong antimicrobial activity and specially against *Neisseria gonorrhoeae*. The manuscript is very comprehensive with some intriguing findings.

Minor comments:

Please replace the WHO, 2016 reference with "Global progress report on HIV, viral hepatitis and sexually transmitted infections, 2021. May 20, 2021".

The reference has been replaced by the more recent report and the estimated number of annual gonorrhoea cases has been updated to 82 million

The same for da Costa-Lourenco et al., 2017 – replace with Unemo, M., Seifert, H. S., Hook, E. W., 3rd, Hawkes, S., Ndowa, F., & Dillon, J. R. (2019). Gonorrhoea. *Nature reviews. Disease primers*, 5(1), 79.

We have also added this more current reference suggested by the reviewer

Row 81-85: Please revise though the conjugative plasmid carrying TetM is not present in a large fraction in all *Neisseria gonorrhoeae* but can differ a lot between regions. The authors should revise the sentence to that it is present in large fractions in some countries. This is also stated by Cehovin et al.

The sentence has been revised according to the reviewer's suggestions to now read on page 4, line 81:

"An example is the acquisition of tetracycline resistance in the form of the tetM gene and its worldwide spread via the gonococcal conjugative plasmid (pTetM), which now has a high prevalence in low income countries (Cehovin & Lewis, 2017; Cehovin et al., 2020)."

Authors should include a reference for Row 102-103 stating "Another group of AQ derivatives comprises the 2-alkyl-4-quinolone N-oxides (AQNOs), which do not function as signalling molecules in *Pseudomonas*."

The references by (Szamosvari and Böttcher, 2017) and (Szamosvari et al., 2020) have been included.

Row 104: Print out the abbreviation for HQNO and NQNO when mentioned in text for the first time.

The abbreviation “NQNO” is detailed in the summary (line 58) and in the introduction section (line 113). The abbreviation “HQNO” is now detailed in the introduction section in line 105.

Row 512-513: Is there a reason the authors used OD550 for Neisseria spp?

The reading of culture optical density at 550nm is a long-established standard for following Neisseria growth in liquid culture and has been used by us and others in the past (see e.g.: Gibbs, C.P., B.-Y. Reimann, E. Schultz, A. Kaufmann, R. Haas, and T.F. Meyer. 1989. Reassortment of pilin genes in Neisseria gonorrhoeae occurs by two distinct mechanisms. Nature. 338:651-652; Hauck, C.R., T.F. Meyer, F. Lang, and E. Gulbins. 1998. CD66-mediated phagocytosis of Opa52 Neisseria gonorrhoeae requires a Src-like tyrosine kinase- and Rac1-dependent signalling pathway. EMBO J. 17:443-454; Malott, R.J., B.O. Keller, R.G. Gaudet, S.E. McCaw, C.C. Lai, W.N. Dobson-Belaire, J.L. Hobbs, F. St Michael, A.D. Cox, T.F. Moraes, and S.D. Gray-Owen. 2013. Neisseria gonorrhoeae-derived heptose elicits an innate immune response and drives HIV-1 expression. Proceedings of the National Academy of Sciences of the United States of America. 110:10234-10239; Roth A, Mattheis C, Muenzner P, Unemo M, Hauck CR. 2013. Innate recognition by neutrophil granulocytes differs between Neisseria gonorrhoeae strains causing local or disseminating infections. Infect Immun. 81: 2358-70. doi: 10.1128/IAI.00128-13; Muenzner, P. and C.R. Hauck. 2020. Neisseria gonorrhoeae Blocks Epithelial Exfoliation by Nitric-Oxide-Mediated Metabolic Cross Talk to Promote Colonization in Mice. Cell Host & Microbe. 27:793-808 e795). Therefore, we have continued to monitor neisserial growth in liquid medium at this wavelength, while growth of other microbes (Klebsiella, Escherichia, Lactobacillus, etc.) was monitored at 600 nm.

Row 533: Please include the source of the NQNO. If this was synthesized, please include a section for this.

The synthesis and high resolution MS-based as well as NMR-based quality control of all the substances used in antibiotic assays (including NQNO) was already detailed in the Supplementary Information S1 (50+ pages including original NMR spectra).

Accordingly, the details of NQNO synthesis are only briefly mentioned in the Material & Methods section on page 24:

“Synthesis of AQNOs

The saturated AQs were synthesized via Conrad-Limpach cyclization and the unsaturated trans-Δ1-NQ by Camps cyclization as described previously (Szamosvari and Böttcher, 2017). The N-oxides were obtained by locking the 4-hydroxyquinolins as ethyl carbonates followed by N-oxidation with mCPBA (m-chloroperoxybenzoic acid) and deprotection. A list of names,

abbreviations, and structures of all synthetic compounds is given in Suppl. Table 4. The details of compound synthesis together with the characterization and confirmation by NMR spectroscopy and high resolution mass spectrometry is described in Suppl. Info S1.

Row 564: Is this OD600?

See also comment above with regard to determination of optical density. We have now clarified this sentence on page 26, line 576 to read:

*“Optical density at 550 nm (*N. gonorrhoeae* and commensal *Neisseriae*) or 600 nm (all other microorganisms) was determined and 4×10^7 bacteria were inoculated in 5 ml of the respective growth medium.”*

Row 630: I suggest that the authors refer to the 500 bp paired-end to have instead 250 bp paired-end reads. Genome determination and comparison section: The authors is strongly advised to upload the raw fastq files to the Bioproject as well.

We thank the reviewer for pointing us to this ambiguous wording. Accordingly, we have corrected this statement in line 660 according to the reviewer’s suggestion. We also uploaded the raw fastq files of the sequencing results to Bioproject number PRJNA728975 at NCBI GenBank.

Row 135: Authors state that they tested other “pathogenic *Neisseriae*”. But I can’t see that the authors have tested *P. aeruginosa* factors against *N. meningitidis*. Why is this?

This is indeed a keen observation by the reviewer. We have refrained from using meningococci in our analysis, as my current co-workers involved in this project (PhD students and M.Sc. students) do not have a basic vaccination with one of the meningococcal vaccines. Since this would be a mandatory pre-requisite according to our institutional biosafety guidelines, we are limited in our current ability to handle viable meningococci. However, we plan to offer vaccination to the next generation of student co-workers to be able to extend our analysis to meningococci in the near future. This would be particularly interesting as the closest relative of gonococci and meningococci, *Neisseria lactamica*, also showed sensitivity towards NQNO suggesting that further testing of these compounds against *N. meningitidis* might be a worthwhile endeavour in the future.

To clearly indicate this focus of our analysis, we have rephrased the respective sentence on page 8, line 155 to now read:

*“The compounds were dissolved in DMSO and applied in increasing concentrations (5 - 50 μ M) to broth cultures of pathogenic *Neisseria gonorrhoeae* or commensal *N. cinerea*.”*

While we could not extend our analysis to meningococci, which occupy a distinct niche in our body, we have tested a wider variety of commensal bacteria of the female vaginal tract (including *Lactobacillus delbrueckii*, *L. gasseri*, *L. paragasseri*, *L. homini*, *L. jensenii* and *Limosilactobacillus vaginalis*) for their susceptibility towards NQNO and also 3Me-NQNO. With these additional data, which are now included as novel panels in Figure 1E and Suppl. Fig. S4B (NQNO) as well as Fig. 8H and Suppl. Fig. S8E (3Me-NQNO), we provide additional evidence for the selective action on gonococci and the natural resistance of commensal vaginal bacteria against these two compounds.

Figure 1A: The authors refer to this electron microscopy scan to conclude that *Neisseria cinerea* is not affected by PAO1. How can the reader discern this? To me, both strains are affected in a similar way including intact cells. Furthermore, I imagine that the authors have chosen a very well representative, even the best, scan photo.

We thank the reviewer for this comment, as it is always difficult to select from the multitude of pictures, which we have acquired. Indeed, the ones we had chosen originally already demonstrate the damaged appearance and the resulting cellular debris of the gonococci treated with *P.aeruginosa* culture supernatants compared to the other samples. However, upon reduction of these micrographs to the size used in the panel these features are harder to discern. Therefore, we have chosen additional micrographs from our analysis of these samples and have used a slightly larger magnification. As can be seen, the gonococci incubated with *P. aeruginosa* culture supernatant are strongly damaged and few intact bacteria remain, while *N. cinerea* is not harmed by the same culture supernatants and both, *N. gonorrhoeae* and *N. cinerea*, show the regular morphology with mainly intact diplococcal structures, when grown under control conditions. The new panel now replaces former Fig. 1A.

Figure S1. Why are different strains used in the different experiments? I.e in figure S1A the authors have used MS11, while in S1B and S1C they have used Ngo 11. I would rather that the authors include all tested *N gonorrhoeae* isolates in the figure.

During our study, we used as many different gonococcal strains as we could to demonstrate the broad applicability of our findings. This is also true for Figure S1, where we show the original results (pictures of the GC agar growth plates) for *N. gonorrhoeae* strain MS11 (which shows intermediate growth inhibition in these cross-streak assays; see Fig. S1B) together with results for gonococcal strain Ngo 11 and strain *N. cinerea*, which show maximal (Ngo11) or minimal (*N. cinerea*) response in these growth assays. These examples of original data were chosen to demonstrate the complete breadth of the observed responses and all the data for all the strains are summarized in Suppl. Figure S1B in a single panel. To present, as suggested by the reviewer, the results for all the strains in the form of depicting the original plates would clearly bust the Figure and would require two additional letter-size pages minimum.

The authors need to include figure texts for supplementary figures.

The Figure legends for the Suppl. Fig. were included in the manuscript and they are found from page 38ff.

Row 163-165: Pefloxacin is not a relevant antibiotic for the treatment of *Neisseria gonorrhoeae*. Please include ciprofloxacin MICs instead.

This sentence is not meant to introduce Pefloxacin as an antibiotic for *Neisseria*, but as a general example of an antibiotic compound, which is in clinical use at concentrations that match the observed minimum inhibitory concentrations (MICs) for NQNO. With this statement we would like to indicate that already our initial hit compound NQNO is effective at concentrations used in other instances of antibiotic therapy. In the case of ciprofloxacin, due to drug resistance, the MICs reported for the *N. gonorrhoeae* strains used in our study vary from 0.004 - <32 mg/l.

In the course of this study, we identify an improved NQNO-derived compound (3-MeNQNO, former Fig.7, now Fig. 8 in the revised manuscript), which shows a dramatic increase in efficacy. We envision that further derivatisation at this position could lead to even further increases in efficacy in the near future (see also our response to reviewer #3, comment #10). Therefore, we do not feel that it is warranted at this stage to include a specific comparison of NQNO efficacy with other currently used antibiotics for the treatment of gonorrhea, as the development of NQNO-derived compounds is only in its infancy and poised to see significant improvements in the short term as demonstrated in our study.

Major comments

The manuscript includes an unreasonable number of different figures. As reader it is very confusing when the authors use A to even H of unrelated figures. E.g how is figure 1A related to 1B? The authors could revise these as e.g figure 1 and 2 instead. This should be applied to all figures.

We agree with the reviewer that our manuscript is rich in primary data, which we display in several Main and Supplementary Figures. However, we think this is a positive and strong aspect of our contribution. In all the Figures we try to assemble panels, which are intimately related. In the example mentioned by the reviewer, Fig. 1A shows the distinct effect of *P. aeruginosa* culture supernatants on commensal (*N. cinerea*) vs. pathogenic (*N. gonorrhoeae*) *Neisseria* by scanning electron microscopy, while Fig. 1B gives the chemical structures of some of the known ingredients of *P. aeruginosa* supernatants, which are in the center of the following experiments. I think the thematical connection is obvious, but of course this can be best appreciated by reading the accompanying text in the results section. As the total amount of Figures is limited, it is not possible to simply subdivide each Main Figure into two or three new Figures. On the other hand, we do not want to reduce the content of the paper, as we

believe that the entirety of all the presented data is of particular value for the reader. Therefore, we have not re-shuffled or reduced Figures.

The authors are very clear that *N. cinerea* is resistant to NQNO. However, the authors use *N. cinerea* data to draw conclusions for all commensal *Neisseria*. This should be revised and clear that NQNO does affect commensals in various degrees, eg see row 198-200.

We have followed this valuable suggestion by the reviewer and adapted the text at several instances, where we had generalized to all commensal *Neisseriae*. We now clearly indicate that NQNO does not affect most commensals, but that some, in particular *N. lactamica*, are as vulnerable as *N. gonorrhoeae*. Furthermore, in our later functional experiments we explicitly refer to *N. cinerea*, which was used as a control throughout the study, but not to the totality of commensal *Neisseriae* :

Page 5, line 116:

“Strikingly, NQNO does not act on most non-pathogenic neisserial species or unrelated lactobacilli, which live as commensals on human mucosal surfaces, and effective doses of NQNO do not harm human epithelial cells in vitro.”

Page 9, line 178:

“While all gonococcal strains were strongly impaired in their growth, most commensal Neisseriae showed negligible sensitivity to NQNO (Fig. 1D). An exception was seen for N. lactamica, a species generally regarded as commensal, but closely related to N. gonorrhoeae (Bennett et al., 2007) (Fig. 1D).”

Page 9, line 200:

“However, ATP levels were unaffected in N. cinerea suggesting that respiratory chain function and ATP synthesis in this commensal Neisseria species was not reduced by NQNO (Fig. 2A).”

The authors could include limitations of the study in Discussion, eg the authors suggest that NQNO could be used via local vaginal administration. However, authors have not tested many of the bacteria in the vaginal microbiota. The authors could still include this hypothesis but include the limitations of the study. Furthermore, using antibacterial compounds freely, especially prior to exposure could lead to rapid resistance in the gonococcal population. The authors can include and discuss around this possibility and the importance of antibiotic stewardship.

We thank the reviewer for this advice, but it is exactly in this section of the discussion section, where we clearly indicate that the used topical application of NQNO in the vivo model would represent a limitation for the clinical use of this compound as the systemic oral or parenteral treatment is in most cases the preferred route of administration. In the same section, we also

refer to the limitations of the used animal model, which does not recapitulate the human disease. Therefore, we think that we frankly pointed to limitations of the study. However, the reviewer makes an important point in that we only tested a single commensal non-neisserial bacterial species from the female vaginal tract for its sensitivity towards NQNO. To address this weak point, we have now conducted additional growth assay with a wide selection of lactobacilli (*Lactobacillus brevis*, *L. delbruecki*, *L. gasseri*, *L. paragasseri*, *L. homini*, *L. jensenii* und *Limosilactobacillus vaginalis*) and tested them not only with NQNO, but also with the improved lead compound 3Me-NQNO.

As you can see in the novel data panels in Figure 1E and Suppl. Fig. S4B (NQNO) as well as Fig. 8H and Suppl. Fig. S8E (3Me-NQNO), none of these Gram-positive commensal bacteria is sensitive towards concentrations of NQNO or 3Me-NQNO, which strongly affect *N. gonorrhoeae*. In addition to these Figure panels, we have also included a sentence in the discussion section on page 21, line 484 to underscore these novel findings:

“Therefore, this class of compounds might serve as a starting point to derive pathogen-selective compounds, sparing beneficial commensal bacteria such as lactobacilli from antibiotic action.”

Finally, although the manuscript is well written, it draws very large conclusions that would need further studies with a larger, up-to-date collection of gonococcal isolates. The tested isolates are mainly lab strains or isolates no longer circulating in the gonococcal population. The gonococcal population have shifted towards azithromycin resistance (due to change in treatment recommendations) and it is worrying that the WHO P (azithromycin resistant strain that have clonally expanded) is resistant to NQNO. The authors could mention how this can be addressed in future studies.

We agree with the reviewer that it will be useful to expand the tested gonococcal strains even further in the future (a total of 14 independent gonococcal strains has been tested by us). However, this is the initial identification and characterization of this class of compounds and their activity towards gonococci (and towards a large number of related and unrelated bacteria (9 commensal Neisseria species; 7 distinct Lactobacilli; 4 distinct further microbes), where the emphasis clearly is not only on detecting effects on more and more isolates, but rather on understanding the mechanistic principles underlying the selective action of these compounds.

We are totally aware of the current shift in the resistance spectrum of *N. gonorrhoeae* in parts of the world towards azithromycin resistance (see e.g. Lu Z, Tadi DA, Fu J, Azizian K, Kouhsari E. (2022) Global status of Azithromycin and Erythromycin Resistance Rates in Neisseria gonorrhoeae: A Systematic Review and Meta-analysis. Yale J Biol Med. Dec 22;95(4):465-478). Therefore, we had not only included WHO P, but also a recent clinical isolate from the UK (NCTC 13799; isolated 2015 in Yorkshire), which we obtained from a public repository (National Collection of Type Cultures, Salisbury, United Kingdom). This strain exhibits high-level resistance to Azithromycin (>256). See also

<https://www.culturecollections.org.uk/products/bacteria/detail.jsp?refId=NCTC+13799&collecton=nctc>). Most importantly, we found that this recent clinical isolate with high level azithromycin resistance is highly susceptible towards NQNO. Indeed, we reported this finding already in the initial version of this manuscript and these results for Ngo NCTC 13799 are shown in Figure 3C of the revised manuscript.

We would like to stress that we tested further relevant antibiotic resistant clinical isolates, in particular the ceftriaxone-resistant strain N231 isolated 2011 in Slovenia (Unemo et al. (2012). Treatment failure of pharyngeal gonorrhoea with internationally recommended first-line ceftriaxone verified in Slovenia, September 2011. Euro surveillance 17), which also showed high susceptibility towards NQNO. Of course, there is an endless list of further strains, which could be tested, but several interesting recent strains were not available to us. For example, we have not been able to obtain access to the ceftriaxone/azithromycin-resistant recent isolates from the UK and Australia (Jennison AV, Whiley D, Lahra MM, Graham RM, Cole MJ, Hughes G, et al. Genetic relatedness of ceftriaxone-resistant and high-level azithromycin resistant *Neisseria gonorrhoeae* cases, United Kingdom and Australia, February to April 2018. Euro Surveill. 2019 doi: 10.2807/1560-7917.Es.2019.24.8.1900118). Of course we will stay alert to include such kind of latest strains in future analyses, but for the moment we find it fair to conclude that even for current antibiotic resistant isolates, including high Azithromycin-resistant strains, NQNO is effective.

Most importantly, our novel data obtained with the WHO P strain (see comment #1 and #2 of reviewer #1 and also the novel Fig. 5) further demonstrate that it is the lack of the zeta/epsilon TA system, but not the azithromycin resistance, which render this strain insensitive to NQNO. Accordingly, introduction of the zeta1/epsilon1-operon into WHO P render this strain sensitive to NQNO, despite the existing azithromycin resistance (see novel Fig. 5). The conclusion that an existing azithromycin resistance does not affect vulnerability towards NQNO is also in complete agreement with the known action of macrolides such as azithromycin, which interfere with protein synthesis, while NQNO does compromise, according to the data presented in this manuscript, the electron transport chain of gonococci.

Therefore, we stay firm with our initial judgement, that there is no link between azithromycin resistance and NQNO resistance and that even a current high-level azithromycin-resistant strain is sensitive towards this compound, as we state on page 11, line 243:

“As WHO-P was the only strain in this collection with known resistance to azithromycin, we tested an additional set of multidrug-resistant strains either highly resistant to azithromycin (Azm) or to 3rd generation cephalosporins (Ceph) (Fig. 3C). While the azithromycin-resistant isolate NCTC 13799 and the ceftriaxone-resistant strain N231 (Unemo et al., 2012) were fully susceptible to NQNO, the cefixime-resistant strain N316 (Unemo et al., 2011) showed intermediate resistance to NQNO (Fig. 3C). Accordingly, azithromycin-resistance does not explain the lack of NQNO-mediated growth inhibition of strain WHO-P.”

**Reviewer #3 (Remarks to the Author):**

The *P. aeruginosa* alkyl quinolone N-oxides (AQNOs) were originally discovered as potential antibiotics in the 1950s. Their low in vivo antibacterial efficacy with activity primarily against Gram positives combined with their cytotoxicity for eukaryotic cells (electron transport inhibition) generally prevented the development of AQNOs as therapeutic agents. Mix et al make some interesting observations on the selective susceptibility of *Neisseria gonorrhoea* for NQNO compared with the resistance of commensal *Neisseria* strains. The selectivity observed was due to the release of a zeta toxin from its cognate antitoxin, the genes for which are carried on a large pTetM conjugative plasmid commonly found in gonococci but not in commensal *Neisseria*. In a female mouse genital tract colonization and infection model, topical NQNO blocked colonization. In vitro screening of a selection of modified AQNOs yielded a 3Me-substituted analogue with nanomolar efficacy. Given that *N. gonorrhoeae* is a major global public health concern and a World Health Organization (WHO) priority antibiotic-resistant pathogen, this is an important topic given the urgent need for new therapeutics for the treatment and prevention of gonorrhoea.

Specific points

1. Line 151 and 191. Why did the authors select the C9 compound NQNO rather than the more commonly investigated C7 compound HQNO? Was there any difference in potency?

In our prior work (Szamosvari et al. (2020) Chem. Commun. 56: 6328), we established that C9 quinolones are equally prevalent as C7 compounds in the culture supernatants of *P. aeruginosa*. We also had observed that quinolone-derivatives with a longer alkyl chain (C10) can show higher potency (Szamosvari et al. (2019) A thiochromenone antibiotic derived from the *Pseudomonas* quinolone signal selectively targets the Gram-negative pathogen *Moraxella catarrhalis*. Chemical Science 10, 6624-6628). Therefore, we have initially selected C9 compounds for our experiments. Ongoing extensive SAR studies now reveal that chain lengths shorter than C9 display reduced potency against *N. gonorrhoeae*. These data will be part of a future study detailing a more extensive SAR (see also answer to comment #10).

2. Fig. S1. (C), the inhibition data for the *P. aeruginosa* pqsH mutant is surprising given that it still makes AQNOs. Although the pqsL mutant doesn't make AQNOs it will still produce NHQ which the authors show has very good activity against *N. gonorrhoeae*.

Cross-streak experiments and *P. aeruginosa* mutants of the pqs system initially guided our screening process, prompting us to explore AQNOs more deeply. However, these results only gave us initial hints, but we would not draw explicit conclusions solely based on this approach. For example, disrupting parts of the pqs biosynthetic gene cluster has a variety of effects including quorum sensing-related effects and production of outer membrane vesicles (OMVs). In this respect, the pqsH mutant is also known for its impaired production of outer membrane vesicles (OMVs), crucial for quorum sensing, as well as packaging and transporting antimicrobial quinolones such as AQNOs. These defects could contribute to the reduced

inhibition zone observed for the *pqsH* mutant (Mashburn LM, Whiteley M. Membrane vesicles traffic signals and facilitate group activities in a prokaryote. *Nature*. 2005 Sep 15;437(7057):422-5. doi: 10.1038/nature03925). As correctly pointed out by the reviewer, the *pqs* mutant still produces Aqs, but lacks the more potent AQNOs. This could account for the presence of a smaller, yet still noticeable zone of inhibition in this strain.

3. Lines 217-219. Why did the authors only treat the hVEC and HeLa cells with a maximum of 10 microM which is very close to the gonococcal MIC of 5 microM and well below the therapeutic dose that would require administration. The claim (line 117) that effective doses do not harm epithelial cells in vitro should be qualified given the low concentrations tested.

We agree with the reviewer that this is a limited dose range, which we tested for its effects on human epithelial cells. Accordingly, we have now repeated these kind of in vitro cytotoxicity tests i) using a wider range of NQNO concentrations including significantly higher concentrations; and ii) extending the investigated cell lines to also include a further human cervical epithelial cell line model for *N. gonorrhoeae* infection, the ME-180 cells (see e.g. Muenzner P, et al. (2005) CEACAM engagement by human pathogens enhances cell adhesion and counteracts bacteria-induced detachment of epithelial cells. *J Cell Biol*. 170: 825-36. doi: 10.1083/jcb.200412151; Muenzner P et al. (2010) Human-restricted bacterial pathogens block shedding of epithelial cells by stimulating integrin activation. *Science*.;329: 1197-201. doi: 10.1126/science.1190892); and iii) extended the cytotoxicity testing to the latest developed compound 3Me-NQNO. The novel data are now presented in graphs showing the growth results for all three distinct human epithelial cell lines for each compound. The graphs are now included in Fig. 2E-F and Suppl. Fig. S5A-B (NQNO) and Suppl. Fig. S8A-D (3Me-NQNO).

4. Lines 274 to 276. There are two similar zeta/epsilon toxin-antitoxin systems on the plasmid, why did the authors only focus on the zeta 1/epsilon 1 system since both would be lost when the plasmid was lost during NQNO conditioning and conversely restored when the plasmid was re-introduced. Deletion of each system individually would be informative. It is also possible that other conjugative plasmid genes are involved. Why not introduce the zeta 1/epsilon 1 system alone on a plasmid into a resistant gonococcus strain?

This is a very important point made by the reviewer and argues along the same line as comments #1 and #2 of Reviewer #1 (see above). Following the suggestion of both reviewers, we have introduced a construct, which exclusively encodes the *zeta1/epsilon1* operon, into the NQNO-resistant and multidrug-resistant strain WHO P. As now shown in novel Fig. 5, the *zeta1/epsilon1*-transformed WHO P is becoming sensitive to NQNO and the level of NQNO sensitivity of WHO P with the newly introduced *zeta1/epsilon1* operon is comparable to the NQNO-sensitivity previously established by transformation of WHO P with the complete pTetM-plasmid. These novel results clearly demonstrate that expression of the Zeta1/Epsilon1 TA pair is sufficient to confer NQNO sensitivity to a NQNO-resistant strain of *N. gonorrhoeae*. These results do not rule out that additional gonococcal determinants contribute to the

bacteriostatic effect of NQNO, but clearly help to demarcate an essential role of Zeta1/Epsilon1 in this process.

To incorporate these important novel data, we have included an additional Main Figure (Fig. 5) and an additional paragraph in the results section on page 14, which now reads:

“The Zeta1/Epsilon1 TA system mediates the increased NQNO sensitivity of gonococci

To unambiguously pinpoint the contribution of the zeta-epsilon TA system, we focused on the multidrug-resistant WHO P strain, which lacks the complete pTetM plasmid including both zeta/epsilon operons and which exhibits NQNO resistance (Fig. 3B and Fig. 4A). To this end, we introduced either the pTetM plasmid or the isolated zeta1/epsilon1 operon into WHO P (Suppl. Fig. S6). Both, the plasmid-encoded (WHO P pTetM A and WHO P pTetM B) and the chromosomally-encoded zeta1/epsilon1 genes (WHO P ζ1/ε1) were expressed at equal levels (Fig. 5A). Most importantly, Zeta1/Epsilon1-expressing derivatives of WHO P became NQNO-sensitive, while the parent strain was fully resistant up to 50 μM (Fig. 5B). Similar to the introduction of pTetM into strain MS11-R2 (Fig. 4F), the presence of the Zeta1/Epsilon1 TA pair resulted in dramatically increased membrane permeability in WHO P upon NQNO exposure (Fig. 5C and D). Our genetic data tightly link the NQNO-susceptibility of gonococci to the presence of the zeta1/epsilon1 operon. Furthermore, the observed membrane permeability phenotype is in line with the known activity of the Zeta toxin. Therefore, we hypothesized that NQNO might promote Zeta1 activity to compromise cell wall synthesis and to stall gonococcal growth.”

5. Line 294. The Syto9 and PI fluorescence in Figs 4B and 4F should be quantified.

We thank the reviewer for this suggestion and have now quantified multiple images from several repetitions of this experiment. The data are now presented in Fig. 4 (*N. gonorrhoeae* strain MS11 and *N. cinerea*) as well as in novel Fig. 5 (*N. gonorrhoeae* strain WHO P and epsilon1/zeta1-derivatives). The quantified fluorescence intensity of individual bacteria depicted in the dot graph now better conveys the overall result, namely that the presence of the zeta1/epsilon1 TA system correlates with increased membrane permeability upon NQNO exposure of the gonococci.

6. Lines 311-313 also 338-339. The authors cannot rule out from the data presented that NQNO may act on both the electron chain and directly on one or both of the TA systems.

This is indeed a very valuable point made by the reviewer. To address this concern, we have now performed additional experiments in *E. coli*, which is not susceptible to NQNO. Already in the initial submission we had shown that the epsilon1 antitoxin is readily expressed in *E. coli*, but it is vulnerable to reactive oxygen (H₂O₂). If NQNO would exert a direct effect on epsilon1 antitoxin stability, we should observe degradation of epsilon1 upon treatment of *E. coli* with NQNO. However, NQNO up to concentrations of 50 μM did not lead to changes in epsilon1

levels. These novel data indicate that NQNO does not directly affect epsilon1 stability and these data are now included in Fig. 6D (formerly Fig.5 in the initial submission).

7. Line 361-2 and Fig. 6C. The authors state that NQNO eradicated the gonococci to below the level that could be detected by microscopy. However, there is a very clear green signal towards the top of the panel

In the immunohistological stainings depicted in former Fig. 6C (now Fig. 7C in the revised manuscript) mentioned by the reviewer, there are clear positive, punctuate signals with a size $<5 \mu\text{m}$ resulting from antibody-stained gonococci. These signals are only detected in the DMSO-treated control sample. They can be best appreciated in the enlargements provided on top of the panel. In both, the DMSO control and the NQNO-treated sample there is a broad background staining of low intensity, which is seen throughout the epithelial layer and which is presumably due to the keratinization of the stratified epithelium of the vaginal tract. However, this low-intensity background staining is clearly distinct from the strong, punctuate staining of individual bacteria and small bacterial clusters, which are present in the DMSO-treated samples. The staining pattern for the DMSO-treated control samples is highly similar to mucosa-associated gonococci on the murine vaginal epithelium, which we have reported before in this *in vivo* model system (see e.g. Muenzner P et al. (2010) Human-restricted bacterial pathogens block shedding of epithelial cells by stimulating integrin activation. *Science*;329: 1197-201. doi: 10.1126/science.1190892).

8. Line 370. Fig 6E should be 6F.

We apologize for this mistake. We have corrected the assignment of panels in this Figure, which is now Fig. 7 of the revised manuscript.

9. Lines 373-374. The authors only tested one strain *in vivo* so cannot claim NQNO also suppressed viability of extended drug resistant isolates *in vivo*.

We disagree with the reviewer at this point, as we have already evaluated the potential of NQNO in our *in vivo* infection model in CEA-transgenic female mice using two distinct strains of *Neisseria gonorrhoeae*. One strain used was our laboratory strain MS11, which we have used throughout this manuscript and also extensively employed in prior publications. This strain has been isolated from a local symptomatic infection, but has been kept and genetically modified in the lab to now encode Erythromycin, Chloramphenicol and Tetracyclin resistance genes. Besides strain MS11, we have also used a recent multidrug-resistant isolate from Austria (Ngo 316; Unemo, M., et al. 2011. First *Neisseria gonorrhoeae* strain with resistance to cefixime causing gonorrhoea treatment failure in Austria. *Euro Surveill*, 16), which is resistant against 3rd generation cephalosporines and also against ciprofloxacin.

All these data have been presented in former Fig. 6, which is now Fig. 7 in the revised manuscript. Based on these data and based on the comment of the reviewer, we have rephrased the respective sentence on page 17, line 396 to conclude that:

“Together, these results demonstrate that NQNO can suppress viability and in vivo growth of gonococci, including in vivo growth of a clinical isolate with extended-drug resistance.”

10. Line 376. There is little in the way of a systematic SAR. Given the increased potency of the 3Me AQNO derivative, the synthesis of further 3 position analogues would be informative as would a shorter 2 position C9 alkyl chain to make the compound more drug-like.

— Already in this initial characterization, we present data for ~30 AQNO and AQ derivatives, which we have generated to systematically explore the consequences of substitutions at different positions around the quinolone ring. With this approach we identify and report that substitution at the 3 position increases potency of NQNO. We agree with the reviewer that would be most interesting to explore further substitutions at this position. Accordingly, we are expanding our collection of synthetic AQNOs right now with an emphasis on the 3 position and the alkyl chain length to come up with a comprehensive SAR study, having so far generated over 100 Aqs and AQNOs. We feel that the planned thorough exploration of the chemical space warrants in-depth discussions on structural details, their chemophysical characteristics, and the development of various synthetic routes giving access to these compounds - topics beyond the scope of the current manuscript. Our primary focus with this contribution is elucidating the mechanism and demonstrating the proof-of-principle that the toxin-antitoxin system is involved in the particular susceptibility of *N. gonorrhoeae* towards NQNO and derived compounds. Therefore, we will present the ongoing in-depth SAR analysis in a follow-up study.

— 11. Line 396-397. The authors only tested 14 repeat subcultures for increased resistance which here could relate either to loss of the conjugative plasmid or a mutation in the target of NQNO – probably the quinone reduction site cytochrome bc₁ sub-unit. Many additional subcultures are likely to be required for resistance to emerge.

The reviewer is correct in that a further prolongation of the subculturing of gonococci at low concentrations of our improved compound 3Me-NQNO might result in the emergence of resistant clones. However, we would like to state that we extended this experiment already significantly beyond the duration of our initial procedure, where we observed the emergence of resistant clones after 8 days of consecutive culturing in the presence of NQNO. Accordingly, we can conclude that 3Me-NQNO is less likely to yield resistant strains and therefore, as we have stated, *“NQNO derivatives with a favorable efficacy and safety profile can be developed”*.

Interestingly, also in the case of the NQNO-resistant isolates, which we obtained after 8 days of growth at low NQNO-concentrations, we did not find any mutations in the cytochrome b₁ subunit. This finding already indicates that there might be strong structural constraints on this protein. In the future, we will embark on a chemical-biological screen with a cross-linker

equipped derivative of NQNO to identify the primary target of NQNO, most likely aided by extended cultivation at low concentrations to generate additional, independent resistant clones.

—

—

**Reviewer Comments:****Reviewer #1 (Remarks to the Author):**

With this manuscript, Boettcher, Hauck, and colleagues provide surprising and convincing evidence on a highly specific antimicrobial activity of 2-alkyl-4-quinolones N-oxides (AQNOs) against 25.2 MDa pTetM plasmid carrying *Neisseria gonorrhoea* variants. The inhibition of respiratory chain cytochromes by AQNOs, as the authors describe, has been described in several reports in the past. However, it is new and exciting, that downstream effects of this interference can exert strain specific lethality in *N. gonorrhoea* but not against other neisserial species. In a convincing approach they started with *Pseudomonas aeruginosa* / *N. gonorrhoea* battle experiments enabling identification of AQ(NO)s as anti-gonococcal compounds secreted by *Pseudomonas aeruginosa*. Synthetic compounds were produced, their bacteriostatic or bactericidal activity was monitored and NQNO was shown to be the most potent candidate. As expected, NQNO treatment caused massive perturbation of cytosolic ATP and NADH levels accompanied with ROS production in gonococcal strains but not in *N. cinerea* which they showed to be resistant against AQ(NO)s. By generating NQNO resistant strains which in fact have lost the entire pTetM plasmid, they concluded that the ROS dependent activation of Type II epsilon zeta TA systems is required for gonococcal sensitivity. In fact, clinical variants of *N. gonorrhoea* strains that they identified to be resistant to NQNO lacked the encoding locus. Although ROS dependent activation of Type II TA systems has been reported in the past, this is, to my knowledge, the first time that unleashing a toxin from its antitoxin eventually would tip the scale in favor of exerting a bactericidal phenotype. Most importantly, they established synthesis of a highly potent NQNO lead compound, demonstrate anti-gonococcal efficacy of NQNOs in a mouse model and provided first insights for a potential therapeutic application of their compounds.

In general, the experimental work is well designed, the manuscript is conclusive and convincing. However, as described below, some aspects should be reconsidered in a more appropriate and/or careful way. If resolved, this would strengthen the impact of the manuscript.

Comment 1) The genetic load region of the 25.2 MDa pTetM plasmid contains two paralogues of the epsilon/zeta system and the authors only demonstrate activation of the epsilon_1/zeta_1 system. Is zeta_2 unleashed similarly upon ROS dependent degradation of epsilon_2? If the authors only provide experimental evidence on the mechanism of activation of zeta_1, care should be taken when generalizing their observation. At least, it would be better if they refer to epsilon_1/zeta_1 throughout the manuscript.

The reviewer is completely right in pointing out that the pTetM plasmid encodes two closely related toxin/antitoxin systems, while we only focus our attention on the zeta1/ epsilon1 TA pair. In response to this comment and also to the reviewer's comment #2 below, we have now transformed the NQNO-resistant and multidrug-resistant strain WHO P not only with the full-size pTetM plasmid (see former Fig. 4D-E), but also with a construct encoding the

zeta1/epsilon1 operon only. This allowed us to i) test the contribution of the zeta1/epsilon1 to NQNO sensitivity in the absence of zeta2/epsilon2 and ii) to test the contribution of the zeta1/epsilon1 to NQNO sensitivity in the absence of other open reading frames contained on the pTetM-plasmid. As shown in novel Fig. 5 and Suppl. Fig. S6, the construct is chromosomally integrated in strain WHO P and the zeta1 toxin / epsilon1 antitoxin are expressed at levels comparable to strains harboring the pTetM plasmid. Most importantly, selective expression of Zeta1/Epsilon1 renders the NQNO-resistant strain WHO P now sensitive to NQNO and this sensitivity is similar to the sensitivity observed after transformation of this strain with the complete pTetM plasmid. Accordingly, we can conclude that Zeta1 toxin / Epsilon1 antitoxin expression is required for the NQNO sensitivity of *N. gonorrhoeae*.

Comment 2) Is NQNO sensitivity of *N. gonorrhoea* indeed exclusively the result of zeta_1 toxin activation? In fact, loss of the entire 25.2 MDa pTetM plasmid was observed when raising strains resistant to NQNO and clinical variants that proved not to be sensitive to their compounds most likely do also lack the plasmid. Can the author exclude that sensitivity to their quinolone compounds is established from the plasmid by any other means than the zeta_1 activity? In fact, TA systems were originally described as plasmid addiction modules that ensure stable plasmid maintenance. It would be interesting to see, whether an catalytically inactive zeta_1 toxin encoded from a stably maintained plasmid also gives rise to resistance. In fact, sensitivity could be due to synergistic effects and not just by zeta_1 toxin unleashing as the authors try to suggest.

Again, the reviewer points to an important caveat of our initial analysis, where we only could demonstrate that determinants carried on the pTetM plasmid are responsible for the NQNO-sensitivity of the investigated gonococcal strains. As described in our response to comment #1 of the reviewer, we have now complemented the NQNO-resistant and multidrug-resistant strain WHO P with a construct that exclusively provides the zeta1/epsilon1 operon. As now shown in novel Fig. 5, the zeta1/epsilon1-transformed WHO P is becoming sensitive to NQNO and the level of NQNO sensitivity of WHO P with a chromosomally encoded zeta1/epsilon1 operon is comparable to the NQNO-sensitivity previously established by transformation of WHO P with the complete pTetM-plasmid. These novel results clearly demonstrate that expression of the zeta1/epsilon1 TA pair is sufficient to confer NQNO sensitivity to a NQNO-resistant strain of *N. gonorrhoeae*. These results do not rule out that additional gonococcal determinants contribute to the bacteriostatic effect of NQNO, but clearly help to demarcate an essential role of zeta1/epsilon1 in this process.

To incorporate these important novel data, we have included an additional Main Figure (new Fig. 5) and novel Supplementary data (Suppl. Fig. 6) together with an additional paragraph in the Results section on page 14, line 315ff. which now reads:

“The Zeta1/Epsilon1 TA system mediates the increased NQNO sensitivity of gonococci

To unambiguously pinpoint the contribution of the zeta-epsilon TA system, we focused on the multidrug-resistant WHO P strain, which lacks the complete pTetM plasmid including both

zeta/epsilon operons and which exhibits NQNO resistance (Fig. 3B and Fig. 4A). To this end, we introduced either the pTetM plasmid or the isolated zeta1/epsilon1 operon into WHO P (Suppl. Fig. S6). Both, the plasmid-encoded (WHO P pTetM A and WHO P pTetM B) and the chromosomally-encoded zeta1/epsilon1 genes (WHO P $\zeta1/\epsilon1$) were expressed at equal levels (Fig. 5A). Most importantly, Zeta1/Epsilon1-expressing derivatives of WHO P became NQNO-sensitive, while the parent strain was fully resistant up to 50 μ M (Fig. 5B). Similar to the introduction of pTetM into strain MS11-R2 (Fig. 4F), the presence of the Zeta1/Epsilon1 TA pair resulted in dramatically increased membrane permeability in WHO P upon NQNO exposure (Fig. 5C and D). Our genetic data tightly link the NQNO-susceptibility of gonococci to the presence of the zeta1/epsilon1 operon. Furthermore, the observed membrane permeability phenotype is in line with the known activity of the Zeta toxin. Therefore, we hypothesized that NQNO might promote Zeta1 activity to compromise cell wall synthesis and to stall gonococcal growth.”

Comment 3) Mechanism of action of zeta toxins: Zeta toxins have been shown to phosphorylate a number of cytosolic, UDP-activated sugar compounds and were proposed to exert toxicity by interfering with bacterial peptidoglycan synthesis in *Neisseria gonorrhoea*. However, a zeta toxin from *Streptococcus pyogenes* was proposed to be a kind of UDP-N-acetylglucosamine stimulated ATP hydrolase (for instance: <https://doi.org/10.3389/fmicb.2017.01130>). On the other hand, a zeta toxin has been identified in eukaryotic *Leishmania donovani* recently (<https://doi.org/10.1002/1873-3468.13429>), an organism that does not produce peptidoglycan at all. Although the authors do not go much into detail when describing the observed gonococcal phenotype, it would be appropriated and fair if the authors would occasionally discuss their observation with a more wholistic view on existing literature.

The reviewer suggests that we should allow a broader discussion of zeta toxin action in various organisms. Clearly, we have already in the initial manuscript pointed to the established mechanism of action of the gonococcal zeta toxin and cited the relevant literature for the gonococcal zeta/epsilon TA system including mechanistic and structural studies (Mutschler et al. (2011). A novel mechanism of programmed cell death in bacteria by toxin-antitoxin systems corrupts peptidoglycan synthesis. *PLoS Biol* 9, e1001033; Rocker et al. (2018). The ng_zeta1 toxin of the gonococcal epsilon/zeta toxin/antitoxin system drains precursors for cell wall synthesis. *Nature Commun* 9, 1686). We also had eluded to the prominent role of zeta/epsilon orthologues in Gram positive bacteria and explicitly cited work on the related staphylococcal zeta/epsilon (PezA/T) TA system (Brzowska, I., and Zielenkiewicz, U. (2013). Regulation of toxin-antitoxin systems by proteolysis. *Plasmid* 70, 33-41. Brzowska, I., and Zielenkiewicz, U. (2014). The ClpXP protease is responsible for the degradation of the Epsilon antidote to the Zeta toxin of the streptococcal pSM19035 plasmid. *J Biol Chem* 289, 7514-7523). Also in the initial submission, we had cited several authoritative reviews on TA systems, which cover the full breadth of potential toxin mechanisms and downstream consequences (Gerdes, K., Christensen, S.K., and Lobner-Olesen, A. (2005). Prokaryotic toxin-antitoxin stress response

loci. *Nature Rev Microbiol* 3, 371-382; Harms, A., Brodersen, D.E., Mitarai, N., and Gerdes, K. (2018). Toxins, Targets, and Triggers: An Overview of Toxin-Antitoxin Biology. *Mol Cell* 70, 768-784; Hayes, F. (2003). Toxins-antitoxins: plasmid maintenance, programmed cell death, and cell cycle arrest. *Science* 301, 1496-1499) as well as additional references, which refer to examples of Type II TA systems (Kolodkin-Gal, I., and Engelberg-Kulka, H. (2006). Induction of *Escherichia coli* chromosomal mazEF by stressful conditions causes an irreversible loss of viability. *J Bacteriol* 188, 3420-3423; Leplae, R., Geeraerts, D., Hallez, R., Guglielmini, J., Dreze, P., and Van Melderen, L. (2011). Diversity of bacterial type II toxin-antitoxin systems: a comprehensive search and functional analysis of novel families. *Nucl Acids Res* 39, 5513-5525; Mutschler, H., and Meinhart, A. (2011). epsilon/zeta systems: their role in resistance, virulence, and their potential for antibiotic development. *J Mol Med (Berl)* 89, 1183-1194). We do not see that further expansion to related proteins, in particular zeta-related proteins of eukaryotes without clear functional assignment (<https://doi.org/10.1002/1873-3468.13429>), would strengthen the discussion section. Therefore and also due to space limitations, we have refrained from further expanding and elaborating on the interesting theme of zeta toxin action in various microorganisms, which might be best addressed in a timely review article on this topic.

Comment 4) Are there any unwanted off-target effects of NQNO in eukaryotes? The authors decided for topical application of NQNO in mice which apparently did not exert any detrimental effect to the mucosa and the entire genital track. Furthermore, NQNO was tested for potential off-target effects on vaginal epithelial cells as well as cervix carcinoma cells in vitro. Can the authors report on pharmacological parameters that would exclude for instance oral administration? For instance, what is the lethal dose for mice or how strong is the hepatic clearance in model systems? Although not essential for the general conclusions drawn in the manuscript, it would give a better view on the potential of NQNOs in anti-gonococcal therapy.

As you might expect, these questions are also of enormous interest to us and, as mentioned by the reviewer, we have tried to clarify the initial (and for us accessible) parameters with our *in vitro* cytotoxicity assays, the *in vivo* histopathology as well as the investigation into compound stability in the presence and absence of serum, which were already included in the initial manuscript. We are not equipped for detailed pharmacokinetic and pharmacodynamic analysis and have only begun to assess critical parameters of NQNO (LogP, mutagenesis (Ames test), interference with cytochrome P450 enzymes, hERG inhibition) with the help of contract research organisations. So far, none of these initial tests indicates any major limitation in the further development of NQNO-derived compounds for *in vivo* use. As these data do not add to the novel insight reported in this manuscript, we will report those data together with the further chemical diversification of NQNO-related compounds in a future manuscript.

Reviewer #2 (Remarks to the Author):

I'm glad for the opportunity to review "Quinolone-N-oxides kill multidrug resistant *N. gonorrhoeae* by unleashing the endogenous zeta toxin" by Mix et al. The authors have shown that 2-nonyl-4-quinolone N-oxide might have strong antimicrobial activity and specially against *Neisseria gonorrhoeae*. The manuscript is very comprehensive with some intriguing findings.

Minor comments:

Please replace the WHO, 2016 reference with "Global progress report on HIV, viral hepatitis and sexually transmitted infections, 2021. May 20, 2021".

The reference has been replaced by the more recent report and the estimated number of annual gonorrhoea cases has been updated to 82 million

The same for da Costa-Lourenco et al., 2017 – replace with Unemo, M., Seifert, H. S., Hook, E. W., 3rd, Hawkes, S., Ndowa, F., & Dillon, J. R. (2019). Gonorrhoea. *Nature reviews. Disease primers*, 5(1), 79.

We have also added this more current reference suggested by the reviewer

Row 81-85: Please revise though the conjugative plasmid carrying TetM is not present in a large fraction in all *Neisseria gonorrhoeae* but can differ a lot between regions. The authors should revise the sentence to that it is present in large fractions in some countries. This is also stated by Cehovin et al.

The sentence has been revised according to the reviewer's suggestions to now read on page 4, line 81:

"An example is the acquisition of tetracycline resistance in the form of the tetM gene and its worldwide spread via the gonococcal conjugative plasmid (pTetM), which now has a high prevalence in low income countries (Cehovin & Lewis, 2017; Cehovin et al., 2020)."

Authors should include a reference for Row 102-103 stating "Another group of AQ derivatives comprises the 2-alkyl-4-quinolone N-oxides (AQNOs), which do not function as signalling molecules in *Pseudomonas*."

The references by (Szamosvari and Böttcher, 2017) and (Szamosvari et al., 2020) have been included.

Row 104: Print out the abbreviation for HQNO and NQNO when mentioned in text for the first time.

The abbreviation “NQNO” is detailed in the summary (line 58) and in the introduction section (line 113). The abbreviation “HQNO” is now detailed in the introduction section in line 105.

Row 512-513: Is there a reason the authors used OD550 for *Neisseria* spp?

The reading of culture optical density at 550nm is a long-established standard for following *Neisseria* growth in liquid culture and has been used by us and others in the past (see e.g.: Gibbs, C.P., B.-Y. Reimann, E. Schultz, A. Kaufmann, R. Haas, and T.F. Meyer. 1989. Reassortment of pilin genes in *Neisseria gonorrhoeae* occurs by two distinct mechanisms. *Nature*. 338:651-652; Hauck, C.R., T.F. Meyer, F. Lang, and E. Gulbins. 1998. CD66-mediated phagocytosis of Opa52 *Neisseria gonorrhoeae* requires a Src-like tyrosine kinase- and Rac1-dependent signalling pathway. *EMBO J*. 17:443-454; Malott, R.J., B.O. Keller, R.G. Gaudet, S.E. McCaw, C.C. Lai, W.N. Dobson-Belaire, J.L. Hobbs, F. St Michael, A.D. Cox, T.F. Moraes, and S.D. Gray-Owen. 2013. *Neisseria gonorrhoeae*-derived heptose elicits an innate immune response and drives HIV-1 expression. *Proceedings of the National Academy of Sciences of the United States of America*. 110:10234-10239; Roth A, Mattheis C, Muenzner P, Unemo M, Hauck CR. 2013. Innate recognition by neutrophil granulocytes differs between *Neisseria gonorrhoeae* strains causing local or disseminating infections. *Infect Immun*. 81: 2358-70. doi: 10.1128/IAI.00128-13; Muenzner, P. and C.R. Hauck. 2020. *Neisseria gonorrhoeae* Blocks Epithelial Exfoliation by Nitric-Oxide-Mediated Metabolic Cross Talk to Promote Colonization in Mice. *Cell Host & Microbe*. 27:793-808 e795). Therefore, we have continued to monitor neisserial growth in liquid medium at this wavelength, while growth of other microbes (*Klebsiella*, *Escherichia*, *Lactobacillus*, etc.) was monitored at 600 nm.

Row 533: Please include the source of the NQNO. If this was synthesized, please include a section for this.

The synthesis and high resolution MS-based as well as NMR-based quality control of all the substances used in antibiotic assays (including NQNO) was already detailed in the Supplementary Information S1 (50+ pages including original NMR spectra). Accordingly, the details of NQNO synthesis are only briefly mentioned in the Material & Methods section on page 24:

“Synthesis of AQNOs

The saturated AQs were synthesized via Conrad-Limpach cyclization and the unsaturated trans- Δ 1-NQ by Camps cyclization as described previously (Szamosvari and Böttcher, 2017). The N-oxides were obtained by locking the 4-hydroxyquinolins as ethyl carbonates followed by N-oxidation with mCPBA (m-chloroperoxybenzoic acid) and deprotection. A list of names,

abbreviations, and structures of all synthetic compounds is given in Suppl. Table 4. The details of compound synthesis together with the characterization and confirmation by NMR spectroscopy and high resolution mass spectrometry is described in Suppl. Info S1.

Row 564: Is this OD600?

See also comment above with regard to determination of optical density. We have now clarified this sentence on page 26, line 576 to read:

*“Optical density at 550 nm (*N. gonorrhoeae* and commensal *Neisseriae*) or 600 nm (all other microorganisms) was determined and 4×10^7 bacteria were inoculated in 5 ml of the respective growth medium.”*

Row 630: I suggest that the authors refer to the 500 bp paired-end to have instead 250 bp paired-end reads. Genome determination and comparison section: The authors is strongly advised to upload the raw fastq files to the Bioproject as well.

We thank the reviewer for pointing us to this ambiguous wording. Accordingly, we have corrected this statement in line 660 according to the reviewer’s suggestion. We also uploaded the raw fastq files of the sequencing results to Bioproject number PRJNA728975 at NCBI GenBank.

Row 135: Authors state that they tested other “pathogenic *Neisseriae*”. But I can’t see that the authors have tested *P. aeruginosa* factors against *N. meningitidis*. Why is this?

This is indeed a keen observation by the reviewer. We have refrained from using meningococci in our analysis, as my current co-workers involved in this project (PhD students and M.Sc. students) do not have a basic vaccination with one of the meningococcal vaccines. Since this would be a mandatory pre-requisite according to our institutional biosafety guidelines, we are limited in our current ability to handle viable meningococci. However, we plan to offer vaccination to the next generation of student co-workers to be able to extend our analysis to meningococci in the near future. This would be particularly interesting as the closest relative of gonococci and meningococci, *Neisseria lactamica*, also showed sensitivity towards NQNO suggesting that further testing of these compounds against *N. meningitidis* might be a worthwhile endeavour in the future.

To clearly indicate this focus of our analysis, we have rephrased the respective sentence on page 8, line 155 to now read:

*“The compounds were dissolved in DMSO and applied in increasing concentrations (5 - 50 μ M) to broth cultures of pathogenic *Neisseria gonorrhoeae* or commensal *N. cinerea*.”*

While we could not extend our analysis to meningococci, which occupy a distinct niche in our body, we have tested a wider variety of commensal bacteria of the female vaginal tract (including *Lactobacillus delbrueckii*, *L. gasseri*, *L. paragasseri*, *L. homini*, *L. jensenii* and *Limosilactobacillus vaginalis*) for their susceptibility towards NQNO and also 3Me-NQNO. With these additional data, which are now included as novel panels in Figure 1E and Suppl. Fig. S4B (NQNO) as well as Fig. 8H and Suppl. Fig. S8E (3Me-NQNO), we provide additional evidence for the selective action on gonococci and the natural resistance of commensal vaginal bacteria against these two compounds.

Figure 1A: The authors refer to this electron microscopy scan to conclude that *Neisseria cinerea* is not affected by PAO1. How can the reader discern this? To me, both strains are affected in a similar way including intact cells. Furthermore, I imagine that the authors have chosen a very well representative, even the best, scan photo.

We thank the reviewer for this comment, as it is always difficult to select from the multitude of pictures, which we have acquired. Indeed, the ones we had chosen originally already demonstrate the damaged appearance and the resulting cellular debris of the gonococci treated with *P.aeruginosa* culture supernatants compared to the other samples. However, upon reduction of these micrographs to the size used in the panel these features are harder to discern. Therefore, we have chosen additional micrographs from our analysis of these samples and have used a slightly larger magnification. As can be seen, the gonococci incubated with *P. aeruginosa* culture supernatant are strongly damaged and few intact bacteria remain, while *N. cinerea* is not harmed by the same culture supernatants and both, *N. gonorrhoeae* and *N. cinerea*, show the regular morphology with mainly intact diplococcal structures, when grown under control conditions. The new panel now replaces former Fig. 1A.

Figure S1. Why are different strains used in the different experiments? I.e in figure S1A the authors have used MS11, while in S1B and S1C they have used Ngo 11. I would rather that the authors include all tested *N gonorrhoeae* isolates in the figure.

During our study, we used as many different gonococcal strains as we could to demonstrate the broad applicability of our findings. This is also true for Figure S1, where we show the original results (pictures of the GC agar growth plates) for *N. gonorrhoeae* strain MS11 (which shows intermediate growth inhibition in these cross-streak assays; see Fig. S1B) together with results for gonococcal strain Ngo 11 and strain *N. cinerea*, which show maximal (Ngo11) or minimal (*N. cinerea*) response in these growth assays. These examples of original data were chosen to demonstrate the complete breadth of the observed responses and all the data for all the strains are summarized in Suppl. Figure S1B in a single panel. To present, as suggested by the reviewer, the results for all the strains in the form of depicting the original plates would clearly bust the Figure and would require two additional letter-size pages minimum.

The authors need to include figure texts for supplementary figures.

The Figure legends for the Suppl. Fig. were included in the manuscript and they are found from page 38ff.

Row 163-165: Pefloxacin is not a relevant antibiotic for the treatment of *Neisseria gonorrhoeae*. Please include ciprofloxacin MICs instead.

This sentence is not meant to introduce Pefloxacin as an antibiotic for *Neisseria*, but as a general example of an antibiotic compound, which is in clinical use at concentrations that match the observed minimum inhibitory concentrations (MICs) for NQNO. With this statement we would like to indicate that already our initial hit compound NQNO is effective at concentrations used in other instances of antibiotic therapy. In the case of ciprofloxacin, due to drug resistance, the MICs reported for the *N. gonorrhoeae* strains used in our study vary from 0.004 - <32 mg/l.

In the course of this study, we identify an improved NQNO-derived compound (3-MeNQNO, former Fig.7, now Fig. 8 in the revised manuscript), which shows a dramatic increase in efficacy. We envision that further derivatisation at this position could lead to even further increases in efficacy in the near future (see also our response to reviewer #3, comment #10). Therefore, we do not feel that it is warranted at this stage to include a specific comparison of NQNO efficacy with other currently used antibiotics for the treatment of gonorrhea, as the development of NQNO-derived compounds is only in its infancy and poised to see significant improvements in the short term as demonstrated in our study.

Major comments

The manuscript includes an unreasonable number of different figures. As reader it is very confusing when the authors use A to even H of unrelated figures. E.g how is figure 1A related to 1B? The authors could revise these as e.g figure 1 and 2 instead. This should be applied to all figures.

We agree with the reviewer that our manuscript is rich in primary data, which we display in several Main and Supplementary Figures. However, we think this is a positive and strong aspect of our contribution. In all the Figures we try to assemble panels, which are intimately related. In the example mentioned by the reviewer, Fig. 1A shows the distinct effect of *P. aeruginosa* culture supernatants on commensal (*N. cinerea*) vs. pathogenic (*N. gonorrhoeae*) *Neisseria* by scanning electron microscopy, while Fig. 1B gives the chemical structures of some of the known ingredients of *P. aeruginosa* supernatants, which are in the center of the following experiments. I think the thematical connection is obvious, but of course this can be best appreciated by reading the accompanying text in the results section. As the total amount of Figures is limited, it is not possible to simply subdivide each Main Figure into two or three new Figures. On the other hand, we do not want to reduce the content of the paper, as we

believe that the entirety of all the presented data is of particular value for the reader. Therefore, we have not re-shuffled or reduced Figures.

The authors are very clear that *N. cinerea* is resistant to NQNO. However, the authors use *N. cinerea* data to draw conclusions for all commensal *Neisseria*. This should be revised and clear that NQNO does affect commensals in various degrees, eg see row 198-200.

We have followed this valuable suggestion by the reviewer and adapted the text at several instances, where we had generalized to all commensal *Neisseriae*. We now clearly indicate that NQNO does not affect most commensals, but that some, in particular *N. lactamica*, are as vulnerable as *N. gonorrhoeae*. Furthermore, in our later functional experiments we explicitly refer to *N. cinerea*, which was used as a control throughout the study, but not to the totality of commensal *Neisseriae* :

Page 5, line 116:

“Strikingly, NQNO does not act on most non-pathogenic neisserial species or unrelated lactobacilli, which live as commensals on human mucosal surfaces, and effective doses of NQNO do not harm human epithelial cells in vitro.”

Page 9, line 178:

“While all gonococcal strains were strongly impaired in their growth, most commensal *Neisseriae* showed negligible sensitivity to NQNO (Fig. 1D). An exception was seen for *N. lactamica*, a species generally regarded as commensal, but closely related to *N. gonorrhoeae* (Bennett et al., 2007) (Fig. 1D).”

Page 9, line 200:

“However, ATP levels were unaffected in *N. cinerea* suggesting that respiratory chain function and ATP synthesis in this commensal *Neisseria* species was not reduced by NQNO (Fig. 2A).”

The authors could include limitations of the study in Discussion, eg the authors suggest that NQNO could be used via local vaginal administration. However, authors have not tested many of the bacteria in the vaginal microbiota. The authors could still include this hypothesis but include the limitations of the study. Furthermore, using antibacterial compounds freely, especially prior to exposure could lead to rapid resistance in the gonococcal population. The authors can include and discuss around this possibility and the importance of antibiotic stewardship.

We thank the reviewer for this advice, but it is exactly in this section of the discussion section, where we clearly indicate that the used topical application of NQNO in the vivo model would represent a limitation for the clinical use of this compound as the systemic oral or parenteral treatment is in most cases the preferred route of administration. In the same section, we also

refer to the limitations of the used animal model, which does not recapitulate the human disease. Therefore, we think that we frankly pointed to limitations of the study. However, the reviewer makes an important point in that we only tested a single commensal non-neisserial bacterial species from the female vaginal tract for its sensitivity towards NQNO. To address this weak point, we have now conducted additional growth assay with a wide selection of lactobacilli (*Lactobacillus brevis*, *L. delbruecki*, *L. gasseri*, *L. paragasseri*, *L. homini*, *L. jensenii* und *Limosilactobacillus vaginalis*) and tested them not only with NQNO, but also with the improved lead compound 3Me-NQNO.

As you can see in the novel data panels in Figure 1E and Suppl. Fig. S4B (NQNO) as well as Fig. 8H and Suppl. Fig. S8E (3Me-NQNO), none of these Gram-positive commensal bacteria is sensitive towards concentrations of NQNO or 3Me-NQNO, which strongly affect *N. gonorrhoeae*. In addition to these Figure panels, we have also included a sentence in the discussion section on page 21, line 484 to underscore these novel findings:

“Therefore, this class of compounds might serve as a starting point to derive pathogen-selective compounds, sparing beneficial commensal bacteria such as lactobacilli from antibiotic action.”

Finally, although the manuscript is well written, it draws very large conclusions that would need further studies with a larger, up-to-date collection of gonococcal isolates. The tested isolates are mainly lab strains or isolates no longer circulating in the gonococcal population. The gonococcal population have shifted towards azithromycin resistance (due to change in treatment recommendations) and it is worrying that the WHO P (azithromycin resistant strain that have clonally expanded) is resistant to NQNO. The authors could mention how this can be addressed in future studies.

We agree with the reviewer that it will be useful to expand the tested gonococcal strains even further in the future (a total of 14 independent gonococcal strains has been tested by us). However, this is the initial identification and characterization of this class of compounds and their activity towards gonococci (and towards a large number of related and unrelated bacteria (9 commensal Neisseria species; 7 distinct Lactobacilli; 4 distinct further microbes), where the emphasis clearly is not only on detecting effects on more and more isolates, but rather on understanding the mechanistic principles underlying the selective action of these compounds.

We are totally aware of the current shift in the resistance spectrum of *N. gonorrhoeae* in parts of the world towards azithromycin resistance (see e.g. Lu Z, Tadi DA, Fu J, Azizian K, Kouhsari E. (2022) Global status of Azithromycin and Erythromycin Resistance Rates in Neisseria gonorrhoeae: A Systematic Review and Meta-analysis. Yale J Biol Med. Dec 22;95(4):465-478). Therefore, we had not only included WHO P, but also a recent clinical isolate from the UK (NCTC 13799; isolated 2015 in Yorkshire), which we obtained from a public repository (National Collection of Type Cultures, Salisbury, United Kingdom). This strain exhibits high-level resistance to Azithromycin (>256). See also

<https://www.culturecollections.org.uk/products/bacteria/detail.jsp?refId=NCTC+13799&collecton=nctc>). Most importantly, we found that this recent clinical isolate with high level azithromycin resistance is highly susceptible towards NQNO. Indeed, we reported this finding already in the initial version of this manuscript and these results for Ngo NCTC 13799 are shown in Figure 3C of the revised manuscript.

We would like to stress that we tested further relevant antibiotic resistant clinical isolates, in particular the ceftriaxone-resistant strain N231 isolated 2011 in Slovenia (Unemo et al. (2012). Treatment failure of pharyngeal gonorrhoea with internationally recommended first-line ceftriaxone verified in Slovenia, September 2011. Euro surveillance 17), which also showed high susceptibility towards NQNO. Of course, there is an endless list of further strains, which could be tested, but several interesting recent strains were not available to us. For example, we have not been able to obtain access to the ceftriaxone/azithromycin-resistant recent isolates from the UK and Australia (Jennison AV, Whiley D, Lahra MM, Graham RM, Cole MJ, Hughes G, et al. Genetic relatedness of ceftriaxone-resistant and high-level azithromycin resistant *Neisseria gonorrhoeae* cases, United Kingdom and Australia, February to April 2018. Euro Surveill. 2019 doi: 10.2807/1560-7917.Es.2019.24.8.1900118). Of course we will stay alert to include such kind of latest strains in future analyses, but for the moment we find it fair to conclude that even for current antibiotic resistant isolates, including high Azithromycin-resistant strains, NQNO is effective.

Most importantly, our novel data obtained with the WHO P strain (see comment #1 and #2 of reviewer #1 and also the novel Fig. 5) further demonstrate that it is the lack of the zeta/epsilon TA system, but not the azithromycin resistance, which render this strain insensitive to NQNO. Accordingly, introduction of the zeta1/epsilon1-operon into WHO P render this strain sensitive to NQNO, despite the existing azithromycin resistance (see novel Fig. 5). The conclusion that an existing azithromycin resistance does not affect vulnerability towards NQNO is also in complete agreement with the known action of macrolides such as azithromycin, which interfere with protein synthesis, while NQNO does compromise, according to the data presented in this manuscript, the electron transport chain of gonococci.

Therefore, we stay firm with our initial judgement, that there is no link between azithromycin resistance and NQNO resistance and that even a current high-level azithromycin-resistant strain is sensitive towards this compound, as we state on page 11, line 243:

“As WHO-P was the only strain in this collection with known resistance to azithromycin, we tested an additional set of multidrug-resistant strains either highly resistant to azithromycin (Azm) or to 3rd generation cephalosporins (Ceph) (Fig. 3C). While the azithromycin-resistant isolate NCTC 13799 and the ceftriaxone-resistant strain N231 (Unemo et al., 2012) were fully susceptible to NQNO, the cefixime-resistant strain N316 (Unemo et al., 2011) showed intermediate resistance to NQNO (Fig. 3C). Accordingly, azithromycin-resistance does not explain the lack of NQNO-mediated growth inhibition of strain WHO-P.”

**Reviewer #3 (Remarks to the Author):**

The *P. aeruginosa* alkyl quinolone N-oxides (AQNOs) were originally discovered as potential antibiotics in the 1950s. Their low in vivo antibacterial efficacy with activity primarily against Gram positives combined with their cytotoxicity for eukaryotic cells (electron transport inhibition) generally prevented the development of AQNOs as therapeutic agents. Mix et al make some interesting observations on the selective susceptibility of *Neisseria gonorrhoea* for NQNO compared with the resistance of commensal *Neisseria* strains. The selectivity observed was due to the release of a zeta toxin from its cognate antitoxin, the genes for which are carried on a large pTetM conjugative plasmid commonly found in gonococci but not in commensal *Neisseria*. In a female mouse genital tract colonization and infection model, topical NQNO blocked colonization. In vitro screening of a selection of modified AQNOs yielded a 3Me-substituted analogue with nanomolar efficacy. Given that *N. gonorrhoeae* is a major global public health concern and a World Health Organization (WHO) priority antibiotic-resistant pathogen, this is an important topic given the urgent need for new therapeutics for the treatment and prevention of gonorrhoea.

Specific points

1. Line 151 and 191. Why did the authors select the C9 compound NQNO rather than the more commonly investigated C7 compound HQNO? Was there any difference in potency?

In our prior work (Szamosvari et al. (2020) Chem. Commun. 56: 6328), we established that C9 quinolones are equally prevalent as C7 compounds in the culture supernatants of *P. aeruginosa*. We also had observed that quinolone-derivatives with a longer alkyl chain (C10) can show higher potency (Szamosvari et al. (2019) A thiochromenone antibiotic derived from the *Pseudomonas* quinolone signal selectively targets the Gram-negative pathogen *Moraxella catarrhalis*. Chemical Science 10, 6624-6628). Therefore, we have initially selected C9 compounds for our experiments. Ongoing extensive SAR studies now reveal that chain lengths shorter than C9 display reduced potency against *N. gonorrhoeae*. These data will be part of a future study detailing a more extensive SAR (see also answer to comment #10).

2. Fig. S1. (C), the inhibition data for the *P. aeruginosa* pqsH mutant is surprising given that it still makes AQNOs. Although the pqsL mutant doesn't make AQNOs it will still produce NHQ which the authors show has very good activity against *N. gonorrhoeae*.

Cross-streak experiments and *P. aeruginosa* mutants of the pqs system initially guided our screening process, prompting us to explore AQNOs more deeply. However, these results only gave us initial hints, but we would not draw explicit conclusions solely based on this approach. For example, disrupting parts of the pqs biosynthetic gene cluster has a variety of effects including quorum sensing-related effects and production of outer membrane vesicles (OMVs). In this respect, the pqsH mutant is also known for its impaired production of outer membrane vesicles (OMVs), crucial for quorum sensing, as well as packaging and transporting antimicrobial quinolones such as AQNOs. These defects could contribute to the reduced

inhibition zone observed for the *pqsH* mutant (Mashburn LM, Whiteley M. Membrane vesicles traffic signals and facilitate group activities in a prokaryote. *Nature*. 2005 Sep 15;437(7057):422-5. doi: 10.1038/nature03925). As correctly pointed out by the reviewer, the *pqs* mutant still produces Aqs, but lacks the more potent AQNOs. This could account for the presence of a smaller, yet still noticeable zone of inhibition in this strain.

3. Lines 217-219. Why did the authors only treat the hVEC and Hela cells with a maximum of 10 microM which is very close to the gonococcal MIC of 5 microM and well below the therapeutic dose that would require administration. The claim (line 117) that effective doses do not harm epithelial cells in vitro should be qualified given the low concentrations tested.

We agree with the reviewer that this is a limited dose range, which we tested for its effects on human epithelial cells. Accordingly, we have now repeated these kind of in vitro cytotoxicity tests i) using a wider range of NQNO concentrations including significantly higher concentrations; and ii) extending the investigated cell lines to also include a further human cervical epithelial cell line model for *N. gonorrhoeae* infection, the ME-180 cells (see e.g. Muenzner P, et al. (2005) CEACAM engagement by human pathogens enhances cell adhesion and counteracts bacteria-induced detachment of epithelial cells. *J Cell Biol*. 170: 825-36. doi: 10.1083/jcb.200412151; Muenzner P et al. (2010) Human-restricted bacterial pathogens block shedding of epithelial cells by stimulating integrin activation. *Science*.;329: 1197-201. doi: 10.1126/science.1190892); and iii) extended the cytotoxicity testing to the latest developed compound 3Me-NQNO. The novel data are now presented in graphs showing the growth results for all three distinct human epithelial cell lines for each compound. The graphs are now included in Fig. 2E-F and Suppl. Fig. S5A-B (NQNO) and Suppl. Fig. S8A-D (3Me-NQNO).

4. Lines 274 to 276. There are two similar zeta/epsilon toxin-antitoxin systems on the plasmid, why did the authors only focus on the zeta 1/epsilon 1 system since both would be lost when the plasmid was lost during NQNO conditioning and conversely restored when the plasmid was re-introduced. Deletion of each system individually would be informative. It is also possible that other conjugative plasmid genes are involved. Why not introduce the zeta 1/epsilon 1 system alone on a plasmid into a resistant gonococcus strain?

This is a very important point made by the reviewer and argues along the same line as comments #1 and #2 of Reviewer #1 (see above). Following the suggestion of both reviewers, we have introduced a construct, which exclusively encodes the *zeta1/epsilon1* operon, into the NQNO-resistant and multidrug-resistant strain WHO P. As now shown in novel Fig. 5, the *zeta1/epsilon1*-transformed WHO P is becoming sensitive to NQNO and the level of NQNO sensitivity of WHO P with the newly introduced *zeta1/epsilon1* operon is comparable to the NQNO-sensitivity previously established by transformation of WHO P with the complete pTetM-plasmid. These novel results clearly demonstrate that expression of the Zeta1/Epsilon1 TA pair is sufficient to confer NQNO sensitivity to a NQNO-resistant strain of *N. gonorrhoeae*. These results do not rule out that additional gonococcal determinants contribute to the

bacteriostatic effect of NQNO, but clearly help to demarcate an essential role of Zeta1/Epsilon1 in this process.

To incorporate these important novel data, we have included an additional Main Figure (Fig. 5) and an additional paragraph in the results section on page 14, which now reads:

“The Zeta1/Epsilon1 TA system mediates the increased NQNO sensitivity of gonococci

To unambiguously pinpoint the contribution of the zeta-epsilon TA system, we focused on the multidrug-resistant WHO P strain, which lacks the complete pTetM plasmid including both zeta/epsilon operons and which exhibits NQNO resistance (Fig. 3B and Fig. 4A). To this end, we introduced either the pTetM plasmid or the isolated zeta1/epsilon1 operon into WHO P (Suppl. Fig. S6). Both, the plasmid-encoded (WHO P pTetM A and WHO P pTetM B) and the chromosomally-encoded zeta1/epsilon1 genes (WHO P $\zeta1/\epsilon1$) were expressed at equal levels (Fig. 5A). Most importantly, Zeta1/Epsilon1-expressing derivatives of WHO P became NQNO-sensitive, while the parent strain was fully resistant up to 50 μ M (Fig. 5B). Similar to the introduction of pTetM into strain MS11-R2 (Fig. 4F), the presence of the Zeta1/Epsilon1 TA pair resulted in dramatically increased membrane permeability in WHO P upon NQNO exposure (Fig. 5C and D). Our genetic data tightly link the NQNO-susceptibility of gonococci to the presence of the zeta1/epsilon1 operon. Furthermore, the observed membrane permeability phenotype is in line with the known activity of the Zeta toxin. Therefore, we hypothesized that NQNO might promote Zeta1 activity to compromise cell wall synthesis and to stall gonococcal growth.”

5. Line 294. The Syto9 and PI fluorescence in Figs 4B and 4F should be quantified.

We thank the reviewer for this suggestion and have now quantified multiple images from several repetitions of this experiment. The data are now presented in Fig. 4 (*N. gonorrhoeae* strain MS11 and *N. cinerea*) as well as in novel Fig. 5 (*N. gonorrhoeae* strain WHO P and epsilon1/zeta1-derivatives). The quantified fluorescence intensity of individual bacteria depicted in the dot graph now better conveys the overall result, namely that the presence of the zeta1/epsilon1 TA system correlates with increased membrane permeability upon NQNO exposure of the gonococci.

6. Lines 311-313 also 338-339. The authors cannot rule out from the data presented that NQNO may act on both the electron chain and directly on one or both of the TA systems.

This is indeed a very valuable point made by the reviewer. To address this concern, we have now performed additional experiments in *E. coli*, which is not susceptible to NQNO. Already in the initial submission we had shown that the epsilon1 antitoxin is readily expressed in *E. coli*, but it is vulnerable to reactive oxygen (H_2O_2). If NQNO would exert a direct effect on epsilon1 antitoxin stability, we should observe degradation of epsilon1 upon treatment of *E. coli* with NQNO. However, NQNO up to concentrations of 50 μ M did not lead to changes in epsilon1

levels. These novel data indicate that NQNO does not directly affect epsilon1 stability and these data are now included in Fig. 6D (formerly Fig.5 in the initial submission).

7. Line 361-2 and Fig. 6C. The authors state that NQNO eradicated the gonococci to below the level that could be detected by microscopy. However, there is a very clear green signal towards the top of the panel

In the immunohistological stainings depicted in former Fig. 6C (now Fig. 7C in the revised manuscript) mentioned by the reviewer, there are clear positive, punctuate signals with a size $<5 \mu\text{m}$ resulting from antibody-stained gonococci. These signals are only detected in the DMSO-treated control sample. They can be best appreciated in the enlargements provided on top of the panel. In both, the DMSO control and the NQNO-treated sample there is a broad background staining of low intensity, which is seen throughout the epithelial layer and which is presumably due to the keratinization of the stratified epithelium of the vaginal tract. However, this low-intensity background staining is clearly distinct from the strong, punctuate staining of individual bacteria and small bacterial clusters, which are present in the DMSO-treated samples. The staining pattern for the DMSO-treated control samples is highly similar to mucosa-associated gonococci on the murine vaginal epithelium, which we have reported before in this *in vivo* model system (see e.g. Muenzner P et al. (2010) Human-restricted bacterial pathogens block shedding of epithelial cells by stimulating integrin activation. *Science*;329: 1197-201. doi: 10.1126/science.1190892).

8. Line 370. Fig 6E should be 6F.

We apologize for this mistake. We have corrected the assignment of panels in this Figure, which is now Fig. 7 of the revised manuscript.

9. Lines 373-374. The authors only tested one strain *in vivo* so cannot claim NQNO also suppressed viability of extended drug resistant isolates *in vivo*.

We disagree with the reviewer at this point, as we have already evaluated the potential of NQNO in our *in vivo* infection model in CEA-transgenic female mice using two distinct strains of *Neisseria gonorrhoeae*. One strain used was our laboratory strain MS11, which we have used throughout this manuscript and also extensively employed in prior publications. This strain has been isolated from a local symptomatic infection, but has been kept and genetically modified in the lab to now encode Erythromycin, Chloramphenicol and Tetracyclin resistance genes. Besides strain MS11, we have also used a recent multidrug-resistant isolate from Austria (Ngo 316; Unemo, M., et al. 2011. First *Neisseria gonorrhoeae* strain with resistance to cefixime causing gonorrhoea treatment failure in Austria. *Euro Surveill*, 16), which is resistant against 3rd generation cephalosporines and also against ciprofloxacin.

All these data have been presented in former Fig. 6, which is now Fig. 7 in the revised manuscript. Based on these data and based on the comment of the reviewer, we have rephrased the respective sentence on page 17, line 396 to conclude that:

“Together, these results demonstrate that NQNO can suppress viability and in vivo growth of gonococci, including in vivo growth of a clinical isolate with extended-drug resistance.”

10. Line 376. There is little in the way of a systematic SAR. Given the increased potency of the 3Me AQNO derivative, the synthesis of further 3 position analogues would be informative as would a shorter 2 position C9 alkyl chain to make the compound more drug-like.

— Already in this initial characterization, we present data for ~30 AQNO and AQ derivatives, which we have generated to systematically explore the consequences of substitutions at different positions around the quinolone ring. With this approach we identify and report that substitution at the 3 position increases potency of NQNO. We agree with the reviewer that would be most interesting to explore further substitutions at this position. Accordingly, we are expanding our collection of synthetic AQNOs right now with an emphasis on the 3 position and the alkyl chain length to come up with a comprehensive SAR study, having so far generated over 100 AQs and AQNOs. We feel that the planned thorough exploration of the chemical space warrants in-depth discussions on structural details, their chemophysical characteristics, and the development of various synthetic routes giving access to these compounds - topics beyond the scope of the current manuscript. Our primary focus with this contribution is elucidating the mechanism and demonstrating the proof-of-principle that the toxin-antitoxin system is involved in the particular susceptibility of *N. gonorrhoeae* towards NQNO and derived compounds. Therefore, we will present the ongoing in-depth SAR analysis in a follow-up study.

— 11. Line 396-397. The authors only tested 14 repeat subcultures for increased resistance which here could relate either to loss of the conjugative plasmid or a mutation in the target of NQNO – probably the quinone reduction site cytochrome bc1 sub-unit. Many additional subcultures are likely to be required for resistance to emerge.

The reviewer is correct in that a further prolongation of the subculturing of gonococci at low concentrations of our improved compound 3Me-NQNO might result in the emergence of resistant clones. However, we would like to state that we extended this experiment already significantly beyond the duration of our initial procedure, where we observed the emergence of resistant clones after 8 days of consecutive culturing in the presence of NQNO. Accordingly, we can conclude that 3Me-NQNO is less likely to yield resistant strains and therefore, as we have stated, *“NQNO derivatives with a favorable efficacy and safety profile can be developed”*.

Interestingly, also in the case of the NQNO-resistant isolates, which we obtained after 8 days of growth at low NQNO-concentrations, we did not find any mutations in the cytochrome b1 subunit. This finding already indicates that there might be strong structural constraints on this protein. In the future, we will embark on a chemical-biological screen with a cross-linker

equipped derivative of NQNO to identify the primary target of NQNO, most likely aided by extended cultivation at low concentrations to generate additional, independent resistant clones.

—

—

31.7.2024

Response to Reviewer Comments on revised manuscript NMICROBIOL-23010150A:

Reviewers Comments:

Reviewer #1 (Remarks to the Author):

The authors have added substantial and significant additional data to this manuscript addressing most of the reviewer's concern and provide convincing arguments in their rebuttal letter if not included.

I would like to congratulate the authors for this outstanding achievement.

We thank the reviewer for the constructive and very helpful criticism and the encouraging comments along the review process.

**Reviewer #2 (Remarks to the Author):**

I would like to express my gratitude to the authors for their thoughtful and comprehensive responses to my comments. I appreciate the effort and diligence applied to address the points raised. However, I have a few additional comments.

1) Row135 comment: Please include a sentence or two in regards of limitations in the discussion section regarding *N. meningitidis*.

We have added the following statement to the discussion section on page 20, line 476:

"The closest relative of gonococci, Neisseria meningitidis, was not evaluated in this study. Given the high degree of homology between meningococci and gonococci, the initial perturbation of the primary metabolism by NQNO might occur in both species."

2) Figure S1B – Weren't all the isolates in Table S1 subjected to this methodology? If that's the case, it would be prudent to extend the supplementary material by an additional two pages, despite it already surpassing 50 pages. Providing the inhibition zones for all strains would significantly enhance the comprehensiveness of our data. A dedicated table detailing these inhibition zones would be invaluable. Lastly, for clarification, the label in Figure B should read '*N. gonorrhoeae* MS11,' not '*N. gonorrhoeae* 11,' correct?

The data shown in Figure S1B were "only" compiled with 12 *N. gonorrhoeae* strains and the 8 different commensal *Neisseria* species depicted in this panel. As this cross-streak assay was the initial screen, we did not repeat this crude analysis (which can be influenced by diverse secreted products of *Pseudomonas aeruginosa*) later in our investigation, when we already had identified a defined secreted product of *P. aeruginosa* (NQNO) as potent anti-gonococcal agent. Accordingly, gonococcal strains acquired later in the course of our study (e.g. the WHO collection or the high-azithromycin resistant strains) were not included in this initial analysis, but rather they were directly tested with the defined, synthetic compounds.

Therefore, Suppl. Table S1, which lists all the *Neisseria gonorrhoeae* strains employed in our study, holds a significantly larger collection of different gonococcal strains than were used in the initial screen.

The label in Fig. S1B is indeed "Ngo 11", as this is the designation for this DGI strain originally obtained from Dr. Unemo, Orebro University, Sweden. This "Ngo 11" strain is distinct from *N. gonorrhoeae* strain MS11, which is used in most of our experiments and which is also present, in addition to Ngo 11, in Fig. S1B. Both strains are listed separately in Suppl. Table S1, which details their origins and holds their unique strain identifiers used in our internal strain collection.

3) Row 163-165: This reviewer notes that the authors do not advocate for pefloxacin as a recommended treatment. The authors state:

“Strikingly, 2-nonyl-4-quinolone (NQ), trans- Δ 1 162 -2-nonyl-4-quinolone (trans- Δ 1 163 NQ) and 2-nonyl-4-quinolone N-oxide (NQNO) completely inhibited growth of *N. gonorrhoeae* (Fig. 1C). While trans- Δ 1-NQNO was considerably less active, NQNO inhibited growth of *N. gonorrhoeae* even at the lowest concentration of 5 μ M (~ 1.5 μ g/ml) (Fig. 1C). This effective concentration is in the range of minimum inhibitory concentrations (MICs) observed for clinically relevant antibiotics such as pefloxacin or spectinomycin (Bengtsson-Palme and Larsson, 2016).”

When the authors use the term “clinically relevant” alongside a genuinely clinically relevant antibiotic for *N. gonorrhoeae*, such as spectinomycin, it may lead readers to infer that pefloxacin is/has been clinically relevant for gonorrhoea treatment. Please clarify this statement.

We thank the reviewer for this advice. With this comparison we wanted to stress that already the initial lead compound (NQNO) has an effective concentration in a range, which is comparable to clinically relevant antibiotics used to treat bacterial infections (not necessarily gonococcal infections). However, we completely understand the reasoning of the reviewer and agree that our statement could be misleading. Therefore, we have omitted this sentence. As we later report about 3Me-NQNO, which has a nanomolar MIC, we do not think that this crude comparison of NQNO and other antibiotics is necessary anymore.

4) Final comment: The isolates that have undergone clonal expansion in recent years are the WHO P-like strains. Meanwhile, high-level resistance to azithromycin appears more sporadically. This doesn't necessarily suggest that the resistance mechanisms to azithromycin are causing the decreased susceptibility to NQNO. It may instead indicate that the WHO P-like genomic backbone, with its myriad traits, is also proliferating. This backbone might encompass a consistent TA system. That said, this reviewer is not advocating for additional experiments. Instead, it would be beneficial if the authors could address this point in the discussion section.

We thank the reviewer for this remark and we have added the following statement to the discussion section on page 22, line 517:

“These desirable properties of NQNO are slightly tarnished by the occurrence of naturally NQNO-resistant strains such as WHO P, which lack the pTetM plasmid. While the occurrence of WHO P-like azithromycin-resistant strains has increased in Europe in recent years (Day et al., 2022), pTetM-positive and therefore NQNO-sensitive strains are highly prevalent in other parts of the world (Cehovin et al., 2018; Agbodzi et al., 2023; Tayimetha et al., 2023).”

The reviewer will be assured that we closely follow resistance development in gonococcal isolates from around the globe and will include additional relevant and available antimicrobial resistant isolates in our future studies.

Reviewer #3 (Remarks to the Author):

The authors have addressed all my key points and added new data to support their conclusions with respect to the second toxin/antitoxin system, the stability of the antitoxin, quantification of membrane damage and extended the concentrations and cell lines used to evaluate toxicity of the quinolone N-oxides. Although the precise molecular basis by which the quinolone N-oxides release the anti-toxin from the toxin remains to be established, the authors have made a very novel and interesting finding of broad interest with significant translational potential in the AMR context.

Minor points:

1) Line 104. Evidence that the AQNOs are not signal molecules in *Pseudomonas* was directly addressed in Rampioni et al (2016) PLOS Pathogens doi: 10.1371/journal.ppat.1006029 rather in the two Szamosvaro et al papers cited.

The reviewer is completely right and we have now added a reference to the paper by Rampioni et al. at the corresponding sentence in the introduction.

2) Lines 348 and 351 and Fig. 6F. 'disappearance' in these sentences is incorrect. The epsilon antitoxin is reduced not absent from these blots.

We thank the reviewer for this correction and have changed the wording from "disappearance" to "degradation":

"Importantly, NQNO-triggered degradation of the Epsilon antitoxin was observed in several gonococcal strains including the multidrug resistant WHO G and WHO N strains as well as the MS11-R2 pTetM A conjugant (Fig. 6F)."

3) Fig. 6 F and G. There is a band on the zeta 1 blots marked with a star. This is presumably a protein that cross-reacts with the antibody used? This should be explained in the legend.

We have observed this cross-reacting band already in the analyses shown in Fig. 5A and Fig. 6A, where we have explained this band (and the asterisk label) in the Figure legend as follows:

“The asterisk indicates a non-specific band reacting with the polyclonal anti-zeta antiserum”.

To clarify the meaning of the asterisk also in panels Fig. 6F and 6G, we have copied the above sentence to the respective further sections of the Fig.6 legend.

Seite 7/27

31.7.2024

Response to Reviewer Comments on revised manuscript NMICROBIOL-23010150A:

Reviewers Comments:

Reviewer #4 (Remarks to the Author):

The manuscript by Mix et al investigates the activity of 2-nonyl-4-quinolone N-oxide (NQNO) against *N. gonorrhoeae*. The study reveals that NQNO demonstrates anti-gonococcal activity without impairing growth of commensal *Neisseria* or viability of human cells. In vivo, application of NQNO resulted in reducing the bacterial burden in the vaginal tract.

Mechanistically, NQNO was found to disrupt the gonococcal electron transport chain with the release of zeta toxin from its epsilon antitoxin. Chemical modification of NQNO yielded a more potent derivative. While most of the experimental work is well designed, and the manuscript is well written, I have outlined some comments and critiques below that should be considered to enhance the manuscript's impact

1) Lines 96-97: “Most antibiotics in current use are derived from bacterial secondary metabolites, which have evolved in the frame of microbial competition.” This sentence requires supporting references to substantiate the claim.

We have now added classic studies (Waksman and Woodruff, 1940) as well as recent reviews (Martinez, 2008), which provide the reader with the current understanding of the role of antibiotics in the microbial world.

Waksman SA, Woodruff HB. The soil as a source of microorganisms antagonistic to diseaseproducing bacteria. *J Bacteriol* 1940; 40: 581–600.

Martínez JL. Antibiotics and antibiotic resistance genes in natural environments. *Science*. 2008 Jul 18;321(5887):365-7. doi: 10.1126/science.1159483. PMID: 18635792.

2) Line 127: please correct the typo to “of this global of concern pathogen”.

We thank the reviewer for this advice and have rephrased the sentence to read:

— *“While TA systems have been proposed as targets for novel anti-infectives (Lioy et al., 2010), our study identifies for the first time a highly selective natural antibiotic and its synthetic derivatives, which unleash an endogenous toxin of Neisseria gonorrhoeae to help fight this high priority pathogen.”*

3) Fig. S1C: difference is small

— Indeed, also cross-streaks using mutant *P.aeruginosa* strains still resulted in inhibition zones. This is not surprising, as these *P.aeruginosa* mutants are compromised in various aspects of quinolone synthesis, yet are still able to secrete other secondary metabolites, which appear also capable of inhibiting gonococcal growth. However, the difference between the quinolonereleasing wildtype and the mutant strains was significant, prompting us to investigate the potential of various alkyl-quinolones in more detail. Clearly, the following analyses demonstrate in a multitude of repetitions that this small difference observed in the initial crude screen (cross-streak assay) was also meaningful.

4) Lines 172-174: “a highly consistent growth inhibition of gonococci by NQNO concentrations > 2.5 μM was observed (Suppl. Fig. S2A)”. According to Suppl. Fig. S2A, HQNO was not tested at the concentration of 2.5 μM (the lowest concentration tested was 5 μM . So, this sentence should be rephrased to “a highly consistent growth inhibition of gonococci by NQNO concentrations \geq 5 μM “.

We did test NQNO at concentrations of 2.5 μM using the MS11 strain (see Fig. 3D, where the gonococci are grown over the course of eight days at 2.5 μM of NQNO) and we did not observe an inhibition at this concentration. However, we did not test all gonococcal strains at 2.5 μM of NQNO and therefore we did not include these data in Suppl. Figs. S2 and S3. Accordingly, we have followed the suggestion of the reviewer and have rephrased the sentence on page 8, line 173 to now read:

“... and a highly consistent growth inhibition of gonococci by NQNO concentrations \geq 5 μM was observed (Suppl. Fig. S2A).”

5) Fig. 3D: there are no standard deviation values shown in the figure.

Indeed, this experiment was conducted twice with similar results (NQNO resistance occurred after >7 days). We have now explicitly added the statement “*Shown is a representative experiment performed twice with similar results*” to the Figure legend. As this experiment mainly served to isolate isogenic, NQNO-resistant derivatives, which were to be used in whole

—

—

genome sequencing to identify the genetic determinants of NQNO-resistance, we did not repeat this approach further. Once we had detected the genetic differences between NQNOresistant and sensitive isogenic strains (Fig. 3F), we focussed on identifying the underlying mechanistic principle (Fig. 4 – Fig. 6).

6) Lines 280-281: “two zeta-epsilon (ζ_1/ϵ_1 and ζ_2/ϵ_2) toxin/anti-toxin (TA) systems are encoded on this extrachromosomal DNA”. Why was ζ_1/ϵ_1 system the only system investigated and ζ_2/ϵ_2 was not investigated?

A similar question was raised by other reviewers and we have addressed this question by reexpression of the isolated epsilon1/zeta1 operon. Briefly, while the enzymatic function of the epsilon1/zeta1 TA system has been elucidated (Rocker A et al. (2018) Nat Commun. 9: 1686) and the known zeta1 toxin activity aligns well with the observed phenotype of the NQNOtreated gonococci (increased membrane permeability as demonstrated in Fig. 4B-D, Fig. 4G-H as well as Fig. 5C-D), the function of the epsilon2/zeta2 TA system is unknown. Furthermore, the zeta1 toxins are highly conserved amongst strains carrying the conjugative plasmid pConj (e.g. 100% sequence identity between the zeta1 toxin from strain 5289 (Pachulec & van der Does (2010) PLoS One 5:e9962) and the zeta1 toxin of strain WHO O), while the zeta2 toxins differ broadly between strains (e.g. 71% sequence identity between the zeta2 toxin from strain 5289 (Pachulec & van der Does. 2010) and the zeta2 toxin of strain WHO O). These facts suggested that if we are looking for a common, conserved determinant encoded on pConj and connected to the observed phenotype, we should test the role of the epsilon1/zeta1 TA system.

Therefore, we have focussed on the zeta1/epsilon1 system on the genetic (Fig. 5) and on the protein level (Fig. 6). Most importantly, we have re-introduced either the complete pConj or the isolated epsilon1/zeta1 operon into the WHO P strain, which is NQNO resistant and which naturally lacks the pConj plasmid (Fig. 5). Upon introduction of this genetic determinant (either in the form of the complete pConj or in the form of the isolated epsilon1/zeta1 operon), the resulting WHO P derivatives became NQNO sensitive and exhibited increased membrane permeability upon NQNO treatment, recapitulating the phenotype of NQNO-treated, pConjpossessing strain MS11. It is important to stress that the re-introduction of the isolated epsilon1/zeta1 operon resulted in epsilon1/zeta1 expression levels similar to the levels observed for the introduced complete pConj plasmid. In addition, the severity of the NQNOinduced growth suppression was also comparable amongst the reconstituted strains, suggesting that the second ϵ/ζ TA system does not contribute significantly to the NQNOinduced phenotype.

We have added this information now to the results section on page 12, line 283ff:

*“Besides components needed for plasmid maintenance and conjugation, two zeta-epsilon ($\zeta1/\epsilon1$ and $\zeta2/\epsilon2$) toxin/anti-toxin (TA) systems are encoded on this extrachromosomal DNA (Pachulec and van der Does, 2010). Ngo $\zeta1/\epsilon1$ is highly conserved amongst different *N. gonorrhoeae* isolates, while the $\zeta2/\epsilon2$ genes exhibit sequence variation (Pachulec and van der Does, 2010).”*

Together, our results not only demonstrate that the lack of the epsilon1/zeta1 TA system correlates with NQNO resistance, but also provide evidence that the possession and expression of the epsilon1/zeta1 operon alone is sufficient to render gonococci susceptible to NQNO.

7) Lines 309-311: “The introduction of the zeta/epsilon-encoding plasmid rendered the resulting pTetM-positive clones sensitive to NQNO (Fig. 4F)”. As shown in Fig. 4F, it seems that the growth of MS11-R2 pTetM A and MS11-R2 pTetM B is normally reduced (OD550 of the negative control (DMSO) did not exceed 0.4 as compared to about 1.4 for MS11-R2). In presence of HQNO, the OD550 of MS11-R2 pTetM A and MS11-R2 pTetM B is increasing by time and it reaches values comparable to that of MS11-R2. A similar trend is shown for MS11R2 (growth increases by time with HQNO treatment). Consequently, it is not reasonable to conclude that “The introduction of the zeta/epsilon-encoding plasmid (pTetM A and pTetM B) rendered the resulting pTetM-positive clones sensitive to NQNO”. The incubation time should be extended beyond 8 hours to observe potential differences between the conjugants and the parent strains. Furthermore, given the low OD550 values observed in the case of the conjugant strains, relying solely on OD values for drawing conclusions may not be sufficient. It is essential to include CFU (colony-forming unit) counts, as even minor discrepancies in OD values may not necessarily reflect significant differences in CFU counts.

The reviewer is perfectly right in observing the reduced growth of strain MS11-R2, when this NQNO-resistant MS11-derivative, which lost the pTetM (pConj) plasmid, is complemented with pTetM. However, the reviewer should also take into account the observed acceleration in growth, when the parent MS11 strain loses this plasmid (compare MS11 parent strain with MS11-R2 in Fig.4F or in Fig. 3A vs. Fig.3E). Accordingly, the pTetM-possessing MS11 parent strain shows a slower growth than the derived strains MS11-R1 and MS11-R2, which have lost the large, 42000 bp pTetM plasmid. The alteration in maximum growth rate might be due to the metabolic costs involved in maintaining multiple copies of such a large plasmid. Taking this into account, the re-introduction of pTetM into MS11-R2 to yield strains MS11-R2 pTetM A and MS11-R2 pTetM B only reverts the growth phenotype of these derived strains back to the level observed for the parent MS11 strain.

We have observed a similar deceleration of gonococcal growth upon introduction of pTetM into the pConj-less WHO P strain (see also the next comment by the reviewer). We speculate that

the propagation of this large plasmid drains energy resources, which are then not available for maximum growth.

The reviewer also points to the far end of the growth curve, when gonococci transit from logarithmic growth into stationary phase. At this point, gonococci will start to undergo autolysis, a well described phenomenon in the growth behaviour of *Neisseria gonorrhoeae* (Dillard JP & Seifert HS (1997) Mol Microbiol. 25: 893-901. doi: 10.1111/j.13652958.1997.mmi522.x; Bos MP et al (2005) Infect Immun. 73: 2222-31. doi: 10.1128/IAI.73.4.2222-2231.2005). Strain *N. gonorrhoeae* MS11, without any treatment, will usually enter into stationary phase after 6-8h of growth in liquid medium and then start to undergo autolysis (see e.g. Fig. 3A). Autolysis is also evident during growth on solid media, where gonococci have to be passaged daily to avoid death by autolysis.

This growth behaviour is not specific for a particular strain of *N. gonorrhoeae*, but is also observed for other gonococcal isolates (see e.g. Suppl. Fig. S3). Accordingly, extending the growth phase in liquid culture beyond 8h would not be helpful. Especially upon prolonged culture, CFU counts would even be less meaningful, as they are influenced by both, the experimental treatment as well as autolysis.

Most importantly, changes in OD550 readings very well reflect differences in CFU. Therefore, the clear differences in OD550 readings observed when susceptible gonococcal strains are grown in the presence of NQNO versus growth in solvent alone (DMSO), reflect large differences in cfu. To underscore this point, we have repeated growth assays with the MS11 parent strain. During this repetition, we have not only continuously monitored growth by OD550 readings, but also determined cfu at 2h, 4h and 6h of growth. As you can appreciate from the growth data, the CFU values closely follow the OD550 readings and the difference between treated (50 μ M NQNO) and control (DMSO) cultures is even slightly larger during logarithmic growth, if CFU-values are compared (Reviewer Fig. 1).

Reviewer Fig. 1: OD550 readings and CFU determination of *N. gonorrhoeae* MS11 provide an equivalent measure of bacterial growth. Growth of *N. gonorrhoeae* MS11 in the presence of 50 μ M NQNO or 1% DMSO was monitored by OD550 readings and by determination of CFU counts upon serial dilution from identical cultures. Shown is a representative experiment.

This also indicates that the OD550 readings, which we supply as a continuous measure of our growth assays, could lead to an underestimation of the differences in CFU (rather than an

overestimation as the reviewer suggests). Therefore, our approach is a reliable and conservative method to demonstrate the effect of the NQNO treatment. The novel data showing the congruence of the OD550 readings and CFU counts are now included in Suppl. Fig. S2B and referred to in the text on page 8, line 174ff:

“The effect of NQNO on growth of gonococcal cultures was corroborated by determining colony forming units (cfu) instead of monitoring optical density (OD550 values) (Suppl. Fig. S2B).”

We would also second the reviewer’s statement *“A similar trend is shown for MS11-R2 (growth increases by time with HQNO treatment)”*. This is indeed what we observe and what we report, as this strain is resistant to NQNO (MS11-R2 has lost the pTetM plasmid). Therefore, the increased growth of this strain in the presence of NQNO (as highlighted also by the reviewer) strongly supports our conclusion that determinants on the pTetM plasmid are responsible for the exquisite sensitivity of *N. gonorrhoeae* towards NQNO.

8) Lines 325-326: Similar to the previous comment, Fig. 5B shows that the growth of WHO P pTetM A, WHO P pTetM B, and WHO P $\zeta 1/\epsilon 1$ is normally reduced (OD550 of the negative control reaches up to 0.4-0.5 only as compared to about 1.0 for WHO P). In presence of HQNO, the OD550 of WHO P pTetM A, WHO P pTetM B, and WHO P $\zeta 1/\epsilon 1$ is increasing by time and it reaches values comparable to that of DMSO (~0.25-4). A similar trend is shown for WHO P (growth increases by time with HQNO treatment). Consequently, it is not reasonable to conclude that *“Most importantly, Zeta1/Epsilon1-expressing derivatives of WHO P became NQNO-sensitive, while the parent strain was fully resistant up to 50 μ M (Fig. 5B)”*. The incubation time should be extended beyond 8 hours to see if there will be significant difference between the conjugants and the parent strains. Also, since the OD550 values are low in case of the conjugant strains, we cannot depend on the OD values alone for drawing conclusions. CFU counts are important to be included since this small difference in OD values may not reflect a significant difference in the CFU counts.

Similar to our response to the previous comment, the data presented in Figure 5 clearly demonstrate that the WHO P strain, which lacks the pConj/pTetM plasmid, is becoming NQNO-sensitive and exhibits increased membrane permeability upon complementation with the pTetM plasmid. Again, we have further supplement these results based on OD550 readings, as suggested by the reviewer and as already supplied for the MS11 strain, with parallel CFU counts. These additional experiments have been performed with the parent WHO P strain as well as the WHO P strain transformed with the $\epsilon 1/\zeta 1$ operon.

Similar to the results presented for *N. gonorrhoeae* MS11 (see comment 7), our results show that the CFU determination mirrors the OD550 readings for both strains (WHO P and WHO P $\epsilon 1/\zeta 1$). However, OD550 readings slightly underestimate the actual differences observed in

CFU readings (Reviewer Fig. 2) suggesting that the prominent effect of NQNO/3Me-NQNO on gonococcal growth observed by OD550 readings as reported by us underestimates the even stronger effect of these compounds on recoverable CFU. These data again demonstrate that the used method of OD550 readings is a valid and conservative read-out for determining the effect of compounds on bacterial growth. Due to the lack of space, we have not included these additional basic assays with strain WHO P and WHO P $\epsilon 1/\zeta 1$ in the manuscript, but provide these data here for reviewing purposes.

Reviewer Fig. 2: OD550 readings and CFU determination of *N. gonorrhoeae* WHO P and WHO P $\epsilon 1/\zeta 1$ provide an equivalent measure of bacterial growth. Growth of *N. gonorrhoeae* WHO P and WHO P $\epsilon 1/\zeta 1$ in the presence of 50 μ M NQNO or 1% DMSO was monitored by OD550 readings and by determination of CFU counts upon serial dilution from identical cultures. Shown are representative experiments.

9) Lines 341-347: The reason for switching to the use of *E. coli* instead of *N. gonorrhoeae* to investigate the degradative effect of reactive oxygen species on Epsilon antitoxin is not clear and should be clarified in the manuscript.

A common strategy to test, if specific additional factors are involved in protein maturation or turnover, is the heterologous expression of the protein in question. In that regard, we chose to use *E. coli* K12, as its growth is not affected by NQNO (Fig. 6D) and it is a standard host to express neisserial proteins (see e.g. Kupsch et al. 1993. EMBO J. 12: 641-650; Muenzner & Hauck. 2020. Cell Host & Microbe. 27:793-808). In this way, we could circumvent the primary effect of NQNO and test a direct action of NQNO on Epsilon stability.

As a step further in this direction, we have produced recombinant Epsilon protein, purified it from *E. coli*, and tested the direct action of NQNO on Epsilon stability (see also the next comment by the reviewer).

10) Fig. 6D illustrates that NQNO treatment did not impact the growth of *E. coli*. It's important to note that the outer membrane of *E. coli* differs from that of *N. gonorrhoeae*, which can lead to variations in the susceptibility of these bacteria to different agents. Some agents may exhibit inhibitory activity against *N. gonorrhoeae* without affecting *E. coli* or with reduced activity against *E. coli* due to permeability. This phenomenon is evident in the case of HQNO, which did not inhibit *E. coli* while displaying inhibitory effects on *N. gonorrhoeae*. Additionally, the results presented in Fig. S1 indicate that *N. gonorrhoeae* exhibits a larger inhibition zone compared to the minimal inhibition observed for *E. coli*. Therefore, the use of *E. coli* in this experiment may not support the conclusion that "NQNO does not act directly on Epsilon stability but rather acts indirectly via NQNO-induced oxidative stress." These experiments should ideally be conducted using *N. gonorrhoeae*, as the effects of NQNO on *E. coli* may be limited due to its reduced activity/permeability against this bacterium.

The reviewer suggests that there are differences in NQNO-membrane permeability between the *E. coli* K12 outer membrane (OM) and the gonococcal outer membrane and that a lower permeability of the *E. coli* OM might prevent NQNO from acting on *E. coli*. However, we did not find any data in the literature that support these ideas. Laboratory strains of *E. coli* such as the one used in this experiment usually show the highest (and not the lowest) outer membrane permeability, when compared to other Gram-negative bacteria. For example, Mortimer & Paddock compared the uptake of quinolones by *E. coli*, *Klebsiella pneumoniae*, and *P. aeruginosa* and did report higher quinolone concentrations in *E. coli* than in the other gramnegative bacteria tested (Mortimer & Paddock (1991) J. Antimicrob. Chemother. 28: 639-653). A similar higher permeability for *E. coli* than other Gram negatives has been described by other researchers (Geddes EJ et al. (2023) Nature. 624: 145-153. doi: 10.1038/s41586-023-067608). Most importantly, Baker et al. showed intracellular labelling of PqsR, the cytosolic receptor of PQS, by a synthetic PQS (isomeric quinolone of HQNO), when PqsR was expressed in *E. coli* (Baker YR et al (2017) Chem. Sci., 8, 7403. doi: 10.1039/c7sc01270e). This published study clearly demonstrates that the *E. coli* outer membrane is generally permeable for HQNO isomeric quinolones.

However, as we did not find data comparing permeability of *E.coli* versus *Neisseria* for quinolone-like compounds, we cannot rule out that NQNO (despite its overall hydrophobicity and its proven diffusibility into the *E.coli* cytoplasm) accumulates less in *E.coli* than in *Neisseria gonorrhoeae*. Therefore, we have conducted an additional experiment with recombinant, purified Epsilon protein in the presence of NQNO. As can be seen in novel Fig. 6F, addition of 50 μ M NQNO to purified Epsilon protein did not affect the stability of the antitoxin protein and Epsilon protein levels remained similar to the solvent control, whereas H₂O₂ treatment led to reduced Epsilon levels. In combination with the results of the experiments shown in Fig. 6B-E, these additional data further support our conclusion that NQNO does not act directly on Epsilon stability, but rather acts indirectly via NQNO-induced oxidative stress in susceptible bacteria such as *N. gonorrhoeae*.

11) Fig. 6D: there are no standard deviation values shown in the figure.

Indeed, this is a representative growth curve. The shown experiment has been conducted three times with similar results. We have amended the Figure legend for this panel, which is now Fig. 6E, accordingly to read:

“Shown is a representative experiment conducted three times with similar results.”

12) Lines 360-362: The proposed mechanism of NQNO action is mediated by unleashing the endogenous Zeta1 toxin from inhibition by its cognate epsilon1 antitoxin, and the enzymatic activity of Zeta1 will lead to disruption of cell wall synthesis and integrity. To further substantiate this conclusion, additional assays demonstrating the disruption of cell wall synthesis and integrity in response to NQNO treatment should be included. Relying solely on electron microscopy may not be sufficient to confirm this mechanism conclusively.

We completely agree with the reviewer, but we do not see his/her point here. In contrast to the statement by the reviewer, we did NOT rely solely on electron microscopy. Indeed, our initial submission included assays, which directly demonstrate the increased membrane permeability of gonococci upon treatment with NQNO. These assays have been conducted with multiple isolates and also with pTetM-deficient as well as the pTetM- or Epsilon/Zeta-reconstituted strains and are exactly the type of assays asked for by the reviewer. The primary data are shown in Figure 4B and 4G as well as Fig. 5C. Upon remarks by reviewers, we have also added quantification of all these membrane integrity assays, which are presented in Fig. 4C and 4H as well as Fig. 5D.

13) I have several concerns regarding the animal experiment. Firstly, the treatment was administered after only 1 hour, which is a very short duration for *N. gonorrhoeae* to attach to the vaginal tract and establish infection. Additionally, this timeframe may not be clinically applicable. Topical application of *Neisseria* is not supported and is unlikely to clear *Neisseria* infections. Moreover, any topical treatment, including mild antiseptic solutions or mild H₂O₂, applied in this model topically is likely to result in a significant reduction in bacterial load.

— We understand the reasoning of the reviewer and we would like to answer the two distinct issues raised (treatment schedule / topical application of the compound) separately.

— First, we are aware that topical application of antibiotics is not a current standard treatment for gonorrhea in medical practice. However, we are presenting an experimental treatment regimen, which demonstrates the efficacy of our novel compounds in an *in vivo* setting. As clearly elaborated in the discussion section, the topical administration of antibiotic compounds is considered one of the potential routes for the administration of microbicides in the context of sexually transmitted diseases and such an application is in medical practice for bacterial manifestations in the genital tract. For example, the local application of clindamycin- or metronidazole-containing gels is the current standard in treating bacterial vaginosis (Austin et al., 2005; Bradshaw and Sobel, 2016). Therefore, our experimental approach might differ from current gonorrhea treatment, but it is not unreasonable to propose that the novel compounds described in our manuscript, which are clearly suitable for such a topical application, could open up novel treatment options in the case of gonorrhea. Such potential novel approaches to treat or prevent gonorrhea connect to the second concern by the reviewer: treatment schedule.

Secondly, the reviewer questions our “prophylactic” treatment approach, where the antibiotic is applied shortly after exposure to the pathogen, a situation also referred to as Post-Exposure Prophylaxis (PEP). In this regard, the reviewer suggests that 1 hour of infection might be too short of a time frame to allow the bacteria to attach to the vaginal tract. Based on *in vitro* work by colleagues (Edwards & Apicella (2005) Cell Microbiol 7: 1197-1211) gonococci show strong adhesion to different human epithelial cells within 15 – 30 minutes after infection. However, such a very rapid pilus-mediated attachment presumably does not contribute to gonococcal colonization in mouse models (due to the lack of the appropriate human pilus receptor in the mouse). Nevertheless, we have shown that Opa-mediated firm attachment of gonococci to epithelial cells occurs within 60 min (Schmitter T et al. (2007) Infect Immun. 75: 4116-26). The Opa-CEACAM interaction is the basis for the epithelial cell attachment of the gonococci in our humanized mouse model (CEA-transgenic animals). We further demonstrate in Fig. 5 that the Opa-CEA interaction is the major contributor to neisserial colonization in this model, as there is a strong increase in gonococcal recovery from CEA-transgenic mice versus wildtype mice. Therefore, the assumption of the reviewer that 1 h infection time before treatment is “very short...to attach to the vaginal tract” is unfounded.

Most importantly, the notion that “this timeframe may not be clinically applicable” does not take into account the current paradigm shift in curbing the rise in STIs. Indeed, there has been an ongoing discussion on the use and efficacy of Post-Exposure-Prophylaxis (PEP) in targeting STIs in highly promiscuous and vulnerable groups. Such unconventional approaches have been tested in settings, where repeated infections in specific patient groups have been observed and they represent promising avenues to counteract the rise in transmission of *Treponema*, *Chlamydia*, and *Neisseria gonorrhoeae*. This recent paradigm shift has culminated in new recommendations by the Centers for Disease Control (CDC) in the USA to apply PEP in situations of unprotected sex in high risk settings BEFORE any clinical manifestations occur and BEFORE an infection is diagnosed (Bachmann, L.H., L.A. Barbee, P. Chan, H. Reno, K.A. Workowski, K. Hoover, J. Mermin, and L. Mena. 2024. CDC Clinical Guidelines on the Use of Doxycycline Postexposure Prophylaxis for Bacterial Sexually Transmitted Infection Prevention, United States, 2024. MMWR Recomm Rep 2024. 73(No. RR-2):1–8.). These recommendations have been endorsed by national medical societies such as the German STI Society (Werner, R.N., A.J. Schmidt, A. Potthoff, P. Spornraft-Ragaller, N.H. Brockmeyer, and German STI Society. 2024. Position statement of the German STI Society on the prophylactic use of doxycycline to prevent STIs (Doxy-PEP, Doxy-PrEP). J Dtsch Dermatol Ges. 22 466-478). We would like to stress that the approach followed by us mimicks such a PEP scenario and it is exactly this high-risk context (men-having-sex-with men, sex-workers), where topically applied gonococcidial compounds might be particularly helpful as elaborated in our discussion.

Accordingly, we have amended a sentence referring to these published recommendations to our rationale for the in vivo experiments introduced on page 16, line 379ff:

“Therefore, we tested if the topical application of NQNO to the genital tract of female mice can prohibit infection. In this way, NQNO could serve as a potential post-exposure prophylaxis, a form of treatment advocated for high risk settings (Bachmann et al., 2024; Werner et al., 2024).”

Moreover, in light of the well-intended advice by the reviewer, we have now conducted additional in vivo experiments with an altered pre-treatment and treatment regimen (see also reviewer comment 26), which would reflect the classical therapeutic approach of a treatment initiated upon established infection. Furthermore, these additional experiments have been conducted using our most advanced NQNO derivative (3Me-NQNO) and have been included in novel Fig. 9. In these experiments, the mice are first infected for 24h, before treatment with 3Me-NQNO is initiated. 24 later (48h after the start of the gonococcal infection), remaining gonococci are determined by colony count and tissue samples are analysed by immunohistology. The novel data in Fig. 9 A and B clearly demonstrate that similar to the previously applied “prophylactic” PEP-mimicking approach also in a “therapeutic” setting the alkyl-quinolone compound (3Me-NQNO) is highly effective in diminishing gonococcal CFUs. The data also show that in several mice (3 out of seven), the inoculated bacteria are

completely cleared within 24 h in line with the strongly improved efficacy of our 2nd generation alkyl-quinolone compound 3Me-NQNO.

14) Fig. 7B: Why is the number of mice inconsistent between groups? For instance, in the experiment of CEAtg mice, DMSO: 12 mice, NQNO (25 μ M): 6 mice, and NQNO (50 μ M): 9 mice. For wild-type mice, DMSO: 12 mice, 25 μ M NQNO: 4 mice, and 50 μ M NQNO: 5 mice. The same comment applies to the second mouse experiment (Fig. 7F).

Our veterinarian-run animal facility is very tidy (IVC-cages), but also very small. Therefore, we can maintain only a limited set of breeding pairs (4-5) at any given time. The transgenic animals are constantly backcrossed to wildtype animals (to avoid having homozygotic transgenic animals). Accordingly, the CEA-transgenic female mice employed in most of the experiments represent on average 25% of the offspring. As the female mice are included in experiments during a short age window only (6 - 8 weeks after birth), we have only a given amount of transgenic female animals (7-8 animals) in each experimental cohort, which are then subdivided between the different treatments. We perform several repetitions (biological replicates) of this experiment, but we stop the experiments, once we observe predefined significant ($p < 0.05$) differences between the experimental groups and the control treatment as mandated by the 3R principles. This can lead to differences in the total number of animals per group.

Based on initial experiments with the first generation of NQNO compounds, we calculated the effect size (Cohen's d) to be between 2 and 3. Based on this, we have performed an a priori power analysis with a minimum probability for the type I-error of $p < 0.05$ (the probability that we reject H_0 despite it is correct) to predict, how large a sample size we need to achieve this significance level. As seen in the next figure (Reviewer Fig. 3), the *a priori* analysis indicated that the sample size should consist of at least 4 animals per group.

Reviewer Figure 3: *A priori* determination for sample size in animal experiments based upon expected effect size of 3 and a type I error (alpha error) probability of $p < 0.05$. The minimum sample size based on this scenario was 4 animals/group

Once we had collected data from repetitions of the experiments such as the one shown in Fig. 7F comparing solvent treatment (DMSO) to the treatment with 50 μ M NQNO in CEAtg mice, we were able to determine the actual effect size to be 3.9. Using this effect size, we performed a *post hoc* power analysis for the given sample (6 animals in the control group; 7 animals in the treatment arm), which indicated that the actual power is 0.9999987 and the type II error probability (β error) is $1 - 0.9999987 = 0.0000013$ (Reviewer Fig. 4)

Reviewer Figure 4: Post hoc analysis of statistical power for the animal experiment shown in Fig. 7F based on observed effect size and actual sample size given a type I error (alpha error) probability of $p < 0.05$.

15) No control antibiotics (e.g. ceftriaxone and azithromycin) were used in both mouse experiments.

We are by our animal research ethics law not allowed to sacrifice mice for repetitions of already known facts. Accordingly, we always include the treatment control group (solvent only) and the experimental groups (NQNO at different concentrations) to being able to clearly detect an effect of the NQNO treatment versus the control treatment. Ceftriaxone and Azithromycin, which are standard antibiotics used in the clinic, are applied via completely different routes (parenteral or oral, respectively) distinct from the current experimental approach (topical treatment). Therefore, such distinct treatments with known outcomes would not add any additional insight.

16) Lines 417-419: “At concentrations of 0.5 μM and higher, 3Me-NQNO was bactericidal and induced rapid killing of the pathogens (Fig. 8E)”. Fig. 8F shows that at the concentrations of 0.5 μM , 3Me-NQNO was bacteriostatic, and did not reduce the bacterial count. The concentration of 2.5 μM resulted in reduction of the bacterial count by about 3 log₁₀ after 3 hours. So, at this concentration only, 3Me-NQNO can be considered bactericidal. Please, modify this sentence accordingly.

We have modified the respective sentence to now read: “Already at a concentration of 2.5 μM , 3Me-NQNO was bactericidal and induced rapid killing of the pathogens (Fig. 8E)”.

17) Fig. 8E and 8F: The figures do not show standard deviation values. How many replicates were used for these experiments?

While the 2-week experiment shown in Figure 8F was repeated only once with similar results, the experiment shown in Fig. 8E was performed three times with similar outcomes and a representative result was shown in Fig. 8E.

The three repetitions of the experiment had very similar outcomes as presented here in Reviewer Fig. 5:

Reviewer Fig. 5: Three repetitions of the growth experiment presented in Fig. 8E. These three graphs are combined now into a single graph and have replaced the former panel in main figure Fig. 8E.

We have modified the Figure legend for Fig. 8F accordingly to indicate that a representative experiment is shown and we have exchanged panel Fig. 8E for a panel combining all three repetitions shown in reviewer Fig. 5 including means \pm SEM.

18) Lines 422-424: Toxicity of 3Me-NQNO was measured up to 10 μ M while it showed activity at the concentrations of 2.5 – 5 μ M. To establish a high therapeutic index for the agent, cytotoxicity should be evaluated at higher concentrations, such as those several folds higher than the active concentration (e.g., 100 μ M or higher). This will help provide a more comprehensive assessment of the agent's safety profile and therapeutic potential.

The reviewer might have overlooked that 3Me-NQNO was indeed active at concentrations below 1 μ M against *N. gonorrhoeae*. These assays were presented in Fig. 8B and in Fig. 8D and show that at concentrations of 0.5 μ M (500 nM) 3Me-NQNO completely inhibited growth of *N. gonorrhoeae*. These data imply, that the concentration of 10 μ M 3Me-NQNO applied to human cells is 20-fold above the 3Me-NQNO MIC value for gonococci. Based on the comment by the reviewer, we have performed additional experiments with one human cell line, where we have extended the range of tested 3Me-NQNO concentrations up to 50 μ M (100-fold MIC concentration). These data are now presented in the new Suppl. Figure S8E and show that there is a large potential therapeutic index for 3Me-NQNO.

19) Lines 494-495: "the experimental infection of female mice does not recapitulate gonorrhea and symptomatic disease has not been observed in this model". This information is inaccurate. *N. gonorrhoeae* female mouse model established by Jerse et al 1999 showed symptoms of gonorrhea such as: 1) Gonorrheal inflammation occurred in over 80% of infected mice, 2) *N. gonorrhoeae* was recovered from vaginal swabs for an average of 12 to 13 days following infection, 3) Vaginal smears showed presence of neutrophils recruited to the site of infection.

Based on the valid comment by the reviewer, we have deleted this statement from the discussion section. The former paragraph on page 21, line 494ff read:

"Local application of NQNO to the vaginal tract of mice eliminated the gonococcal inoculum. Clearly, the experimental infection of female mice does not recapitulate gonorrhea and symptomatic disease has not been observed in this model (Jerse, 1999). However, the initial

colonization of the mucosal surface by Neisseria gonorrhoeae can be followed in CEA transgenic mice (Muenzner et al., 2010; Muenzner and Hauck, 2020) and it is during this phase of mucosal establishment that NQNO was applied locally to the genital tract. While a systemic administration of antibiotics is a preferred route in medical practice, sexually transmitted diseases such as HIV or HPV have been targeted by topically applied microbicides (Notario-Perez et al., 2017)."

and the respective paragraph on page 22, line 531 is now shortened to:

"Local application of NQNO to the vaginal tract of mice prohibited gonococcal infection in a post-exposure prophylaxis setting and topical treatment with 3Me-NQNO was able to eradicate established infections. While a systemic administration of antibiotics is a preferred route in the treatment of gonorrhea, allowing also monitoring of patient compliance, sexually transmitted diseases such as HIV or HPV have been targeted by topically applied microbicides (Notario-Perez et al., 2017)."

20) Lines 500-510: discuss the use of topically applied antimicrobials for sexually transmitted diseases, including *N. gonorrhoeae*. However, it's important to note that the case of *N. gonorrhoeae* is different due to specific criteria for anti-gonococcal therapeutics. For *N. gonorrhoeae*, patient compliance is crucial, and there are certain criteria that should be considered. These criteria include the route of administration, which should be either oral or intramuscular (IM), and the treatment duration. For urogenital *N. gonorrhoeae*, the treatment duration should be 1 day for the oral route and up to 3 days for anorectal and oropharyngeal *N. gonorrhoeae* (for the oral route), or 1 day for all types of *N. gonorrhoeae* infections (for the IM route). These criteria reflect the specific challenges and requirements for treating *N. gonorrhoeae* effectively.

As the reviewer correctly states, we include this prospective future use of NQNO-derived compounds as topical anti-gonococcal agents in the discussion section. We are aware that the current usage of anti-gonococcal drugs differs and that for current antibiotics, which are used therapeutically, patient compliance is a critical parameter. Nevertheless, this should not limit the discussion of potentially novel ways of application, such as prophylactic topical treatments, once highly selective (and therefore not harmful to the beneficial lactobacilli flora) and easily applied formulations would become available. In short, the reviewer is correct about current treatment, but our discussion is about the potential future application of compounds with a novel activity spectrum, which would open up novel possibilities in countering gonorrhea.

The open-minded approach to search for additional avenues to curb the spread of STIs in particular patient groups is reflected in the current move to include post-exposure prophylaxis (PEP) to fight a group of notorious STI-agents, including *Neisseria gonorrhoeae*, in routine

medical treatment. See also comment 14) and our discussion of PEP in light of the new recommendations by the Centers for Disease Control (CDC) (Bachmann, L.H., L.A. Barbee, P. Chan, H. Reno, K.A. Workowski, K. Hoover, J. Mermin, and L. Mena. 2024. CDC Clinical Guidelines on the Use of Doxycycline Postexposure Prophylaxis for Bacterial Sexually Transmitted Infection Prevention, United States, 2024. . MMWR Recomm Rep 2024. 73 (No. RR-2):1–8) and the German STI Society (Werner, R.N., A.J. Schmidt, A. Potthoff, P. Spornraft-Ragaller, N.H. Brockmeyer, and German STI Society. 2024. Position statement of

the German STI Society on the prophylactic use of doxycycline to prevent STIs (Doxy-PEP, Doxy-PrEP). . J Dtsch Dermatol Ges. 22 466-478).

We also would like to stress that the clinical application is not the focus of our study and is out of scope for such an initial explorative research into the potent and selective efficacy of NQNO-derived compounds and into the analysis of their mode-of-action. Indeed, our study reports the exciting discovery of a novel principle of targeting endogenous TA systems by naturally occurring secondary metabolites and their improvement by chemical diversification. As numerous pathogens harbor TA systems and as the underlying principle might also be relevant to other microbes, we do not feel that the discussion needs to be confined to the current standard treatment of a particular microbial species. Nevertheless, we are thankful for this advice by the reviewer, and we have modified a statement with regard to topical application of microbicidal compounds and the distinction to current standard treatment of gonorrhea in our discussion section on page 22, line 533:

“While a systemic administration of antibiotics is a preferred route in the treatment of gonorrhoea, allowing also monitoring of patient compliance, sexually transmitted diseases such as HIV or HPV have been targeted by topically applied microbicides (Notario-Perez et al., 2017). Furthermore, local application of clindamycin- or metronidazole-containing gels are current standard therapies for bacterial vaginosis (Austin et al., 2005; Bradshaw and Sobel, 2016). Therefore, vaginal administration of a NQNO-containing gel preparation might also be feasible to control gonococcal infections. In particular, local application of NQNO or 3MeNQNO as exemplified in our animal models would create a novel therapeutic and potentially post-exposure prophylaxis option in addition to standard systemic antibiotic therapy.”

21) I would recommend adding a paragraph discussing the limitations of the study such as the pre-existing resistance (presence of *N. gonorrhoeae* strains with resistance to NQNO e.g. WHO P and strategies to overcome this resistance).

We think it is always a good idea to hint to the limitations of a research study. Therefore, we have included an additional statement discussing the occurrence of pre-existing NQNO resistance and its worldwide prevalence in the discussion section on page 22, line 517:

“These desirable properties of NQNO are slightly tarnished by the occurrence of naturally NQNO-resistant strains such as WHO P, which lack the pTetM plasmid. While the occurrence of WHO P-like azithromycin-resistant strains has increased in Europe in recent years (Day et al., 2022), pTetM-positive and therefore NQNO-sensitive strains are highly prevalent in other parts of the world (Cehovin et al., 2018; Agbodzi et al., 2023; Tayimetha et al., 2023).”

22) A general comment on the materials and methods section: most of the experimental protocols presented do not have citations. References for each experiment need to be added.

This is an interesting comment by the reviewer, but we do not see a benefit in adding more citations. In our experience, such type of method references will put the strain on readers to follow back chains of references before the critical information is located. Instead, we rather prefer to stick with the Nature Microbiology guidelines, which state that “The Methods section should be written as concisely as possible but should contain all elements necessary to allow interpretation and replication of the results.” Accordingly, rather than just referring to previous papers, we try to detail all essential information to interpret and replicate the experiments in the Methods section of the manuscript.

23) Lines 629-635: “Determination of NADH levels”. The experiment also lacks a positive control (an agent that is known to reduce the NADH levels in the bacteria) which provides the level of confidence in the results obtained and ensure that the results are reproducible and robust. The same comment applies to the experiments of “Detection of reactive oxygen species (ROS)”

The reviewer probably missed the use of Antimycin A, a known inhibitor of complex III, which we have employed as a positive control throughout our experiments. This inhibitor, which leads to increases in reactive oxygen species as well as to reductions in NADH and ATP levels, is included as a positive control in Fig. 2A (ATP levels), Fig. 2B (NADH levels), 2C and 2D (ROS production), Fig. 3H (ROS production), Fig. 3I (NADH levels), and Fig. 6G (Epsilon degradation).

24) Lines 647-662: “Selection for NQNO and 3Me-NQNO resistance”. The proposed method requires citation, and the experiment also lacks a positive control (an agent that is

known to be susceptible to resistance development such as rifampicin, tetracycline or ciprofloxacin) which confirms that the assay is right, and the results obtained are reproducible and robust. Additionally, in this method, bacteria are exposed daily to the same conditions (conditioning with 2.5 μM NQNO or 0.1 μM 3Me-NQNO for 8 h) and then evaluated for sensitivity towards different concentrations of NQNO or 3Me-NQNO. This technique may not enhance the development of bacterial resistant mutants since bacteria are exposed to exactly the same conditions every day. The most commonly used method is through serial passages where the bacterial inoculum is prepared from cultures with a subinhibitory concentration of the test agent (these concentrations are not the same every day and may be different according to the MICs of each passage).

Our goal was to test for the onset of resistance i.e. how fast resistance can develop for these compounds and not to reach the maximum resistance level for a given compound. Sub-MIC level selection (at constant sub-MIC passages) and above MIC level selection (as described by the reviewer) are both valid methods for resistance testing and also high-level resistances can arise independent of the method used (Wistrand-Yuen E., Knopp, M., Hjort, K. et al. Evolution of high-level resistance during low-level antibiotic exposure (2018) Nature Communications 9, 1599. doi: 10.1038/s41467-018-04059-1). The study by Wistrand-Yuen et al. is now cited in the Methods section.

In addition, the MIC did not change for 3Me-NQNO over 14 days, which is why there was no possibility to increase the concentration of exposure during the passages. For NQNO the MIC only increased towards the very end of the exposure. Since the sub-MIC concentration has to be adapted to the actual MIC of a compound, there is no possibility to add a control antibiotic within Figure 3D or 8F. Instead, this would be an entirely independent experiment on its own, which is merely a reproduction of previous work and does not add additional insight. As we observe the development of resistance against NQNO, the data presented in Fig. 3D validate that the approach of constant sub-MIC passages as used by us and others is adequate for detecting resistance development.

25) Lines 898-899: A single dose of 17- β -estradiol was administered. To establish the gonococcal infection, three injections of 17 β -estradiol 3-benzoate are commonly administered (one injection every other day) or a slow-release 17 β -estradiol 3-benzoate pellet is implanted to establish a continuous higher estradiol level which is crucial for *N. gonorrhoeae* infection. Also, mice were infected 4 days after estradiol dose. The estradiol level decreases gradually by time. So, after 4 days, the level will be very low given that a single dose was administered. So, these conditions are not ideal to establish a successful gonococcal colonization. Mice should be inoculated with *N. gonorrhoeae* 2 days after estradiol treatment and receive a second dose of estradiol in the same day of infection to ensure that levels of estradiol remain high.

As described in the reference cited (Muenzner et al., 2010) the dose of 17- β -estradiol is dissolved in corn oil and administered subcutaneously. There, the oil serves as a slow release reservoir, which similar to a slow release pellet allows a continuous release of the lipophilic hormone over the course of the experiment. Unfortunately, the description in the cited reference was not precise and 17- β -estradiol in corn oil was indeed administered twice: on day -4 and on day -2 before infection on day 0.

In light of the additional treatment schedule, which we have included in the revised version (see also our answer to comment 26) below), we now provide an expanded and more detailed description of the two *in vivo* treatment regimes, which reflect a "prophylactic" and a "therapeutic" approach, in the Material and Methods section on page 38, line 931ff.

26) Streptomycin is not included in the animal protocol described. Streptomycin is important in the gonococcal infection model to prevent the growth of commensal Gram-negative bacteria which, if present, prevent the gonococcal colonization. Vancomycin is also injected to mice suppress Gram-positive flora permitting successful colonization of *N. gonorrhoeae*.

As the reviewer correctly states, additional antibiotics on top of trimethoprim have been included by researchers in the pre-treatment regimen. The idea behind the application of a cocktail of antibiotics is to suppress growth of the endogenous microbial flora, which can expand during estrogen treatment and which could interfere with gonococcal colonization. However, the necessity for such an expanded antibiotic treatment depends on the used mouse strain and the composition of the community of commensal bacteria in the given mouse colony.

We had compared the recovery of gonococci from mice in our colony treated solely with trimethoprim vs. mice receiving trimethoprim & streptomycin vs. mice receiving trimethoprim & streptomycin & vancomycin in the past. All of these treatments prevented overgrowth of the commensal flora and allowed recovery of gonococci 24h after infection to a similar extent. As trimethoprim is given via the drinking water, we had initially observed that some mice strongly reduce their liquid intake upon trimethoprim addition (we monitor the decline in the volume of the provided drinking water). This might be the reason, why trimethoprim addition alone does not work for some investigators. As a consequence, we have started to slightly sweeten the drinking water containing trimethoprim, dramatically increasing the compliance of the mice to take in trimethoprim. Gonococci are naturally resistant against trimethoprim, while this antibiotic acts against gram-negative as well as gram-positive bacteria. Accordingly, this treatment gives gonococci apparently a sufficient advantage during the initial colonization of the vaginal tract, which is clearly documented by the recovery of these bacteria following infection in our previously published and also in the herein reported study.

We would like to stress, that we plate the swab material recovered from the infected animals both on GC plates containing antibiotics (to select for the used gonococcal strain) as well as on non-selective, antibiotic-free GC agar as described in Material&Methods. Therefore, we are

able to detect the presence and the abundance of commensal bacteria in the vaginal tract of the employed mice. In rare instances (~1 in 30 mice), we have observed an overabundance of Gram-positive commensal bacteria in individual mice (10-fold higher numbers of bacteria recovered on antibiotic-free GC plates than on GC plates containing antibiotics) and have excluded such mice from further analysis. We have also determined the identity of these commensal bacteria by 16S rRNA sequencing (*Streptococcus spec.*; *Rothia nasimurium*), but we do not have an explanation for their sporadic occurrence currently.

Accordingly, for the prophylactic setting, where NQNO-compounds are applied in a PEPmimicking manner, the mild antibiotic regimen allows the recovery of gonococci and also allows the efficacy of test compounds to be revealed.

However, following the advice of the reviewer from comment 13), that the conventional, therapeutic treatment of gonorrhea starts once the infection is established, we have now also conducted experiments with a more aggressive antibiotic pre-treatment of the animals to allow recovery of gonococci after prolonged infection. In this regard, the drinking water of the animals is supplemented with trimethoprim from day -4 as before. On top of that antibiotic treatment, the animals receive injections with estrogen and streptomycin/chloramphenicol, respectively, on day -4 and day -2 before they are infected with gonococci on day 0. 24h after the infection, the mice are then treated vaginally with solvent (DMSO) or with our test compound 3Me-NQNO. Another 24h later, CFUs are determined by vaginal swaps and tissue samples are taken for histology. The data from these “therapeutic” treatment approaches with 3Me-NQNO are now presented in novel Fig. 9A and B. These experiments clearly demonstrate that topically applied 3Me-NQNO also functions in this “therapeutic” setting and diminishes (and in a few cases completely eradicates) gonococci from the female genital tract.

27) Lines 908-910: “samples collected were plated onto GC agar containing chloramphenicol (10 µg/ml) and erythromycin (7 µg/ml)”. The antibiotics added to the plates does not contain antifungal agent like nystatin to inhibit *Candida* which can be present in the vaginal fluid. Also, colistin is frequently added as one of the antibiotics to the GC plates to inhibit Gram-negative bacteria, including *Pseudomonas* species.

Most likely due to the barrier facility housing our animals and the use of IVC cages, in more than 15 years of conducting these types of infection experiments with female mice we have never observed *Candida* growth on the GC plates without antibiotics recovering the swap material. This observation underscores our previous point that such additional antibiotic applications might only be necessary under conditions, where the endogenous flora present in the mouse colony under question mandates such an additional treatment.

28) *N. cinerea* is not affected by NQNO or 3Me-NQNO unlike *N. gonorrhoeae*. Based upon the proposed mechanism, it is very important to show the genetic differences between *N. cinerea* and *N. gonorrhoeae* including the electron transport systems and Zeta/epsilon toxin/antitoxin systems and show the differences which make *N. cinerea* resistant to these agents.

— This is an interesting suggestion by the reviewer, as we exactly follow this line of thought in our manuscript to exploit genetic differences in our effort to pinpoint the molecular determinants for the NQNO-sensitivity of *N. gonorrhoeae*. However, instead of starting with *N. cinerea* as a reference point (where 1000s of genetic differences between this species and *Neisseria gonorrhoeae* can be expected) we employed isogenic, resistant strains isolated after antibiotic conditioning (Fig. 3). Accordingly, and as detailed in Fig. 3F, we were able to detect a limited amount of genetic changes correlating with NQNO-sensitivity, allowing us to rapidly identify the $\epsilon 1/\zeta 1$ TA system, which is not present in the resistant strains and which is also not present in *Neisseria cinerea*, as the critical genetic determinant.

—

Reviewer Comments:

I would like to thank the authors for addressing some of my concerns. However, two issues still need to be addressed.

First, the difference between the outer membranes of *E. coli* and *Neisseria*, and the impact this has on the conclusions of the experiment. I was surprised that the authors did not find relevant literature to support this distinction, so I am including some references (1-3) that may help in selecting the correct strains and designing an appropriate experiment. The outer membranes of *E. coli* and *Neisseria* differ significantly in composition, function, and ability to allow the passage of drugs or compounds. *Neisseria* has a lipooligosaccharide (LOS), while *E. coli* has lipopolysaccharide (LPS). LOS is shorter and less complex. The porins in *Neisseria* differ from those in *E. coli*. The *E. coli* strain used in the study does not have the highest permeability and possesses intrinsic resistance mechanisms, including the outer membrane and efflux systems, which prevent most inhibitors from crossing (4) (Randall et al., 2013, *Antimicrob Agents Chemother* 57(1): 637–639). When these intrinsic resistance mechanisms are artificially compromised, such as by permeabilizing the outer membrane or deleting the efflux systems, agents can cross and demonstrate antibacterial activity. To confirm the conclusion that NQNO treatment does not affect *E. coli*, experiments should be conducted using *E. coli* strains with compromised outer membranes and efflux systems (4) (Randall et al., 2013, *Antimicrob Agents Chemother* 57(1): 637–639).

The second major issue is the animal experiment, which remains seriously flawed. While this reviewer acknowledges the potential benefits of prophylaxis, it is important to note that neither prophylactic nor treatment measures for urogenital *Neisseria* infections involve topical application, as it is ineffective. Another significant issue is the lack of positive controls. The authors' explanation for not including a positive control antibiotic in the treatment group raises concerns about the experiment's rigor and validity. Positive controls in drug discovery are essential for validating the experimental setup's ability to detect an effect. Without positive control, it is unclear whether the treatment's lack of effectiveness/clearance is due to the compound itself or flaws in the experimental design. Given that the experimental setting used is non-standard, including a positive control is crucial to establish a valid baseline. The claim that animal research ethics laws prohibit the sacrifice of mice for known facts suggests a misunderstanding of ethical guidelines.

- 1. Wolf-Watz, H.; Elmros, T.; Normark, S.; Bloom, G. D., Cell envelope of *Neisseria gonorrhoeae*. A comparative study with *Escherichia coli*. *Br J Vener Dis* 1976, 52 (2), 142-5.**
- 2. Morse, S. A.; Cacciapuoti, A. F.; Lysko, P. G., Physiology of *Neisseria gonorrhoeae*. *Adv Microb Physiol* 1979, 20, 251-320.**
- 3. Morse, S. A., *Neisseria gonorrhoeae*: physiology and metabolism. *Sex Transm Dis* 1979, 6 (1), 28-37.**
- 4. Randall, C. P.; Mariner, K. R.; Chopra, I.; O'Neill, A. J., The target of daptomycin is absent from *Escherichia coli* and other gram-negative pathogens. *Antimicrob Agents Chemother* 2013, 57 (1), 637-9.**

Author Responses

Major concern 1) *E.coli* outer membrane might be impermeable to NQNO:

This “concern” is a smoke bomb, as the reviewer’s initial critique (comment #10) questioned our statement that “NQNO does not directly affect the epsilon antitoxin”. In the revised manuscript, we have addressed this point by conducting experiments using purified epsilon toxin treated with NQNO *in vitro* (see novel Fig. 6F and read our response to comment #10 in our previous response-to-reviewer letter). Now the reviewer introduces an unrelated, inflated discussion about the differences in the outer membrane of *E.coli* versus the outer membrane of gonococci, a well-known fact that has no relevance to the original question:

- a) our study focuses on why NQNO works in *N. gonorrhoeae*, not why it doesn’t work in *E. coli*;
- b) we performed additional, *in vitro* experiments with purified components, confirming that there is no direct action of NQNO on the epsilon-antitoxin. These clear-cut *in vitro* results eliminate any concerns about membrane permeability or efflux pump activity influencing the outcome of our mechanistic investigation. Accordingly, this experiment allows us to affirm that “NQNO does not directly affect the epsilon antitoxin”.

Therefore, this “major concern” by the reviewer is nothing more than a made-up sideshow. It is baffling that this reviewer pontificates about differences in bacterial outer membranes, when we not only cite several studies, which report the penetration of quinolones (such as NQNO) into *E. coli* cells (e.g. a study exploring their role as quorum sensing signals (Baker YR et al., 2017, Chem. Sci., 8, 7403. doi:

10.1039/c7sc01270e)), but also provide data from cell-free, *in vitro* approaches. It is either a failure on the part of the reviewer to grasp our straightforward biochemical approach (then he/she might not qualify as a reviewer for these types of experiments and his/her comment **should be disregarded**) or it is a deliberate misuse of her/his position to obstruct our manuscript's publication by throwing a red herring (then this comment **must be disregarded**).

Major concern 2) "animal experiment, which remains seriously flawed"

The reviewer’s second major concern is based on incorrect and misleading statements that do not reflect current scientific standards.

Most prominently, she/he demands additional positive controls for an experimental treatment of mice, which is intended for the proof-of-principle demonstration of a new species-specific bactericidal compound acting via a novel mechanism. In stark contrast to what the reviewer repeatedly insists, the **use of an additional control antibiotic is NOT standard practice** when assessing the *in vivo* efficacy of novel antibacterial candidates. Examples from previous studies published in your Journal or in your high standard sister journals serve as examples:

- a) studies on Lolamycin, which was tested against *Klebsiella pneumoniae* and others in an acute pneumonia model (Muñoz, K.A. et al. (2024) Nature 630: 429–436. doi: 10.1038/s41586-024-07502-0)
- b) studies on Zosurabalpin, which was tested against peritoneal –instilled

Acinetobacter baumannii in a sepsis model (Zampaloni, C. et al. (2024) Nature, 625: 566–571. doi: 10.1038/s41586-023-06873-0)

c) studies on elimination of *Staphylococcus aureus* by Epifadin-producing *S. epidermidis* in a cotton rat nasal colonization model (Torres Salazar, B. O. et al. (2024) Nature Microbiology, 9: 200–213. doi: 10.1038/s41564-023-01544-2

d) studies on Iboxamycin against multiple pathogenic bacteria in a neutropenic mouse thigh infection model (Mitcheltree, M. J. et al. (2021) Nature, 599: 507–512. doi: 10.1038/s41586-021-04045-6

e) studies on Abaucin (Liu G. et al. (2023) Nat Chem Biol. 19: 1342–1350. doi: 10.1038/s41589-023-01349-8), which was tested against *Acinetobacter baumannii* in a dorsal wound infection model.

f) studies on Lugdunin (Zipperer et al. (2016) Nature 535: 511–516 doi: 10.1038/nature18634), which was tested against *Staphylococcus aureus* in a mouse skin infection model.

None of these studies included a "positive control antibiotic" with known activity in their *in vivo* animal models. There are countless studies, published in other, well-respected journals, reporting early pre-clinical work on the effectiveness of novel antibiotic candidates. In general, these studies use mock versus compound treatment to assess efficacy, but **NO additional "positive control" with an already approved and wellknown antibiotic** as mandated by this reviewer.

We list here three further prominent examples published in other respected journals investigating the *in vivo* efficiency of novel experimental compounds. Again, the study designs (mock treatment vs. compound treatment) essentially reflect the study layout used in our manuscript:

Wu, K.J.Y. et al. (2024) Science, 383(6684), 721–726. doi: 10.1126/science.adk8013.

Martin, J. K. et al. (2020) Cell, 181(7), 1518–1532. doi: 10.1016/j.cell.2020.05.005

Di, Y.P. et al. (2020) Sci Adv, 6(18), eaay6817. doi: 10.1126/sciadv.aay6817.

We would also like to point out investigations, which explore the use of topical microbicides in the murine vaginal infection model with *Neisseria gonorrhoeae* by Dr. Jerse. The research question behind these studies essentially mirrors our own investigation and the reviewer explicitly referred to the methodology used by Dr. Jerse in his initial assessment of our manuscript. Importantly, both studies **do NOT include the oral or parenteral or topical application of a known antibiotic as a "positive control"!**

Pilligua-Lucas M, Tkavc R, Bash SK, North BB, Weitzel MB, Jerse AE. (2023) Sex Transm Infect. 99: 409–415. doi: 10.1136/sextrans-2022-055596.

Spencer SE, Valentin-Bon IE, Whaley K, Jerse AE. (2004) J Infect Dis. 189: 410–9. doi: 10.1086/381125.

Also from a logical standpoint, the reviewer's desire for an additional "positive control" is not justified, as our data clearly demonstrate the efficacy of our novel antibiotic against *Neisseria gonorrhoeae* in an animal model using both prophylactic and therapeutic approaches (the

latter conducted according to the reviewer's suggestions from the initial review). It seems that the reviewer herself/himself got lost in her/his own logic, when she/he now writes. "Without positive control, it is unclear whether the treatment's lack of effectiveness/clearance is due to the compound itself...". Clearly, there is no "lack of effectiveness" in our case. However, even if there would be a "lack of effectiveness", an additional positive control would not help in any way to understand if such a "lack of effectiveness is due to the compound itself".

Obviously, to evaluate the effectiveness of an experimental treatment you need a negative control / mock treatment / placebo treatment (see also all the above cited *in vivo* studies). This is even more important in the case of the experimental protocol requested by the reviewer and originally formulated by Dr. Jerse ((Jerse AE. (1999) *Infect Immun.* 67(11):5699-708. doi: 10.1128/IAI.67.11.5699-5708.1999), which employs a massive pre-treatment of the animals with hormone and 3 different antibiotics. Accordingly, it is pertinent to include a mock treatment as a negative control, where all the animals undergo this extensive pretreatment, to demonstrate that the effect is not due to the extensive pretreatment. Our data in novel Figure 9A clearly show that, when compared to the relevant Mock-treated control, 3Me-NQNO completely eradicates bacteria in about half of the infected animals within 24h and significantly reduces bacterial levels (by 2 log₁₀) in the remaining 3Me-NQNO-treated animals. Including a well-known antibiotic would not provide additional insights and there is no antibiotic that can do better than clearing the infection within a day.

In this context, **the reviewer ridicules our ethical concerns** performing repeat experiments / positive controls with known antibiotics as "a misunderstanding of ethical guidelines". Ethical guidelines relevant for animal experimentation include the 3R principles (*Reduce, Refine, Replace*), which are nowadays mandated by regulatory authorities and funding bodies, with the important aim of minimizing the number of animals used in research. Conducting repeat experiments with a well-known antibiotic (already in use in the clinic) with a different mode of action and a different route of administration would provide no additional insights and would heavily violate the 3R principle of "Reduce". Dismissing these concerns and questioning the rigor of our experiments is clearly inappropriate and reflects yesterday's mindset that defines "experimental rigor" by the number of sacrificed animals.

With regard to the conduct of animal experiments, the reviewer insinuates that we would use a non-standard animal model ("Given that the experimental setting used is non-standard..."). This **misrepresentation of our work** is repeated in her/his newest statement, despite the fact that we have now followed the approach mentioned by the reviewer (Jerse AE. (1999) *Infect Immun.* 67(11):5699-708. doi: 10.1128/IAI.67.11.5699-5708.1999) for the additional animal experiments included in the revised manuscript. It is obvious, that this reviewer defines the model used by Dr. Jerse as the "Standard experimental setting", though there have been important adaptations to this protocol since 1999 based on scientific evidence. The most important rational alteration is the switch from using wildtype mice (which are still continued to be used by Dr. Jerse and some other investigators in their studies and which this reviewer obviously believes to be the "standard animal experiment") to using humanized mice, which express the human receptor for the gonococcal Opa adhesins on the vaginal mucosa. This deliberate alteration has been introduced by us in 2010 (Muenzner P et al (2010) *Science.* 329(5996):1197-201. doi: 10.1126/science.1190892). Clearly, humanized mice support at least some of the human-specific interactions of these highly specialized bacteria. This is significant as several critical virulence factors of pathogenic *Neisseriae* (such as adhesins, ironacquisition proteins,

complement-interfering proteins) only interact with human, but not murine counterparts. Our choice of CEACAM-transgenic mice was based on the finding that gonococcal Opa adhesins selectively engage human members of the CEACAM family, but that gonococci (and other CEACAM-binding human bacterial pathogens) do not bind to murine CEACAM family members (Voges M et al (2010) BMC Microbiol. 2010 Apr 20;10:117. doi: 10.1186/1471-2180-10-117). CEACAM-transgenic mice have since been used not only by us, but also by other researchers investigating gonococcal genital infections in the mouse model (Yu Q et al (2019) PLoS Pathog.

15(12):e1008136. doi: 10.1371/journal.ppat.1008136; Muenzner P & Hauck CR (2020) Cell Host Microbe. 27(5):793-808.e5. doi: 10.1016/j.chom.2020.03.010; Islam EA (2016) Mucosal Immunol. 9(4):1051-64. doi: 10.1038/mi.2015.122). In all these studies it has been found that the presence of the human receptor facilitates gonococcal colonization of the murine vaginal and uterine mucosa, **making CEACAM-transgenic, humanized mice a more appropriate model than wildtype mice** for gonococcal vaginal infection. CEACAM-transgenic mice are now also employed in studies concerning other CEACAM-binding human pathogens, again improving bacterial colonization of various mucosal sites and revealing species-specific aspects not seen in "standard" wildtype mice (Sheikh A et al. (2024) Proc Natl Acad Sci U S A. 121(38):e2410679121. doi: 10.1073/pnas.2410679121; Pajon R et al (2015) Vaccine.

33(11):1317-1323. doi: 10.1016/j.vaccine.2015.01.057; Johswich KO et al (2013) PLoS Pathog. 9(7):e1003509. doi: 10.1371/journal.ppat.1003509). Therefore, it is absurd to disparage our approach as "non-standard", when the standard from 1999 with employing regular wildtype mice is outdated.

Finally, when faced with the effectiveness of the topical treatment using NQNO and 3Me-NQNO, the reviewer firmly attests "topical application ... is ineffective"! This sentence reflects the response by the holy inquisition to Kopernikus' findings: "No matter what you find, the Sun will always circle Earth!". But to phrase it in a more serious manner: The statement of the reviewer that "topical application ... is ineffective"! is a **blatant denial of experimental results and scientifically unsound**.

To put the reviewer's demeanor in context, consider research on gonococcal vaccines. Despite past failures, colleagues such as Dr. Jerse, Dr. Russell, and Dr. Gray-Owen continue to investigate gonococcal vaccine candidates using murine vaginal infection models, either with wildtype (Jerse/Russell) or CEA-transgenic (Gray-Owen) mice. Should their grant applications be denied or their manuscripts be rejected with the blatant statement "vaccination against gonococci ...is ineffective"? Of course not! Just as vaccine research should not be dismissed due to past challenges, the potential for novel topical treatments, especially those with new mechanisms of action like NQNO and 3Me-NQNO, should not be brushed off by dogmatic beliefs. The compounds described in our study have the added advantage of selectively targeting *N. gonorrhoeae* while sparing beneficial lactobacilli, an important improvement compared to earlier microbicides.

Together, our study and the included *in vivo* experiments provide a strong proof of principle for a novel species-selective antibiotic that is potent and safe when applied topically, opening new avenues for future clinical research. Although our study is not intended to present a final clinical drug candidate, it marks a significant step forward in developing effective treatments for gonococcal infections.

Summing up all the evidence provided above, the categorical statement of the reviewer that our “animal experiment [...] remains seriously flawed” and requires a “positive control” is completely unfounded. It is obvious that this reviewer, for whatever reason, is guided by strong personal sentiments against our work and that this reviewer’s opinions do not reflect current standards in microbiology and infection biology. In light of the overwhelming support from three expert reviewers and our detailed response, including extensive new data addressing all 28 points raised, it is difficult to rationalize how the unsupported and far-fetched “major concerns” by this fourth reviewer can justify rejection.

We very much appreciate that you have given us the opportunity to respond to this reviewer's comments.